palaeontology/evolution

Archosauria, Triassic, Euparkeria, anatomical network analysis, sensory and feeding evolution

**Author for correspondence:**
Roland B. Sookias
e-mail: sookias.r.b@gmail.com

# The craniomandibular anatomy of the early archosauriform *Euparkeria capensis* and the dawn of the archosaur skull

Roland B. Sookias[1,2], David Dilkes[3], Gabriela Sobral[4], Roger M. H. Smith[5,6], Frederik P. Wolvaardt[5], Andrea B. Arcucci[7], Bhart-Anjan S. Bhullar[8,9] and Ingmar Werneburg[10,11]

[1]Museum für Naturkunde, Leibniz-Institut für Evolutions- und Biodiversitätsforschung, Invalidenstraße 43, 10115 Berlin, Germany
[2]Department of Earth Sciences, University of Oxford, South Parks Road, Oxford OX1 3AN, UK
[3]Department of Biology, University of Wisconsin Oshkosh, Oshkosh, WI 54901, USA
[4]Staatliches Museum für Naturkunde, Rosenstein 1, 70191 Stuttgart, Germany
[5]Evolutionary Studies Institute, University of the Witwatersrand, 1 Jan Smuts Avenue, Braamfontein 2000, Johannesburg, South Africa
[6]Iziko South African Museum, PO Box 61, Cape Town, South Africa
[7]IMIBIO CONICET Universidad Nacional de San Luis, Av Ejercito de los Andes 950, 5700 San Luis, Argentina
[8]Department of Earth and Planetary Sciences, 210 Whitney Ave., Yale University, New Haven, CT 06511, USA
[9]Yale Peabody Museum of Natural History, 170 Whitney Ave., New Haven, CT 06511, USA
[10]Senckenberg Center for Human Evolution and Palaeoenvironment (HEP) at Eberhard-Karls-Universität, Sigwartstraße 10, 72076 Tübingen, Germany
[11]Fachbereich Geowissenschaften der Eberhard-Karls-Universität Tübingen, Hölderlinstraße 12, 72074 Tübingen, Germany

RBS, 0000-0002-5189-4011; GS, 0000-0002-5001-4406; FPW, 0000-0001-8117-558X; IW, 0000-0003-1359-2036

Archosauria (birds, crocodilians and their extinct relatives) form a major part of terrestrial ecosystems today, with over 10 000 living species, and came to dominate the land for most of the Mesozoic (over 150 Myr) after radiating following the Permian–Triassic extinction. The archosaur skull has been essential to this diversification, itself diversified into myriad forms. The archosauriform *Euparkeria capensis* from the Middle Triassic (Anisian) of South Africa has been of great interest since its initial description in 1913, because its anatomy shed light on the origins and early evolution of crown Archosauria and potentially approached that of the archosaur common ancestor.

*Euparkeria* has been widely used as an outgroup in phylogenetic analyses and when investigating patterns of trait evolution among archosaurs. Although described monographically in 1965, subsequent years have seen great advances in the understanding of early archosaurs and in imaging techniques. Here, the cranium and mandible of *Euparkeria* are fully redescribed and documented using all fossil material and computed tomographic data. Details previously unclear are fully described, including vomerine dentition, the epipterygoid, number of premaxillary teeth and palatal arrangement. A new diagnosis and cranial and braincase reconstruction is provided, and an anatomical network analysis is performed on the skull of *Euparkeria* and compared with other amniotes. The modular composition of the cranium suggests a flexible skull well adapted to feeding on agile food, but with a clear tendency towards more carnivorous behaviour, placing the taxon at the interface between ancestral diapsid and crown archosaur ecomorphology, corresponding to increases in brain size, visual sensitivity, upright locomotion and metabolism around this point in archosauriform evolution. The skull of *Euparkeria* epitomizes a major evolutionary transition, and places crown archosaur morphology in an evolutionary context.

## 1. Introduction

Archosauria, the 'ruling reptiles', form a major part of modern vertebrate biodiversity, represented by some 10 000 extant species of birds and crocodilians [1]. Archosaurs formed the primary component of the larger-bodied vertebrate terrestrial fauna of the Mesozoic from at least the Late Triassic to the Cretaceous [2–5]. During the Triassic, archosaurs entered ecological niches previously filled by non-mammalian therapsids. This faunal transition was a landmark in the evolution of the Earth's vertebrate fauna, and the diversification of Archosauria is an outstanding example of an evolutionary radiation occurring over an extended geological timescale [3,6]. Understanding this transition and the characteristics of Archosauria that allowed the clade to radiate so spectacularly is of major importance to vertebrate palaeontology and evolutionary biology.

The skull serves to contain and protect some of the most important structures in the vertebrate body—the feeding, sensory and information processing apparatus [7]. Since its initial development as at least three structures with separate origins [8], the skull has evolved in myriad directions, becoming extensively modified to yield such disparate forms as the highly kinetic cranium of snakes [9], and the much reduced cartilaginous structure seen in sharks [7]. The archosaur skull, in particular, has also seen extensive modification from its presumed initial form, yielding forms as varied as the edentulous elongated beak of hummingbirds [10], the immensely powerful jaws of hypercarnivorous theropods [11] and the uniquely flexible grazing-adapted skull of ornithischians [12]. An increased importance of vision over olfaction is in particular seen in flying archosaurs (birds and pterosaurs) [13], corresponding to the relatively larger postrostral region (and thus reduced face) connected with miniaturization and paedomorphosis in birds and their immediate relatives [14,15]. Concurrently, an increase in brain size is seen in theropods and in birds in particular [16], and both this and the evolutionary flexibility of the mechanical aspects of the skull, especially the beak [17], are plausibly at least partly responsible for the astonishing success of birds today. Understanding the starting point of these spectacular changes and to what degree they are ancestral for Archosauria is thus of great biological interest.

*Euparkeria capensis* is a relatively small taxon represented by over 10 individuals from a single locality in the Anisian of South Africa (Subzone B of the *Cynognathus* Assemblage Zone). Even following its initial description and discovery in the early twentieth century, the taxon was considered of major scientific interest along with other 'pseudosuchian' (at the time including *Euparkeria*) taxa because it shed light on the origins of dinosaurs and birds [18]. The taxon also specifically helped to elucidate the morphology and taxonomy of other Triassic archosaurs [19]. Today *Euparkeria* is usually placed phylogenetically on the stem of Archosauria relatively close to the crown, frequently as the sister taxon to Archosauria (e.g. [20–34]; figure 1). Due in part to its position in cladistic phylogenies, *Euparkeria* (N.B. genus names are used alone after the first mention in the case of monospecific genera) has been considered to approach potentially the morphology of the ancestral archosaur [37,38]. Furthermore, *Euparkeria* is often used as an outgroup in phylogenetic analyses of crown archosaur taxa [39–51]. *Euparkeria* is thus key in understanding the success and diversification of crown archosaurs and the sequence of character changes prior to and during this radiation. Its morphology provides a potential analogue for the ancestral condition in Archosauria, and understanding its cranial anatomy in the evolutionary context promises to shed light on the incredible diversity of archosaur skull morphology and the source from which it sprang.

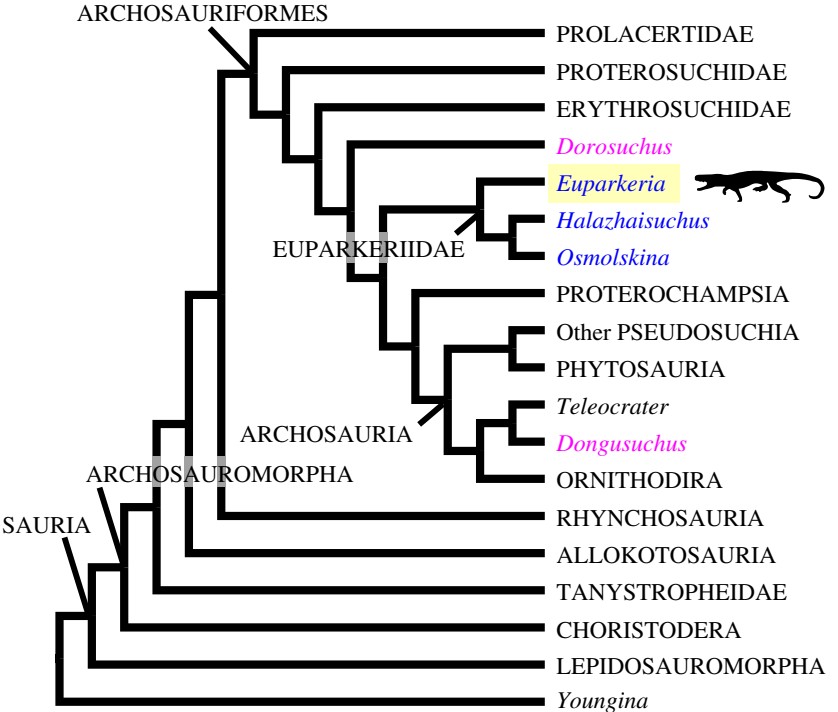

**Figure 1.** Schematic phylogeny based on those of Nesbitt [35], Ezcurra [36] and Sookias [32], showing the position of *Euparkeria capensis* (blue text, yellow background), other euparkeriids (blue text), taxa previously considered euparkeriids (pink text) and other key early diapsid taxa. Although Proterochampsidae was found to be the sister taxon to Archosauria by Ezcurra [36] and in one analysis of Nesbitt [35], many analyses have placed *Euparkeria* as the sister taxon to crown Archosauria, including the main analysis of Nesbitt [35], and the analysis of Nesbitt [31] upon which the analyses of Sookias *et al.* [33,34] and Sookias [32] were based.

*Euparkeria* was originally described by Broom [18,52], with this work augmented by Haughton [53]. A detailed monograph describing the taxon was produced by Ewer [54]. Subsequently, the braincase was described in detail by Gower & Weber [38] and this description was elaborated on using computed tomography (CT) data and all specimens by Sobral *et al.* [55]. While the original cranial section of the description of Ewer [54] was relatively thorough, a redescription and full documentation of the material is long overdue. Cladistic phylogenetics was yet to be fully developed when the material was described by Ewer [54], and many aspects of morphology now considered potentially phylogenetically informative were left undescribed. Furthermore, digital photography and CT scanning, as well as further preparation of some of the material in the intervening years, permit far more information to be collected and presented than was previously possible. Here, we redescribe and document the skull and mandible of *Euparkeria* using all available material and CT scans of several specimens. The braincase is not redescribed because of its recent extensive treatment by Sobral *et al.* [55], but a brief summary and reconstructions are, however, provided.

## 1.1. Taxonomic background

The genus *Euparkeria* and the only known species *Euparkeria capensis* were both erected by Broom [18,52]. Today, all material from the type site not belonging to the rhynchosaur *Mesosuchus browni* is assigned to *Euparkeria* based on morphology. Previously, the larger specimen in SAM-PK-6047 consisting of postcranial material (and thus not described in this paper) was assigned by Broom [18] to a new genus and species *Browniella africana*. However, this assignment was based on size and morphological features that appear to have been doubtfully identified (a restricted ilium is probably attributable to damage, and absence of a second pubic foramen doubtful). Haughton [53] synonymized *Euparkeria* and *Browniella*, and this synonymy has not been questioned by subsequent workers (e.g. [38,54]).

## 1.2. Geological setting

Subzone B of the *Cynognathus* Assemblage Zone [56,57], within which all type and referred material was found, is the uppermost biozone of the Beaufort Group (figure 2*c*). Subzone B is defined by the first appearance of the mastodonsaurid temnospondyl *Xenotosuchus* (='*Parotosuchus*') *africanus* [57]. Other tetrapod taxa in Subzone B include the archosauriform *Erythrosuchus*, the dicynodont *Kannemeyeria*, and

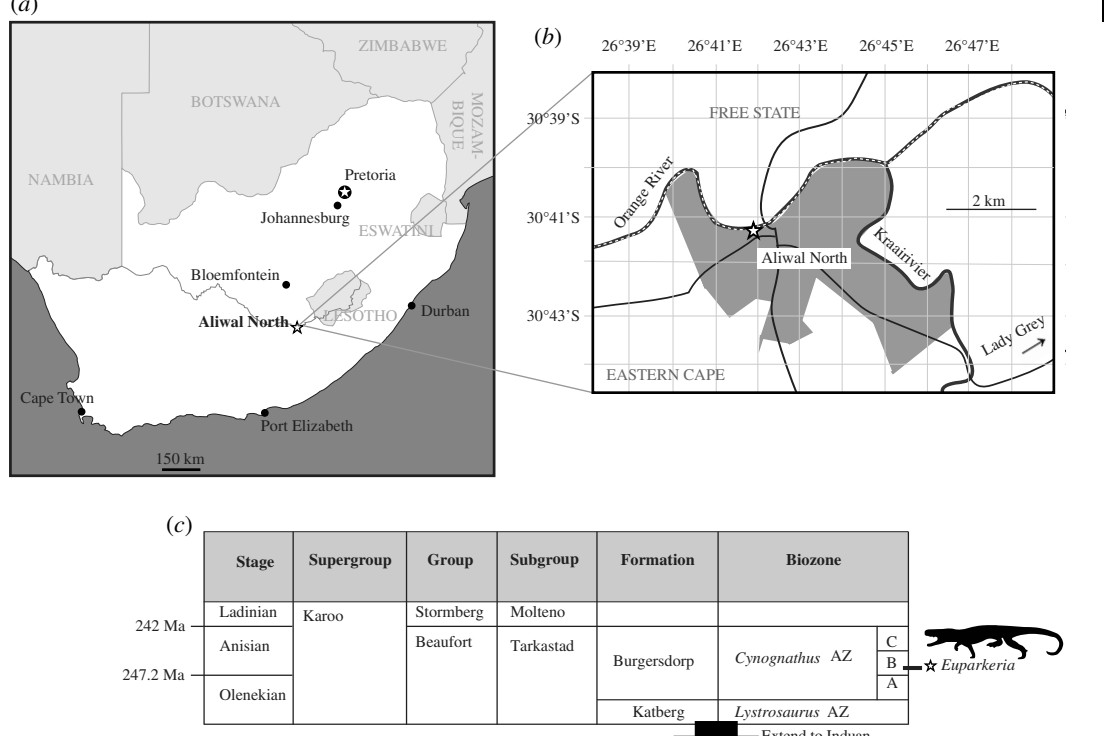

**Figure 2.** Maps showing (*a*) location of Aliwal North within South Africa and (*b*) the approximate position of the locality thought to have yielded all *Euparkeria capensis* fossils (a stone quarry adjacent to a small stream—'Krietfontein Spruit')—this location is based on the newly rediscovered notes of Mr Alfred Brown, and differs from the position identified by Ewer [54] which was described as being on the way out of Aliwal North in the direction of Lady Grey; (*c*) schematic stratigraphic column showing current understanding of the correlation of *Cynognathus* Assemblage Zone Subzone B, within which the locality is placed. In (*a*) and (*b*) Aliwal North and the locality respectively are marked with a star. In (*a*), pale grey is South Africa, medium grey indicates surrounding countries, dark grey is sea, and the line passing through South Africa and Aliwal North is the Orange River. In (*b*) grey indicates settlement, thin black lines are roads, the thicker black lines are rivers, and the dashed line along the Orange River is the province boundary.

the cynodonts *Trirachodon*, *Cynognathus* and *Diademodon* [56]. The *Cynognathus* Assemblage Zone has historically been considered to be entirely Early Triassic in age, but more recent work suggests Subzones B and C are in fact Middle Triassic (Anisian), with Subzone B potentially early Anisian [56] (figure 2*c*). An Anisian age assignment for Subzone B (e.g. [56]) is based on long-range vertebrate biostratigraphy, as direct dates are lacking. This biostratigraphic evidence is primarily the dominance of the vertebrate assemblage by a kannemeyeriid dicynodont, kannemeyeriids being widely considered a typical Middle Triassic group. Further evidence is that Subzone B is placed stratigraphically between subzones that have been independently assigned to the late Olenekian (Subzone A) and the late Anisian (Subzone C), again using vertebrate biostratigraphy [56]. There has been some recent suggestion of a Carnian age based on SHRIMP U-Pb dating of the Quebrada de los Fósiles Formation (Argentina) to 235.8 ± 2 Ma, and biostratigraphic correlation of the *Cynognathus* Assemblage Zone (in particular Subzone B) with the overlying Río Seco de La Quebrada Formation (based on the presence of *Cynognathus*, *Diademodon* and *Kannemeyeria*) [58], but this is not yet broadly accepted, given the major discordance with previous understanding and the well-constrained Anisian age of the overlying Perovkian Land Vertebrate Faunochron [59].

# 2. Methods

## 2.1. Examination of fossil material

All fossil material was examined first hand, with those photographs taken by R.B.S. having used a Nikon D5100 digital single-lens reflex camera with either a 50 or 18–55 mm lens. An image bank of all photographs used in the description, and a photogrammetric model of the skull of the holotype (SAM-PK-5867) from surface light scans have been uploaded to the Zenodo repository (doi:10.5281/

zenodo.3887056). Measurements were taken with a digital caliper accurate to 0.5 mm, and a full table of measurements for SAM-PK-5867 is given in the electronic supplementary material.

## 2.2. Comparisons

Comparisons were made with specimens directly observed where possible, and otherwise from photographs and the literature. As mentioned above, generic names are used alone after their first mention for monospecific genera, and otherwise, the specific name is given.

## 2.3. CT scanning

SAM-PK-5867 and SAM-PK-6047A were scanned using X-ray CT at the Evolutionary Studies Institute, University of the Witwatersrand, Johannesburg, South Africa. Scanning was undertaken with an X Tek HMX ST 225 (Nikon Metrology Inc.), comprising 3000 projections, using a tungsten target with gain 4. Files were reconstructed using CT Pro 3D software (Nikon Metrology, Inc.). Scan settings were as follows: SAM-PK-5867—70 kV, 140 µA, 1000 ms, 57.50 µm voxel size, 1.8 mm Al filter; SAM-PK-6047A—120 kV, 95 µA, 2000 ms, 60.10 µm voxel size, 1.2 mm Cu filter. UMZC T.692 was CT scanned at the Natural History Museum, London, UK, with an X Tek HMX ST 225 (Nikon Metrology Inc.), comprising 3142 projections, using a tungsten target with gain 4. Scan settings were as follows: 210 kV, 250 µA, 500 ms, 98.8 µm voxel size, 2.5 mm Cu filter. All scans were post-processed, segmented and examined using VG Studio Max 2.1 and 2.2, and myVGL 3.3.2 (Volume Graphics, Heidelberg, Germany). The scans of UMZC T.692 are archived at the Zenodo repository (doi:10.5281/zenodo.3887056), and those of other specimens will be archived following completion of work by B.-A.S.B. and are available upon request. Surface renderings of all CT-scanned specimens have also been made available as mesh files at Zenodo (doi:10.5281/zenodo.3887056).

## 2.4. Anatomical network analysis

An anatomical network analysis (AnNA) was performed following the protocol of Werneburg *et al.* [60]. An anatomical network was coded in a spreadsheet, in which each contact between bones was coded as '1', and the absence of a connection was coded as '0' (see Anatomical Network Matrix spreadsheet in electronic supplementary material). Data were analysed in R 3.5.3 [61] using the package *igraph* 1.0.0 [62]. Dendrograms were generated using agglomerative hierarchical clustering [63,64], illustrating the hierarchical grouping of nodes (i.e. connections between elements). Strong anatomical modules were identified by the highest $Q$-value (best partition identified by a community detection algorithm better than what is expected at random). The dendrogram was cut at $Q_{max}$ to identify $Q$-modules. The statistical significance of each cluster in the dendrogram was calculated and is indicated by circles in the dendrogram. The following parameters were computed (see Esteve-Altava *et al.* [65]): $N$, number of bones; $K$, number of links (connections between bones); $D$, density of connections, which has been interpreted in past works as a proxy of morphological complexity, with a higher value meaning a greater complexity of the skull; $C$, mean clustering coefficient, which has been interpreted in the past as a proxy of anatomical integration, with higher values meaning greater integration; $L$, shortest path length, which has been interpreted in the past as proxy of integration, with smaller values meaning greater integration; $H$, heterogeneity of connections, which has been interpreted as a proxy of anisomerism (i.e. irregularity, differentiation or specialization) with higher values meaning greater anisomerism.

## 2.5. Phylogenetic analysis

A phylogenetic analysis was carried out using the matrix of Sookias [32], including all taxa and characters and (as in Sookias [32]) using *Youngina capensis* as an outgroup, but rescoring character 90 (ectopterygoid single [0] or double [1] headed) to 1 (double headed) for *Euparkeria* based on new information (see below). The modified matrix and TNT code used to process it is given as electronic supplementary material. A traditional search was conducted with TNT 1.5 [66] using tree bisection–reconnection (TBR) branch swapping and 1000 replicates. Standard and GC bootstrap values and decay indices (Bremer support; using the Bremer script) were calculated for each node. The same was also conducted *a priori* excluding '*Turfanosuchus shageduensis*', as it has been considered a *nomen dubium* [34].

## 2.6. Institutional abbreviations

AMNH, American Museum of Natural History, New York, USA; BP, Evolutionary Studies Institute, University of the Witwatersrand, Johannesburg, South Africa [formerly Bernard Price Institute]; BSCUB, Biological Sciences Collection of the University of Birmingham, Birmingham, UK; GPIT,

Paläontologische Sammlung, Eberhard-Karls-Universität Tübingen, Tübingen, Germany [formerly Geologisch-Paläontologisches Institut Tübingen]; NHMUK, Natural History Museum, London, UK; PULR, Paleontología, Universidad Nacional de La Rioja, La Rioja, Argentina; PVSJ, División de Paleontología de Vertebrados del Museo de Ciencias Naturales y Universidad Nacional de San Juan, San Juan, Argentina; SAM, Iziko South African Museum, Cape Town, South Africa; UMZC, University Museum of Zoology Cambridge, Cambridge, UK; ZPAL, Institute of Paleobiology, Polish Academy of Sciences, Warsaw, Poland.

# 3. Results

## 3.1. Systematic palaeontology

Archosauromorpha Huene 1946 [67] *sensu* Gauthier *et al.* [68]

Archosauriformes Gauthier *et al.* 1988 [69] *sensu* Nesbitt 2011 [31]

Euparkeriidae Huene, 1920 [69] *sensu* Sookias & Butler, 2013 [72]

**Phylogenetic definition.** The most inclusive clade containing *Euparkeria capensis* Broom 1913 [52] but not *Crocodylus niloticus* Laurenti 1768 [70] or *Passer domesticus* Linnaeus 1758 [71], following Sookias & Butler 2013 [72].

*Euparkeria capensis* Broom, 1913 [52].

**Type and only species**. *Euparkeria capensis* Broom, 1913 [52].

**Diagnosis**. As for type and only species.

*Euparkeria capensis* Broom, 1913 [52].

### 3.1.1. Synonymy

'*Browniella africana*' Broom, 1913 [18].

### 3.1.2. Revised diagnosis (cranial only; modified from Sookias [32])

Relatively small (holotype SAM-PK-5867 femur length 55.7 mm; largest femur SAM-PK-10671 65.5 mm as preserved) archosauriform with relatively deep (not dorsoventrally flattened; figure 3) skull and recurved, serrated teeth, diagnosable by the following autapomorphies: posterodorsal process (= postnarial process) of the premaxilla is primarily vertical, curves slightly anteriorly towards its dorsal tip and is rounded (in lateral view) distally; the presence of two (as opposed to one in some other taxa, e.g. *Postosuchus kirkpatricki* [73]) inner ridges on the dorsal process of the maxilla within the antorbital fossa; a unique configuration of palatal teeth consisting of a broad field of small teeth along the medial margin of the pterygoid with a thinner field curving anterolaterally, a straight field on the palatine and two small vomerine teeth.

*Euparkeria* is distinguished from *Osmolskina czatkowicensis* [74] by possessing exoccipitals discrete from the opisthotics, a premaxilla lacking any slight overhang/downturn and which is clearly attached by a faceted articulation to the maxilla (unlike the weak attachment in *Osmolskina*) and not separated from it by an additional antorbital opening (unlike in *Osmolskina*), and a tapered and posteriorly curved nasal process of the maxilla (contrasting with the subquadrangular process in *Osmolskina*).

*Euparkeria* is differentiated from many other archosauromorph taxa by a roughly straight ventral margin of the upper jaw in lateral view, with no pronounced downward curvature of the maxilla or premaxilla, unlike in *Osmolskina* and in many early pseudosuchians (e.g. *Batrachotomus kupferzellensis* [75], *Ornithosuchus woodwardi* [76]). The orbit of both *Euparkeria* and *Osmolskina* is also relatively circular (height : width ratio 0.87 in SAM-PK-5867), compared with that of many pseudosuchians (e.g. *Batrachotomus* [75], height : width 2.29, *Ornithosuchus* [76], height : width 1.20). The straightness of the ventral margin of the upper jaw and the orbit shape is arguably more similar to some avemetatarsalian archosaurs (e.g. *Herrerasaurus ischigualastensis* [77], orbital height : width 0.98; *Eoraptor lunensis* [78], height : width 1.07) than to early terrestrial pseudosuchians, but the autapomorphic posterodorsal process of the premaxilla and arrangement of palatal dentition differentiates *Euparkeria* from ornithodirans, and in dinosaurs, the postfrontal is lost.

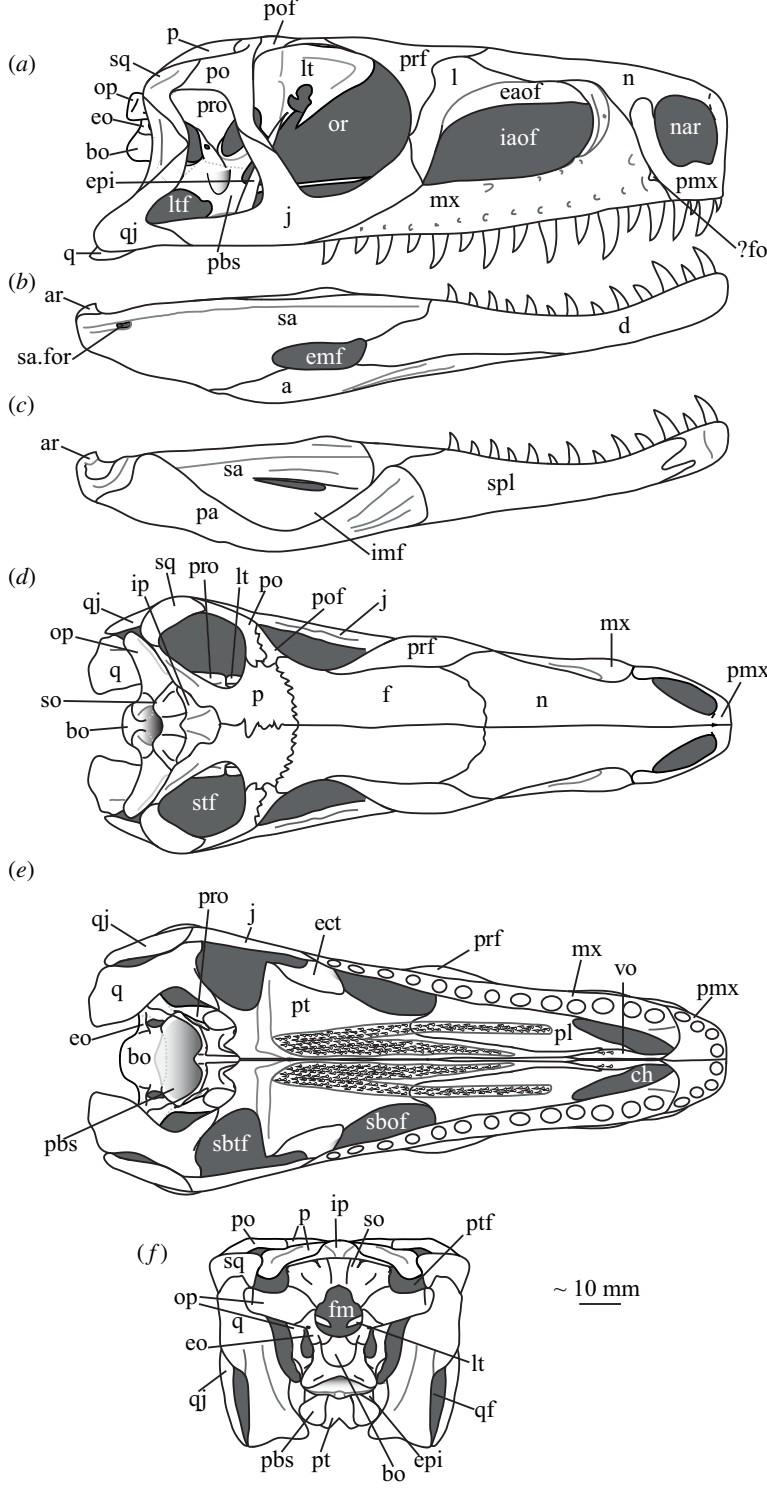

**Figure 3.** Reconstruction of the skull and mandible of *Euparkeria capensis*. Cranium (*a*) and right mandible (*b*) in right lateral view; (*c*) left mandible in medial view; and cranium in (*d*) dorsal, (*e*) ventral and (*f*) posterior view. ?for, possible small subnarial foramen; a, angular; ar, articular; bo, basioccipital; ch, choana; d, dentary; eaof, external antorbital fenestra; emf, external mandibular fenestra; eo, exoccipital; epi, epipterygoid; f, frontal; fm, foramen magnum; iaof, internal antorbital fenestra; imf, internal mandibular fenestra; ip, interparietal; j, jugal; l, lacrimal; lt, laterosphenoid; ltf, lateral temporal fenestra; mx, maxilla; n, nasal; nar, external naris; or, orbit; op, opisthotic; p, parietal; pa, prearticular; pbs, parabasisphenoid; pmx, premaxilla; po, postorbital; pof, postfrontal; pro, prootic; pt, pterygoid; ptf, posttemporal fenestra; q, quadrate; qf, quadrate foramen; qj, quadratojugal; sa, surangular; sa.for, surangular foramen; sbof, suborbital fenestra; sbtf, subtemporal fenestra; so, supraoccipital, spl, splenial; sq, squamosal; stf, supratemporal fenestra.

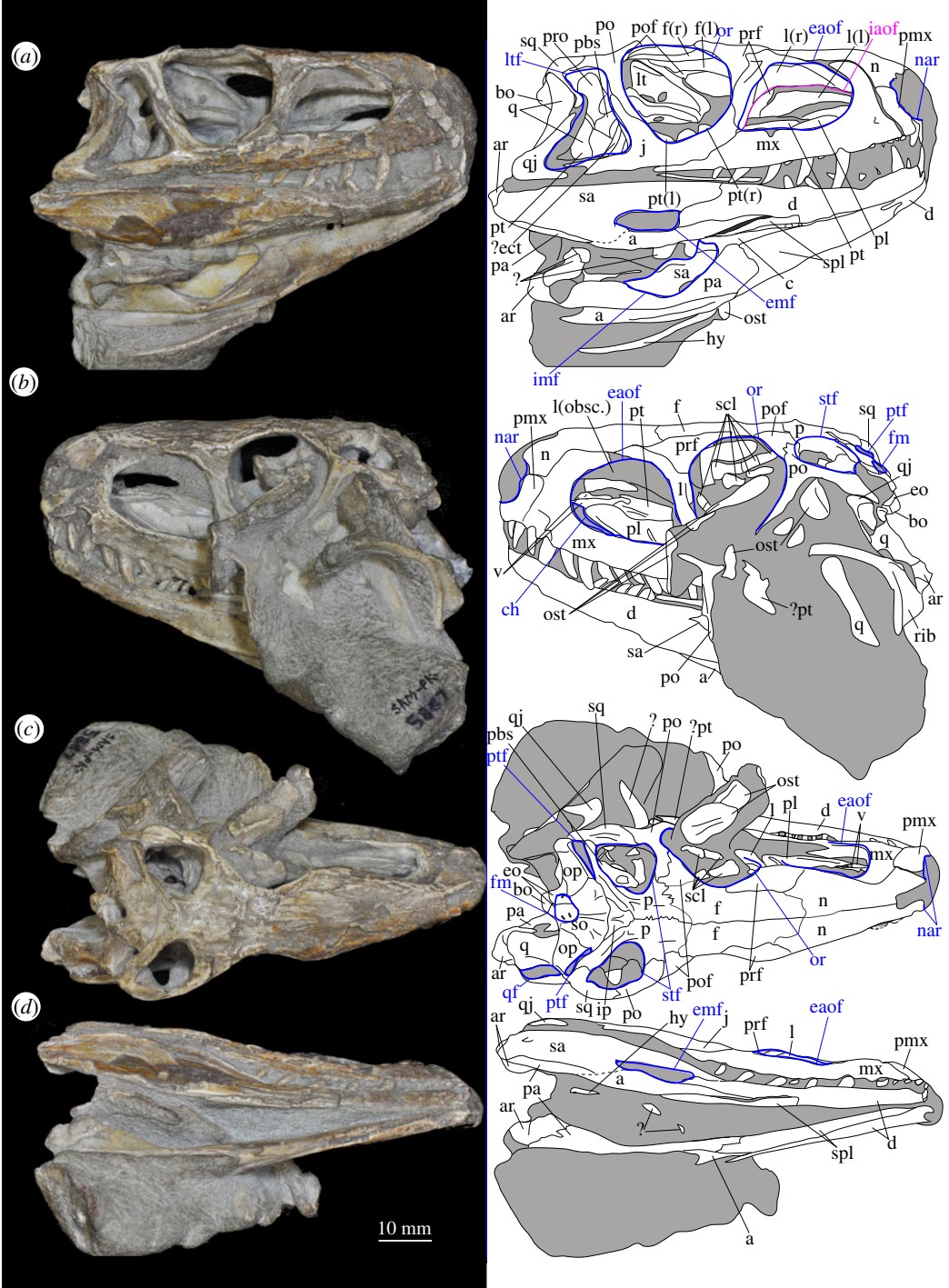

**Figure 4.** Holotype skull SAM-PK-5867 of *Euparkeria capensis* in (*a*) right lateral, (*b*) left lateral, (*c*) dorsal and (*d*) ventral view. ?, unidentified element or uncertainty in identification when preceding element; a, angular; ar, articular; bo, basioccipital; c, coronoid; ch, choana; d, dentary; eaof, external antorbital fenestra; ect, ectopterygoid; emf, external mandibular fenestra; eo, exoccipital; f(r/l), frontal (right/left); fm, foramen magnum; hy, hyoid; iaof, internal antorbital fenestra; imf, internal mandibular fenestra; ip, interparietal; j, jugal; l, lacrimal; lt, laterosphenoid; ltf, lateral temporal fenestra; mx, maxilla; n, nasal; nar, external naris; obsc., obscured by matrix; or, orbit; ost, osteoderms; p, parietal; pa, prearticular; pl, palatine; pmx, premaxilla; pof, postfrontal; po, postorbital; pt(r/l), pterygoid (right/left); ptf, posttemporal fenestra; q, quadrate; qf, quadrate foramen; qj, quadratojugal; rib, rib; sa, surangular; scl, sclera; so, supraoccipital; spl, splenial; stf, supratemporal fenestra; v, vomer. White, bone; grey, matrix. Blue text and lines indicate foramina and fenestrae. The same scale applies to all images.

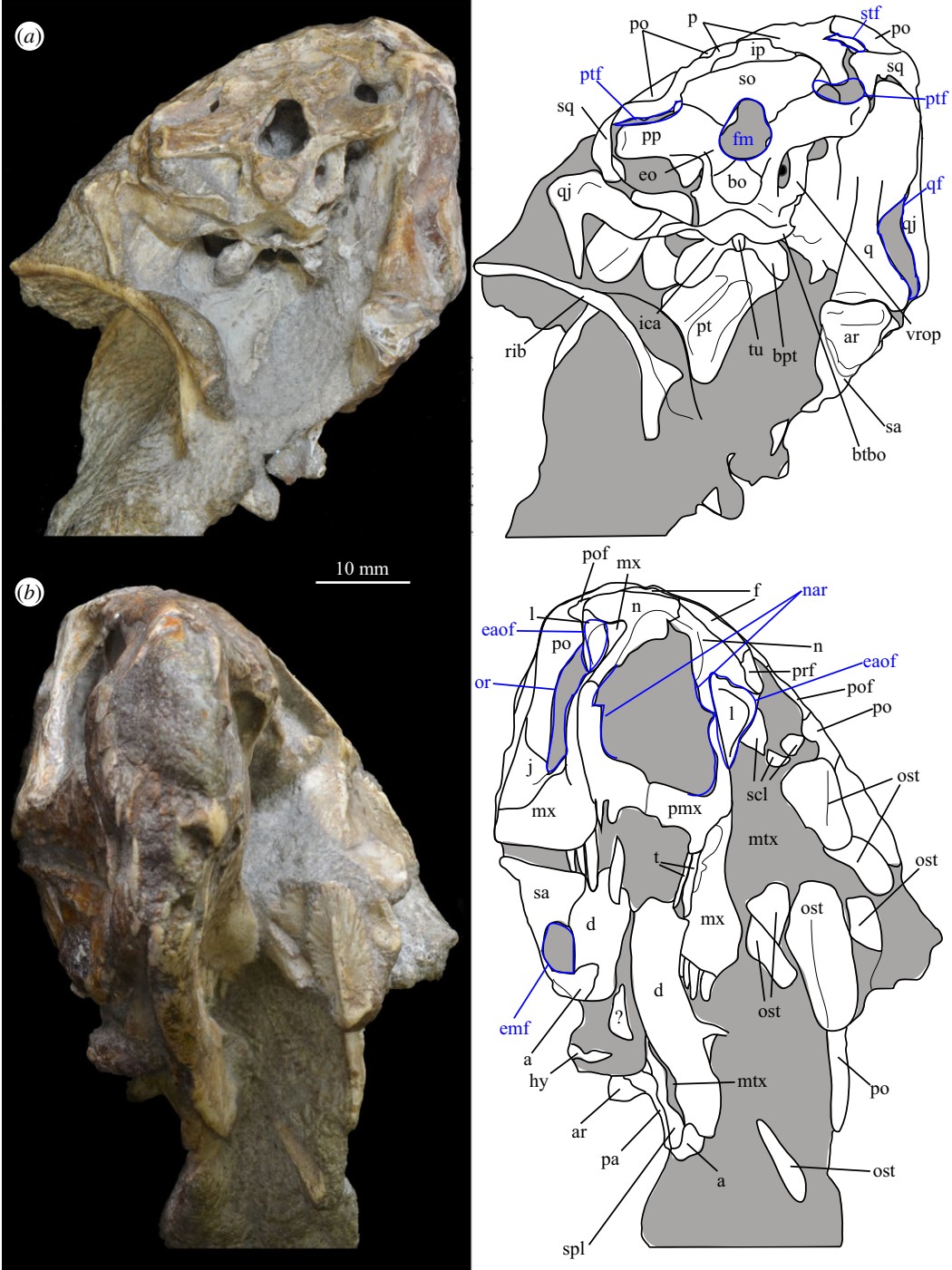

**Figure 5.** Holotype skull SAM-PK-5867 of *Euparkeria capensis* in (*a*) posterior and (*b*) anterior view with interpretive drawings. ?, unidentified element; a, angular; ar, articular; bo, basioccipital; bpt, basipterygoid process; btbo, basal tuber of the basioccipital; d, dentary; eaof, external antorbital fenestra; emf, external mandibular fenestra; eo, exoccipital; f, frontal; fm, foramen magnum; hy, hyoid; ip, interparietal; j, jugal; l, lacrimal; mx, maxilla; n, nasal; nar, naris; or, orbit; ost, osteoderms; p, parietal; pa, prearticular; pmx, premaxilla; po, postorbital; pof, postfrontal; pp, paroccipital process; prf, prefrontal; pt, pterygoid; ptf, posttemporal fenestra; q, quadrate; qf, quadrate foramen; qj, quadratojugal; rib, rib; so, supraoccipital; sa, surangular; scl, sclera; spl, splenial; sq, squamosal; stf, supratemporal fenestra; t, tooth; tu, tuber; vrop, ventral ramus of opisthotic. White, bone; grey, matrix. The same scale applies to (*a*) and (*b*).

### 3.1.3. Etymology

*Euparkeria,* from *eu-* (Ancient Greek εʊ-, true, good, well), Parker (in honour of Prof. W. Kitchen Parker who served as an inspiration to R. Broom) and *-ia* (Latin, substantive suffix); *capensis* (Latin, of the Cape,

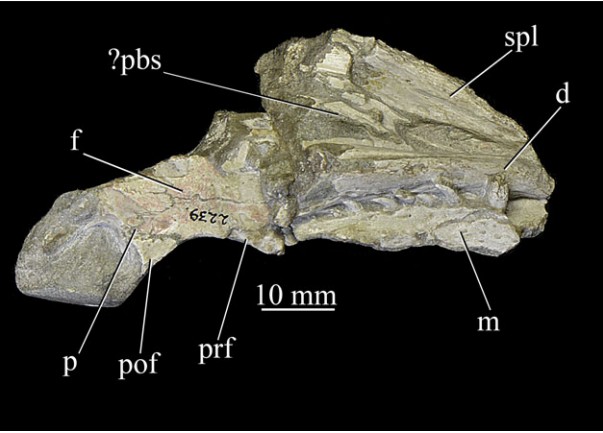

**Figure 6.** Specimen AMNH 2239 of *Euparkeria capensis* showing partial skull roof and jaws in dorsal and left approximately ventrolateral view, respectively. ?pbs, possible parabasisphenoid; d, dentary; f, frontal; m, maxilla; p, parietal; pof, postfrontal; prf, prefrontal; spl, splenial.

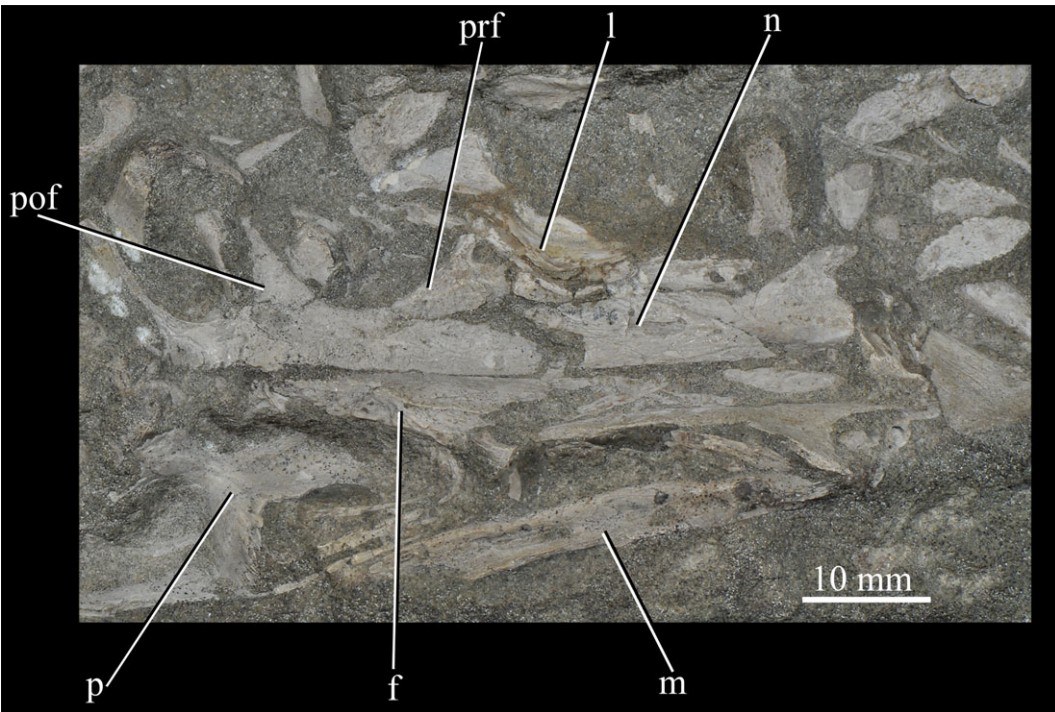

**Figure 7.** Skull of *Euparkeria capensis* specimen GPIT/RE/12913 in dorsal view. f, frontal; l, lacrimal; m, maxilla; n, nasal; p, parietal; pof, postfrontal; prf, prefrontal.

presumably referring to the Cape of Good Hope and its namesake the Cape Province, in which the town of Aliwal North was situated at the time of description).

### 3.1.4. Holotype

SAM-PK-5867 (specimen 1 of Haughton [53]; figures 4 and 5), complete skull except for part of premaxilla, articulated except for the left mandible which is not articulated with the quadrate and the surangular of which is partially disarticulated.

### 3.1.5. Referred specimens that include cranial material (postcranial material not listed)

All other specimens were collected at the same locality and around the same time and by the same collectors as the holotype (see below).

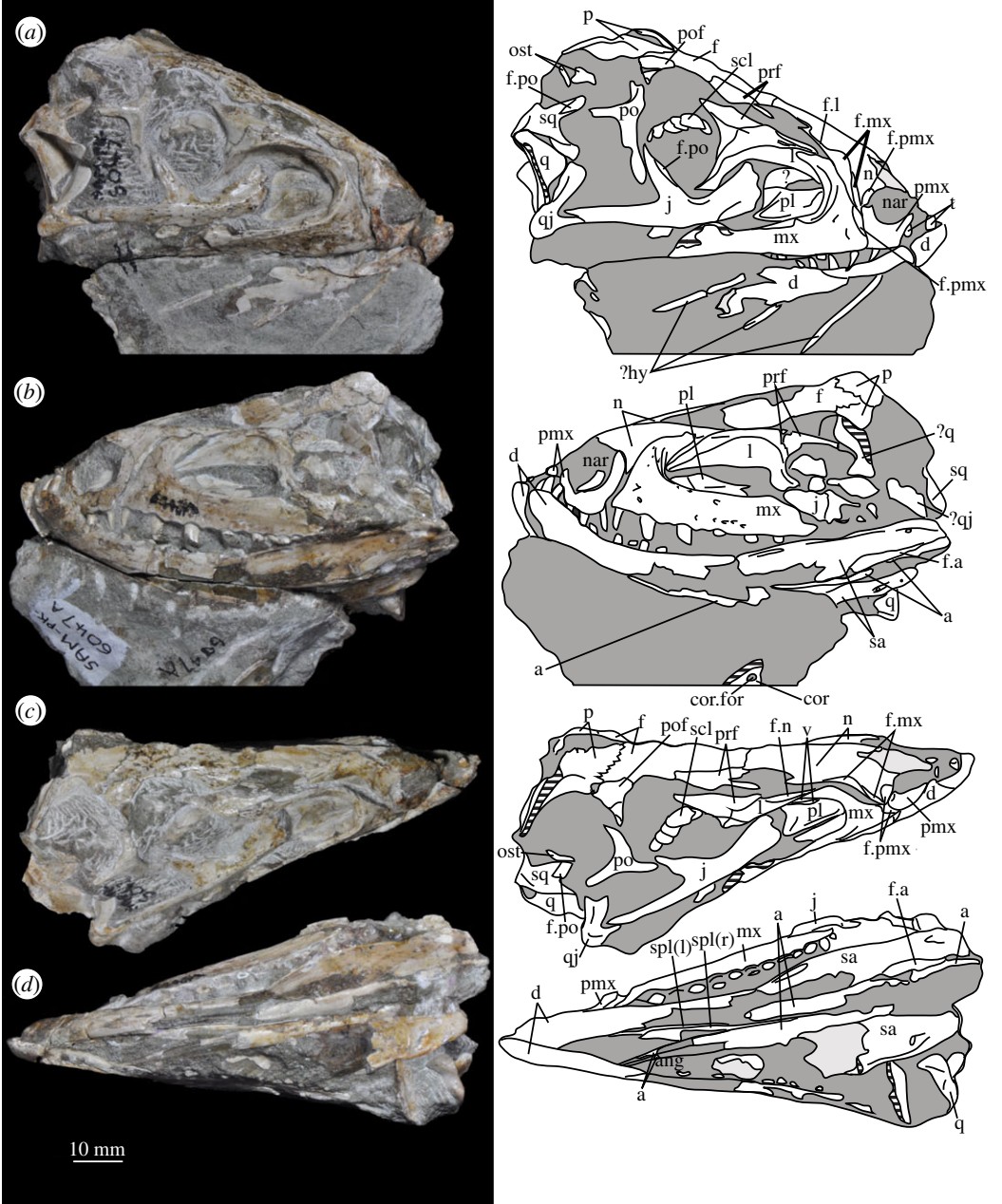

**Figure 8.** Skull of *Euparkeria capensis* specimen SAM-PK-6047A in (*a*) right lateral, (*b*) left lateral, (*c*) dorsal and (*d*) ventral view. ?, unidentified element or uncertainty in identification when preceding element; a, angular; ar, articular; cor, coronoid; cor.for, coronoid foramen; d, dentary; f., facet for; f, frontal; hy, hyoid; j, jugal; l, lacrimal; mx, maxilla; n, nasal; nar, external naris; ost, osteoderms; p, parietal; pl, palatine; pmx, premaxilla; pof, postfrontal; po, postorbital; q, quadrate; qj, quadratojugal; sa, surangular; scl, sclera; sq, squamosal; v, vomer. Dark grey, matrix; light grey, matrix with bone fragments, cross-hatching, broken surface. The same scale applies to all images.

 **AMNH 2239** (figure 6), partial right skull and mandible including right and left maxillae, parietals, postfrontals, prefrontals, nasals, dentaries, surangulars, angulars and splenials, right lacrimal, and possibly part of the braincase.

 **GPIT/RE/12913** (formerly SAM 7708 and GPIT 1681/2; figure 7), flattened skull roof, including right and left nasals, frontals and parietals, left lacrimal and postfrontal, left and partial right prefrontal, right maxilla, and other cranial fragments which cannot be identified with certainty.

 **SAM-PK-6047A** (specimen 2 of Haughton [53]; figures 8 and 9), almost complete, semiarticulated skull lacking the posterior end of the right mandible.

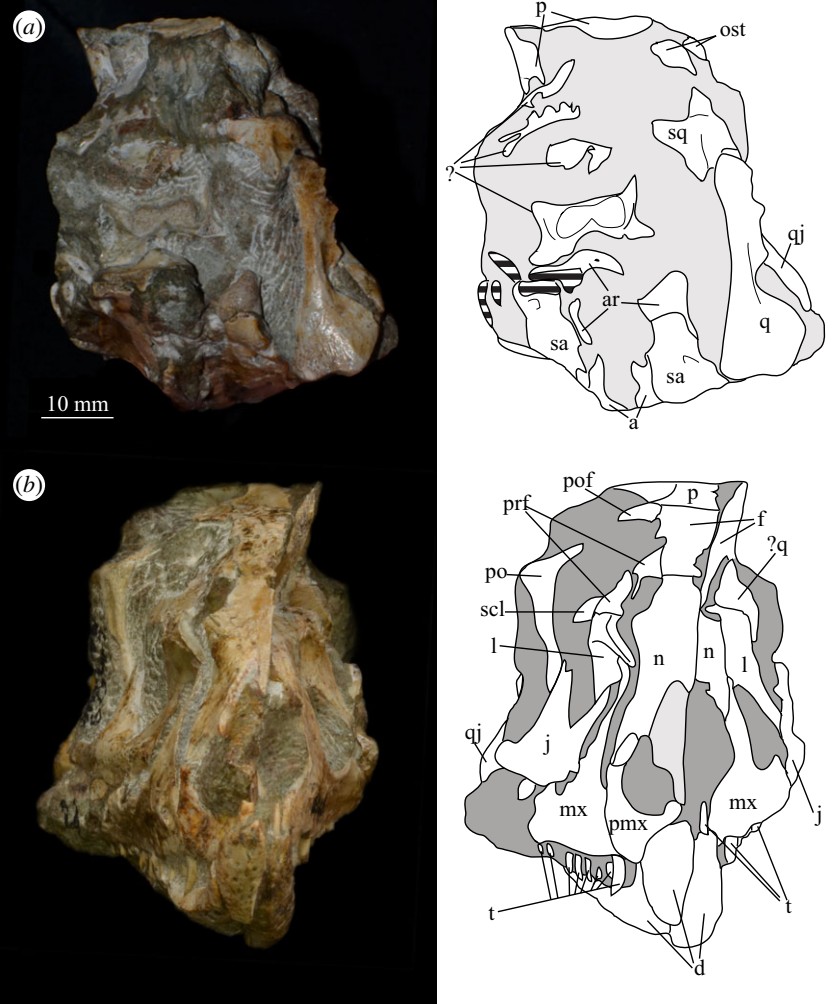

**Figure 9.** Skull of SAM-PK-6047A in (*a*) posterior and (*b*) anterior view with interpretive drawings. ?, unidentified element; a, angular; ar, articular; d, dentary; j, jugal; l, lacrimal; mx, maxilla; n, nasal; ost, osteoderm; t, tooth; p, parietal; po, postorbital; pof, postfrontal; prf, prefrontal; sa, surangular; scl, sclera; sq, squamosal; t, tooth. White, bone; dark grey, matrix; light grey, matrix with bone fragments; cross-hatching, broken surface. The same scale applies to (*a*) and (*b*).

**SAM-PK-6048** (specimen 3 of Haughton [53]; figure 10), most of right maxilla and premaxilla lacking posterodorsal process, partial left nasal, partial right mandible, partial right and left frontals and parietals, left postfrontal, partial left postorbital.

**SAM-PK-6050** (specimen 6 of Haughton [53]; figures 11 and see 26*e,f*), well-preserved left and right maxillae and anterior mandible, partial right jugal, partial palate including partial left and right palatines and pterygoids, other less well-preserved cranial elements possibly including posterior mandibles.

**SAM-PK-7699** (figure 12), left mandible, left and possible right parietal and possible left nasal.

**SAM-PK-13664** (see figures 42*a*, 43*a* and 44*a,b*), right pterygoid and ectopterygoid, and left palatine exposed in ventral view.

**SAM-PK-13665** (specimen 5 of Haughton [53]; figure 13), crushed but largely complete skull in the block, exposed on the left side with most elements roughly in articulation.

**SAM-PK-13666** (figure 14), slightly distorted, largely complete right hand side of skull exposed medially and left mandible exposed laterally.

**SAM-PK-13667** (figure 15), crushed partial skull including left and right mandibles and right maxilla.

**SAM-PK-K8050** (figures 16 and 17), four crushed skulls exposed in dorsal view including nasals, frontals, parietals, postfrontals, prefrontals, quadrates, squamosals, jugals and dentaries. The block also contains a skull and postcranium of the rhynchosaur *Meosuchus browni* (specimen SAM-PK-K8051).

**UMZC T.692** (formerly R 527; figures 18 and 19 and see 46*b–d*), cranial remains of two individuals. Individual A includes frontals, parietals, right and left prefrontal, right postfrontal, right and left

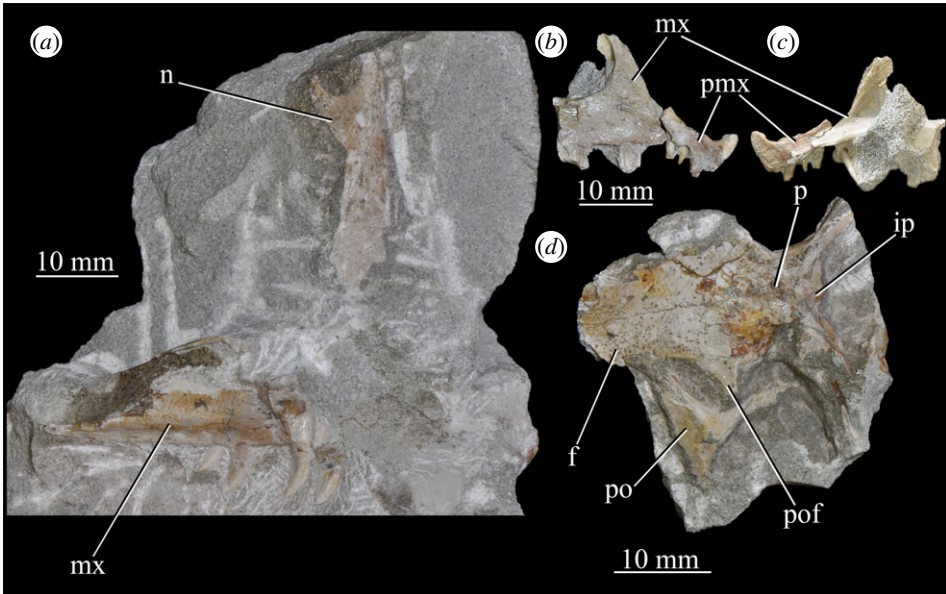

**Figure 10.** Specimen SAM-PK-6048 of *Euparkeria capensis*. (*a*) Block with nasal (dorsal view) and maxilla (right lateral view); articulated partial premaxilla and maxilla in right lateral (*b*) and (*c*) medial view; (*d*) partial skull roof in dorsal view. f, frontal; ip, interparietal; mx, maxilla; n, nasal; p, parietal; pmx, premaxilla; po, postorbital; pof, postfrontal.

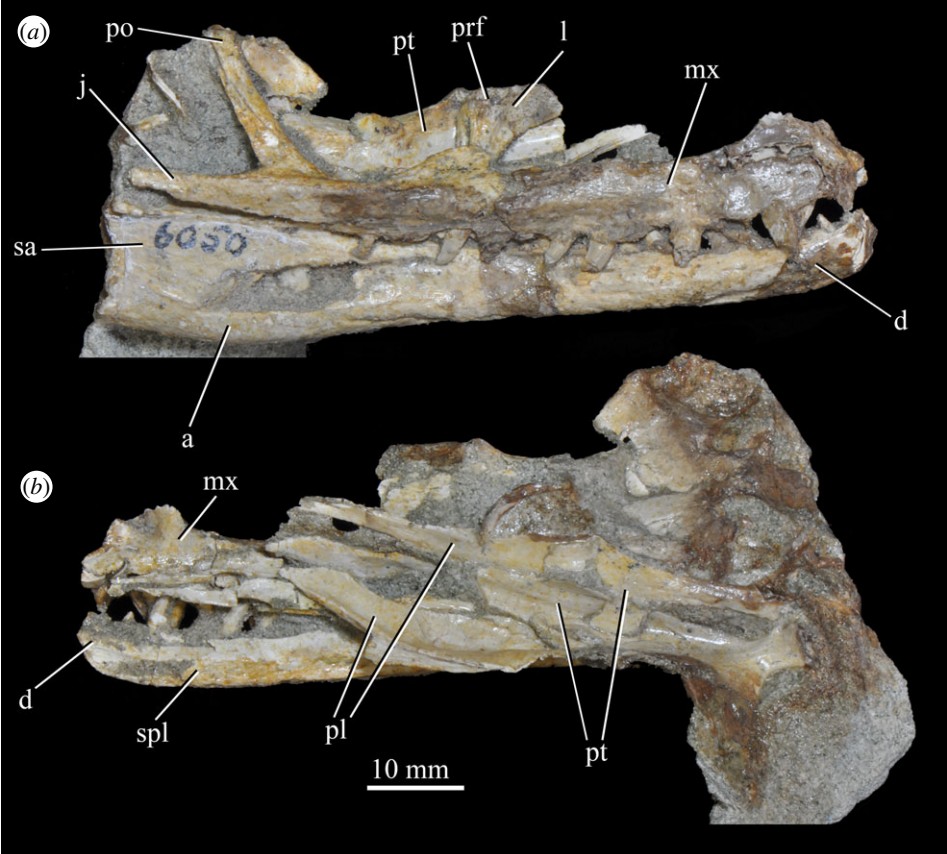

**Figure 11.** Right-hand side of jaws and cheek region and both sides of the palate of specimen SAM-PK-6050 of *Euparkeria capensis* in (*a*) lateral and (*b*) medial view. a, angular; d, dentary; j, jugal; l, lacrimal; mx, maxilla; sa, surangular; spl, splenial; pl, palatine; po, postorbital; prf, prefrontal; pt, pterygoid. The same scale applies to (*a*) and (*b*).

lacrimal, right postorbital, largely complete braincase, left pterygoid and ectopterygoid, partial right and left palatine and vomers, left and right hyoid, left partial maxilla, left jugal, right quadrate, and anterior of the left and posterior of the right mandible. Individual B includes left maxilla, anterior of the left and

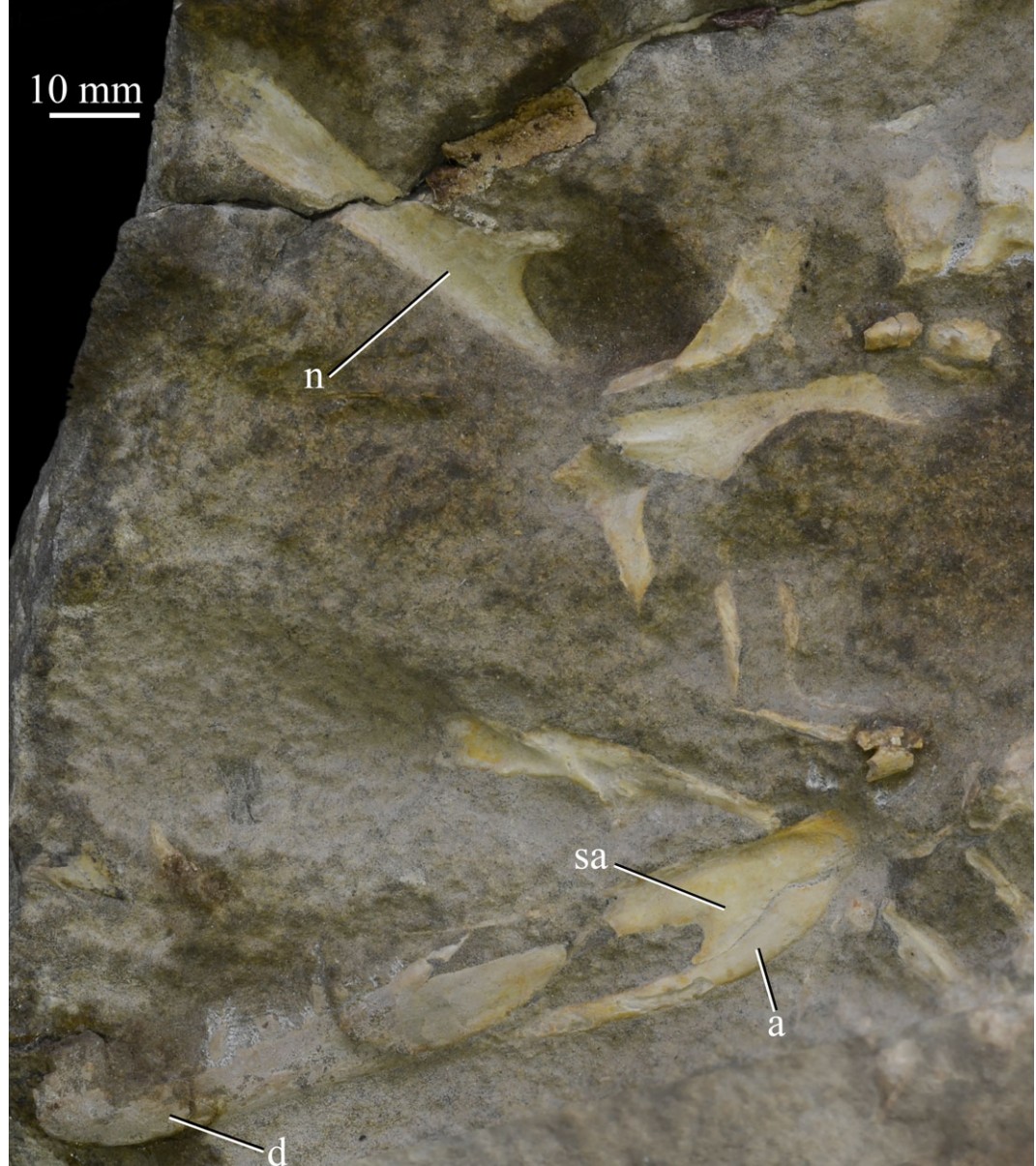

**Figure 12.** Cranial (ventral view) and mandibular (left lateral view) elements of specimen SAM-PK-7699 of *Euparkeria capensis*. a, angular; d, dentary; n, nasal; sa, surangular.

posterior of the right mandible, part of anterior of the left skull roof including nasal, posterior of the left jugal, parietals and left quadrate.

### 3.1.6. Specimens previously referred as cranial material

AMNH 19351 (electronic supplementary material, figure S1), labelled as containing 'fragment of jaw', but fragments of bone unidentifiable. Referral to either *Euparkeria* or *Mesosuchus* is plausible based on its provenance, and the material is thus here considered Archosauromorpha indet.

### 3.1.7. Horizon and type locality

There has never been any doubt that all specimens were collected from near Aliwal North (now in Walter Sisulu Municipality [79]), Eastern Cape, South Africa, and the locality was described as being 'in the *Cynognathus* zone of the Karroo system' by Prof. David M. S. Watson [80] (figure 2). However, the recent analysis of detailed hand-written notes in the original journals of Mr Alfred Brown has changed and clarified our knowledge of the locality. These journals (and the fossils—see below) have been in the SAM collections since 1921, a year after the death of Brown, but have never been fully documented or

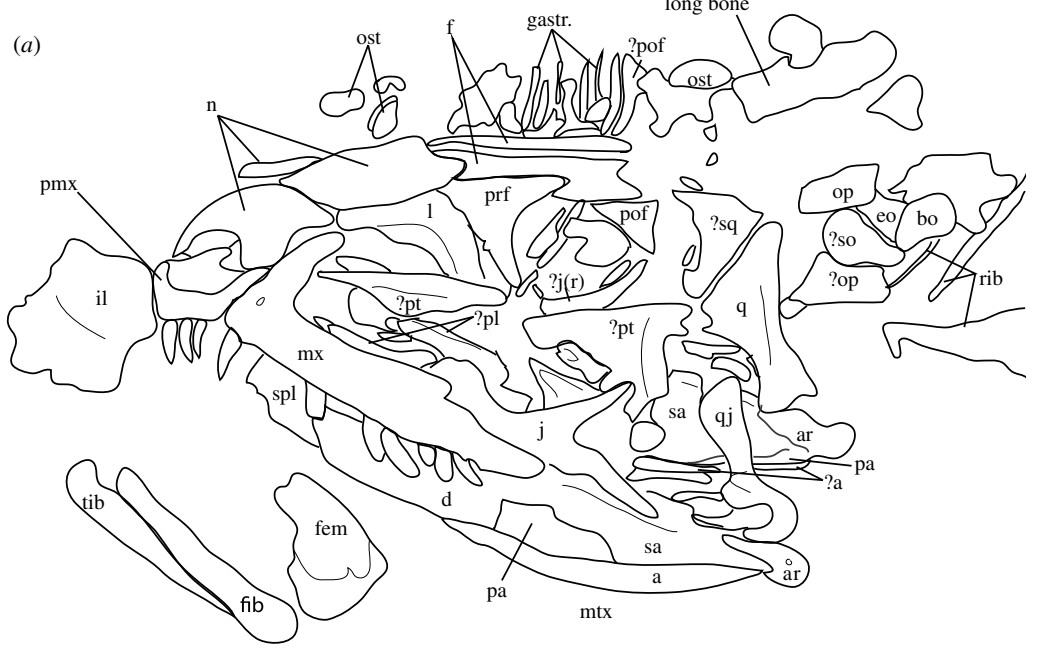

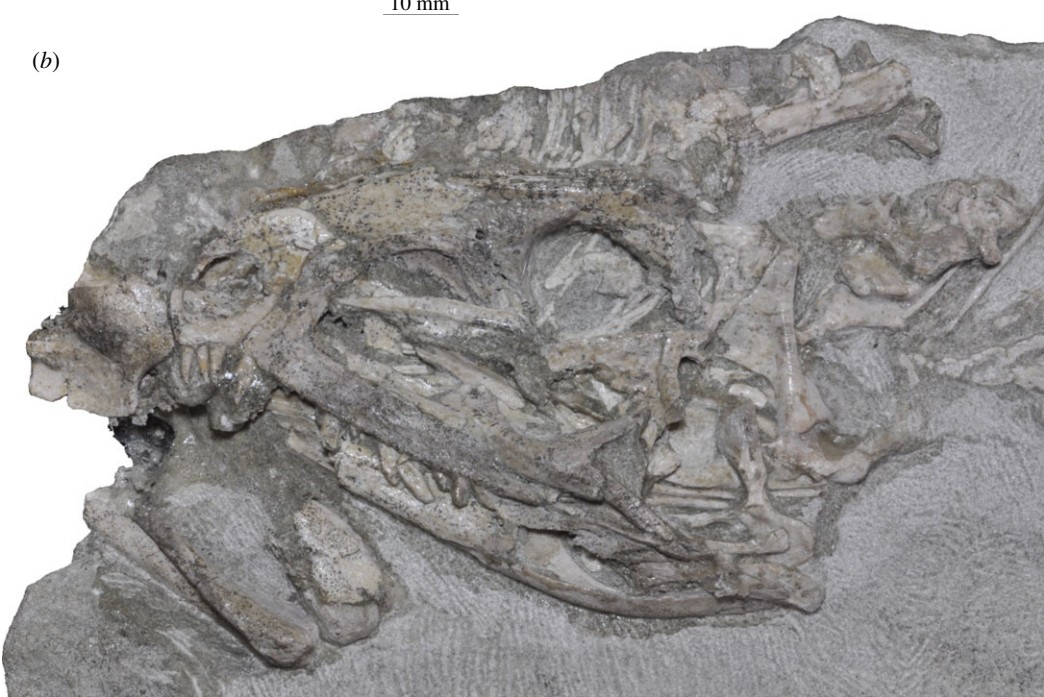

**Figure 13.** Cranium of *Euparkeria capensis* specimen SAM-PK-13665 in left lateral view. (*a*) Interpretative line drawing; (*b*) photograph. ?, uncertain identification; a, angular; ar, articular; bo, basioccipital; d, dentary; eo, exoccipital; f, frontal; fem, femur; fib, fibula; gastr., gastralia; il, ilium; j (r), jugal (right); l, lacrimal; mtx, matrix; n, nasal; op, opisthotic; ost, osteoderms; pa, prearticular; pl, palatine; pmx, premaxilla; pof, postfrontal; prf, prefrontal; pt, pterygoid; q, quadrate; rib, rib; sa, surangular; so, supraoccipital; sq, squamosal; tib, tibia.

transcribed. Brown's notes indicate that the first specimens came from the quarry of Mr Alexander Alcock, and were found initially (noted on 22 July 1907) during the preparation of slabs in the workshop by a worker—Mr Gibbs (no first name is known); Mr Alcock subsequently informed Brown. Further specimens were noted as being discovered on 21 July 1912 in the quarry (by now in possession of Mr James Webster) itself (Brown notes that they were 'Found in the sandstone bed of this sluit considerable fossil remains of a fine saurian'), and working after-hours, a quarryman then cut out the blocks for Brown. The block containing SAM-PK-K8050

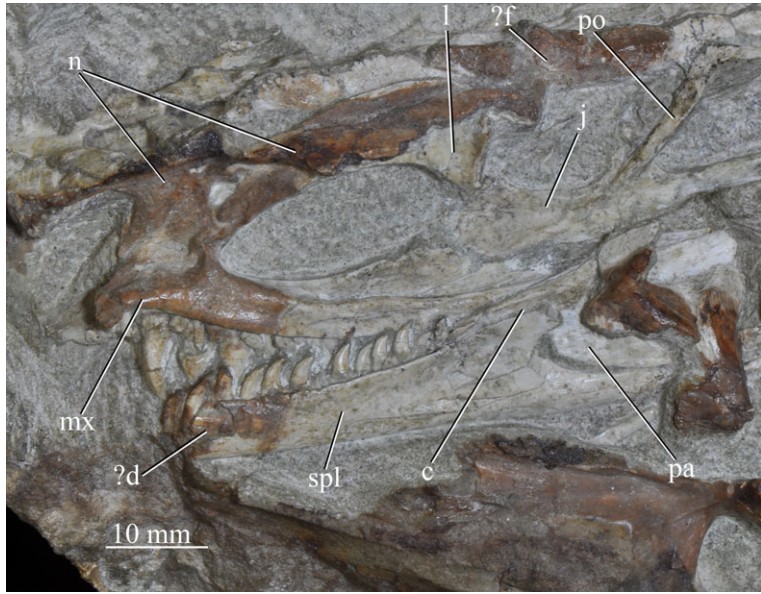

**Figure 14.** Right hand side of the cranium and left mandible of *Euparkeria capensis* specimen SAM-PK-13666 in medial and lateral view, respectively. ?, uncertain identification; c, coronoid; d, dentary; f, frontal; j, jugal; l, lacrimal; mx, maxilla; n, nasal; pa, prearticular; po, postorbital; spl, splenial.

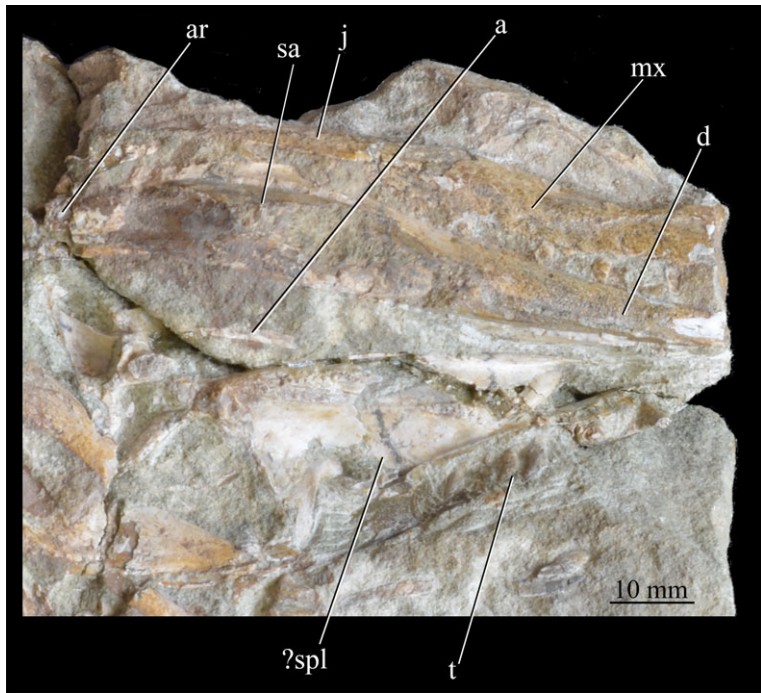

**Figure 15.** Cranial material of *Euparkeria capensis* specimen SAM-PK-13667 in the right lateral (right jaw, upper) and medial (left jaw, lower) view. ?, uncertain identification; a, angular; ar, articular; d, dentary; j, jugal; mx, maxilla; sa, surangular; spl, splenial; t, tooth.

and *Mesosuchus browni* specimen SAM-PK-K8051 must have been part of the 1907 discovery, as it was examined and noted by Watson in 1911 [52,80]; which other specimens were found during either of the two discoveries is still unknown pending further study of Brown's journals. These notes also confirmed Haughton's [53] name for the locality ('Krietfontein Spruit'—'spruit' meaning 'stream' in Afrikaans). As suggested by Ewer [54], this was probably Brown's personal name for the stream running away from 'Krietfontein'. Krietfontein is recorded in the town records referring to a cluster of mineral springs. The

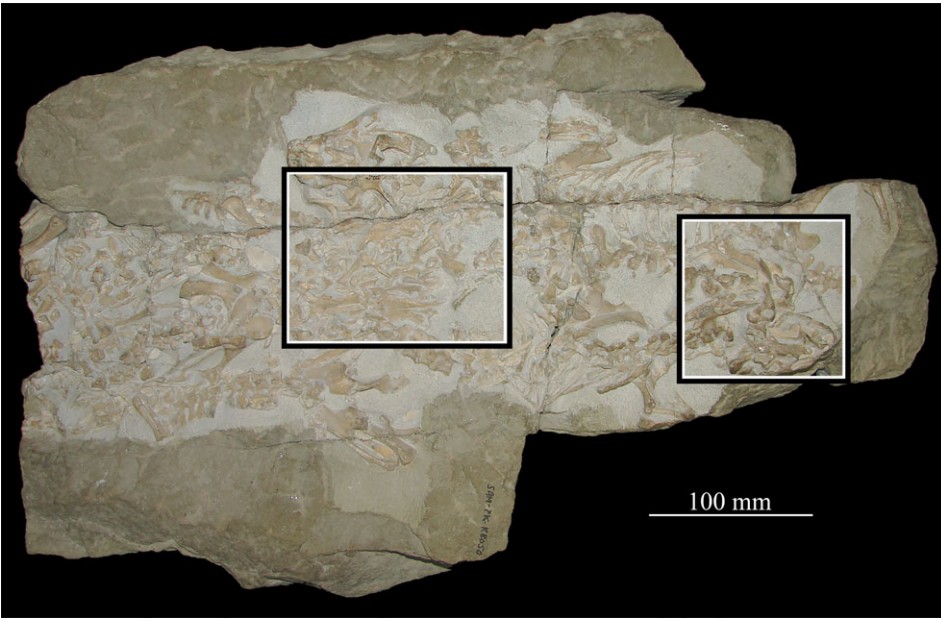

**Figure 16.** Entire block including *Euparkeria capensis* specimen SAM-PK-K8050 and *Mesosuchus browni* SAM-PK-K8051. Areas with cranial material of *Euparkeria capensis* bounded by boxes, and expanded and labelled in figure 17. Skulls and postcranial are in broadly dorsal view, but crushed dorsoventrally.

authors (R.M.H.S. and F.P.W.) have confirmed that the springs continue to source the stream alongside which the *Euparkeria* quarry is located.

After Brown's death in 1920, specimens from his collection were presented to the South African Museum in 1921, prior to Haughton's work in 1922 [53]. According to Ewer [54], these specimens are labelled as 'A. Brown collection, Krielfontein, Aliwal North' ('Kriet-' was seemingly accidentally misspelt with 'l' in the SAM records due to misreading of Brown's handwriting). Further material was also presented by Higgins to the SAM in 1924 and 1925 [54], with that presented in 1924 labelled 'Higgins collection, Krielfontein, Aliwal North' and that in 1925 labelled 'A. W. Higgins collection, Quarry, Commonage, Aliwal North'—both seemingly from the same locality as the Brown collection specimens. Haughton [53] reported the locality as 'Krietfontein Spruit on the Aliwal North Commonage', and indicated that all specimens collected to date (1922) were from one locality. Based on a communication from a certain Colonel de Wet, Ewer [54] suggested that some or even all of the specimens labelled 'A. Brown collection' may have been physically collected by Higgins for Brown, but this is not supported by the notes of Brown; given Higgins and Brown were good friends, it seems unlikely he would have made an oversight. In 1911, Brown also directly presented a specimen (now UMZC T.692) to Watson at University College London, from where it was transferred to the University Museum of Zoology in Cambridge. Two SAM specimens (then SAM 7708 and 7698) were presented by the SAM to the Institut für Geologie und Paläontologie at the University of Tübingen (now GPIT/RE/12913—which includes cranial material; see above—and GPIT/RE/15029). Four specimens were also acquired by the American Museum of Natural History (AMNH 2238, 2239, 5548 and 19351); these are thought to have been bought, probably as part of a purchase of '200 South African fossils' in 1913 from Robert Broom, who had visited Aliwal North in 1911/1912 and presumably acquired the specimens directly from Mr Brown.

In December 2019, R.M.H.S. and F.P.W followed the directions from Browns journal to an abandoned quarry site next to the stream running along the western side of the English graveyard in the Aliwal North townlands. Here was found sandstone outcropping in the old quarry face identical to the matrix of the *Euparkeria* blocks, and a newly discovered *in situ* long bone (not currently assignable beyond Archosauromorpha indet) also matched the preservation style of the *Euparkeria* specimens. Based on this we concur with the information gleaned from Brown's notes, and consider that the *Euparkeria*-bearing sandstone blocks in the collections of Iziko South African Museum all came from Mr Alcock's (and later Mr Webster's, following the records of the Aliwal North Municipal Museum and Brown's notes) dimension stone quarry in Krietfontein Spruit, on Aliwal North Commonage, some 150 m

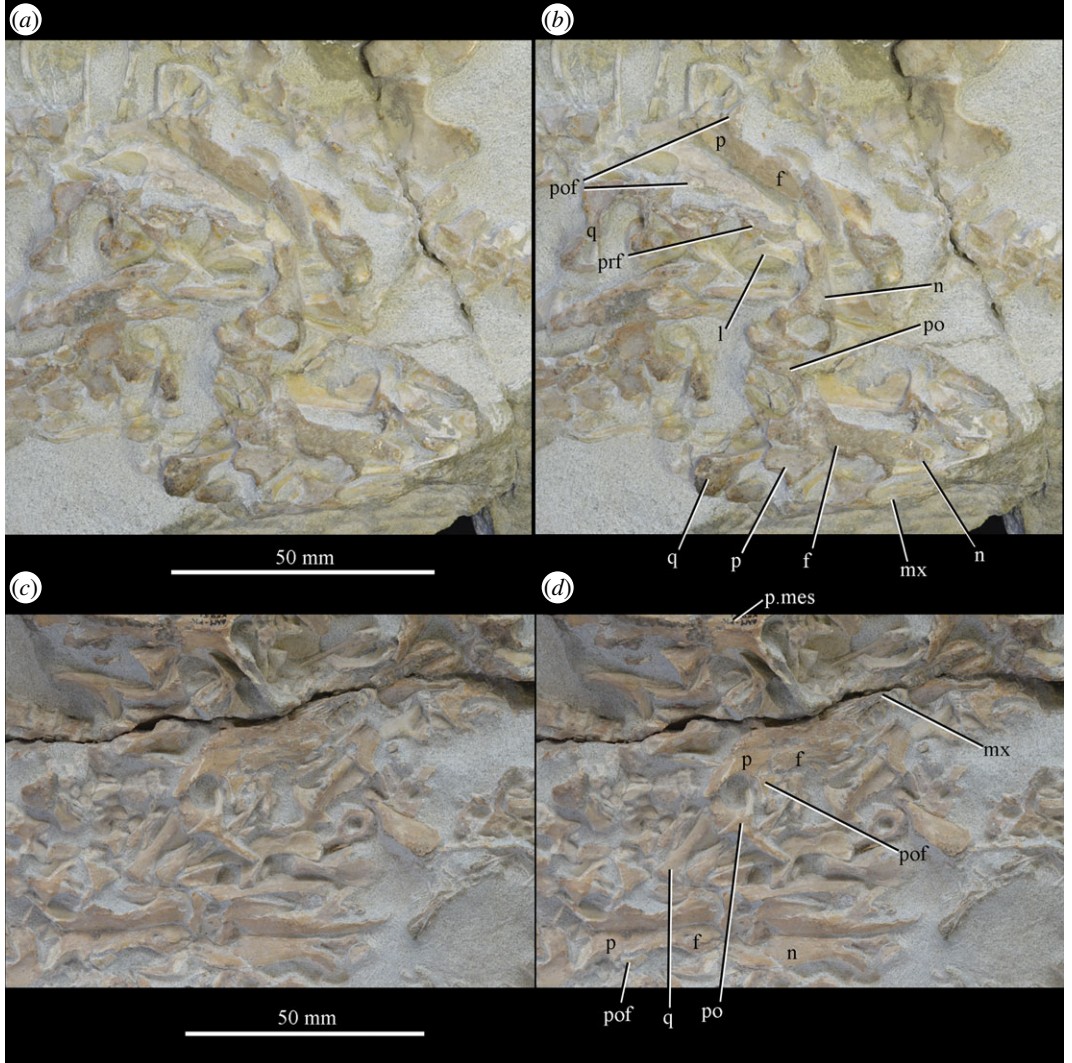

**Figure 17.** Close-up images of areas showing cranial material of *Euparkeria capensis* of block including SAM-PK-K8050 and *Mesosuchus browni* SAM-PK-K8051, without (*a,c*) and with (*b,d*) labelling. (*a,b*) are the area surrounded by the left-hand box and (*c,d*) the right-hand box in figure 16. f, frontal; l, lacrimal; mx, maxilla; n, nasal; p, parietal; p.mes, parietal of *Mesosuchus browni*; po, postorbital; pof, postfrontal; prf, prefrontal; q, quadrate. Upper scale applies to (*a,b*) and lower to (*c,d*).

upstream from the confluence with the Orange River, near the southern end of the old railway bridge. The quarry floor is approximately 23 stratigraphic metres lower in *Cynognathus* Subzone B than that determined by Ewer [54] (see below) and is close to, but still above, the transition with Subzone A. The bones are preserved in pale-olive fine-grained trough cross-bedded sandstone at the top of a laterally extensive greater than 8 m thick, three-storeyed, channel sandstone body which is probably the same unit that has been informally referred to as the 'middle marker' [81] or 'Eldorado member' [82] of the Burgersdorp Formation. Based on this re-exploration, a detailed work on the locality and its depositional environment is currently in preparation. An area Ewer [54], (who was unaware of Brown's journals) took to be the *Euparkeria* locality was based upon what now proves to be erroneous directions from a local resident. An exposure of the appropriate sediments 'on one of the commonages over the brow of a hill on the left of the road to Lady Grey just outside Aliwal North' (Ewer [54], p. 381). This area is some 5 km away from the Krietfontein spruit quarry, approximately 23 m higher in the succession, and the sandstone has no lithological similarity with the *Euparkeria* matrix nor does it show any signs of quarrying activity.

### 3.1.8. Notes on associated specimens

SAM-PK-K8050 contains the remains of at least four individuals of *Euparkeria* in close proximity, and UMZC T.692 and GPIT/RE/12913 contain at least two individuals. UMZC T.692 and the block containing SAM-PK-

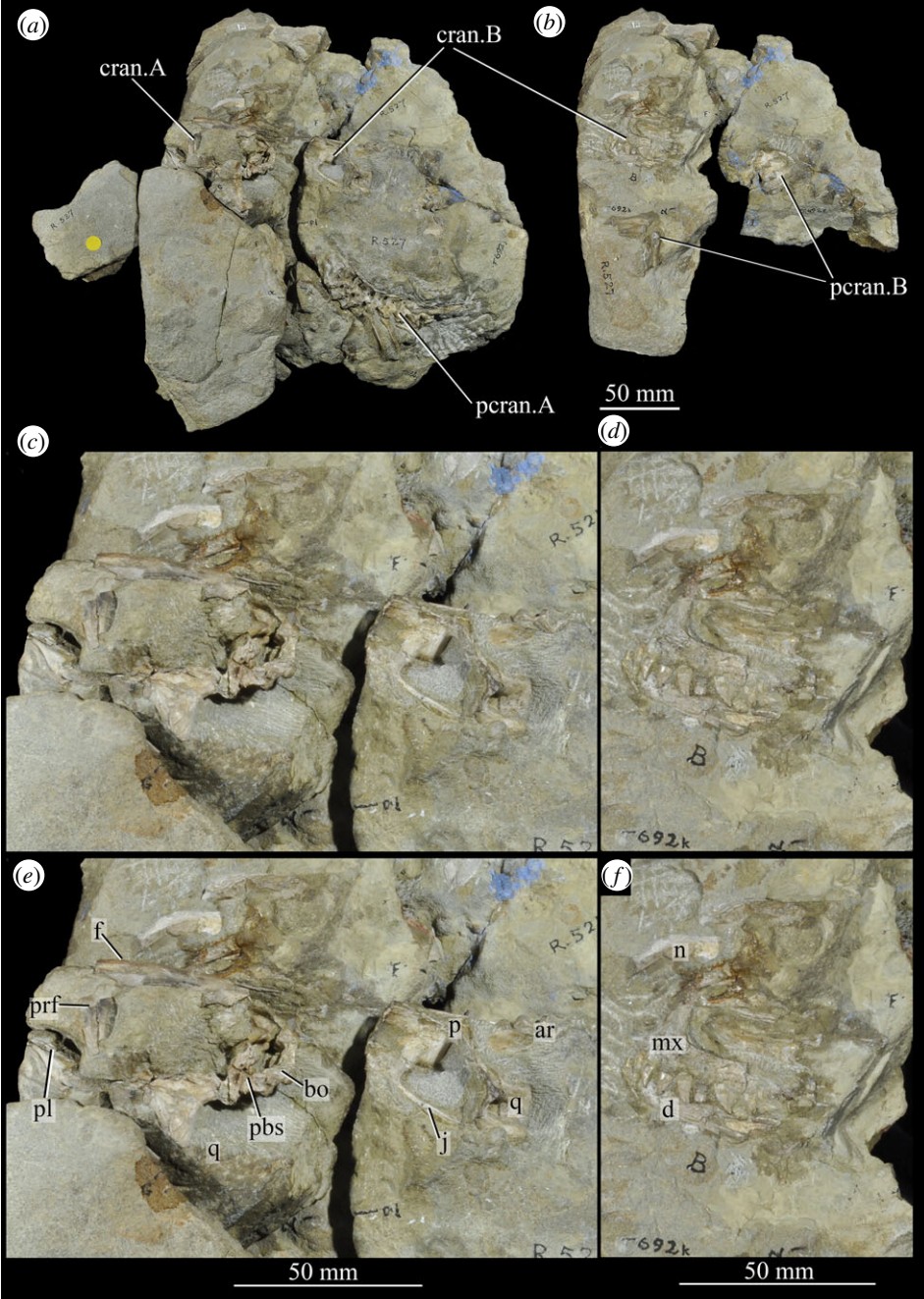

**Figure 18.** Overview of *Euparkeria capensis* specimen UMZC T.692, which contains two individuals, showing (*a*) all blocks fully assembled with skulls in left lateral view, (*b*) the same view but with blocks containing individual A removed to expose the anterior skull and forelimb of individual B, and close-ups of the skull of individual A and posterior of the skull of individual B in left lateral view, without (*c*) and with (*e*) labelling, and of the anterior of the skull of individual B without (*d*) and with (*f*) labelling. ar, articular; cran.A, cranium of individual A; cran.B, cranium of individual B; d, dentary; f, frontal; j, jugal; mx, maxilla; n, nasal; p, parietal; pbs, parabasisphenoid; pcran. A, postcranium of individual A; pcran.B, postcranium of individual B; pl, palatine; prf, prefrontal; q, quadrate. Scale below (*b*) applies to (*a*,*b*), and that below (*c*,*e*) applies to (*c*,*e*) and that below (*d*,*f*) applies to (*d*,*f*).

K8050 also contain remains of *Mesosuchus* (most of the skeleton, and a marginal tooth row, respectively). There is no reason (e.g. differences in form and size between elements or presence of overlapping elements) to consider that any other specimens, including the holotype, include elements from more than one individual. It has been suggested by Ewer [54] based on the aggregation evidenced in some specimens that all specimens were found in close proximity and may have been hibernating together. Certainly, all specimens were all found at one site, but there remains no direct evidence that the separate specimens were found in close proximity, although this is plausible.

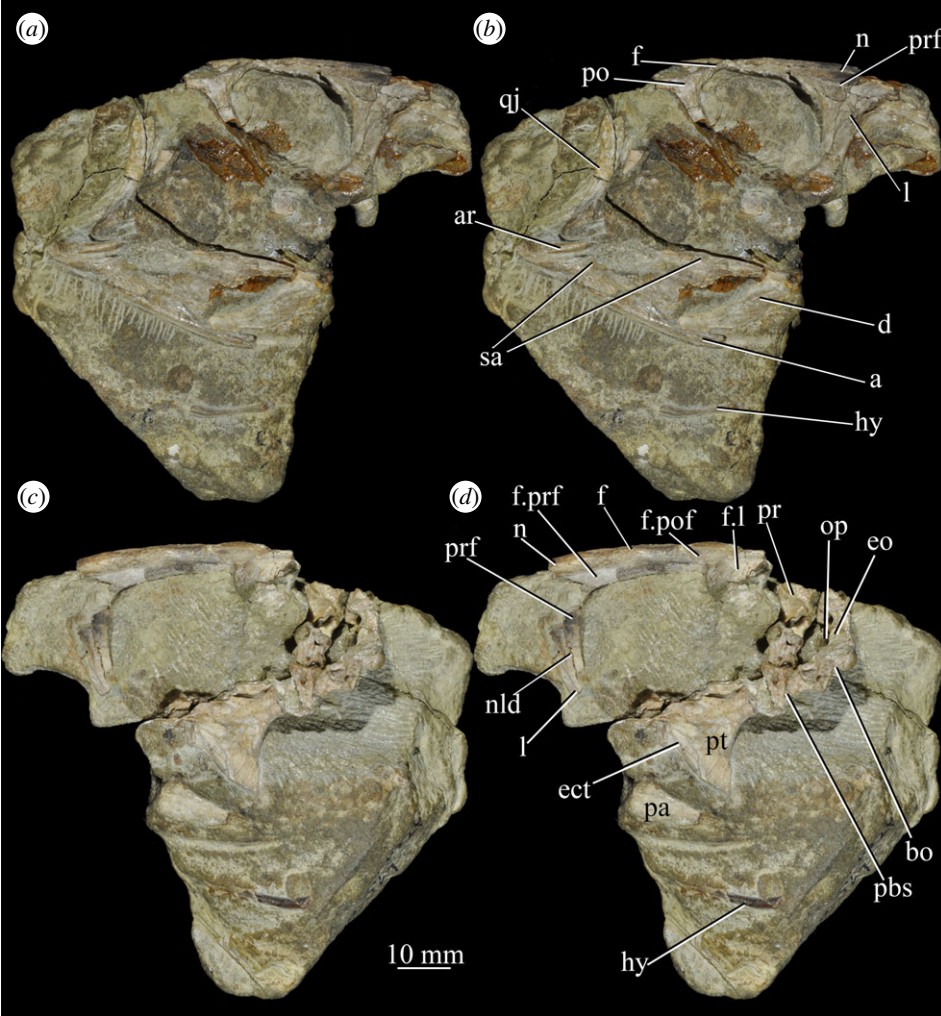

**Figure 19.** Posterior of the skull of *Euparkeria capensis* specimen UMZC T.692 individual A, which preserves much of the braincase in (*a*,*b*) right and (*c*,*d*) left lateral views—(*a*) and (*c*) are unlabelled, while (*b*) and (*d*) are labelled. a, angular; ect, ectopterygoid; eo, exoccipital; f, frontal; f., facet for; hy, hyoid; l, lacrimal; n, nasal; nld, nasolacrimal duct; op, opisthotic; pa, prearticular; pbs, parabasisphenoid; po, postorbital; pof, postfrontal; pr, prootic; prf, prefrontal; pt, pterygoid; qj, quadratojugal; sa, surangular. The same scale applies to all images.

## 3.2. Description and comparisons

### 3.2.1. Overview and major cranial openings

The skull of *Euparkeria* is relatively short anteroposteriorly, and is relatively tall dorsoventrally at its anterior end (figures 3, 4 and 8), contrasting with long-snouted basal archosauriforms such as *Proterosuchus fergusi* [83] or *Chanaresuchus bonapartei* [80], where the snout tapers dorsoventrally towards its anterior end. The skull of *Euparkeria* is more similar in proportions to basal archosauriforms such as *Osmolskina* [74] and *Erythrosuchus africanus* [81], and crown group taxa such as *Ornithosuchus* [76]. In dorsal and ventral view (figures 4*c*,*d* and 8*c*,*d*), the skull has a broadly triangular outline, being transversely expanded posteriorly and tapering anteriorly. However, the rostrum does not taper as strongly relative to the posterior end of the skull as in some other early stem and crown archosaurs (e.g. *Sphenosuchus acutus* [82], *Chanaresuchus* [80], *Proterosuchus fergusi* [83]). In transverse section, the skull outline approaches a mediolaterally compressed trapezium posteriorly (i.e. wider ventrally than dorsally), becoming more oval anteriorly.

The external naris is positioned just posterior to the anterodorsal margin of the rostrum, its dorsal border formed by the nasal, and its posterior, anterior and ventral borders by the premaxilla (figures 3, 4 and 8, nar). The external naris is subcircular in outline. A bar formed by the premaxilla, nasal and maxilla separates the external naris and external antorbital fenestra.

*Euparkeria* possesses external and internal antorbital fenestrae (figures 3, 4*a*–*c* and 20, eaof, iaof; *sensu* Witmer [84]). Anteriorly, dorsally and to a lesser extent posteriorly, these fenestrae bound an antorbital

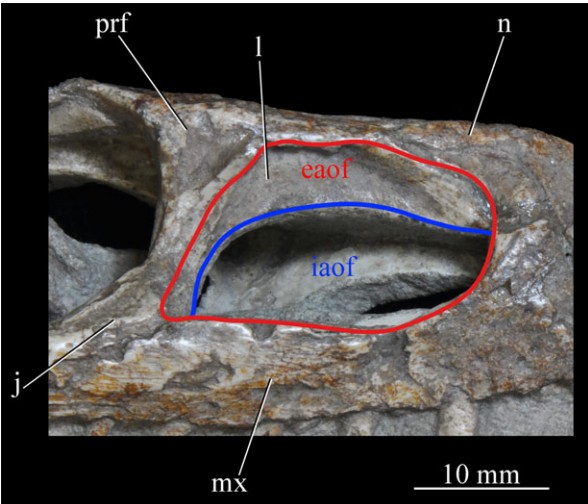

**Figure 20.** Right antorbital fenestra of *Euparkeria capensis* specimen SAM-PK-5867 in lateral view. eaof, external antorbital fenestra; iaof, internal antorbital fenestra; j, jugal; l, lacrimal; mx, maxilla; n, nasal; prf, prefrontal. External antorbital fenestra outlined in red, internal antorbital fenestra outlined in blue.

fossa extending onto the maxilla and lacrimal. The external antorbital fenestra is bounded by the maxilla ventrally, anteriorly and anterodorsally, the jugal posteroventrally, and the lacrimal posteriorly and dorsally. The ventral margin of the external antorbital fenestra is slightly below that of the naris, and its dorsal margin is slightly above that of the naris. The external antorbital fenestra is anteroposteriorly elongated and suboval in outline, and is drawn into a point posteroventrally. The external antorbital fenestra is separated from the orbit by the antorbital bar. The internal antorbital fenestra, bordered by the same elements as the external fenestra, is similar in shape to the external antorbital fenestra but is proportionately less tall dorsoventrally.

The orbit of *Euparkeria* (figures 3 and 4*a*,*b*, or) is subcircular and its posterior margin is placed anterior to the occipital condyle by 22% of the distance from the occipital condyle to the tip of the rostrum. The orbit is bordered by the frontal dorsally, the postfrontal posterodorsally, the postorbital posteriorly, the jugal ventrally, and the jugal, lacrimal and prefrontal anteriorly. The ventral margin of the orbit is slightly ventral to that of the antorbital fenestra and the dorsal margin is just dorsal to the dorsal margin of the antorbital fenestra. The orbit is separated from the infratemporal fenestra and the supratemporal fenestra (see below) by the postorbital bar.

*Euparkeria* possesses a dorsoventrally elongated, laterally opening infratemporal fenestra (figures 3 and 4*a*, ltf; =lateral temporal fenestra), dorsomedial to which is a smaller, dorsally opening, subcircular supratemporal fenestra (figures 3 and 4*b*,*c*, stf). The infratemporal fenestra is bordered by the quadratojugal posteroventrally, the jugal anteroventrally, the postorbital anterodorsally and the squamosal posterodorsally. In lateral view, the infratemporal fenestra is pentagonal, with a straight dorsoventrally extending anterior margin, straight dorsal and ventral margins, and a posterior margin composed of a ventral section that is posteroventral-to-anterodorsally slanted and a dorsal section that is slanted in the opposite direction (thus forming a '>'-shaped posterior margin in right lateral view). The infratemporal fenestra is thus anteroposteriorly longest at its base, becoming narrower dorsally (though anterior and posterior edges do not contact, unlike in many pseudosuchians including in crocodylomorphs [31]), reaching its narrowest point at around two-thirds of its height, and then re-expanding dorsally.

The supratemporal fenestra is bordered by the squamosal posterolaterally, by the postorbital anterolaterally and by the parietal medially. The lateral margin of the supratemporal fenestra is just ventral to the level of the dorsal margin of the orbit, and the medial margin is just dorsal to the level of the dorsal margin of the orbit. In dorsal view, the supratemporal fenestra is teardrop shaped, with the point of the teardrop directed posterolaterally. The infratemporal fenestra and supratemporal fenestra are separated from each other by the postorbital–squamosal bar. A shallow supratemporal fossa surrounds the supratemporal fenestra along the anterior two-thirds of its margin.

The posttemporal fenestra forms an elongated slit in posterior view, curving slightly dorsally at its medial end (figures 3, 4*c* and 5*a*, ptf). The ventral margin of the posttemporal fenestra is formed by the paroccipital process of the opisthotic, the medial margin by the supraoccipital and prootic, the dorsal margin by the parietal and the lateral margin is formed by the squamosal.

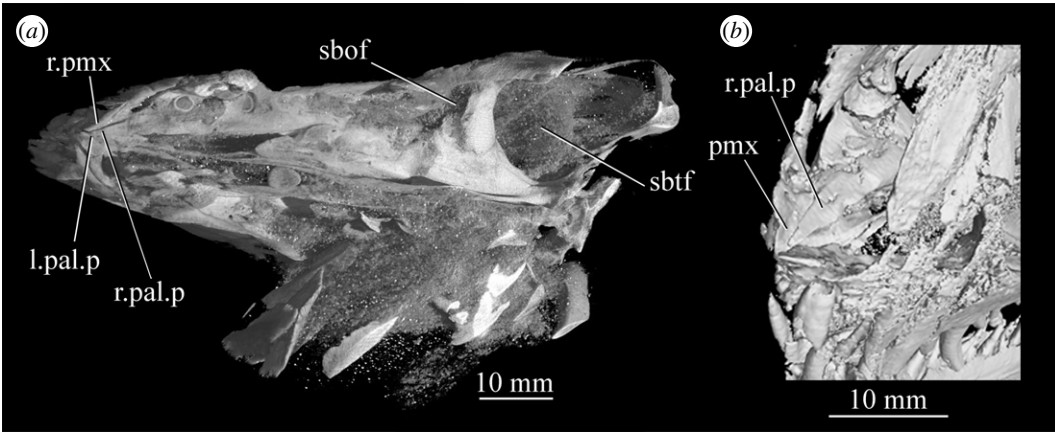

**Figure 21.** CT images showing anterior palate and palatal fenestrae of *Euparkeria capensis*. (*a*) Palate of SAM-PK-5867 in dorsal view and (*b*) surface rendering of the anterior palate of SAM-PK-5867 viewed through naris in left anterolateral and dorsal view. l., left; mx, maxilla; pal.p, palatal process of maxilla; pmx, premaxilla; r., right; sbof, suborbital fenestra; sbtf, subtemporal fenestra.

A quadrate foramen is present between the quadrate medially and the quadratojugal laterally (figures 3 and 5*a*, qf), forming a dorsoventrally elongated ellipsoid in posterior view. Both medial and lateral margins of the foramen (and thus the foramen itself) are medially bowed in posterior view. The exact size and form of the quadrate foramen in life is difficult to assess; the reconstruction of Ewer [54] appears to underestimate its size in posterior view. As preserved, a small part of the foramen is visible in lateral view.

The foramen magnum (figures 3 and 5*a*, fm) is a subcircular opening in the back of the skull formed by the exoccipitals laterally, and also ventrally with the exception of a small contribution by the basioccipital around the midline, and by the supraoccipital dorsally. The dorsal margin of the foramen magnum is level with the dorsal margin of the infratemporal fenestra. The widest point of the foramen magnum is ventral to its dorsoventral centre.

The choana (figures 3 and 4*b*, ch) is bordered posterolaterally and posteromedially by the palatine, laterally by the maxilla, anterolaterally by the premaxilla and anteromedially by the vomer. The posterior of the choana is anteroposteriorly level with the midpoint of the antorbital fenestra, while the anterior of the choana is anterior to the anterior margin of the antorbital fenestra. The choana is a mediolaterally compressed ovaloid in shape, tapering strongly posteriorly.

A small suborbital fenestra is present between the pterygoid (posteromedially), palatine (anterolaterally) and ectopterygoid posteriorly (figures 3 and 21, sbof). It is visible from CT scan data only and it is not exposed in any specimen, and appears to have been a posterolaterally–anteromedially elongated oval. A large subtemporal fenestra (figures 3 and 21*a*, sbtf) is present in the palate below the infratemporal fenestra, bordered anteriorly and medially by the pterygoid, posteriorly by the quadrate, anterolaterally by the jugal and posterolaterally by the quadratojugal. It is elliptical, wider anteriorly than posteriorly, with an anteromedial–posterolateral long axis.

### 3.2.2. Summary of braincase morphology

The braincase of *Euparkeria* was described separately in much detail by Gower & Weber [38], based on one specimen, and later fully redescribed from all specimens—including the internal anatomy and inner ear—by Sobral *et al.* [55] (figure 22). Thus the braincase morphology will not be described in detail here, but a brief overview of the external anatomy is given, and a separate reconstruction is shown in figure 22.

In overall shape, the braincase is more verticalized than that of most non-archosaurian archosauromorphs (e.g. *Mesosuchus* [85]), approaching more closely that of early crown archosaurs (e.g. *Xilousuchus* [86]); i.e. the relative positions of the basipterygoid processes, basal tubera and occipital condyle lie in progressively higher planes. All elements are clearly separately ossified except the basisphenoid and parasphenoid, which are co-ossified as the parabasisphenoid. The braincase lacks a pronounced midline fossa at the anterior end of the median pharyngral recess, unlike many crown taxa (see [31,55]). The exoccipitals form the lateral margins of the foramen magnum, and approach each other but do not quite touch along its ventral margin; dorsally, they are widely separated. The exoccipitals are pierced by the foramen for cranial nerve (CN) XII. The basioccipital

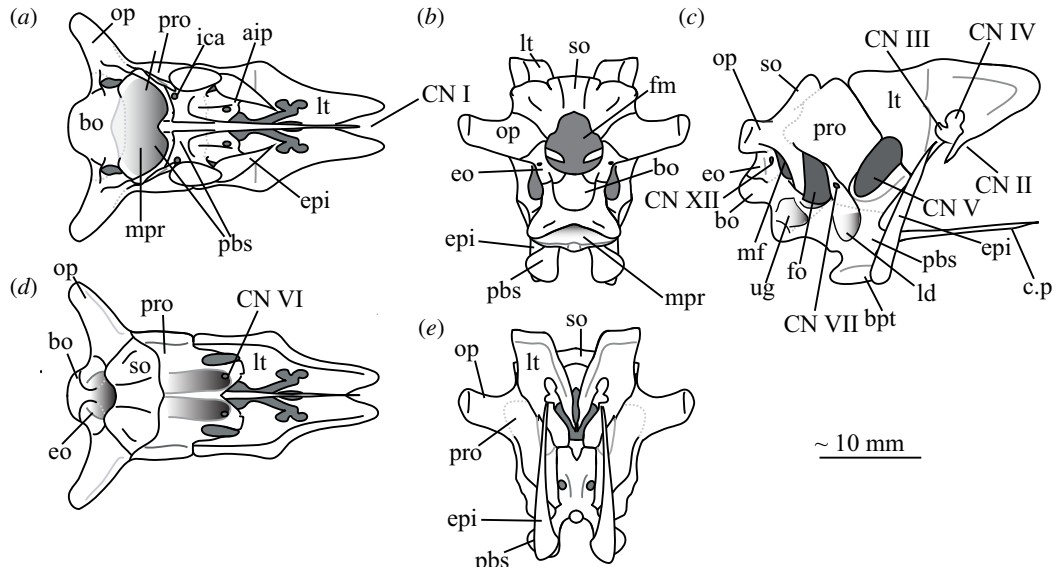

**Figure 22.** Reconstruction of the braincase of *Euparkeria capensis* in (*a*) ventral, (*b*) posterior, (*c*) right lateral, (*d*) dorsal and (*e*) anterior view. aip, anterior inferior process; bpt, basipterygoid process; bo, basioccipital; CN, cranial nerve (followed by a number); c.p, cultriform process; eo, exoccipital; epi, epipterygoid; fm, foramen magnum; fo, fenestra ovalis; ica, internal carotid artery foramen; ld, lateral depression; lt, laterosphenoid; mf, metotic foramen; mpr, median pharyngeal recess; op, opisthotic; pbs, parabasisphenoid; pro, prootic; so, supraoccipital; ug, unossified gap.

forms the occipital condyle, the ventral margin of the foramen magnum and the posterior part of the basal tubera. The anterior parts of the basal tubera are formed by the parabasisphenoid, as are the anterior part of the braincase floor, and the basipterygoid processes. The supraoccipital forms the posterior braincase roof and dorsal margin of the foramen magnum. It is entirely visible in posterior view, articulating anterodorsally with the parietals. The opisthotics form the paroccipital processes— which are gently posterolaterally directed and straight—and the ventral ramus of the opisthotic forms the anterior wall of the metotic foramen and posterior wall of the fenestra ovalis. The prootics form the lateral walls of the braincase anterior to the fenestra ovalis and posterior to the foramen for CN V. They articulate with the laterosphenoids anteriorly, which in turn form the lateral walls of the braincase anterior to CN V. There is a large lateroventral opening in the side of the laterosphenoid, through which CN III and IV passed, with CN II passing anteroventrally to these in front of the slender process. Dorsolaterally, the laterosphenoid widens to form capitulate processes. The anterior roof of the braincase would have been formed by the parietal and frontal.

The foramen magnum is oval, slightly higher than wide (seemingly exaggerated by deformation in SAM-PK-5867; see Overview and major cranial openings). The foramen ovalis and metotic foramen are also higher than wide, with the latter curving backwards at its dorsal and ventral extremes. The foramina for CN XII and VII (the latter in the prootic anterior to the fenestra ovalis) are small openings in the side of the braincase, whereas that for CN V is large with a clear depression for the Gasserian ganglion. The openings for CN II–IV on the laterosphenoid are also relatively large. CN VI exited the braincase anteroventrally through small foramina in the dorsum sellae. The carotids enter the hypophyseal fossa through foramina on the posteromedial base of the basipterygoid processes.

The middle and inner ear anatomy of *Euparkeria* is well preserved and was described in detail by Sobral *et al.* [55]. The fenestra ovalis is dorsoventrally expanded when compared with less derived taxa such as *Prolacerta broomi* [87] or *Captorhinus laticeps* [88], formed mostly by the prootic and opisthotic, with a contribution of the parabasisphenoid in the ventral and anteroventral portions of the lateral border, and of the basioccipital posteromedially. At about midheight on the fenestra ovalis, the lagenar crests mark the separation of the vestibular and cochlear regions. The lagenar recess is longer than in stem saurian taxa (e.g. *Youngina capensis* [89]), indicating a relatively long cochlea, and its ventralmost portion lies in an unossified space delimited by the braincase floor medially and the parabasisphenoid laterally. Most of the medial wall of the otic capsule was unossified, except for a short medial expansion of the ventral ramus of the opisthotic posteriorly, which is marked by the perilymphatic notch. The metotic foramen was also elongate and large compared with non-saurian eureptiles [88,89]. The ventral portion is wider than the dorsal, with nervous and vascular elements

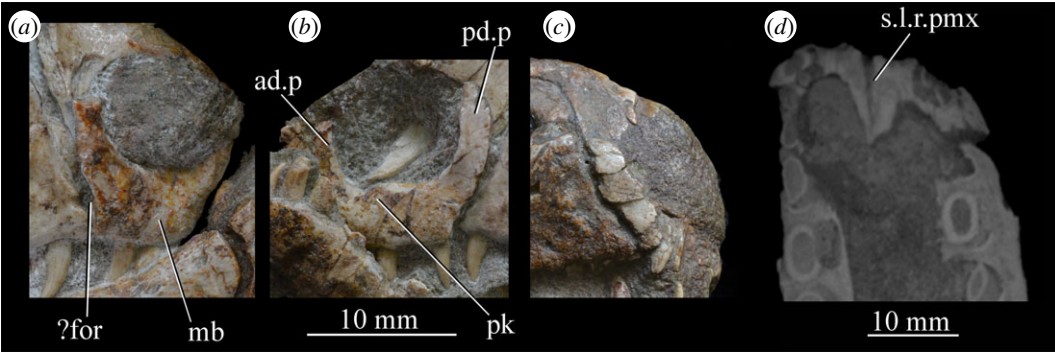

**Figure 23.** Examples and details of the premaxilla of *Euparkeria capensis*. Right (*a*) and left (*b*) premaxilla of SAM-PK-6047A in lateral view; (*c*) right premaxilla of SAM-PK-5867 in lateral view; (*d*) transverse view of CT scan of the premaxillae of SAM-PK-5867, showing their anterior palatal contact. ?for, potential foramen; ad.p, anterodorsal process; mb, main body (of premaxilla); pd.p, posterodorsal process; pk, low peak; s.l.r.pmx, palatal contact between left and right maxillae. The scale below (*b*) applies to (*a–c*), and that below (*d*) to (*d*) only.

presumably crossing dorsally, and the perilymphatic sac sitting ventrally. This regionalization indicates it was partially used as a pressure-relief mechanism. Together, these characteristics point to a refined sense of hearing (see Sobral *et al*. [55] and discussion in the current work). *Euparkeria* possesses relatively thin and elongate semicircular canals of subequal lengths. The anterior canal is the longest of the three, slightly flexing the common crus posteriorly, while the lateral one is the shortest, entering the anterior ampulla posterolaterally. The floccular fossa on the medial side of the prootic is deep and well developed. These features suggest increasingly sensitive gaze stabilization methods compared with other non-archosaurian archosauriforms and archosauromorphs (see Discussion).

### 3.2.3. Dermal roofing bones

#### 3.2.3.1. Premaxilla (figure 23)

The posterior portions of the premaxillae are preserved in SAM-PK-5867 (in articulation with the maxillae; figures 4 and 23*c*) and the entire bones are present in SAM-PK-6047A (slightly displaced from their original articulations; figures 8 and 23*a,b*), with the left premaxilla of the latter being the most complete example. SAM-PK-6048 also preserves a right premaxilla lacking the posterodorsal process (figure 10*b,c*). The premaxilla comprises a toothbearing main body ventral to the external naris, an anterodorsal process forming the anterior margin of the external naris, and a posterodorsal process forming the posterior margin of the external naris. The premaxilla articulates posteriorly with the maxilla, and anterodorsally (via the anterodorsal process) and posterodorsally (via the posterodorsal process) with the nasal. It is not downturned, unlike in several archosauriform taxa (e.g. *Proterosuchus fergusi* [83]; *Erythrosuchus* [81]; *Prolacerta broomi* [90]; in SAM-PK-6048 it has been partially disarticulated, giving the impression of being downturned—figure 10).

The premaxilla contacted its antimere along the ventral midline, forming the anteriormost part of the palate (figure 23*d*). The premaxilla would have also contacted its antimere anteriorly (figure 24) and anterodorsally to form the tip of the rostrum, though both premaxillae are not fully articulated in any specimen. Based on the shape of each individual premaxilla, the rostrum would have been rounded anteriorly in dorsal view. The vomer contacted the premaxilla posteromedially on the dorsal surface of the palatal contribution of the premaxillae (figure 25).

The main body of the premaxilla is an anteroposteriorly elongated bar that is narrower mediolaterally than dorsoventrally. The alveolar margin is nearly straight and slopes slightly from posterodorsally to anteroventrally at about 5° to the horizontal in lateral view. The lateral surface of the main body is dorsoventrally and anteroposteriorly convex. The dorsal margin of the main body is generally concave, forming the narial margin, and merges anteriorly and posteriorly with the bases of the antero- and posterodorsal processes.

Around half way between these processes, the dorsal margin of the main body in SAM-PK-6047A (figure 23*b*, pk) is drawn upwards to a low peak, similar to the less pronounced convexity in, e.g. *Postosuchus kirkpatricki* ([73], fig. 3). There is no narial fossa in *Euparkeria*, nor is there a ventrally opening subnarial gap as is seen in saurischians or crocodylomorphs (see Nesbitt [31], character 11—scored as '?' by Nesbitt). On the right-hand side of SAM-PK-6047A there appears to be a small subnarial foramen (see

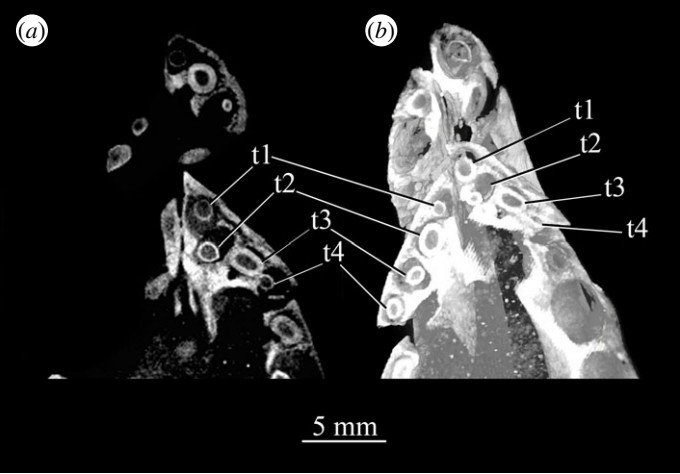

**Figure 24.** CT slice (*a*) and reconstruction (*b*) showing premaxillary teeth of *Euparkeria capensis* specimen SAM-PK-6047A in cross-section in dorsal view. t1–4, premaxillary teeth 1–4.

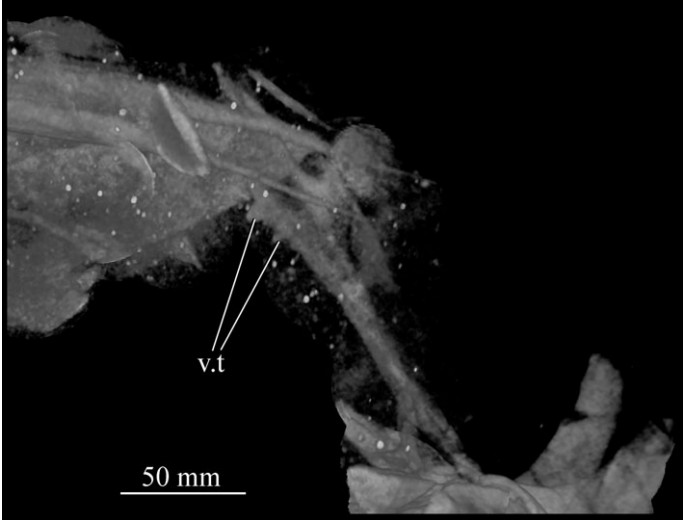

**Figure 25.** CT image of premaxillae and vomers of *Euparkeria capensis* specimen SAM-PK-6047A in the right lateral view. v.t, vomerine teeth.

Nesbitt [31], character 12—scored as absent by Nesbitt) on the anterior border of the maxilla (figure 23*a*, ?for), which would have been closed anteriorly by the premaxilla. This opening is not slitlike, unlike in, e.g. *Saurosuchus galilei* (PVSJ 32; [31]), and approaches more closely the morphology of *Batrachotomus* [75] in that it extends further posteriorly and is more circular in lateral view. This is distinct from the anteriorly directed foramen on the maxilla (see below). Preservation impedes the assessment of this feature in other specimens. The external surface of the premaxilla lacks notable foramina.

The anterodorsal and posterodorsal processes of the premaxilla (figure 23, ad.p, pd.p) are directed approximately perpendicular to the alveolar margin. The anterodorsal process arcs up posteriorly from its base, and is thicker mediolaterally and shorter anteroposteriorly than the posterodorsal process. The dorsal tip of the anterodorsal process has been lost during preservation in all specimens, but would presumably have contacted the nasal. Judging from the position of the nasal, the anterodorsal process cannot have been substantially proximodistally longer than the anteroposterior length of the main body of the premaxilla. The posterodorsal process is longer than the anteroposterior length of the main body of the premaxilla, and arcs up anteriorly from its base, unlike in *Osmolskina* [74] where it is straight. The posterodorsal process differs from that of many other archosauriform and crown archosaur taxa in being dorsally/anterodorsally rather than posterodorsally directed (e.g. *Proterosuchus fergusi* [83]; *Erythrosuchus* [81]; *Rauisuchus tiradentes* [91]; *Herrerasaurus* [77]). The posterodorsal process's vertical, distally rounded morphology was previously suggested as diagnostic of *Euparkeria* [44] and this is supported here.

As is visible on the right side of SAM-PK-6047A (figure 23*a*), the posterodorsal process forms an overlapping contact with the nasal: the medial surface of the posterior part of the tip of the process fits into a posterodorsally rounded, depressed facet on the nasal (see Nasal). This is similar to the situation in erythrosuchids [81,92], *Gracilisuchus stipanicicorum* [93] and *Eoraptor lunensis* [78], but differs from other basal theropods and sauropodomorphs where the process is thin and does not reach the nasal [31], and from aetosaurs [31], *Xilousuchus sapingensis* [94] and *Arizonasaurus babbitti* [95], where the process is thick but also does not reach the nasal. The facet on the nasal borders the external naris anteroventrally. The form of the suture with the maxilla can be seen on the right side of SAM-PK-6047A (figure 23*a*) due to the premaxilla being slightly displaced from its original articulation. In lateral view, the suture is anteriorly convex ventrally and anteriorly concave dorsally. The premaxilla overlaps the maxilla laterally along this suture, with this overlap most extensive at the dorsoventral midpoint of the anteriorly concave part of the suture. There is no slot for the posterodorsal process of the premaxilla on the maxilla, unlike in *Arizonasaurus babbitti* [95] or *Xilousuchus sapingensis* [94].

CT data for SAM-PK-5867 and SAM-PK-6047A show four premaxillary teeth (figure 24; *contra* Nesbitt [31] who counted three but agreeing with Ewer [54]), as is common among non-archosaurian archosauriforms and basal archosaurs (e.g. *Chanaresuchus* [80], *Osmolskina* [74], 'rauisuchians' [75]). Three premaxillary teeth are present in *Gracilisuchus* [93] and ornithosuchids [22,76]. *Erythrosuchus* [81] and *Prolacerta* [90] have five and *Proterosuchus fergusi* has eight [31], reflecting its elongated snout. The anterior end of the premaxilla is not well preserved in SAM-PK-5867, but two teeth can be identified with certainty on the right side and one tooth on the left. The premaxillary teeth, like the maxillary teeth, are recurved and pointed at their apices, and their crowns are slightly wider mesiodistally than labiolingually. The premaxillary teeth are, however, less labiolingually compressed than the anteriormost maxillary teeth and are less strongly recurved. Except for the posteriormost right premaxillary tooth of the holotype, they also appear to lack denticles, but this may be due to their being obscured by matrix.

### 3.2.3.2. Maxilla (figure 26)

Both maxillae are preserved in SAM-PK-5867 and SAM-PK-6047A (figure 26*a*–*d*), and SAM-PK-6050 contains disarticulated left (figure 26*e,f*) and right maxillae probably from one individual. Maxillae are preserved in UMZC T.692 (right maxilla in medial view is preserved in individual A—figure 27; left maxilla in individual B—figure 18*d,f*). The aforementioned examples form the basis of the description. Specimens SAM-PK-6048, SAM-PK-K8050, SAM-PK-13665, SAM-PK-13666, SAM-PK-13667, GPIT/RE/12913 and AMNH 2239 also preserve examples of the maxilla. The main body of the maxilla is a horizontal bar, longer anteroposteriorly than wide mediolaterally (as in most basal archosaurs and non-archosaurian archosauriforms but contrasting with the wide maxillae of, e.g. *Stagonolepis* [96] and *Effigia* [97]) forming the ventral margin of the external antorbital fenestra and bearing the maxillary teeth.

The dorsal process (=ascending process; figure 26, d.p) of the maxilla forms the anterior margin of the external and internal antorbital fenestrae and curves posteriorly in a concave arc, tapering to a tip similar to many other archosauriforms (e.g. *Erythrosuchus* [81], *Batrachotomus kupferzellensis* [75]), but unlike the blunt process of *Osmolskina* [74] or the non-tapering process of *Stagonolepis robertsoni* [96]. A palatal process (figure 26*e*, pal.p) arcs anteromedially from the anterior end of the main body. The maxilla articulates anteriorly with the premaxilla, anterodorsally (via the dorsal process) with the nasal, at the tip of the dorsal process with the lacrimal, and posteriorly with the jugal. Anteromedially, the maxilla just contacts its antimere via the palatal process, with the ventrolateral surfaces of the palatal processes contacting the premaxillae. The palatine contacts the ventral portion of the medial surface of the maxilla for much of the length of the maxilla.

The dorsal margin of the main body of the maxilla forming the ventral margin of the antorbital fenestra is concave in lateral view, with the anterior and posterior ends of the main body dorsoventrally level with one another. In lateral view, the alveolar margin of the maxilla is subhorizontal, showing no sinusoidal curvature as in *Erythrosuchus* [81]. The alveolar margin is scalloped above each tooth in lateral view (clear in the left maxilla of SAM-PK-6047A—figure 26*c*).

The lateral surface of the maxilla is best preserved in SAM-PK-6047A (figure 26*c,d*) and lacks ornamentation with the exception of subtle horizontal striations towards the posterior end. In SAM-PK-6047A an 'anterior maxillary foramen' (see Modesto & Sues [90] for a discussion of homologies) is present (figure 26*c*, am.for), opening anterolaterally below the anteroposterior midpoint of the dorsal process and half way up the main body of the maxilla. This foramen was considered homologous with

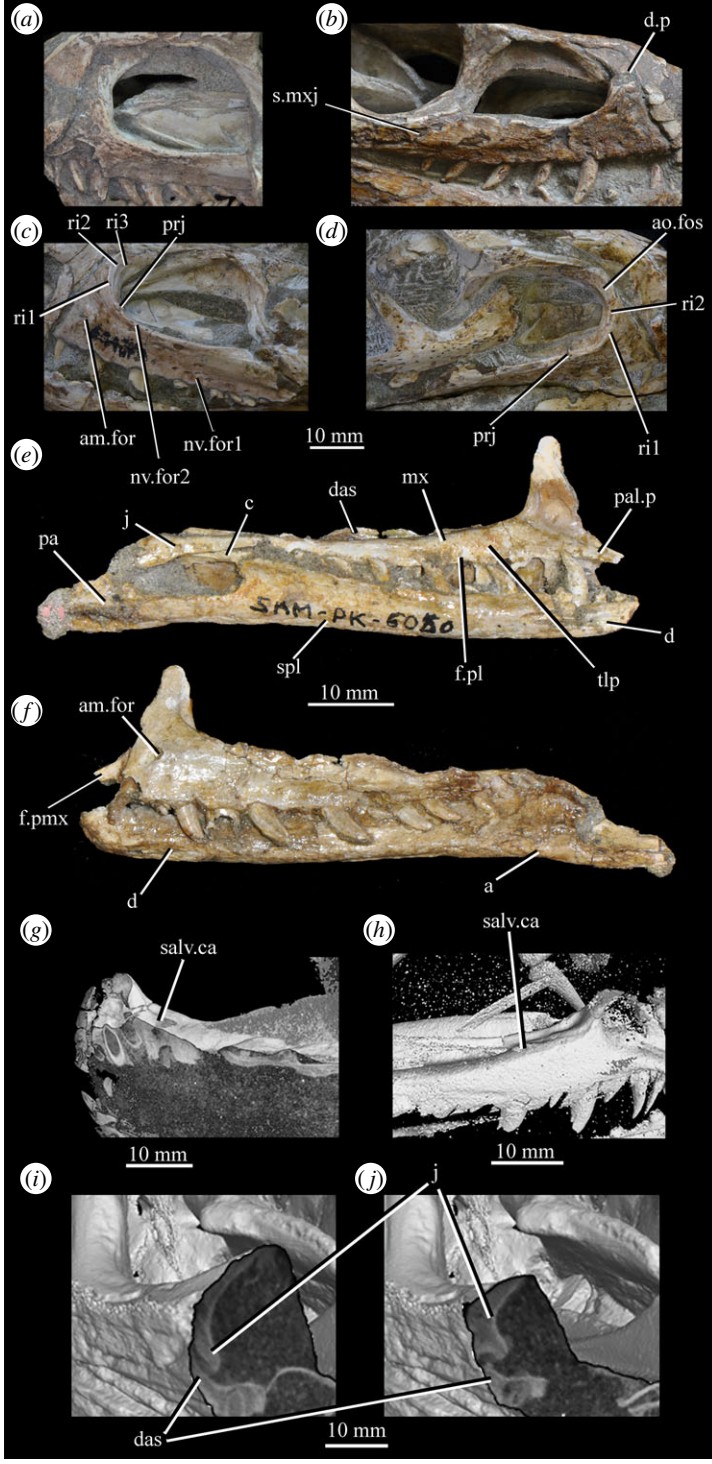

**Figure 26.** Examples and details of the maxilla of *Euparkeria capensis*. (*a*) Left maxilla of SAM-PK-5867 in lateral view; (*b*) right maxilla of SAM-PK-5867 in lateral view; (*c*) left maxilla of SAM-PK-6047A in lateral view; (*d*) right maxilla of SAM-PK-6047A in lateral view; left maxilla of SAM-PK-6050 in (*e*) medial and (*f*) lateral view; CT scan renderings of (*g*) right maxilla of SAM-PK-5867 in medial and slightly posterodorsal view, with sagittal section to show entrance and path of the superior alveolar canal, (*h*) left maxilla of SAM-PK-6047A in medial and slightly dorsal view to show the entrance of the superior alveolar canal and (*i,j*) right maxilla of SAM-PK-5867 in right lateral view with more anterior (*i*) and more posterior (*j*) cross-section to show articulation with jugal. a, angular; am.for, anterior maxillary foramen; ao.fos, antorbital fossa; c, coronoid; d, dentary; das, dorsally ascending sheet; d.p, dorsal process; f.pl, facet for the palatine; f.pmx, facet for premaxilla; j, jugal; mx, maxilla; nv.for1, lower row of neurovascular foramina; nv.for2, upper row of neurovascular foramina; pa, prearticular; pal.p, palatal process; prj, projection; ri, ridge (see text for numbering); s.mxj, suture between maxilla and jugal; salv.ca, entrance of superior alveolar canal; spl, splenial; tlp, thickened lower portion. Upper scale applies to (*a–d*), scale between (*e*) and (*f*) to both (*e*) and (*f*), those below (*g*) and (*h*) to those images, respectively, and that below (*i*) and (*j*) to both (*i*) and (*j*).

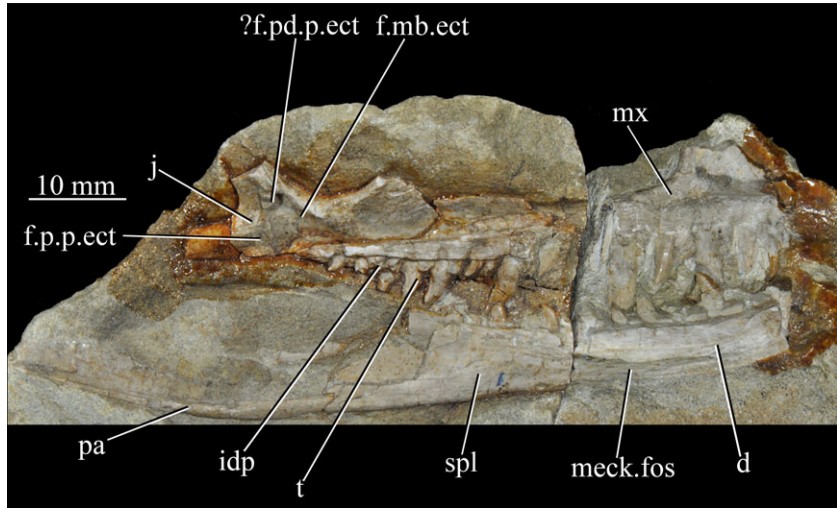

**Figure 27.** Left jaws of *Euparkeria capensis* specimen UMZC T.692 individual A in medial view. d, dentary; f.mb.ect, facet for main body of ectopterygoid head; ?f.pd.p.ect, probable facet for posterodorsal process of ectopterygoid; f.p.p.ect, facet for posterior process of ectopterygoid; idp, interdental plate; j, jugal; meck.fos, Meckelian fossa; mx, maxilla; pa, prearticular; spl, splenial; t, tooth.

foramina in *Prolacerta* and *Proterosuchus fergusi* by Nesbitt [31], and is probably homologous with what is identified as the superior alveolar canal exit foramen in *Osmolskina* [74] based on position.

At least nine posterolaterally opening neurovascular foramina form an anterodorsally directed row along the main body, just above the alveolar margin (figure 26*c*, nv.for1), extending to just anterior to the posteriormost tooth; they are irregularly shaped and spaced. A series of four smaller, laterally opening foramina (figure 26*c*, nv.for2), border the anteroventral margin of the antorbital fenestra, with a larger posterolaterally opening foramen posterior to the last of this series. Several other irregularly shaped and spaced smaller foramina are present between the lower and upper series. A small projection protrudes from the dorsal margin of the maxilla in SAM-PK-6047A (figure 26*d*, prj) at approximately one-third of the distance from the anterior margin of the fenestra to the posterior end of the maxillary contribution to the margin, but is absent in other specimens of *Euparkeria*, perhaps due to preservation. A similar projection is present in one specimen of *Erythrosuchus* (SAM-PK-K1098) but is absent in other specimens and in other taxa (e.g. *Batrachotomus* [75], *Herrerasaurus* [77], *Osmolskina* [74]). One of the four foramina is placed at the base of this projection in the left-hand side of SAM-PK-6047A, and two small foramina appear to be present on the right-hand side.

The more anterior and ventral part of the lateral surface of the dorsal process is flat and featureless, not bulging laterally as in *Erythrosuchus* [81]. A ridge (figure 26*c*,*d*, ri1) marks the anterior margin of the external antorbital fenestra, potentially homologous, given its position, with the bulbous ridge present in other taxa such as *Postosuchus kirkpatricki* [31] and the sharp ridge of, e.g. *Coelophysis bauri* [31]. Two further ridges, one inner (i.e. closer to the margin of the internal antorbital fenestra) and one outer, are visible within this fossa on the left side of SAM-PK-6047A (figure 26*c*, ri2, ri3; the inner ridge is not visible on the right side), arcing to follow the curvature of the dorsal process. A single ridge, in roughly the same position as the outer ridge of the two in *Euparkeria*, is present in *Postosuchus kirkpatricki* (TTU-P 9000 shown in Nesbitt [31], fig. 15), but the presence of two ridges is autapomorphic.

The medial surface of the main body of the maxilla can be seen in SAM-PK-6050 (figure 26*e*), UMZC T.692 (figure 27) and SAM-PK-13666 (figure 14). The ventral two-thirds of the medial surface posterior to the dorsal process is rounded and bulges medially (figure 26*e*, tlp). Dorsal to this transversely thickened lower portion, the main body narrows to form a laterally placed, dorsally ascending sheet (the apex of which forms the border of the external antorbital fenestra; figure 26*e*, das), which merges with the posterior margin of the dorsal process anteriorly. Posteriorly, this sheet and the dorsal surface of the transversely thickened lower portion form a facet for the anteroventral lateral side of the jugal (figure 26*i*), and towards the posterior extreme of the maxilla, the jugal extends dorsal and lateral to this sheet, cupping it in place (figure 26*j*). A posterodorsally open foramen for the entrance of the superior alveolar canal positioned close to the anterior portion of the sheet, as in *Osmolskina* (see [74], fig. 6), is visible in the CT scans (figure 26*g*,*h*), but is obscured or damaged in specimens where the medial surface of the maxilla is exposed.

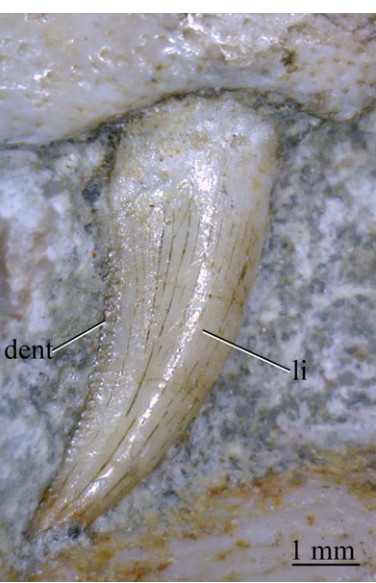

**Figure 28.** Right maxillary tooth of *Euparkeria capensis* specimen SAM-PK-6047A in lateral view. dent, denticles; li, line.

The faint palatine facet (figure 26*e*, f.pl) consists of a shallow, posterodorsally to anteroventrally elongated depression on the transversely thickened lower portion, centred at the anteroposterior midpoint of the maxilla posterior to the dorsal process. A large, subcircular fossa, which would have surrounded the posterolateral part of the main nasal chamber, is present at the base of the dorsal process, primarily on the anterior half of the process; a similar depressed area is present in *Osmolskina* [74], but less strongly defined anteriorly and dorsally. The anteromedially directed palatal process of the maxilla (visible in all views in SAM-PK-6050; figure 26*e*, pal.p) has a ventral margin flush with the ventral margin of the main body of the maxilla and a dorsal margin that curves down and inwards from the anterior margin of the dorsal process in a slightly dorsally concave curve. The ventral two-thirds of the palatal process are expanded transversely in comparison with the sheet-like dorsal third of the process.

The maxilla–jugal suture extends anterodorsally to posteroventrally in lateral view (figure 26*b*, s.mxj). The maxilla overlaps the jugal laterally, with the maxilla articulating with a depressed facet on the jugal, with no slot and process articulation as is seen in aetosaurs (see [31]). The articulation between the maxilla and the lacrimal consists of the posterodorsal tip of the dorsal process being received by a ventral facet on the lacrimal (figure 26*d*). The dorsal process of the maxilla overlies the nasal laterally, with the lateral surface of the nasal excavated to form a facet to receive the medial surface of the maxilla (see Nasal). The distal end of the palatal process of the maxilla articulated only at their very tips with the opposite process at the midline in a simple point articulation (see figure 21*a*, although the left process has been moved anteriorly post-mortem). Ventrolaterally and anteriorly the palatal processes contacted the premaxillae (figure 21*b*), with a facet consisting of a simple ventrolaterally open concave area along their length (figure 26*f*, f.pmx)

The left maxilla of SAM-PK-5867 preserves 12 teeth, with an empty alveolus posterior to the distalmost tooth; 14 alveoli (one empty anterior to the distalmost tooth) are present in SAM-PK-6047A. This is a similar number to that in many other archosauriforms (e.g. *Osmolskina* with 12 [74], *Chanaresuchus* with 12 [80], *Erythrosuchus* with 11 [81]), but fewer than in some long-snouted taxa (e.g. *P. fergusi* with 20 [83], *Prolacerta* with 19—BP-1-471). The teeth are thecodont. The roots are bluntly rounded basally, expand in diameter until they reach the crown, and show no true striations, but do show thin lines running proximodistally down the tooth (figure 28, li). The crowns are homodont and recurved, tapering mesiodistally and labiolingually to their apices, and slope diagonally from anterodorsally to posteroventrally along their long axes in lateral view. The recurvature of the crowns does not change consistently along the row, but the teeth slope more strongly from anterodorsally to posteroventrally posteriorly in the row. The crowns are wider mesiodistally than labiolingually, tapering to carinae mesially and distally, and their labial and lingual surfaces are mesiodistally convex. The crowns are serrated mesially and distally along the carinae from close to their base to their apices (figure 28). The denticle density is around 8 mm$^{-1}$ (first right maxillary tooth, SAM-PK-6047A, dorsalmost eight posterior denticles), though the denticles decrease in apicobasal height (and thus increase in density) apically.

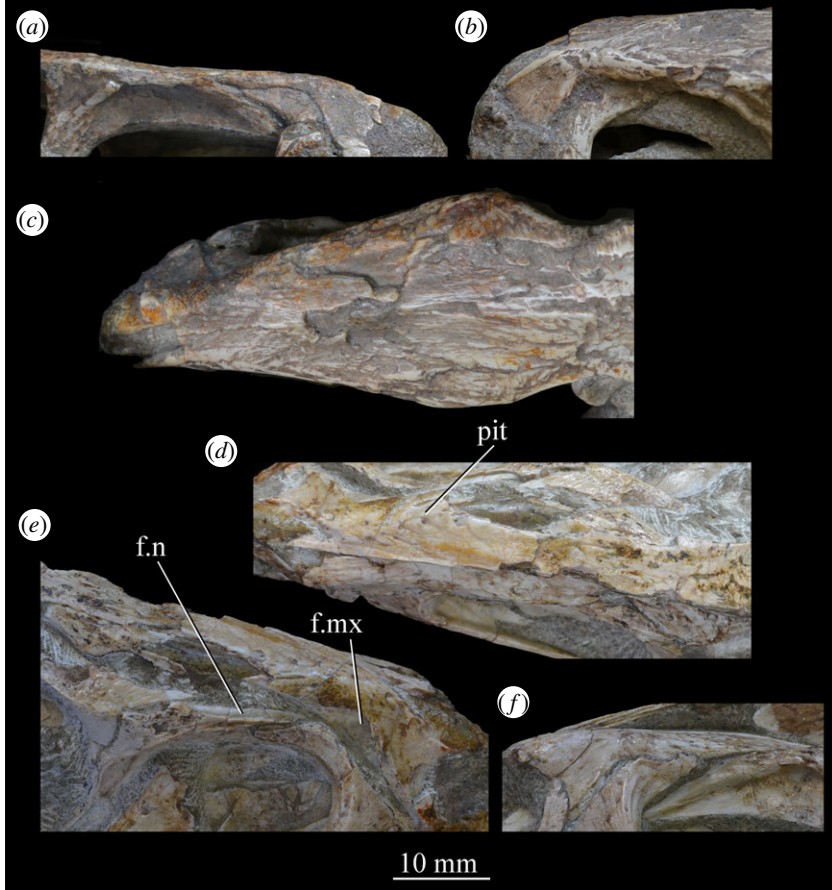

**Figure 29.** Examples and details of the nasal of *Euparkeria capensis*. Nasals of SAM-PK-5867 in (*a*) right lateral, (*b*) left lateral, and (*c*) dorsal view, and of SAM-PK-6047A in (*d*) dorsal, (*e*) right lateral, (*f*) left lateral view. f.mx, maxillary facet; f.n, nasal facet; pit, pit. The same scale applies to all images.

The apicobasally tallest and mesiodistally broadest crown is tooth three in SAM-PK-5867 and on the left side in SAM-PK-6047A; crown length and breadth decrease anterior and posterior to this point. The maxillary teeth differ from the premaxillary teeth in being more labiolingually compressed, possessing serrations, and in being more strongly recurved. There is no diastema between the premaxillary and maxillary teeth. Alternating replacement is plausible as teeth are alternately fully erupted and incompletely erupted/absent in SAM-PK-6047A, and a similar pattern is also apparent in SAM-PK-5867, especially when CT data are examined. The replacement teeth show no features that differentiate them from fully erupted teeth except that they and their roots are shorter.

Interdental units (=interdental plate + interdental septum—the two structures are not distinguishable in *Euparkeria*—see [98]) are present as described in detail by Senter [98]. They are visible on the medial side of the mandible in UMZC T.692 (figure 27, idp). These units are separate ossifications from the maxilla, implanted dorsally within it like the teeth, and extending medially between the teeth and then anteriorly and posteriorly lingual to the teeth immediately anterior and posterior to the unit. They are subrectangular in lingual view, extending around one-third of the way across the tooth anterior and posterior (they are not fused, unlike in, e.g. *Polonosuchus silesiacus* [99]), and for around one-quarter of the exposed apicobasal height of the tooth.

### 3.2.3.3. Nasal (figure 29)

Both nasals are preserved in SAM-PK-5867 (figure 29*a*–*c*) and in SAM-PK-6047A (figure 29*d*–*f*), SAM-PK-6048 (figure 10*a*), SAM-PK-K8050 (figure 17) and GPIT/RE/12913 (figure 7), and a right and left (individual A), and probable second left (individual B) nasal are partially preserved in UMZC T.692 (figure 18); these specimens form the basis of the description. Specimens AMNH 2239, SAM-PK-13665, SAM-PK-13666 and SAM-PK-7699 also preserve examples of the nasals. The nasals form around 40% of the anteroposterior length of the skull roof. Anterolaterally, the nasals contact the

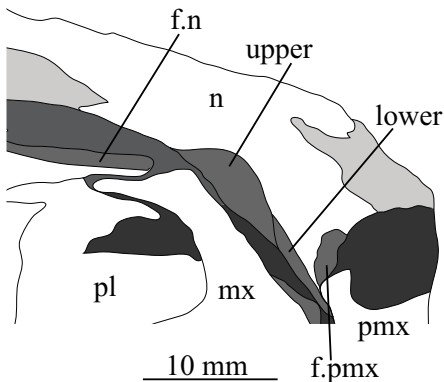

**Figure 30.** Line drawing showing the maxillary and premaxillary facets of the nasal from *Euparkeria capensis* specimen SAM-PK-6047A in right lateral view. f.n, lacrimal facet for nasal; f.pmx, facet for premaxilla; lower, lower section of maxillary facet of nasal; mx, maxilla; n, nasal; pl, palatine; pmx, premaxilla; upper, upper section of maxillary facet of nasal. White, bone; dark grey, matrix; light grey, broken surface; medium grey, articulatory facet.

premaxilla and maxilla; they curve down anteriorly to meet the premaxilla anterior and posterior to the external naris, and thus form the dorsal margin of the naris and the dorsalmost parts of the bars anterior and posterior to the naris. Posteriorly, the nasals contact the frontals, and medially, they contact their antimeres. Ventrolaterally at their posterior ends, the nasals contact the lacrimals. Posterolaterally, the nasals contact the prefrontals relatively extensively (unlike the point contact in *Ornithosuchus* [76] and lack of contact in *Riojasuchus tenuiceps* [100,101]), and are excluded from the lateral margin of the skull by the prefrontals posterior to the start of this contact.

In dorsal view, the nasals are widest just anterior to their suture with the frontals at the posterior margin of the external antorbital fenestra. The nasals narrow anteriorly, then expand before contacting the premaxillae. The dorsal surface of the nasal is very slightly convex both anteroposteriorly and mediolaterally, with no posterior depressed area as in some 'rauisuchians' and crocodylomorphs (e.g. *Batrachotomus* [75], *Sphenosuchus acutus* [31]). This convexity becomes more pronounced anteriorly, with the bone sloping down laterally and anteriorly to contact the premaxilla and maxilla. The external surface of the nasal is smooth apart from 5 to 10 posteromedially/posterolaterally opening small pits (figure 29, pit) on the dorsal surface (see right-hand nasal in SAM-PK-6047A) following a rough posterolateral to anteromedial diagonal.

SAM-PK-6047A shows details of the overlapping contact between the nasal and the lacrimal. It consists of a dorsolaterally open, mediolaterally concave facet covering most of the dorsal surface of the lacrimal (figure 29, f.n; with the exception of a ridge along the lateral edge of the lacrimal), which receives the ventromedial surface of the nasal. As in most non-eusaurischian archosauriforms [102,103], no posterolateral process of the nasal envelops part of the lacrimal. The nasal–frontal suture forms an anteriorly convex arc in dorsal view (figure 4*c*)—this is similar to *Osmolskina* [74], but is not seen in, e.g. *Erythrosuchus* where the suture line is not curved (excepting interdigitations) [81]. The fine detail of the nasal–frontal suture is not well preserved in any specimen, but based on SAM-PK-6047A (right-hand side), it appears to be very gently interdigitated, with at least two slight interdigitations per side (figure 8*c*); this contrasts with *Osmolskina* [74] and most crown archosaurs [31] which lack interdigitations, but the interdigitations in *Euparkeria* are much less pronounced than in *Erythrosuchus* [81] and proterochampsids [80,104] but similar in number, and less pronounced and fewer in number than in *Proterosuchus fergusi* [83] (which has at least seven on each side). The degree of interdigitation and number is less than that of the frontal–parietal suture (see below).

A broad depressed facet along the anterior part of the ventral margin of the nasal receives the anteromedial surface of the maxilla (figure 29, f.mx). The facet for the maxilla is separated into two parts: the lower part of the facet (figure 30, lower) is less strongly excavated/depressed than the upper part (figure 30, upper). The upper part is broader and rounded anteriorly. The lateral surface of the nasal immediately adjacent and anterodorsal to the dorsal section of the maxillary facet is thickened, and forms the anterodorsal margin of the external antorbital fenestra. The posterodorsal process of the premaxilla is received by a depressed, posterodorsally rounded facet on the anterolateral surface of the nasal (figure 30, f.pmx). The suture anterodorsal to the naris between the nasal and the premaxilla is not clearly preserved in any specimen. The nasal slightly overlies the prefrontal at their contact, and there is a simple appression, with no interdigitations or true facet. In

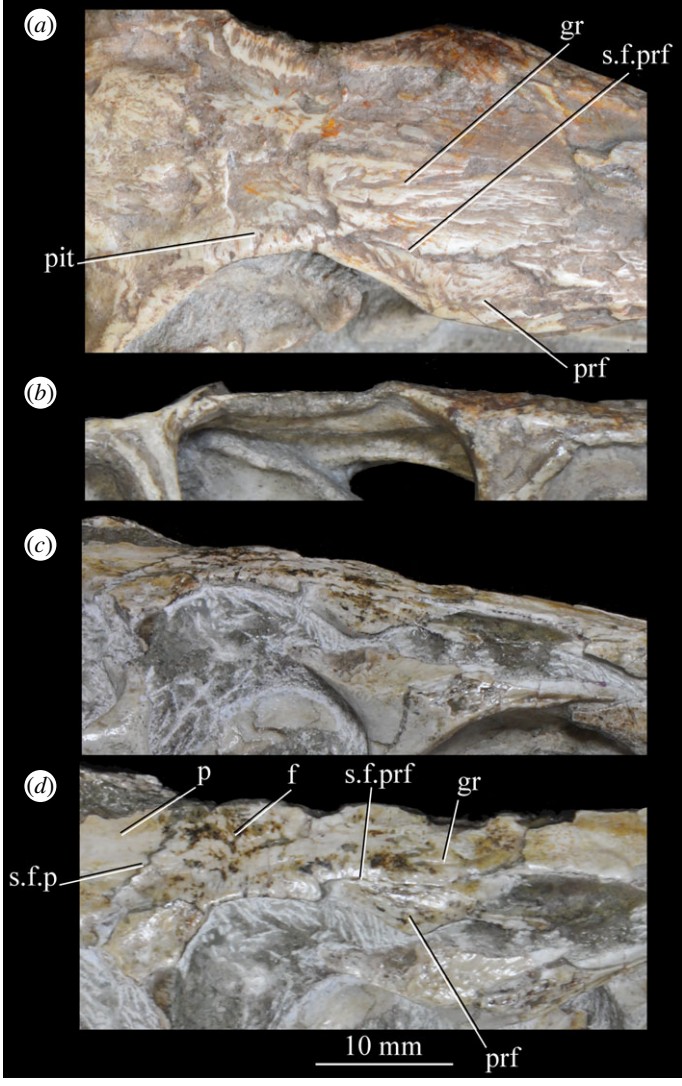

**Figure 31.** Examples and details of the frontal of *Euparkeria capensis*. Frontals of SAM-PK-5867 in (*a*) dorsal and (*b*) right lateral view and of SAM-PK-6047 in right lateral (*c*) and dorsal (*d*) view. f, frontal; gr, groove; p, parietal; pit, pit; prf, prefrontal; s., suture between. Scale below (*d*) applies to (*a*–*d*).

dorsal view, the suture curves laterally in a broad arc. The median contact between the nasals is a non-interdigitating suture along the midline.

### 3.2.3.4. *Frontal* (figure 31)

Both frontals are preserved in SAM-PK-5867 (figure 31*a,b*), in SAM-PK-6047A (figure 31*c,d*) though much of the left frontal is lost, SAM-PK-6048 (figure 10), SAM-PK-K8050 (figure 17), GPIT/RE/12913 (figure 7) and in individual A are preserved in UMZC T.692 (figures 18 and 19), and these specimens form the basis of the description. Specimens SAM-PK-13665, SAM-PK-13666 and SAM-PK-7699 also preserve examples of the frontals. The frontals of individual A are preserved in UMZC T.692. The frontals are anteroposteriorly elongated, subrectangular and form most of the skull roof above the orbits (they are excluded from the orbital margin anteriorly by the prefrontals and posteriorly by the postfrontals). The frontals are approximately equal in length to the nasals and longer than the parietals. Anteriorly, the frontals contact the nasals, posteriorly the parietals, anterolaterally the prefrontals and posterolaterally the postfrontals. In dorsal view, the frontals expand slightly mediolaterally anterior to their suture with the parietals to reach their maximum width at the point at which they start to contribute to the orbital margin (figure 4*c*). The frontals narrow anteriorly along the orbital margin, then more strongly when excluded from the orbital margin by the prefrontals. They do not narrow consistently towards the midline anteriorly as in some pseudosuchians (e.g.

*Batrachotomus* [75]), and are not as wide at their posterior ends relative to their anterior ends as in *Erythrosuchus* [81]. Contact with the postorbitals is prevented by parietal–postfrontal contact. The supratemporal fossa (see above) does not extend onto the frontals.

The dorsal surface of the frontals of SAM-PK-5867 (figure 31*a*) shows radiating grooves originating from near the centre of the posterior third of the bone (figure 31*a*, gr), which are also seen in *Chanaresuchus* [80]). In SAM-PK-6047A (figure 31*d*) ornamentation is absent except for two to three larger grooves extending posteriorly from the anterior margins of the bones (figure 31*d*, gr). This difference may be biological or preservational. There is no parietal fossa, unlike in, e.g. *Erythrosuchus* [81]. The dorsal surfaces of the frontals are flat to slightly convex with the exception of the posterolateral projections on the frontal between the postfrontals and the parietals, which are slightly concave. The lateral margins of the frontals above the orbits are rounded in cross-section and slightly thickened, with faint pitting (figure 31*a*, pit) just medial to the margin on the dorsal surface. There is no midline ridge or ridges as seen in some pseudosuchians [31].

Posteriorly, the frontals form a suture with the parietals extending posterolaterally from the midline to the lateral skull margin (figure 31*a,d*), with less than 10 short interdigitations (visible clearly in SAM-PK-6047A—figure 31*d*, s.f.p: maximum 1.4 mm, with frontal length at the midline of 16.5 mm, with eight interdigitations visible on the left-hand side and six on the right). There is no marked anterior projection of the parietals along the midline, *contra* the reconstructions of Ewer [54] and Nesbitt [31]; instead, the parietals simply taper gradually anteriorly towards the midline (figure 4*c* and 8*c*). The postfrontal is received by an anteroposteriorly elongated elliptical facet on the posterolateral part of the frontal, continuing onto the parietal (figure 19, f.pof). The medial frontal–postfrontal suture is non-interdigitating, curving posteriorly from the lateral skull margin.

Most of the frontal–prefrontal suture forms a shallow medially convex curve, continuing posteriorly from the nasal–prefrontal suture; there is no interdigitation excepting two small lateral meanders about half way along its length in SAM-PK-6047A (figure 31*d*, s.f.prf). In SAM-PK-6047A, the frontal–prefrontal suture bends sharply back on itself at its posteriormost point, and arcs from posteromedial to anterolateral briefly (with a posteriorly convex curvature) before reaching the orbital margin; in SAM-PK-5867, this change in direction appears less abrupt (figure 31*a*, s.f.prf), but the suture is unclear. The midline suture between the frontals is not clearly visible in any specimen, but based on SAM-PK-5867 and the slightly disarticulated frontals in one of the individuals in SAM-PK-8050 (figure 17*a,b*) appears to lack the unusual, pronounced sinuosity seen posteriorly in *Osmolskina* [74].

### 3.2.3.5. Parietal (figure 32)

Examples of the parietals are preserved in the holotype SAM-PK-5867 (figure 32*a,c*) and SAM-PK-6047A (figure 32*b*), and these form the basis of the description. Examples of the parietal are also preserved in SAM-PK-6048, SAM-PK-K8050, SAM-PK-13665, SAM-PK-13666, SAM-PK-7699, AMNH 2239, UMZC T.692 and GPIT/RE/12913. The parietals are unfused, and form the posterior of the skull table and the anteromedial, medial and posteromedial margins of the supratemporal fenestrae. Anteriorly the parietals contact the frontals, medially the parietals contact one another, and anterolaterally the parietals contact the medial margin of the postorbitals and the posteromedial margin of the postfrontals. Posteromedially, the parietals contact the interparietal, and posteriorly and posteroventrally, they contact the supraoccipital.

Posterolaterally, the parietals taper to form posterolateral processes (=posterolateral wings) which project at around 45° to the midline and curve slightly in a very shallow anterolaterally concave arc to form the posteromedial margins of the supratemporal fenestrae (figure 32, pl.p). The occipital margin formed by the parietals is thus V-shaped in dorsal view; this is similar to the morphology of *Osmolskina* [74] other Triassic non-archosaurian archosauromorphs such as *Mesosuchus* (SAM-PK-6536) or *Howesia browni* [105] and to some Triassic archosaurs (e.g. *Batrachotomus* [75]), but differs from the mediolaterally straight processes seen in some crocodylomorphs [106,107]. The distal tips of the posterolateral processes do, however, curve very slightly anteriorly, differing from *Osmolskina* [74] and *Mesosuchus* (SAM-PK-6536). The posterolateral processes contact the supraoccipital posteriorly and the squamosals at their tips, with their ventral margins contacting the plate-like portion of the interparietal (see below).

The dorsal surfaces of the parietals are flat with a slight mediolateral convexity anterolaterally and a slight mediolateral and anteroposterior concavity posteromedially. In SAM-PK-6047A, three grooves extend posteriorly from the posterior tips of interdigitations at the parietal–frontal suture (figure 32*b*, gr). In SAM-PK-5867, there are faint striations (figure 32*a*, str) radiating anteriorly and medially from roughly the centre of the dorsal surface of the main body of the bone. The dorsal surfaces of the

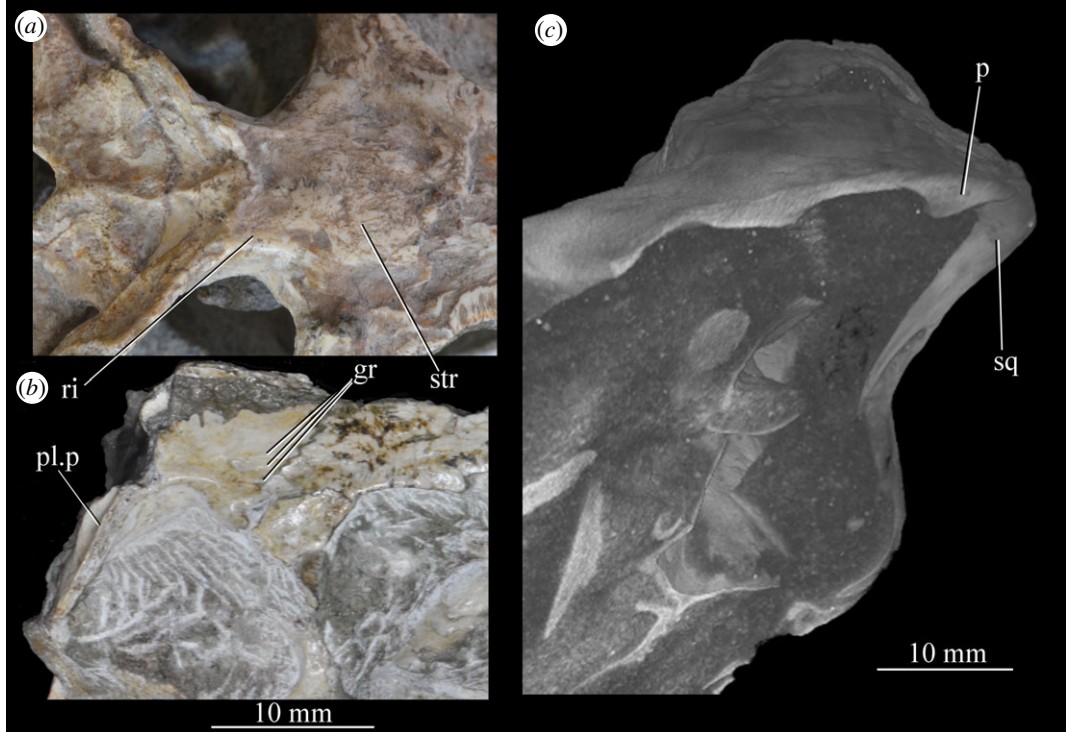

**Figure 32.** Examples and details of the articulation of the parietal of *Euparkeria capensis*. (*a*) Parietals of SAM-PK-5867 in dorsal view; (*b*) right parietal of SAM-PK-6047A in dorsal view; and (*c*) CT scan cross-section of SAM-PK-5867 in posterior view showing articulation of the parietal with squamosal. gr, groove; p, parietal; pl.p, posterolateral process; ri, ridge; sq, squamosal; str, striations.

posterolateral processes form a rounded ridge (figure 32*a*, ri) which continues along the lateral margin of the main body of the bone, forming the margin of the supratemporal fossa. No pineal fossa is present (unlike in *Erythrosuchus* [81] or *Proterosuchus fergusi* [83]), nor a pineal foramen (unlike in *Proterosuchus fergusi* [83] and *Prolacerta* [90]). Neither a supratemporal fossa *sensu* Nesbitt [31] (i.e. a single fossa centred at the midline separated from the supratemporal fenestra by raised strips of bone) nor sagittal ridge is present, unlike in some pseudosuchians [31].

The midline suture between the parietals is one of the most strongly interdigitating sutures on the skull (the longest interdigitation is around 20% of the minimum mediolateral width of the main body of the parietal in SAM-PK-5867; Ewer [54] does not show the full extent of this interdigitation). This contrasts strongly with *Osmolskina* and proterochampsids, where the suture lacks interdigitations when it is present [74,104,108]. The sutures with the postfrontal and postorbital are gently interdigitating. The suture with the interparietal is regularly, but gently interdigitating. The suture with the squamosal is non-interdigitating, but in cross-section, the parietal can be seen to be cupped ventrally and laterally by a dorsomedially open facet on the squamosal (figure 32*c*).

### 3.2.3.6. *Interparietal* (figure 33)

An interparietal (=postparietal) is present (unlike in pseudosuchians and avemetatarsalians [31], proterochampsids [104], *Prolacerta* [90] and *Mesosuchus* [109]) and is preserved in SAM-PK-5867 (figure 33; forming the basis of the description), SAM-PK-6048 and SAM-PK-K8050. It is a single, unpaired median element. Its anterior portion is roughly rhombus-shaped in dorsal view, similar to that of *Erythrosuchus* [81] (but differing from the more rounded interparietal of *Proterosuchus fergusi* [83]) tapering to a point anteriorly, posteriorly and laterally. The part of the rhombus anterior to the lateral corners is shorter anteroposteriorly than that posterior to them. Posteriorly, the interparietal widens laterally either side of the raised central rhomboid portion to form a broad plate which abuts the posteromedial edges of the parietal posterolateral processes. This plate is so distinct from the rest of the interparietal that it has been mistaken for part of the parietal, but CT data reveal no suture between this plate and the interparietal, but a clear suture between the plate and the parietals; although now part of the interparietal, this area may represent the embryonic tabular (figure 33, ?tab), based on the position of these bones in other taxa [110]. The interparietal has interdigitating sutures

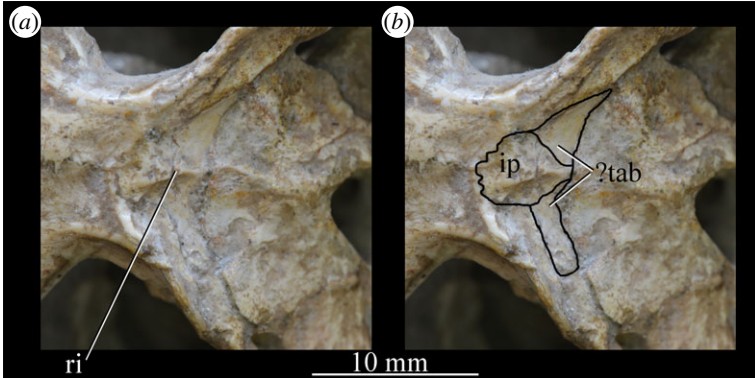

**Figure 33.** Interparietal of *Euparkeria capensis* specimen SAM-PK-5867 in dorsal view (*a*) with and (*b*) without boundaries of elements indicated. ?tab, possible embryonic tabulars; ip, interparietal; ri, ridge.

with the parietals anterolaterally on the margin of the rhomboid anterior portion and non-interdigitating sutures with posterolateral processes of the parietal and with the supraoccipital posteroventrolaterally.

Anteriorly, the dorsal surface is convex anteroposteriorly and mediolaterally. Posteriorly, the lateral margins of the bone are depressed ventrally, while the medial part forms a sharply defined posteriorly tapering ridge (figure 33, ri) with a flattened dorsal surface (not figured or described by Ewer [54]). The sides of this ridge are concave anteroposteriorly and mediolaterally.

### 3.2.4. Circumorbital series and anterior cheek

#### 3.2.4.1. *Prefrontal* (figure 34)

Both prefrontals are well preserved in SAM-PK-5867 (figure 34*a,b*), SAM-PK-6047A (figure 34*c,d*) and individual A in UMZC T.692 (figure 34*e*), and these form the basis of the description. Examples of the prefrontal are also preserved in SAM-PK-6050, SAM-PK-K8050, SAM-PK-13665, SAM-PK-7699, AMNH 2239 and GPIT/RE/12913. The prefrontal forms the anterodorsal orbital margin, and the skull table directly anteromedial to that point. The prefrontal is an anteroposteriorly elongated ellipse in dorsal view, forming the lateral margin of the skull roof. There is no ventromedial process *sensu* Gower & Walker [111], unlike in aetosaurs and crocodylomorphs, and the prefrontal does not contact the palate. A ventral ramus (figure 34, v.r) descends at its lateral margin, forming, along with the lacrimal, the dorsal portion of the antorbital bar. Medially, posteriorly and posteromedially the prefrontal contacts the frontal, anteriorly and anteromedially it contacts the nasal, and the anteroventral surface of the main body (figure 34, mb) of the bone and the anterior face of the ventral ramus contact the lacrimal.

Faint longitudinal striations on the dorsal surface of the prefrontal in SAM-PK-6047A and SAM-PK-5867 radiate from its lateral extreme (figure 34, str). The lateral margin of the bone is rounded in anterior/posterior view and slightly thickened dorsoventrally along the orbital margin. Unlike in *Erythrosuchus* [81] and *Garjainia prima* [112], where the prefrontal bulges laterally, there is no notable discontinuity in the skull margin between the frontal and prefrontal. The dorsal surface is slightly convex anteroposteriorly and mediolaterally. The ventral ramus of the prefrontal narrows anteroposteriorly as it extends ventrally. The posterior surface of the ventral ramus is convex (where it forms part of the internal margin of the orbit) and concave (forming the curved anterodorsal rim of the orbit) dorsoventrally. The lateral surface of the prefrontal is convex but becomes less strongly so anteriorly.

On the left-hand side of specimen A of UMZC T.692, foramina for the nasolacrimal canal are visible (figure 34*e*, for; see [98]). These foramina are visible externally as small anteriorly and laterally open notches extending mediolaterally across the anterior surface of the descending ramus of the prefrontal, situated one above the other. The lower notch is roughly the same size as the upper, and is separated from the ventral margin of the upper notch by a distance roughly equal to the combined height of both notches. From these foramina, the nasolacrimal canal extends medially and anteriorly, in a posterodorsally convex arc through the lacrimal as described below (under Lacrimal), and by Senter [98]. Only a single, dorsoventrally expanded notch appears to be present in SAM-PK-5867, but this may be an artefact of preservation. The entrance area is damaged in SAM-PK-6047A, preventing assessment.

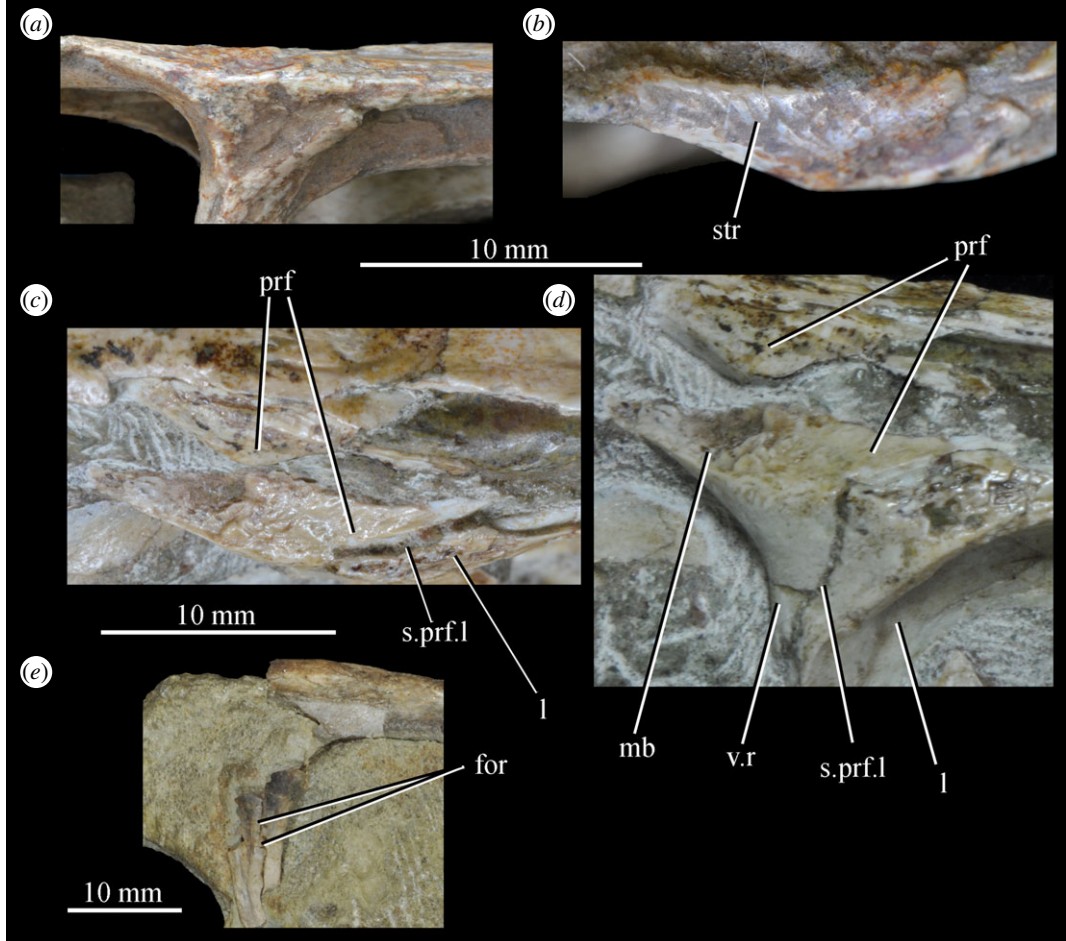

**Figure 34.** Examples and details of the prefrontal of *Euparkeria capensis*. Right prefrontal of SAM-PK-5867 in (*a*) lateral and (*b*) dorsal view; right prefrontal of SAM-PK-6047A in (*c*) dorsal and (*d*) lateral view; (*e*) partial left prefrontal of UMZC T.692 individual A in lateral view. for, foramina for nasolacrimal duct; l, lacrimal; mb, main body (of prefrontal); prf, prefrontal; s., suture between; str, striation; v.r, ventral ramus. The same scale (under images) applies to (*a*) and (*b*), and that under (*c*) applies to (*c*) and (*d*).

The dorsal suture of the prefrontal with the frontal and nasal meanders gently mediolaterally but is not truly interdigitating. The prefrontal–lacrimal suture is not interdigitating. In lateral view, this suture extends from posteroventral to anterodorsal (figure 34*d*, s.prf.l), from the ventral tip of the prefrontal up until the anteroposterior midpoint of the dorsal margin of the antorbital bar. Ventrally, the suture line is slightly posterodorsally convex in lateral view, while dorsally, it is posterodorsally concave. The anterior part of the main body of the prefrontal overlaps the medial part of the dorsal surface of the lacrimal, and the ventral ramus of the prefrontal is overlain laterally at its anterior margin by the ventral ramus of the lacrimal (figure 34*d*). Unlike in *Shansisuchus shansisuchus* [113,114], the prefrontal does not exclude the frontal from the orbital margin.

### 3.2.4.2. Postfrontal (figure 35)

Postfrontals are present (unlike in proterochampsids [80,104], Dinosauria [103], Crocodylomorpha [20], *Vancleavea campi* [28], *Effigia okeeffeae* [97] and *Shuvosaurus inexpectatus* [115]). Both postfrontals are preserved in SAM-PK-5867 (figure 35*a*,*b*), and a right postfrontal is preserved in UMZC T.692 (individual A; figure 35*d*) and in SAM-PK-6047A (figure 35*c*), and are the primary source for the description. SAM-PK-6048, SAM-PK-6050, SAM-PK-K8050, SAM-PK-13665, AMNH 2239 and GPIT/ RE/12913 also preserve examples of the postfrontal. The postfrontal is triangular in dorsal view (figure 35*a*–*d*). Its medial margin extends anteroposteriorly, contacting the frontal and excluding it from the orbital margin and forming a point contact with the parietal posteromedially. Its posterior margin extends mediolaterally contacting the postorbital. Its anterolateral (=orbital) margin connects the other margins and forms part of the orbital rim. The lateralmost corner is further from the

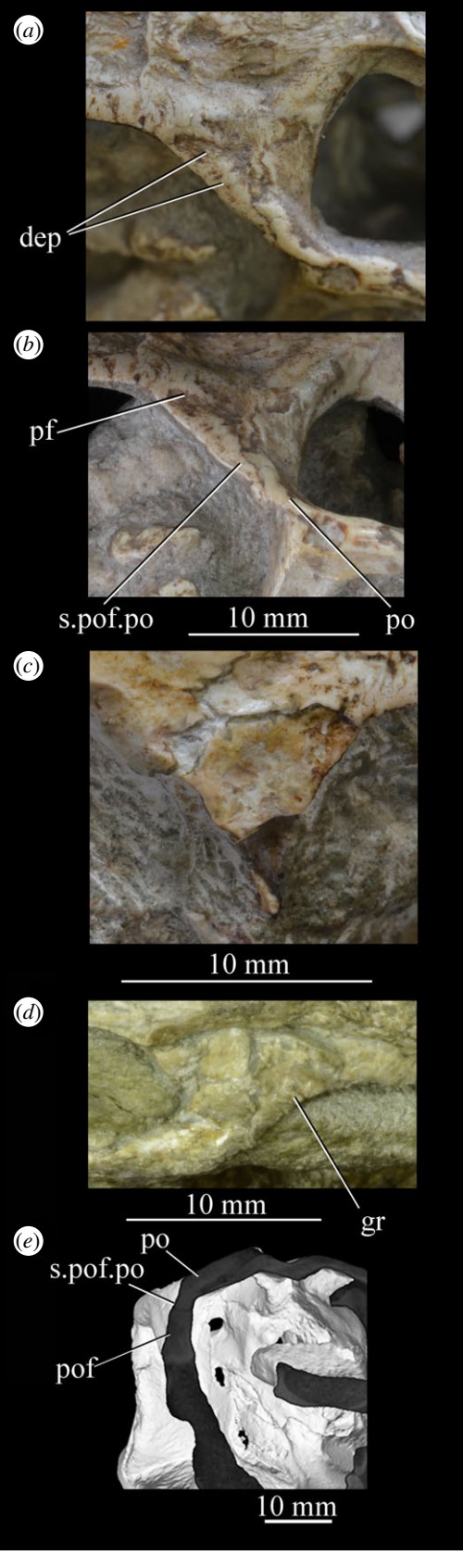

**Figure 35.** Examples and details of the postfrontal of *Euparkeria capensis*. Left postfrontal of SAM-PK-5867 in (*a*) dorsal and (*b*) dorsolateral view; (*c*) right postfrontal of SAM-PK-6047A in dorsolateral view; (*d*) right postfrontal of UMZC T.692 individual A in dorsal view; (*e*) CT image showing cross-section of the postfrontal–postorbital suture in right anterolateral view. dep, depressions; gr, groove; po, postorbital; pof, postfrontal; s., suture between. The upper scale applies to (*a,b*).

anterior or posterior corners than the anterior and posterior corners are from one another, and is deflected ventrally to form a thin flange that forms the anterodorsal part of the postorbital bar (the posterodorsal part of the bar being formed by the postorbital).

In SAM-PK-5867, a number of small (less than 5% of the size of the postfrontal in any direction) depressions are irregularly spaced across the external surface of the postfrontal (figure 35a, dep). In UMZC T.692, grooves (each with a width under 10% of the length of the orbital margin of the postfrontal) extend posteromedially from the orbital margin of the postfrontal, disappearing less than half way to the posterior/medial edges of the bone (figure 35d, gr). SAM-PK-6047A lacks these depressions and grooves. In SAM-PK-6047A, the centre of the dorsal surface of the postfrontal is slightly concave anteroposteriorly and mediolaterally. Around this central concave area, the surface is convex, forming a rim. In SAM-PK-5867 and UMZC T.692, the entire external surface of the bone is slightly convex or flat.

In lateral view, the postfrontal–postorbital suture extends ventrally, meandering slightly (figure 35b, s.pf.po). In cross-section, the postfrontal extends ventral to the postorbital and is concave posterodorsally, receiving the convex anteroventral surface of the postorbital (figure 35e).

### 3.2.4.3. Postorbital (figure 36)

Both postorbitals are preserved in SAM-PK-5867 (figures 4 and 36c,d) and in SAM-PK-6047A (right articulated—figure 8, left disarticulated and isolated—figure 36a,b) and a right postorbital is preserved in UMZC T.692 (individual A; figure 19), and are the primary descriptive sources. Examples of the postorbital are also preserved in SAM-PK-6048, SAM-PK-6050, SAM-PK-13666 and SAM-PK-K8050. The postorbital is composed of three rami which intersect to form a main body (figure 36, mb): a ventral ramus (the longest; figure 36, v.r), an anteromedial ramus (the second longest; figure 36, am.r) and a posterior ramus (the shortest; figure 36, p.r). The ventral ramus forms most of the dorsal portion of the postorbital bar excepting the small anterodorsal contribution by the postfrontal. It, therefore, forms over 80% of the posterior orbital margin, the contribution being greater than reconstructed by Ewer [54]. The anteromedial ramus forms a small part of the posterodorsal margin of the orbit before contacting the postfrontal, again *contra* Ewer [54], who figured the anteromedial ramus as excluded by the postfrontal from the orbital margin.

Further anteriorly, the anteromedial ramus of the postorbital curves medially posterior to the postfrontal (thus contacting the postfrontal along its anterior margin), forming the anterolateral margin of the supratemporal fenestra apart from the small contribution by the parietal ventromedially, and contacting the parietal medially. This medially curving section of the anteromedial ramus is anteroposteriorly thinner than dorsoventrally tall, i.e. it forms a (relatively thick) vertical sheet. The posterior ramus of the postorbital forms the anteroventral half of the upper temporal bar (= postorbital–squamosal bar) and contacts the squamosal posteriorly. The postorbital–squamosal suture (figure 36, s.po.sq) extends diagonally from anterodorsally to posteroventrally in lateral view, and thus, the posterior ramus of the postorbital forms over half of the dorsal margin of the infratemporal fenestra, and less than half of the lateral margin of the supratemporal fenestra (figures 4a and 36).

There is no clear ornamentation on the external surface of the postorbital, but the external surface of the main body is gently rugose. Most of the surface of the anteromedial ramus is dorsolaterally convex, but that adjacent to the supratemporal fenestra is depressed to form part of the supratemporal fossa. The postorbital lacks the bulges or bosses seen in erythrosuchids [81,112,116].

The posterior ramus of the postorbital is received by a dorsoventrally concave facet on the squamosal (see Squamosal), which tapers posteriorly and has a rounded tip, matching the form of the posterior ramus of the postorbital. The posteromedial surface of the ventral ramus of the postorbital articulates with the anterolateral surface of the dorsal process of the jugal. There is a concavity and a convexity on the facet on each element, thereby creating an S-shaped articulation in transverse section (figure 36d, s.j.po); the concavity on the jugal and convexity on the postorbital is lateral, and the convexity on the jugal and concavity on the postorbital is medial. The tip of the ventral ramus of the postorbital is tapered in comparison with the blunt end projecting into the orbit seen in some loricatans and theropods [31].

### 3.2.4.4. Jugal (figure 37)

Both jugals are preserved in SAM-PK-5867 (figures 4 and 37a) and SAM-PK-6047A (figures 8 and 37b,c), but the left jugal is incomplete in both specimens; the jugals are visible in lateral view, with the exception of the left jugal of SAM-PK-6047A which is isolated (figure 37c). The left jugal is visible in medial view in individual A of UMZC T.692 (figure 27), and this and the preceding specimens form the basis of this

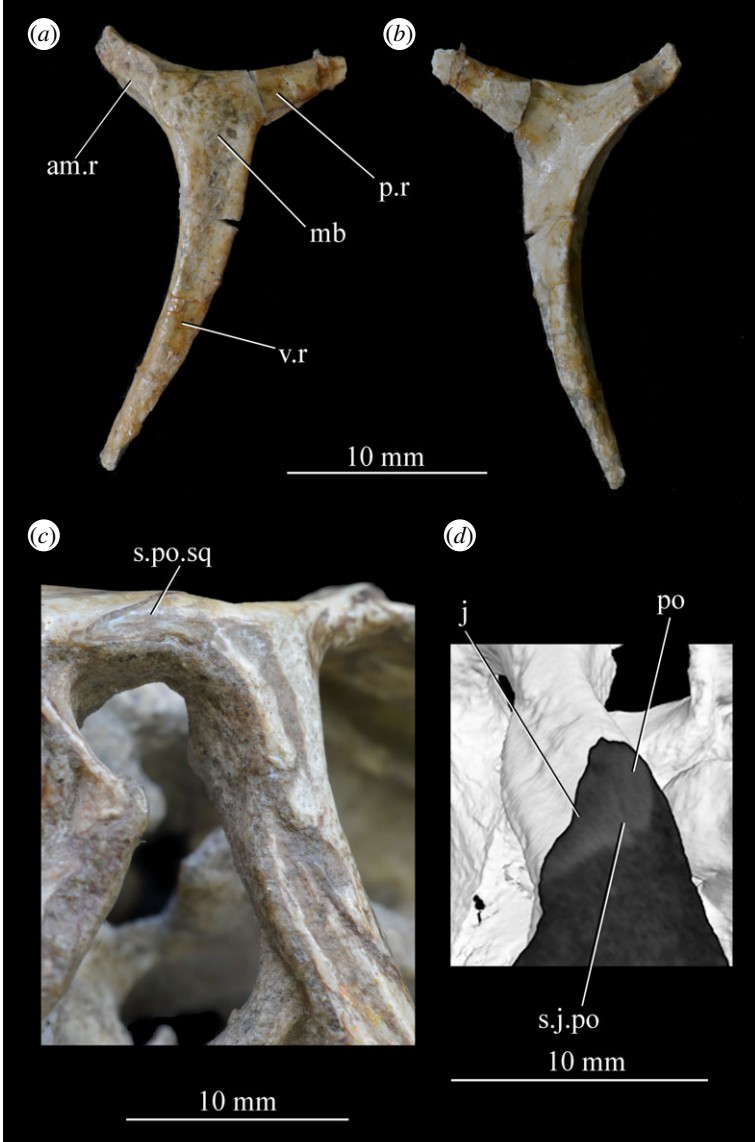

**Figure 36.** Examples and details of the postorbital of *Euparkeria capensis*. Left postorbital of SAM-PK-6047A in (*a*) lateral and (*b*) medial view; right postorbital of SAM-PK-5867 in lateral view (*c*); (*d*) CT image of SAM-PK-5867 showing cross-section through ventral ramus of the right postorbital (right ventrolateral view) to show articulation of postorbital with jugal. am.r, anteromedial ramus; j, jugal; mb, main body; po, postorbital; p.r, posterior ramus; s., suture between; sq, squamosal; v.r, ventral ramus. Top scale applies to (*a,b*).

description. Individual B of UMZC T.692 also includes a posterior left jugal, and examples of the jugal are also preserved in SAM-PK-6050, SAM-PK-13665, SAM-PK-13666, SAM-PK-13667 and SAM-PK-K8050. The jugal forms the ventrolateral margin of the skull between the maxilla, with which it sutures anteriorly, and the quadratojugal, with which it sutures posteriorly. The jugal forms the majority of the lower temporal bar and the entire ventral margin of the infratemporal fenestra, unlike in some theropods where the quadratojugal forms most of the bar [31]. The lower temporal bar of *Euparkeria* also differs from the bar of *Erythrosuchus* in that less of the bar is formed by the quadratojugal [81]. The bar is closed as in most Archosauriformes [31,117], but contrasting with many non-archosauriform archosauromorphs (e.g. *Proterosuchus fergusi*—SAM-PK-K10603; *Prolacerta* [90]) and lepidosauromorphs, with an open temporal bar probably ancestral for Sauria [117,118].

The main body of the jugal (figure 37, mb.j) is primarily horizontal, unlike in some aetosaurs and other crown taxa where it slopes down posteriorly [31,119,120]. Anteriorly, the dorsal margin of the jugal extends anterodorsally to suture with the lacrimal, forming the lower half of the antorbital bar and thus part of the posterior margin of the external antorbital fenestra (figure 20), unlike in some crown archosaurs where the jugal terminates posterior to the fenestra [31], but not extending as far anteriorly as in, e.g. *Erythrosuchus* [81] or *Chanaresuchus* [80]. The ventral margin of the jugal—and

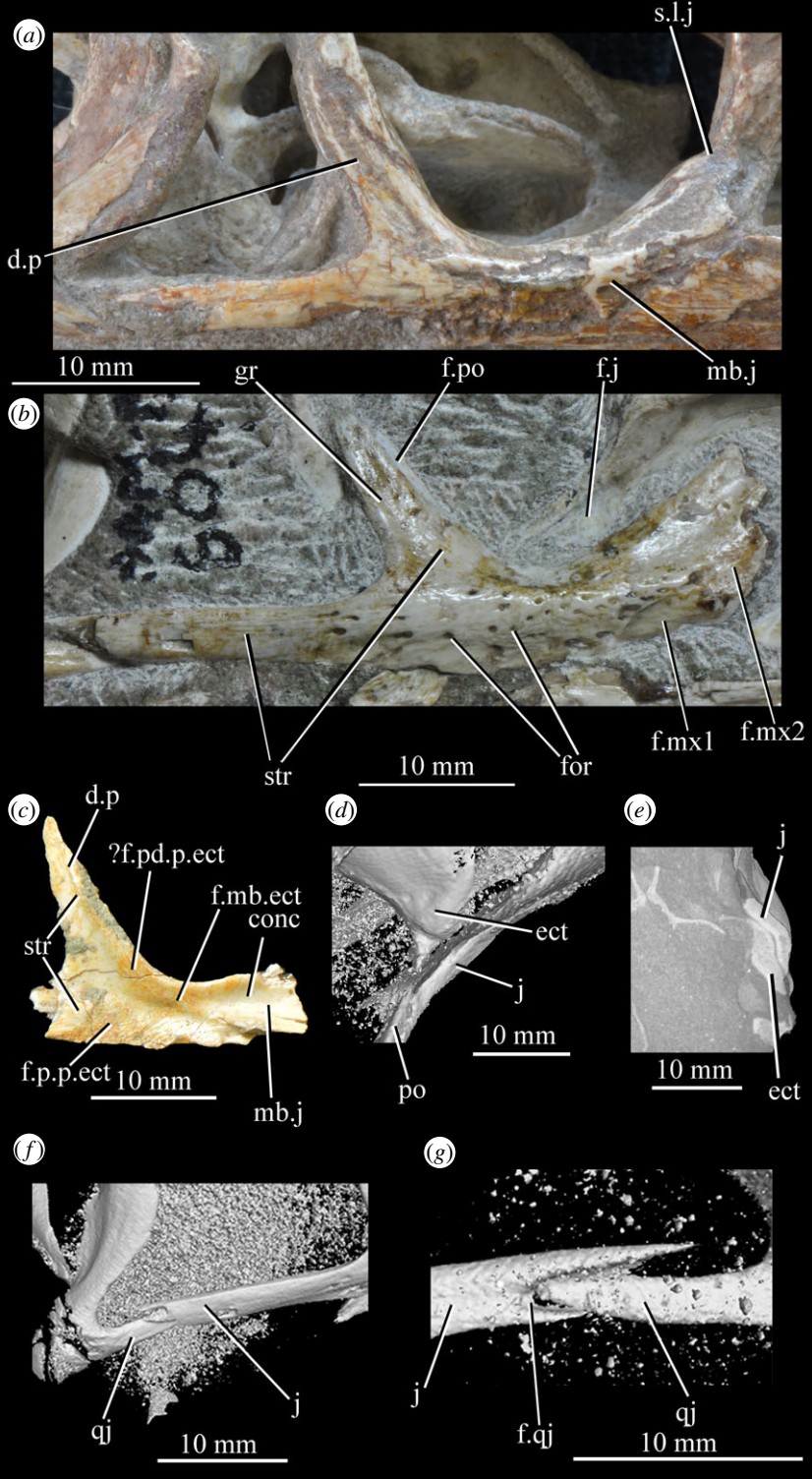

**Figure 37.** Examples and details of the jugal of *Euparkeria capensis*. Right jugal of (*a*) SAM-PK-5867 and (*b*) SAM-PK-6047A in lateral view; (*c*) left jugal of SAM-PK-5867 in medial view; (*d*) CT rendering showing articulation of right jugal with ectopterygoid in SAM-PK-5867 in dorsal view; (*e*) CT scan image showing contact between jugal and ectopterygoid on the right-hand side in SAM-PK-5867—plane is transverse, view posterior; CT surface renderings of SAM-PK-6047A showing right jugal–quadratojugal articulation in (*f*) right posterolateral and slightly dorsal view, and (*g*) medial view. ?, uncertainty; conc, concavity; d.p, dorsal process of the jugal; ect, ectopterygoid; f., facet for; for, foramen; gr, groove; j, jugal; l, lacrimal; mb.ect, main body/anterior part of the head of the ectopterygoid; mb.j, main body of the jugal; mx, maxilla; pd.p, posterodorsal process; po, postorbital; p.p.ect, posterior process of ectopterygoid; qj, quadratojugal; s., suture between; str, striations.

thus the jugal as a whole—also deflects dorsally towards its anterior end as it overlaps the maxilla. The dorsal process of the jugal (figure 37, d.p) arises from near the anteroposterior centre of the bone, and sutures dorsally with the ventral process of the postorbital, thereby forming the lower half of the postorbital bar (see Postorbital). Medially, the ectopterygoid sutures with the jugal, connecting the posterior of the palate to the side of the skull (figure 37d,e).

The lateral surface of the main body of the jugal is dorsoventrally and, less strongly, anteroposteriorly convex. This dorsoventral convexity has been characterized by Nesbitt [31] as a broad, rounded lateral ridge as in, for example, *Ornithosuchus* [76] or *Batrachotomus* [75], and homologous with the clearer, bulbous ridge of some other 'rauisuchians' (e.g. [121]) and the sharper ridge of, for example, proterochampsids [104]. The ventrolateral bulging of the jugal seen in erythrosuchids (e.g. [81]) may be homologous with this 'ridge', though not treated as such by Nesbitt [31]. Over 20 small foramina (largest foramen around 1 mm in diameter; figure 37, for) are present in a field reaching from just behind the dorsal process of the jugal to the base of the jugal contribution to the antorbital bar. These foramina become more numerous anteriorly. Posteriorly, the foramina open posteriorly, in the centre of the field they open laterally, and anteriorly they open anteriorly.

Striations (figure 37, str) extend anteroposteriorly along the posterior end of the lateral surface of the main body of the jugal. Most of these striations begin at the posterior margin of one of the more posterior foramina on the lateral surface. A field of irregular, anteroposterior striations about 3 mm long, with only small dorsoventral (less than 1 mm) separation between adjacent striations, is also present on the lower half of the main body of the jugal in SAM-PK-6047A, immediately anterior to the base of the dorsal process (figure 37b, str, right line). The lateral surface of the dorsal process of the jugal is in general anteroposteriorly convex, but a shallow (less than 1 mm deep in SAM-PK-6047A) dorsoventrally elongated groove extends along the process (figure 37b, gr), beginning around one-quarter of the way up the process from its ventral extremity. Ventrally, this groove is nearer the posterior than the anterior margin of the dorsal process, but the anterior and posterior margins of the process are equidistant from the groove dorsally, as the section of the dorsal process anterior to the groove narrows anteroposteriorly. The posterior margin of the dorsal process is very slightly convex (not markedly so as in some crocodylomorphs [31]).

The medial surface of the jugal is visible in SAM-PK-5867 on the left side (figure 37c), in UMZC T.692 individual A on the left side (figure 27, j), and in the CT scans of SAM-PK-5867 (figure 37d,e) and SAM-PK-6047A (figure 37f,g). Anteriorly, the medial surface of the main body is concave (figure 37c, conc). Posteriorly, it is flatter, with facets for the head of the ectopterygoid visible (see below). The medial surface of the dorsal process is slightly concave along its length. Faint striations extend up the dorsal process, and posterodorsally and posteroventrally from the facet for the main body of the ectopterygoid (figure 37c, str).

The jugal articulates with the maxilla via two facets (figure 37, f.mx1, f.mx2) at the anteroventral extremity of the lateral surface of the main body of the jugal contrasting with the single facet seen in, for example, *Osmolskina* [74]. The lower and more posterior of these facets has a posterodorsally convexly curved rim along its posterodorsal margin. The upper and more anterior of these facets is separated from the lower facet by a low ridge continuous with the central ridge of the main body of the jugal. The posterodorsal border of the upper facet is a low straight ridge that extends diagonally anterodorsally from the anterior extremity of the more ventral, convex area on the main body of the jugal. The lateral surface of the upper facet is flat, whereas that of the lower facet slopes from dorsolateral to ventromedial, but is flat anteroposteriorly. The jugal laterally overlaps the ventral extremity of the lacrimal and contacts the prefrontal dorsally. A slightly concave facet, expanded into a flange, on the anteroventral surface of the posteroventral extremity of the lacrimal (figure 37b,f,j), received the flat to the convex surface of the jugal.

The jugal articulates with the shallow facet on the postorbital via an anterolaterally opening, dorsoventrally elongated facet on the dorsal two-thirds of the anterolateral surface of the dorsal process (figure 37b, f.po; one-third to one-quarter of the width of the dorsal process of the jugal at its widest). The articulation between the jugal and quadratojugal consists of a deep elongated concave facet on the medial side of the jugal which received the lateral side of the anterior end of the quadratojugal (figure 37f,g, f.qj). The facet tapers anteriorly following the form of the quadratojugal. The jugal and ectopterygoid articulation is visible in the CT scans of SAM-PK-5867 (figure 37d,e) and SAM-PK-6047A, and the facets for the ectopterygoid are visible on the medial side of the isolated left jugal in SAM-PK-6047A (figure 37c, f.mb.ect, f.p.p.ect, ?f.pd.p.ect). The laterally convex anterior part of the head of the ectopterygoid was received by a simple, anteroposteriorly elongated, concavity on the medial side of the jugal (figure 37c, f.mb.ect). Posteriorly the ectopterygoid divided into posterior

and posterodorsal processes (see below). A striated facet for the posterior process, tapering posteriorly, is visible on the jugal (37*c*, f.p.p.ect). In UMZC T.692 individual A, a facet for the posterodorsal process of the ectopterygoid is present in the form of a rugose depression (figure 27, ?f.pd.p.ect) posterodorsal to the facet for the anterior part of the head; in SAM-PK-6047A, this facet is less clear, but may be represented by an area of striation (figure 37*c*, ?f.pd.p.ect).

### 3.2.4.5. Lacrimal (figure 38)

Both lacrimal are preserved in SAM-PK-5867 (figures 4 and 38*a*) and SAM-PK-6047A (figures 8 and 38*c*), and the right lacrimal is preserved in UMZC T.692 specimen A (figure 38*b*), and these form the basis of this description. Examples of the lacrimal are also preserved in AMNH 2239, GPIT/RE/12913, SAM-PK-6050, SAM-PK-13665, SAM-PK-13666 and SAM-PK-8050. The lacrimal is posterodorsally convexly curved along its length; it resembles capital $\Gamma$ with a rounded corner between the two rami. The lacrimal articulates posteroventrally with the jugal and posteriorly and posterodorsally with the prefrontal, forming together with these bones the antorbital bar separating the orbit from the antorbital fenestrae. Dorsally, the lacrimal articulates with the nasal and anterodorsally with the tip of the dorsal process of the maxilla. The lacrimal has two distinct lateral surfaces: a dorsolateral surface and a medially inset ventrolateral surface, separated from one another by a ridge (figure 38, dl.sr, vl.sr, ri).

The anterior part of the ventrolateral surface (figure 38, vl.sr) of the lacrimal forms most of the dorsal border of the internal antorbital fenestra and all of the antorbital fossa except for the contribution by the tip of the ascending process of the maxilla. This configuration is similar in most archosauriforms except erythrosuchids [81,122], some 'rauisuchians' [121] and theropods [44], where the maxilla forms most of the dorsal margin of the fenestra and thus the fossa. The lacrimal does not overhang a pocket/fossa laterally as in basal saurischians [103]. The posterior and ventral parts of the ventrolateral surface of the lacrimal form most of the ventral half of the antorbital bar, differing from most other archosauriforms where the lacrimal usually forms the majority of the antorbital bar (e.g. figs. 11–13 in Nesbitt [31]). The dorsolateral surface of the lacrimal (figure 38, dl.sr) forms the posterior two-thirds of the dorsal margin of the external antorbital fenestra, the anterodorsal part of the lateral surface of the antorbital bar and the small lacrimal contribution to the edge of the skull roof.

The ventrolateral surface of the lacrimal is concave. The dorsolateral surface of the lacrimal is slightly convex in all directions. The ventrolateral surface lacks ornamentation, but the dorsolateral surface has posteroventrally to anterodorsally oriented striations (figure 38*c*, str), and small (around 0.1 mm in diameter) pits (figure 38*c*, pit). On the right-hand side of individual A of UMZC T.692, the path of the nasolacrimal duct within the lacrimal is exposed (figure 38*b*, nl.ca), as described by Senter [98]; the path can also be seen in the CT scans of SAM-PK-5867 and SAM-PK-6047A (figure 38*e,f*). The duct curves dorsally and anteriorly from the entrance to the duct at the contact between the ventral ramus of the prefrontal and the lacrimal in a posterodorsally convex arc (following the lacrimal as a whole). The dorsoventral height of the duct is around 10% of the total dorsoventral height of the lacrimal.

The lacrimal–jugal suture (figure 37*a*, s.l.j)—placed dorsal to the ventral margin of the orbit, as in most non-saurischian archosauromorphs [31,44]—consists of the anterodorsal end of the jugal overlying the flat to the concave expanded ventral part of the lateral surface of the anteroventral extreme of the lacrimal (figure 37*b*, f.j; see Jugal). In lateral view, the suture with the prefrontal does not interdigitate and extends posteroventrally to anterodorsally up the antorbital bar (see Prefrontal). In dorsal view, the suture of the prefrontal and lacrimal (figure 34, s.prf.l) is non-interdigitating, extending posterolaterally to anteromedially with an angle of around 20° to directly mediolateral. The lacrimal–nasal suture consists of an anteroposteriorly elongated and depressed rounded facet (figure 29*e*, f.n) on the anterodorsal part of the lacrimal, receiving the lateral part of the nasal. The facet is separated from the lateral skull margin by a thin strip of non-depressed bone. Based on CT data, the lacrimal–maxilla articulation consists of the anterior end of the lacrimal projecting medial to the dorsal process of the maxilla, and more posteriorly very briefly contacting the dorsal process before the lacrimal replaces the maxilla at the lateral side of the skull more posteriorly and becomes exposed laterally (figure 38*d*).

## 3.2.5. Posterior cheek

### 3.2.5.1. Squamosal (figure 39)

Both squamosals are preserved in SAM-PK-5867 (figure 39*a–c*) and the right is preserved in SAM-PK-6047A (figure 39*d,e*), and these form the basis of this description. Several squamosals are also

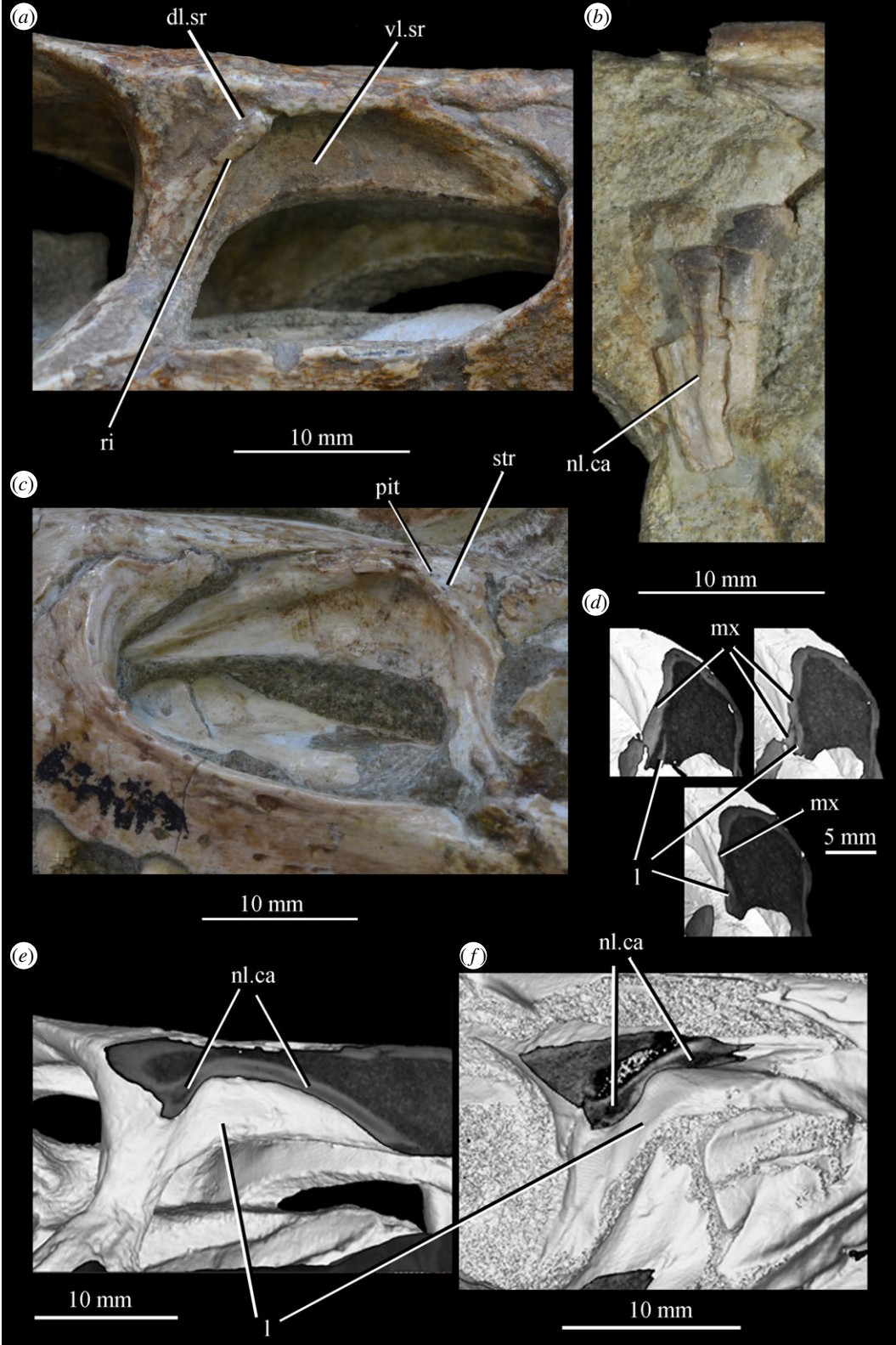

**Figure 38.** Examples and details of the lacrimal of *Euparkeria capensis*. (*a*) Right lacrimal of SAM-PK-5867 in lateral view; (*b*) left lacrimal of UMZC T.692 in lateral view; (*c*) left lacrimal of SAM-PK-6047A in lateral view; (*d*) CT renderings of SAM-PK-5867 in right anterolateral view, with coronal section showing lacrimal–maxilla articulation—top left section most posterior, and bottom most anterior; CT renderings of the right-hand side of (*e*) SAM-PK-5867 and (*f*) SAM-PK-6047A with cross-section in the sagittal plane to show the path of the nasolacrimal canal. dl.sr, dorsolateral surface; l, lacrimal; mx, maxilla; nl.ca, nasolacrimal canal; pit, pit; ri, ridge; str, striations; vl.sr, ventrolateral surface.

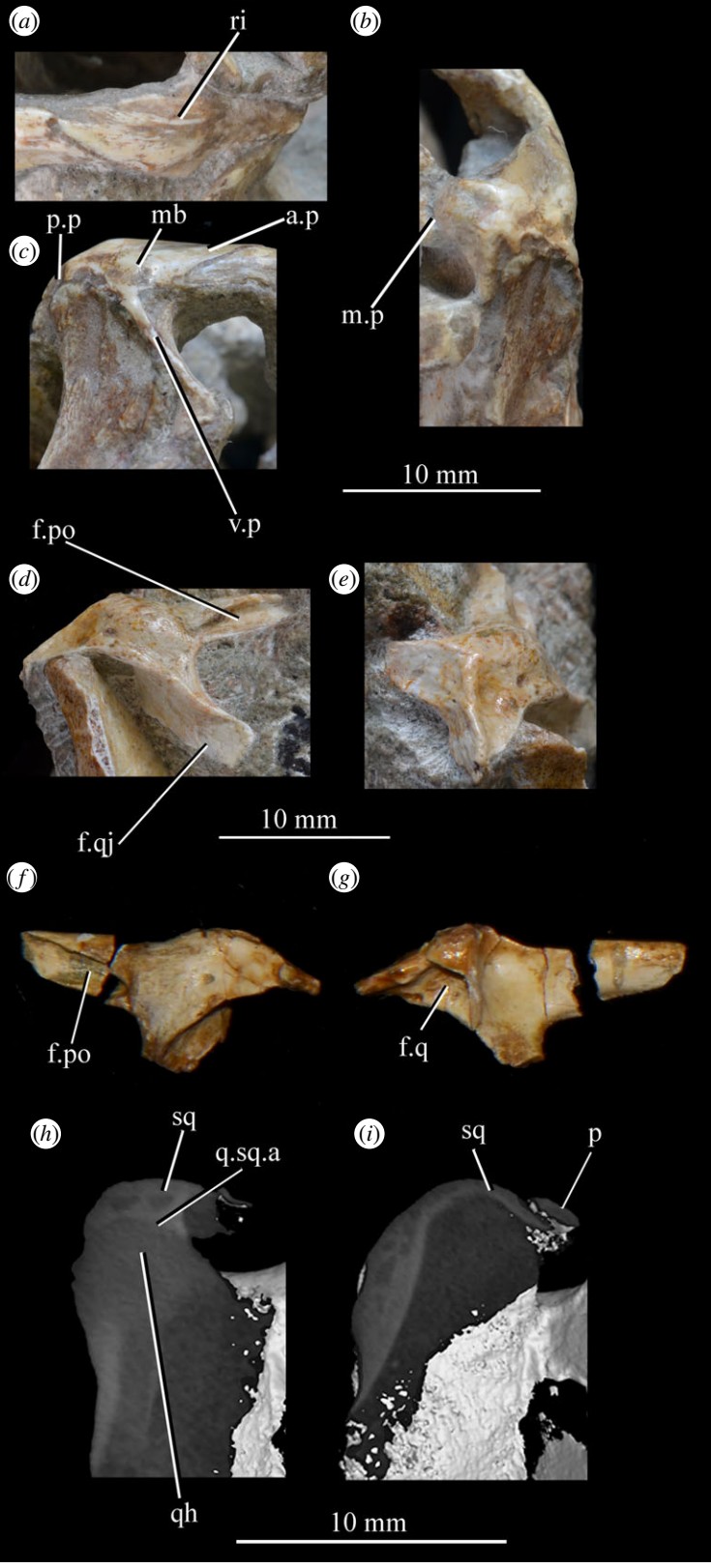

**Figure 39.** Examples and details of the squamosal of *Euparkeria capensis*. Left squamosal of SAM-PK-5867 in (*a*) dorsal view; right squamosal of SAM-PK-5867 in (*b*) posterodorsal and (*c*) lateral view; right squamosal of SAM-PK-6047A in (*d*) lateral and (*e*) posterior view; left squamosal of SAM-PK-6047A in (*f*) lateral and (*g*) medial view; CT image of SAM-PK-5867 showing cross-section in the sagittal plane showing dorsal quadrate–squamosal (*h*) and squamosal–parietal (*i*) articulation. a.p, anterior process; f., facet for; mb, main body of squamosal; m.p, medial process; p, parietal; p.p, posterior process; po, postorbital; q, quadrate; qh, quadrate head; qj; quadratojugal; q.sq.a, quadrate–squamosal articulation; ri, ridge; sq, squamosal; v.p, ventral process. Upper scale applies to (*a–c*), middle scale to (*d–g*), lower scale to (*h,i*).

preserved in SAM-PK-K8050, and one is probably preserved in SAM-PK-13665. The squamosal forms the dorsolateral extremity of the lateral surface of the skull, the posterolateral margin of the supratemporal fenestra and the dorsal half of the posterior margin and the posteriormost part of the dorsal margin of the infratemporal fenestra. The rhombus-shaped main body of the squamosal (figure 39, mb) has distinct anterior, posterior, ventral and dorsal corners and faces dorsolaterally. A process extends from each of the corners of the main body (figure 39, a.p, m.p, p.p, v.p). The anterior process (figure 39, a.p) articulates with the postorbital to form the upper temporal bar. In dorsal view, the squamosal contribution to the upper temporal bar is much less than the width of the supratemporal fenestra, as in most archosauriforms excepting some 'sphenosuchians' and *Orthosuchus stormbergi* [31].

The posterior process (figure 39, p.p) is ventromedially deflected, articulating along its ventral margin with the quadrate head. This process forms the lateral margin of the posttemporal fenestra. The ventral process (figure 39, v.p) extends anteroventrally to articulate with the quadratojugal (with which it forms the posterior margin of the infratemporal fenestra) and with the quadrate posteriorly and medially. The ventral process lacks an anterior process, unlike in some pseudosuchians [31]. The ventral process is thicker than one-quarter of its length, as in most archosauriforms except sauropodomorphs, basal theropods [31] and *Silesaurus opolensis* [123], but it is not as thickened as in erythrosuchids (where it is thickened to form a pleat—[81,122]) or proterosuchids [83]. The medial process (extending from the dorsal corner of the rhombus; figure 39, m.p) is deflected medially to articulate with the tip of the posterolateral process of the parietal and thus form the dorsal margin of the posttemporal fenestra.

The external surface of the squamosal is convex apart from the lateral surface of the posterior process, which is concave. The squamosal lacks ornamentation. A ridge (clearly visible on the left side of SAM-PK-5867—figure 39a, ri) extends from the base of the posterior process to the tip of the anterior process, shifting first laterally and then medially toward its anterior end, following the convexity of the squamosal. The width of the ridge is around 10% of the maximum width of the squamosal posteriorly and widens to around 20% of the maximum width of the squamosal anteriorly. The ridge becomes lower anteriorly, before disappearing. There is no pronounced laterally expanded ridge, however, as seen in, e.g. *Polonosuchus silesiacus* [121] or *Batrachotomus* [75]. There is no ridge on the ventral process, unlike in, e.g. *Polonosuchus* [121]. There are no foramina or fossae unlike the deep fossa in *Polonosuchus* [121] and *Postosuchus kirkpatricki* [31], nor is there a groove on the dorsolateral edge, unlike in some crocodylomorphs [31]. The supratemporal fossa does not extend onto the squamosal, unlike in basal crocodylomorphs and proterochampsids [31].

The squamosal–postorbital articulation consists of an anteroposteriorly elongated facet on the lower part of the lateral surface of the anterior process of the squamosal (figure 39d,f, f.po), which receives the posterior process of the postorbital. The facet opens laterally and anteriorly, and tapers posteriorly (the dorsal margin of the facet slopes anterodorsally to posteroventrally, whereas the ventral margin of the facet is approximately horizontal) to a rounded tip. The articulation between the ventral process and the quadratojugal consists of the ventral end of the ventral process, which expands ventrally into a mediolaterally broad, posteriorly concave plate (figure 39, f.qj), receiving the anteromedial extremity of the equally mediolaterally broadened dorsal tip of the anterodorsal process of the quadratojugal. As preserved in SAM-PK-5867, the tip of the ventral process of the squamosal is roughly level with the anteroposterior centre of the ventral margin of the infratemporal fenestra, but would probably have been slightly further posterior in life because the squamosal and quadratojugal are slightly disarticulated. In SAM-PK-5867, the anteroposterior width of the infratemporal fenestra anterior to the tip of the ventral process of the squamosal (the narrowest point of the infratemporal fenestra) is roughly one-quarter of the ventral anteroposterior width of the infratemporal fenestra (the widest point of the infratemporal fenestra), though again this would have probably been slightly wider in life.

The articulation between the ventral process of the squamosal and the quadrate occurs via the mediolaterally expanded plate-like head of the quadrate overlying the equally expanded posteromedial surface of the ventral process of the squamosal. The articulation between the posterior process of the squamosal and the quadrate consists of the concave surface of the squamosal (figure 39g, f.q) cupping the convex head of the quadrate medially, laterally and dorsally; the two bones are firmly appressed, and the squamosal can be seen to closely follow the contours of the quadrate, though there are no true interdigitations (figure 39h). The dorsal margin of the squamosal protrudes slightly over the margin of the quadrate in posterior view. The distal tip of the posterior process does not contact or cup the quadrate, and rather simply projects just posterior to it. In lateral view, the posterior end of the squamosal extends posterior to the head of the quadrate, unlike in non-proterchampsid non-archosaurian archosauriforms excluding *Vancleavea* [28,31]. The tip of the posterolateral process of the parietal was cupped posteriorly and dorsally by the medial process of the squamosal (figure 39i), with the posteromedial surface of the squamosal concave and the anterolateral

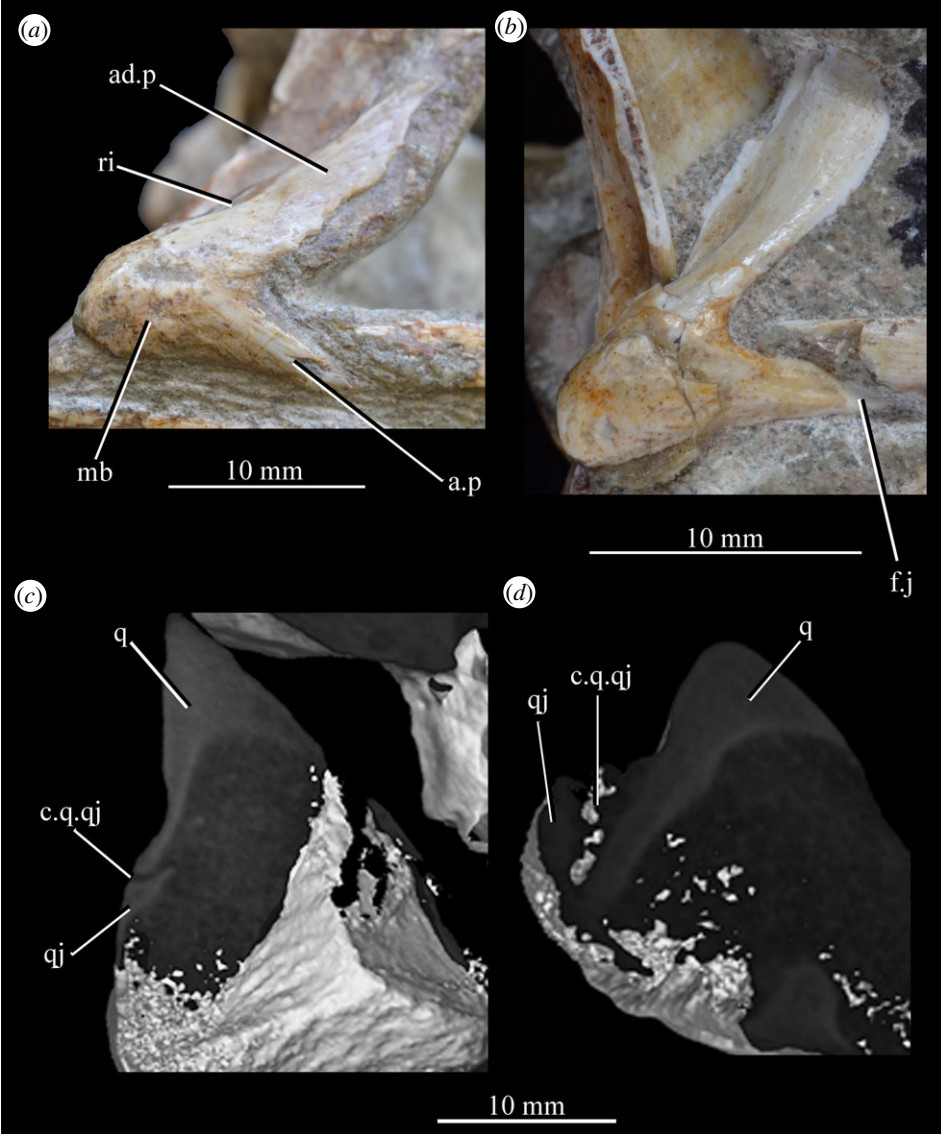

**Figure 40.** Examples and details of the quadratojugal of *Euparkeria capensis*. Right quadratojugal of (*a*) SAM-PK-5867 and (*b*) SAM-PK-6047A in lateral view; CT images showing upper (*c*) and lower (*d*) quadratojugal–quadrate articulations, with cross-section in the approximately transverse plane. ad.p, anterodorsal process; a.p, anterior process; c., contact between; f.j, facet for jugal; mb, main body; q, quadrate; qj, quadratojugal; ri, ridge.

surface of the process of the parietal slightly convex; there is no real facet on the squamosal, however, with the bones simply closely appressed.

### 3.2.5.2. Quadratojugal (figure 40)

Both quadratojugals are preserved in SAM-PK-5867 (figures 4 and 40*a*) and on the right side in SAM-PK-6047A (figure 40*b*), and these form the basis of the description. Quadratojugals are also preserved in SAM-PK-13665, SAM-PK-13666 and probably SAM-PK-K8050. The quadratojugal forms the posterolateral corner of the skull in lateral view. The quadratojugal has an anterodorsal process (figure 40, ad.p) extending dorsally and anteriorly (at around 40° to the horizontal) to articulate with the ventral process of the squamosal and form most of the posterior margin of the infratemporal fenestra (unlike in some pseudosuchians where it forms under 80% of this margin [31]), and an anterior process (figure 40, a.p) that articulates with the medial side of the jugal (see Jugal) and forms the ventral margin of the infratemporal fenestra (the lower temporal bar). The quadratojugal is overall shaped roughly like an arrowhead or boomerang in lateral view, with the point of the arrow

posteroventral; this is described as L-shaped by Nesbitt [31], and is the typical condition for archosauriforms with a closed lower temporal bar, except in phytosaurs where it is subtriangular (i.e. the processes are extremely shortened) [31].

The anterodorsal process is wider than the anterior process, forming an anterolaterally to posteromedially broad plate, tapering distally. The anterior process is thick mediolaterally (not platelike) and tapers dorsoventrally in lateral view. The lateral flange of the quadrate contacts the posteromedial part of the anterodorsal process of the quadratojugal. The lateral margin of the quadrate also contacts the medial surface of the main body of the quadratojugal immediately dorsal to the lateral quadrate condyle. These two contacts are separated by the quadrate foramen.

The external surface of the main body of the quadratojugal (figure 40, mb; i.e. the posterolateralmost extremity of the element in lateral view) is convex in all directions, and the external surface of the anterior process is dorsolaterally convex in posterior view. There is no lateral temporal fossa marked by ridges, unlike in proterochampsids [28,104], nor bulbous, or boss-like expansions as in erythrosuchids [81,116]. A central ridge (figure 40*a*, ri) extends up the external surface of the anterodorsal process, dividing it into a more medial, dorsally facing, and more lateral, laterally facing, section. The quadratojugal lacks ornamentation.

The articulation of the anterior process of the quadratojugal with the jugal consists of the tapering anterior process contacting the posteroventral surface of the posterior process of the jugal and overlapping it medially to be received by a facet (see Jugal), and thereby forming the ventral margin of the posterior part of the temporal bar. There is also a small facet on the quadratojugal itself, consisting of an embayed region on the anterodorsal part of the anterior process, which received a convexity of the jugal (figure 40*b*, f.j). This is similar to the arrangement in proterochampsids [80] and erythrosuchids [81] (though the jugal tapers to a process posteriorly), but differs from proterosuchids, where the quadratojugal overlaps the jugal laterally [83] in specimens with closed lower temporal bars, and the ventral margin of the temporal bar in lateral view is thus formed by the jugal. The quadratojugal–squamosal articulation consists of the broad tip of the anterodorsal process of the quadratojugal being received by a ventrolaterally expanded, posterolaterally concave, flange at the ventral tip of the ventral process of the squamosal (figure 39*d*). The upper quadratojugal–quadrate articulation is loose, with the expanded lateral flange of the quadrate overlying the external surface of the anterodorsal process on the medial side of the central ridge from just under two-thirds of the way up the anterodorsal process. The anterior surface of the flange of the quadrate is convex in the transverse section and the surface of the quadratojugal which it overlaps is concave (figure 40*c*). The lower quadratojugal–quadrate articulation consists of a laterally projecting flange of the quadrate, posterolaterally slightly convex in the transverse section, which articulates with a slightly concave surface on the anteromedial surface of the quadratojugal; the bones are not firmly appressed (figure 40*d*).

### 3.2.6. Palatal complex

#### 3.2.6.1. *Vomer* (figures 25 and 41)

The entire posterior part of the left vomer and most of the right are visible in SAM-PK-5867 in dorsal view through the antorbital fenestra (figure 41*c*). Left and right vomers are exposed in dorsal and ventral view in SAM-PK-6050 and were described by Gow [124] (figure 41*a,b*). The aforementioned specimens are used in this description. UMZC T.692 (individual A) preserves the anterior palate including vomers, and SAM-PK-13665 and AMNH 2239 preserve some of the anterior palate exposed, but details are hard to discern. The vomers of SAM-PK-5867 and SAM-PK-6047A can be seen in all views in the CT scans, though resolution means details of their form remain uncertain. The vomer is an anteroposteriorly elongated element forming the thin strip of palate between the anterior end of the palatine and the skull midline, as well as the palate anterior to the pterygoid and palatine and posterior to the maxillary and premaxillary contributions (which the vomers overlie). The vomer thus formed around half of the medial border of the choana. The vomers are mediolaterally thin anteriorly, widening posteriorly to a maximum width immediately anterior to their contacts with the palatines. Posterior to this they narrow slightly, but not to the degree figured by Gow [124].

The vomers contact the pterygoids posteriorly at their posterolateral tips, but appear to exclude the pterygoid from the midline at its anterior tip (rather than vice versa as figured by Gow [124]). Posterolaterally, the vomers contacted the palatines in an extensive, non-interdigitating suture. The vomers contact the dorsal surface of the palatal processes of the maxillae in a simple contact with the vomers overlying the maxillae. The vomers extended anteriorly to contact the small palatal

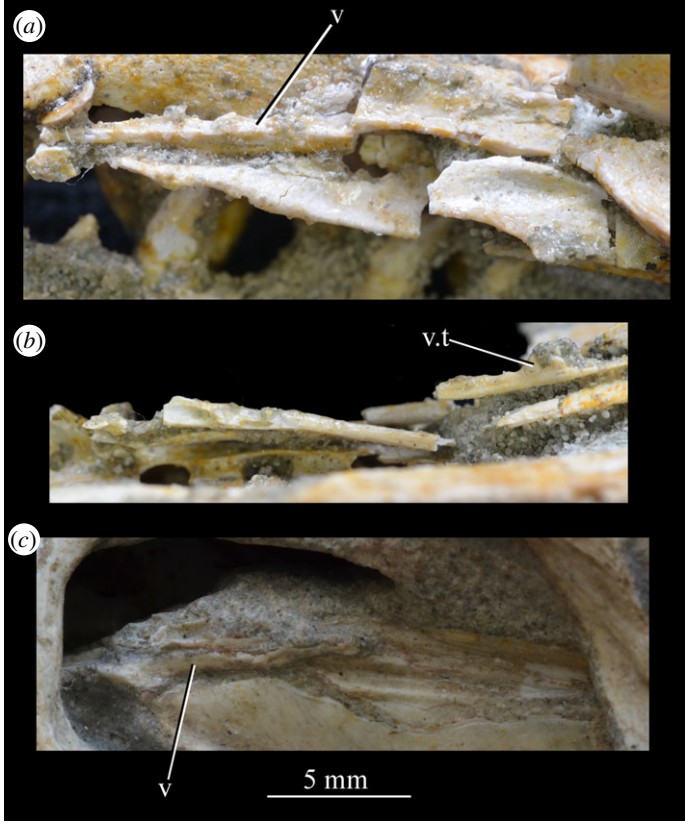

**Figure 41.** Examples of the vomer of *Euparkeria capensis*. Vomer of SAM-PK-6050 in (*a*) dorsal and (*b*) ventral view; (*c*) vomers of SAM-PK-5867 in dorsal view. v, vomer; v.t, vomerine teeth. The same scale applies to (*a–c*).

contribution of the premaxillae, as hypothesized by Gow [124] and as is also the case in *Chanaresuchus* (though not in *Ornithosuchus*, where the vomers end on the maxillae—Walker [76]). The contact is again simple, with the vomers appressed to the dorsal surface of the premaxillae and extending between the anterodorsal processes of the premaxillae at the midline.

The dorsal surface of the vomer is flat, with no ornamentation or foramina. The ventral surface of the vomer was figured by Gow ([124], fig. 1E) as bearing two pointed, slightly recurved, teeth situated just anterior to the start of the articulation with the pterygoid and palatine, at the transversely widest section of the vomer. These structures, which do appear to be teeth, can be identified on the left vomer only in SAM-PK-6050 (figure 41*b*); structures in a similar position can be identified in the CT scans of SAM-PK-6047A (figure 25), but no teeth can be observed in SAM-PK-5867, probably due to preservation and resolution. It can be concluded that *Euparkeria* has at least two vomerine teeth; fewer teeth than seen in *Prolacerta* (two tooth rows—[125]), proterosuchids (seven—[126]) and proterochampsids (14 in *Chanaresuchus*—[80]) but more than in the edentulous vomer of *Garjainia prima* [127] or crown archosaurs (e.g. [44,121]). The teeth are simple, pointed structures and are not recurved. They are elliptical in cross-section, slightly longer anteroposteriorly than mediolaterally, and—like the other palatal teeth—they appear to extend directly from the bone with no notable sockets.

### 3.2.6.2. *Palatine* (figure 42)

The palatine is best seen in SAM-PK-13664 (right-hand side in ventral view; figure 42*a*), and can be seen on both sides in dorsal view in SAM-PK-5867 (figure 42*b*), SAM-PK-6047A (figure 42*c*) and SAM-PK-6050 (figure 11*b*); these form the primary basis of this description. UMZC T.692 (individual A) preserves partial palatines and palatines are also preserved in SAM-PK-13665 and SAM-PK-13666. The palatine forms the palate posterior to the vomers and anterolateral to the pterygoids. The palatine is an anteroposteriorly elongated element, wider than the vomer mediolaterally but around the same width as the pterygoid. The palatine contacts the vomer anteromedially, the pterygoid posteriorly and posteromedially (no palatine–pterygoid fenestra is present, unlike in ornithosuchids [22,76]), and the maxilla laterally. The palatine does not contact the ectopterygoid posteromedially (*contra* Ewer [54]);

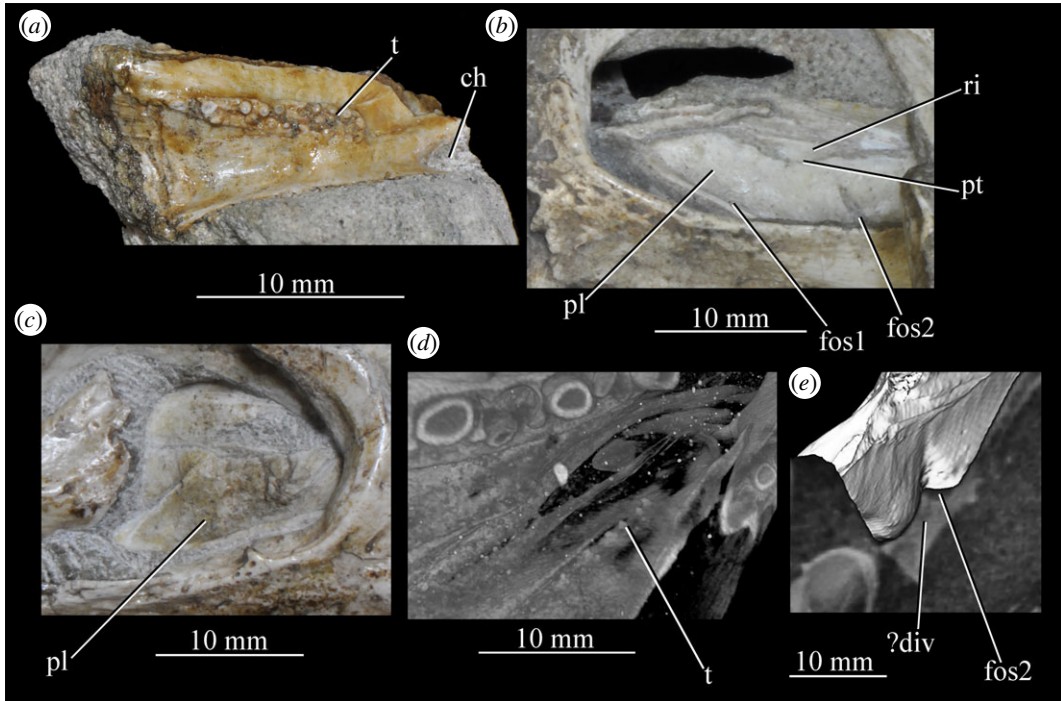

**Figure 42.** Examples and details of the palatine of *Euparkeria capensis*. (*a*) Left palatine of SAM-PK-13664 in ventral view; (*b*) left palatine of SAM-PK-5867 in laterodorsal view; (*c*) right palatine of SAM-PK-6047A in laterodorsal view; (*d*) CT image of palatines of SAM-PK-5867 in right posteroventral view; and (*e*) CT image of left palatine of SAM-PK-5867 in posterior and slightly dorsolateral view, with cross-section showing posterior, narrow fossa and potential diverticulum into bone. ?div, potential diverticulum; ch, choana; fos1, anterior, broad fossa; fos2, posterior, narrow fossa; pl, palatine; pt, pterygoid; ri, ridge; t, tooth.

this contact is also absent in other stem and crown archosaurs (*Proterosuchus fergusi* [126], *Prolacerta* [125], *Chanaresuchus* [80], *Garjainia prima* [127], *Herrerasaurus ischigualastensis* [77], crocodylomorphs [128]). The anterior margin of the palatine is gently anteriorly concavely curved to form the posterior margin of the choana. Most of the posterior half of the medial margin of the choana is formed by the palatine.

The palatine shows numerous (more than 20) small (largest around 0.5 mm in diameter), pointed and slightly recurved (their tips are damaged in SAM-PK-13664, but are visible in CT scans of SAM-PK-5867 and SAM-PK-6047A) palatal teeth (figure 42, t) on a raised, anteroposteriorly elongated elliptical area of the ventral surface roughly at the mediolateral centre of the bone. The teeth are arranged randomly. This is similar to the irregular field present in *Osmolskina* [74], but differs from the narrower, more regular tooth field of *Proterosuchus fergusi* [126] and *Chanaresuchus* [80,104], the row of larger, more strongly recurved teeth seen in *Prolacerta* [125] and from the edentulous palatine of erythrosuchids [81,127] and crown archosaurs (e.g. [44,121]).

The tooth field is around three teeth wide at its widest and tapers to a single tooth in width anteriorly and posteriorly. In SAM-PK-13664, this tooth bearing area begins some 3 mm posterior to the posteriormost choanal margin (closer than in the reconstruction of Ewer [54]) and ends around 12 mm posterior to this. The tooth field narrows mediolaterally posteriorly, with the lateral margin moving medially to meet the medial margin. Medial to the tooth field is the pterygoid facet. Lateral to the anterior of the tooth field the ventral surface of the palatine is flat, sloping dorsally and becoming slightly mediolaterally concave as it approaches the choana. At around one-third of the length of the tooth field posteriorly, the ventral surface of the palatine lateral to the tooth field becomes pronouncedly mediolaterally concave and widens, with the lateral margin of the tooth field moving medially and the lateral margin of the palatine moving laterally.

Anterior to the tooth field is an area of flat bone bordered medially by the pterygoid facet and laterally by the area of the ventral surface immediately posterior to the choana. The pterygoid facet and the area posterior to the choana are dorsally displaced in comparison with the ventral surface immediately anterior to the tooth field. Both borders arc in laterally convex arcs from posterolateral to anteromedial. The surface anterior to the tooth field is slightly mediolaterally concave. The choana is not bordered by a ridge, unlike in *Polonosuchus* and some other pseudosuchians [31,121].

The dorsal surface of the medial two-thirds of the palatine is visible in SAM-PK-5867 (figure 42*b*), and the entire palatine is visible in both this specimen and SAM-PK-6047A with CT data. *Euparkeria* shows the typical archosauriform morphology of a dorsal fossa approaching the posterior choanal margin (in aetosaurs and some other pseudosuchians a flat area of bone separates the two [31]; figure 42, fos1): the dorsal surface is slightly concave, with a strongly concave strip extending anteroposteriorly just medial to the medial choanal margin, with the choanal margin itself formed into a flattened ridge. Some way posterior to the choana is another, thin, posterodorsally opening fossa extending posterolaterally to anteromedially (figure 42, fos2). The fossa is blind, but there is a potential minor diverticulum from the fossa into the bone (figure 42*e*, div?). A similar feature is tentatively identified by Borsuk-Białynicka & Evans [74] in *Osmolskina* to be the point of entry of some branches of the medial palatal artery, but neither taxon shows an exit foramen.

Articulation with the pterygoid is via a facet along the medial margin of the ventral surface of the palatine, with the palatine thus dorsal to the pterygoid (see Pterygoid). This facet is an anteroposteriorly elongated, mediolaterally concave area set slightly dorsally to the rest of the ventral surface of the bone, bordered laterally by the palatine tooth field. The anterolateral margin of the facet curves from posterolateral to anteromedial in a laterally convex arc. Dorsally (in SAM-PK-5867), the contact between the pterygoid and the palatine is a simple, non-interdigitating suture (see Pterygoid). The vomer contacts the palatine in a simple appressed contact with no interdigitations (see Vomer). The articulation with the maxilla consists of a shallow, depressed facet on the medial surface of the main body of the maxilla extending from around 20–50% of the distance posteriorly from the posterior margin of the dorsal process of the maxilla to the posterior tip of the main body of the maxilla (figure 26, f.pl); the convex lateral surface of the palatine fits into this depression.

### 3.2.6.3. Ectopterygoid (figure 43)

The ectopterygoid is clearly seen in SAM-PK-13664 (figure 43*a*), where it is exposed ventrally and laterally and in individual A in UMZC T.692 in dorsal view (figure 43*b*). The ectopterygoid is also preserved in SAM-PK-5867 and can be viewed in CT scans (figure 43*d–g*). The ectopterygoid forms a small posterolateral part of the palate. The ectopterygoid connects the pterygoid to the side of the skull, suturing with the dorsal surface of the pterygoid (unlike in pseudosuchians and other stem archosaurs, where it sutures to the ventral surface, but similar to the situation in dinosaurs [31,77]) posteromedially and the jugal laterally. The ectopterygoid does not appear to have reached the maxilla. The ectopterygoid does not articulate with the posterior of the palatine medially as is reconstructed by Ewer [54].

The main body of the ectopterygoid is a plate that dorsally overlays the anterolateral part of the posterior end of the palatal ramus of the pterygoid. Medially, the ectopterygoid extends to about half way between the lateralmost extent of the palatal flange of the pterygoid and the edge of the pterygoid tooth field. Laterally, the ectopterygoid narrows anteroposteriorly then expands again into an anteroposteriorly elongated head (figure 43, h), elliptical in lateral view. Posteriorly, this head divides into a posterordorsal process and a directly posterior process, both tapering distally (figure 43, pd.p, p.p). The head articulated with the jugal. There does not appear to have been an articulation with the maxilla; although the anteriormost part of the ectoptergoid head is anteroposteriorly level with the posteriormost tip of the maxilla in SAM-PK-5087, there is no facet on either element, nor does the ectopterygoid form a separate process or head for articulation with the maxilla. *Euparkeria* can, thus, contra Nesbitt [31], Sookias [32], Sookias *et al*. [33,34], Butler *et al*. [129] etc. be described as possessing a 'double headed' ectopterygoid. The ectopterygoid is broadly similar in *Osmolskina czatkowicensis* (ZPAL RV408), which also shows a posterodorsal projection. *Erythrosuchus*, however, rather shows an anterodorsal projection [81]. Several pseudosuchians, including two crocodylomorphs, also show this posterodorsal projection [31], but most crown archosaurs do not show this double head (e.g. *Alioramus altai* [130]; *Caiman latirostris*—ZMB 36627). The jugal does not show a clearly visible secondary facet for the posterodorsal process (figure 37*c*), unlike, for example, *Rauisuchus* [91]. Although scored as absent in all non-archosaurian archosaurmorphs [31], in most cases, the dorsomedial part of the ectopterygoid is obscured or not fully described (e.g. in *Doswellia kaltenbachi* [131], in proterochampsids [108]), and thus further work, including CT investigation, is needed to fully assess the distribution of this feature.

The anterior two-thirds of the ventral surface of the main body of the ectopterygoid is concave in all directions, with the point of maximum concavity just medial to the point where the ectopterygoid is shortest anteroposteriorly (i.e. the narrowed shaft of the bone connecting to the head laterally; figure 43*a*, pmc). This concavity is the pterygoid facet (as illustrated in *Plateosaurus engelhardti*—[31],

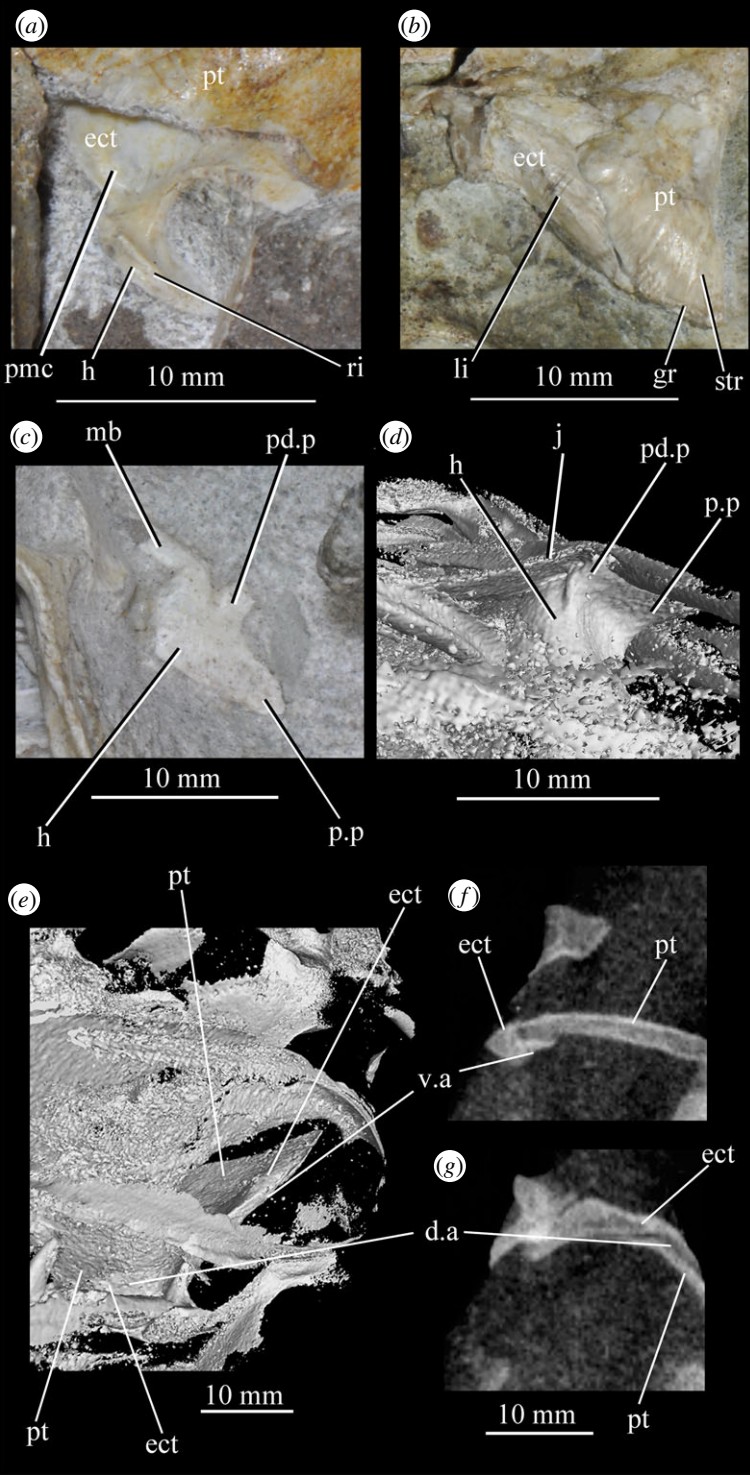

**Figure 43.** Examples and details of the ectopterygoid of *Euparkeria capensis*. (*a*) right ectopterygoid of SAM-PK-13664 in ventral view; (*b*) left ectopterygoid of UMZC-T.692 individual A in left dorsolateral view; (*c*) left ectopterygoid of SAM-PK-5867 in lateral view showing the articulatory surface of the head for jugal; (*d*) CT scan image of the right ectopterygoid of SAM-PK-5867 in dorsomedial view; (*e*) CT scan image of the right ectopterygoid of SAM-PK-5867 in articulation with the pterygoid anteroventral view; (*f*) more posterior CT cross section through ectopterygoid-pterygoid articulation on right hand side of SAM-PK-5867 showing ventral articulation with pterygoid; (*g*) more anterior CT cross section through etopterygoid articulation on right hand side of SAM-PK-5867 showing dorsal articulation with pterygoid. d.a, dorsal ectopterygoid-pterygoid articulation; ect, ectopterygoid; gr, groove; h, head; j, jugal; li, line of inflection between surfaces; mb, main body of the ectopterygoid; pd.p, posterodorsal process; pmc, point of maximum ventral concavity; p.p, posterior process; pt, pterygoid; ri, ridge; str, striations; v.a, ventral ectopterygoid-pterygoid articulation. The scale below (*g*) applies to (*f,g*).

fig. 22) and is not homologous with the distinct fossa in tetanurans (see [31]). The posterior part of the ventral surface of the main body of the ectopterygoid (posterior to the narrow shaft) is convex. The dorsal surface of the main body of the ectopterygoid is convex mediolaterally, with a sharp line of inflection between mediodorsally and laterodorsally facing surfaces extending anteroposteriorly down the mediolateral centre of the bone (figure 43b, li). The dorsal surface, and the ectopterygoid as a whole, does not arch dorsally between the pterygoid and the jugal as in dinosaurs [31]. The ventral surface of the head shows no notable features except for a thin blunt ridge extending posterolaterally to anteromedially (figure 43a, ri). The ridge starts at around 10% of the anteroposterior length of the head from its posterior margin and reaches to its anterior end, but decreases in height and increases in width slightly anteriorly. The ectopterygoid lacks teeth or foramina.

The ectopterygoid–pterygoid articulation is a simple, non-interdigitating and irregular suture connecting the posteromedial margin of the ectopterygoid with the anterolateral margin of the posteriorly expanded part of the palatal ramus of the pterygoid bordering the main tooth field, with a slight ventral overlap of the pterygoid by the ectopterygoid at the suture (figure 43b). Following Sereno & Novas [77], this can be described as the ectopterygoid articulating ventrally to the pterygoid (rather than dorsally, as in dinosaurs). As in most archosauriforms, the ectopteryoid–pterygoid contact is long, as opposed to the state in *Prolacerta* [125] and proterochampsids [80,104] where the contact is restricted to the anterior of the posteriorly expanded part of the pterygoid (see [28]). The ectopterygoid–jugal contact consists of the dorsolaterally convex (in coronal section) ectopterygoid being received by a concave facet on the jugal (see Jugal).

### 3.2.6.4. Pterygoid (figure 44)

The pterygoid is best observed in SAM-PK-13664 (figure 44a,b), where a disarticulated pterygoid with only the dorsal surface obscured is preserved. Both pterygoids are also preserved in SAM-PK-5867 and their anterior ends are visible dorsally through the antorbital fenestra (figure 42b), and are visible in full using CT data (figures 21a and 44c). SAM-PK-6047A preserves both pterygoids in articulation, but they are only visible using CT data. These specimens and the left pterygoid of UMZC T.692 (individual A; figures 19 and 43b) provide the basis of the description. The left pterygoid is also partially exposed in SAM-PK-13665 and is preserved in UMZC T.692, and left and right partial pterygoids are preserved in SAM-PK-6050. The pterygoid forms the posteromedial part of the palate. The pterygoid articulates with the vomer anterolaterally and palatine posterior to this, with the ectopterygoid posterolaterally, with the braincase (basisphenoid) posteriorly and with the quadrate posteriorly via the quadrate ramus (figure 44, q.r). The palatal ramus of the pterygoid (figure 44, pal.r) is narrow anteriorly, forming only the medialmost part of the palate. Posterior to this, the pterygoid widens into a plate, suturing laterally with the ectopterygoids. A vertical sheet (figure 44a, vs) rises from the medial margin of the palatal ramus, forming a lateral wall to the interpterygoid vacuity. Posteriorly, the pterygoids are fused at the midline. The quadrate ramus of the pterygoid is a mediolaterally broad plate-like process projecting dorsolaterally (at about 15° to the vertical) from the posteromedial extreme of the palatal ramus (figure 44, q.r).

The ventral surface of the pterygoid holds an anteroposteriorly elongated, ovaloid (with pointed anterior and posterior tips) field of more than 50 palatal teeth covering most of the more medial part of the ventral surface (figure 44, t) that corresponds to field T3 *sensu* Welman [126]. The teeth are arranged randomly. These teeth differ in no notable way from those on the palatine, and are thus much smaller than the marginal teeth, slightly recurved and pointed. The medial margin of this tooth field extends along the medial margin of the pterygoid. Some 40% of the anteroposterior length of the tooth field is posterior to the anteriormost point of the ectopterygoid.

Roughly level with the anteriormost point of the ectopterygoid, a narrower tooth field splits off the main tooth field (figure 44, t2); this smaller field extends from posteromedial to anterolateral at around 45° to the midline. This field approaches the posteromedial end of the palatine tooth field anterolaterally, but the fields do not appear to be continuous. Both tooth fields are displaced ventrally in relation to the rest of the ventral surface of the pterygoid. The pterygoid dentition of *Euparkeria* thus differs from that of some non-archosaurian archosauromorphs in being composed of irregular tooth fields, rather than rows of teeth (as is seen in *Chanaresuchus* [80] and *Osmolskina* [74]). *Proterosuchus fergusi* [126] and *Prolacerta* [90] also show irregular tooth fields, but the teeth of the latter taxon are larger and recurved, and the arrangement differs in both; erythrosuchids [81,127] and most crown archosaurs lack pterygoid dentition.

Additionally, the tooth fields of *Euparkeria* diverge anteriorly like the rows or fields of other non-archosaurian archosauromorphs [74,80,90,126], but this divergence is relatively more anterior. Behind

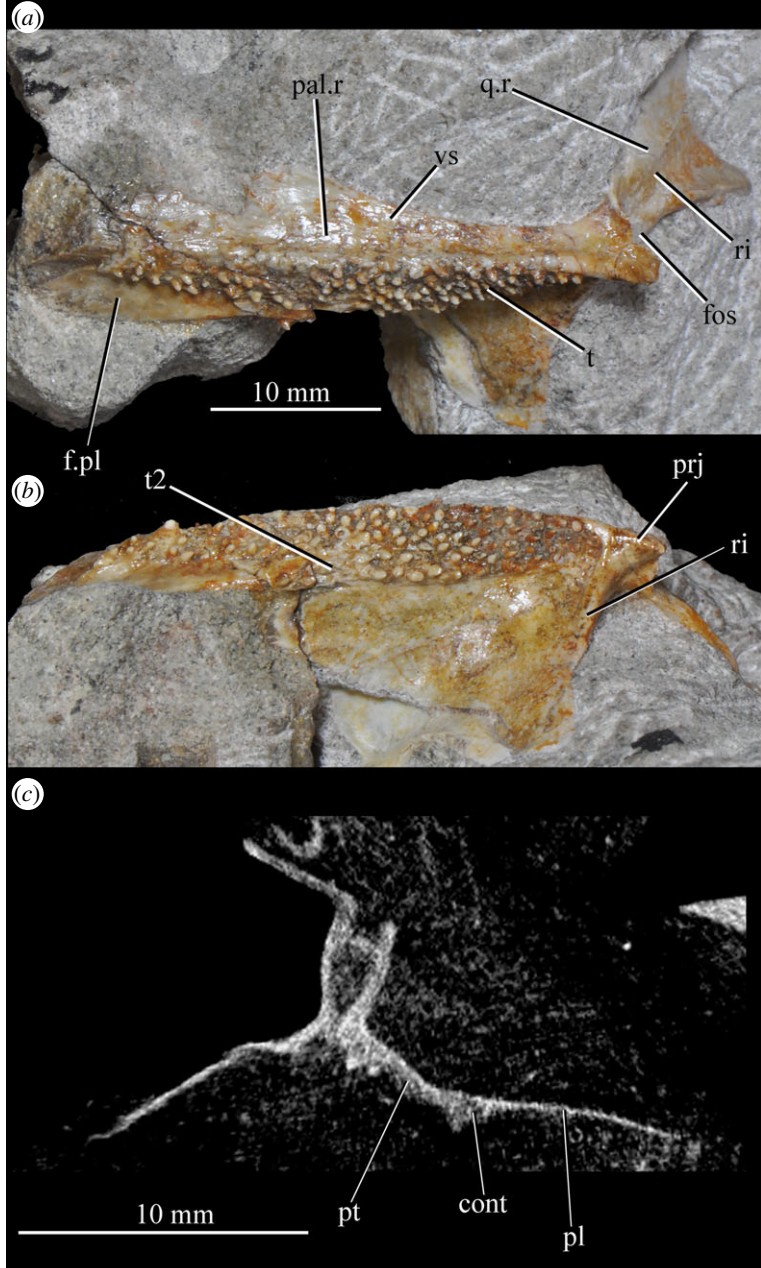

**Figure 44.** Examples and details of the pterygoid of *Euparkeria capensis*. Right pterygoid of SAM-PK-13664 in (*a*) ventromedial and (*b*) ventral views; (*c*) coronal CT slice through pterygoids of SAM-PK-5867 showing pterygoid–palatine articulation. cont, pterygoid–palatine contact; fos, fossa; f.pl, palatine facet; pl, palatine; prj, projection; pt, pterygoid; q.r, quadrate ramus of pterygoid; ri, ridge, t, main pterygoid tooth field; t2, narrow secondary pterygoid tooth field, vs, vertical sheet. The scale below (*a*) applies to (*a,b*).

the posterior tip of the main tooth field, the ventrally displaced surface of the pterygoid forms a projection, narrower mediolaterally than deep dorsoventrally (figure 44, prj). The ventral surface of the pterygoid lateral to the tooth field is concave in all directions, with the maximum point of concavity around half way along the suture with the ectopterygoid. A ridge (figure 44*b*, ri) borders the posterior edge of this lateral flange of the pterygoid, narrowing and becoming lower anterolaterally.

The dorsal surface of the posterior half of the ptergyoid can be seen in UMCZ T.692 (individual A). This surface is primarily convex. Striations (figure 43*b*, str) extend posteromedially from the anterior margin of the lateralmost part of the mediolaterally expanded posterior part of the pterygoid. A slight groove, the width of which is around 5% of the mediolateral width of the widest point of the pterygoid, extends anteroposteriorly near the medial extremity of the dorsal surface. The dorsal surface of the anterior half of the pterygoid (figure 43*b*, gr). It is flat apart from a rounded ridge extending from posteromedial to anterolateral at around 5° to the midline (figure 42*b*, ri).

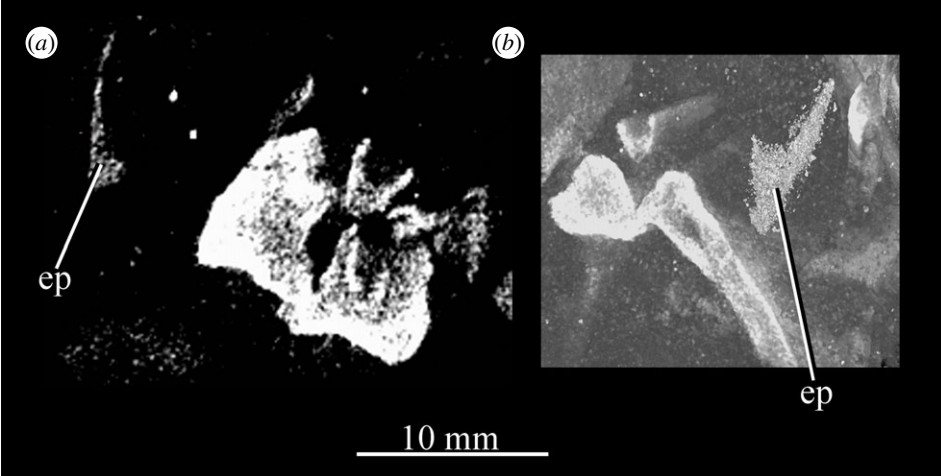

**Figure 45.** CT images of right epipterygoid of *Euparkeria capensis* specimen SAM-PK-6047A: (*a*) coronal slice in anterior view; rendering in posterolateral view with coronal section. ep, epipterygoid.

In posterior view, just dorsal to the projection behind the tooth field, is the base of the quadrate ramus of the pterygoid, which extends dorsolaterally (figure 44*a*, q.r). The posterior surface of the base of the quadrate ramus is concave, and the anterior surface convex. This concavity, together with the projection of the palatal ramus of the quadrate below it, creates a fossa at the base of the process that opens posteriorly (figure 44, fos). This fossa received the basipterygoid process of the basisphenoid. The form of the rest of the quadrate ramus is reminiscent of, as described by Ewer [54], a fishtail, with the long posterior margin of the 'tail' directed dorsolaterally. A central ridge divides the two flanges of the ramus (i.e. the upper and lower parts of the 'tail'; figure 44, ri).

The articulation of the pterygoid with the palatine consists of a depressed facet on the palatine that receives the anterolateral part of the pterygoid (figure 44*c*, cont). The part of the anterolateral surface of the pterygoid that articulates with this facet can be seen lateral to the anterior part of the main tooth field in SAM-PK-13664 (figure 44*a*, f.pl). It is flat and smooth, and is slightly dorsal to the part of the ventral surface of the pterygoid on which the tooth field sits. The pterygoid–quadrate articulation consists of a broad overlapping articulation between the broad end of the fishtail-like quadrate process of the pterygoid and the pterygoid process of the quadrate (see Quadrate).

The pterygoid–ectopterygoid articulation is a simple, non-interdigitating suture extending first anteromedially to posterolaterally, and then arcing more strongly laterally (see Ectopterygoid). The pterygoid–vomer (see Vomer) articulation consists of a simple appression, with the vomer articulating medially and anteriorly with the anterior process of the pterygoid. The pterygoid–epipterygoid articulation would presumably have consisted of the ventral broad foot of the epipterygoid being simply appressed to the centre of the posteromedial part of the dorsal surface of the palatal wing of the pterygoid (see Epipterygoid), but the two are not preserved in contact.

### 3.2.6.5. Epipterygoid (figure 45).

The epipterygoid can only be observed in the CT scan of SAM-PK-6047, having been lost in SAM-PK-5867 during preparation. It is a small element ascending from the dorsolateral base of the quadrate ramus of the pterygoid, and is dorsoventrally elongated and oval in transverse section. It would presumably have contacted the braincase in life, but its disarticulation means the exact point of contact is uncertain; based on its position in relation to the posterior of the orbit (the braincase is largely lost/damaged in SAM-PK-6047A), it would have contacted the anterior of the prootic or posterior of the laterosphenoid. Dorsally, it narrows to a thin tip, and ventrally, it expands into a broad foot, which articulates with the pterygoid. In posterior view, the element projects in a ventromedial to dorsolateral, laterally convex arc. In posterior view, the dorsomedial margin is more strongly curved than the ventrolateral margin.

### 3.2.6.6. Quadrate (figure 46)

The left and right quadrates are preserved in SAM-PK-5867 (figures 4, 5 and 46*a,e*), two left quadrates from different individuals are present in UMZC T.692 (figures 18, 19 and 46*b–d*), and a damaged right

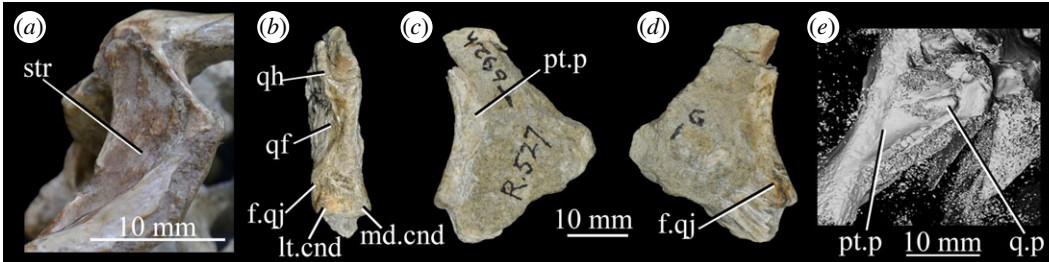

**Figure 46.** Examples and details of the quadrate of *Euparkeria capensis*. (*a*) right quadrate of SAM-PK-5867 in right lateral view; left quadrate of UMZC T.692 individual A in (*b*) posterior, (*c*) medial and (*d*) lateral view; and (*e*) CT surface rendering of the left quadrate of SAM-PK-5867 in articulation with pterygoid in left posteromedial view. f.qj, quadratojugal facet; lt.cnd, lateral condyle; md.cnd, medial condyle; pt.p, pterygoid process of quadrate; qf, quadrate foramen; qh, quadrate head; q.p, quadrate process of pterygoid; str, striations. The scale below (*a*) applies to (*a*) only, that between (*c*) and (*d*) applies to (*b*–*d*), and that below (*e*) to (*e*) only.

quadrate is preserved in SAM-PK-6047A (figure 8); these are the primary basis of the description. Examples of the quadrate are also preserved in SAM-PK-13665 and SAM-PK-K8050. The quadrate is a dorsoventrally expanded bone that forms the lateral margin of the posterior end of the skull. In posterior view (figure 46*b*), the quadrate shaft is expanded transversely dorsally and ventrally; the dorsal end of the quadrate (the head—figure 46*b*, qh) is wider than the ventral end, being expanded laterally to form an elliptical plate. The shaft is narrowed at its centre, forming a quadrate foramen (as in most archosauromophs excluding aetosaurs, some crocodylomorphs and *Vancleavea* [28,132]) between its lateral margin and the medial margin of the quadratojugal, which contacts the quadrate above and below this point.

The ventral end of the shaft is divided into lateral and medial condyles (figure 46, lt.cnd, md.cnd) that articulate with the cotyle fossae of the mandible (as in all archosauromorphs excepting *Shuvosaurus* and *Effigia*, where the quadrate forms cotyles and the articular forms condyles [133]). The medial condyle is wider mediolaterally than the lateral condyle. Anteromedially, the quadrate contacts the pterygoid via the pterygoid wing of the quadrate. The quadrate contacts the quadratojugal along two surfaces laterally—one surface placed more dorsally contacted via the anterolateral surface of the quadrate and one placed more ventrally contacted via the posterolateral surface of the quadrate. The quadrate head contacts the posteroventral part of the squamosal, and is separated from the prootic (unlike in crocodylomorphs and avians [82,111]) and opisthotic (unlike in phytosaurs [31]) by the squamosal.

In lateral or medial view (figure 46*a*), the posterior margin of the quadrate is concave and the anterior margin convex, and the shaft extends from anterodorsal to posteroventral at around 25° to the vertical. This posteroventrally directed slope is typical of most archosauromorphs excluding derived aetosaurs, some shuvosaurids and some theropods where the quadrate is anteroventrally directed [97]. In dorsal and ventral views, the posterior margin is convex and the anterior margin is concave. Irregular striations extend down the shaft (figure 46, str), fanning out laterally and medially from the centre of the shaft towards its ends. The quadrate lacks other ornamentation (e.g. no rugose, dorsoventrally orientated crest as in *Polonosuchus* and *Postosuchus kirkpatricki* [31]) and foramina (unlike in some crocodylomorphs [31,107]) or pits (as in *Vancleavea* [28] and possibly *Erythrosuchus* [81]) though the quadrate and quadratojugal form the quadrate foramen.

The lower quadrate–quadratojugal articulation can be seen in the left quadrate UMZC T.692h: the lateral margin of the quadrate expands into a posteriorly convex flange dorsolateral to the lateral condyle (figure 46, f.qj). This flange is overlain by, and articulates with, the main body of the quadratojugal (see Quadratojugal). The nature of the upper articulation with the quadratojugal can be seen in SAM-PK-5867 (figures 40*c,d* and 46*a*); the medial surface of the lower part of the lateral flange of the quadrate head posteriorly overlies the dorsomedial part of the posterior surface of the anterodorsal process of the quadratojugal.

The pterygoid process arises from the medial edge of the quadrate just level with the ventral margin of the quadrate foramen. The pterygoid process extends from posteroventrolateral to anterodorsomedial and consists of a broad plate, '>'-shaped in lateral view. The articulation between the pterygoid process of the quadrate and the quadrate ramus of the pterygoid consists of a broad articulation (figure 46*e*), with the pterygoid ramus of the quadrate concave dorsomedially in coronal section to receive the ventrolaterally convex quadrate ramus of the pterygoid (the midline of the 'fishtail' of the quadrate ramus—figure 44*a*, ri—being the point of maximum convexity).

The quadrate–squamosal articulation consists of the laterally expanded elliptical plate of the quadrate head being received by the concave posteroventromedial surface of the squamosal (visible in the

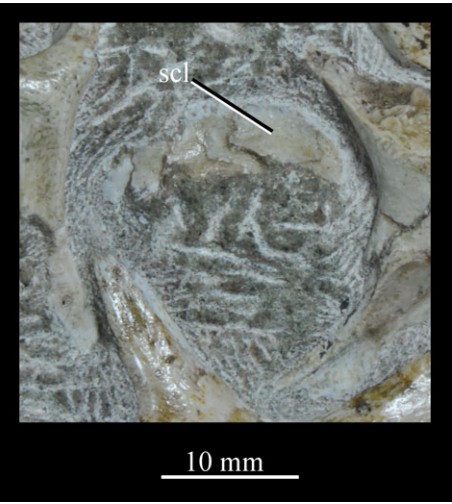

**Figure 47.** Close-up of the scleral ring on the right-hand side of *Euparkeria capensis* specimen SAM-PK-6047A in lateral view. scl, sclera.

disarticulated squamosal in SAM-PK-6047A—figure 39*f,g*) between the ventral and posterior processes of the squamosal, which form a rim around the head of the quadrate, securing it in place.

### 3.2.7. Sclera

Sclera are preserved in position in the right orbit of the skull in SAM-PK-6047A, forming the dorsal part of a sclerotic ring (figure 47), having been lost in preparation in SAM-PK-5867. They are plates around 4 mm in anteroposterior length and 3 mm in dorsoventral depth with irregularly shaped margins, tapered anterior and posterior ends, and flat or very slightly convex lateral surfaces. The sclera lack pitting or ornamentation. The dorsalmost sclerum overlies those anteroventral and posteroventral to it laterally, with this pattern of the more dorsal sclerum overlying that below it continuing around the ring. Sclera are present in *Prolacerta* [90], but are not well preserved enough to allow their form to be compared. *Proterosuchus fergusi* also possesses sclera (e.g. BP-1-4016); *contra* Cruickshank [83], these do not appear to be fewer in number or relatively larger than those of *Euparkeria*.

No sclera have been identified from proterochampsids [80,104], *Osmolskina* [74] or erythrosuchids [81,122], but are present in many crown archosaurs (e.g. [97,134]), and may well have failed to be preserved in stem archosaur taxa in which they are unknown. Our measurements of the scleral ring on the right-hand side of SAM-PK-6047A correspond as follows to those of Schmitz & Motani [134] taken from a cast of SAM-PK-5867 (taken prior to removal of the scleral ring to expose the internal cranial anatomy), which were used to infer scotopic morphology (i.e. nocturnality) in *Euparkeria*: orbital length 15.2 mm (21.1 mm in [134]); external scleral ring diameter 13.9 mm (14.35 mm in [134]); internal scleral ring diameter 11.3 mm (8.82 mm in [134]).

### 3.2.8. Mandible

The mandible is around the same length as the skull measured from the tip of the rostrum to the posterior of the quadrate (figures 3, 4 and 8). It deepens from its posterior end to a maximum dorsoventral height of approximately 15% of the total length at around one-third of the distance from the posterior to anterior ends. The mandible's anterior end is deflected dorsally.

An anteroposteriorly elongated oval external mandibular fenestra is present (figures 3, 4*a*, emf), as in all basal archosauriforms. A medial internal mandibular fenestra (figures 3, 4*a*, imf) has blunt anterior, posterior and ventral corners and a more rounded dorsal margin. The anterior and posterior corners of the internal mandibular fenestra are nearer the dorsal than the ventral margin of the opening.

#### 3.2.8.1. *Articular* (figures 48 and 49)

The left and right articulars are preserved in articulation in SAM-PK-5867, and visible in posterior and ventral views, the left articular is also visible medially but the right articular is partially obscured (figures 4, 5 and 48*a,b*). The right articular is preserved in SAM-PK-6047A (figure 48*g,h*); those of individuals A and B are preserved in UMZC T.692, with that of individual B visible in all views

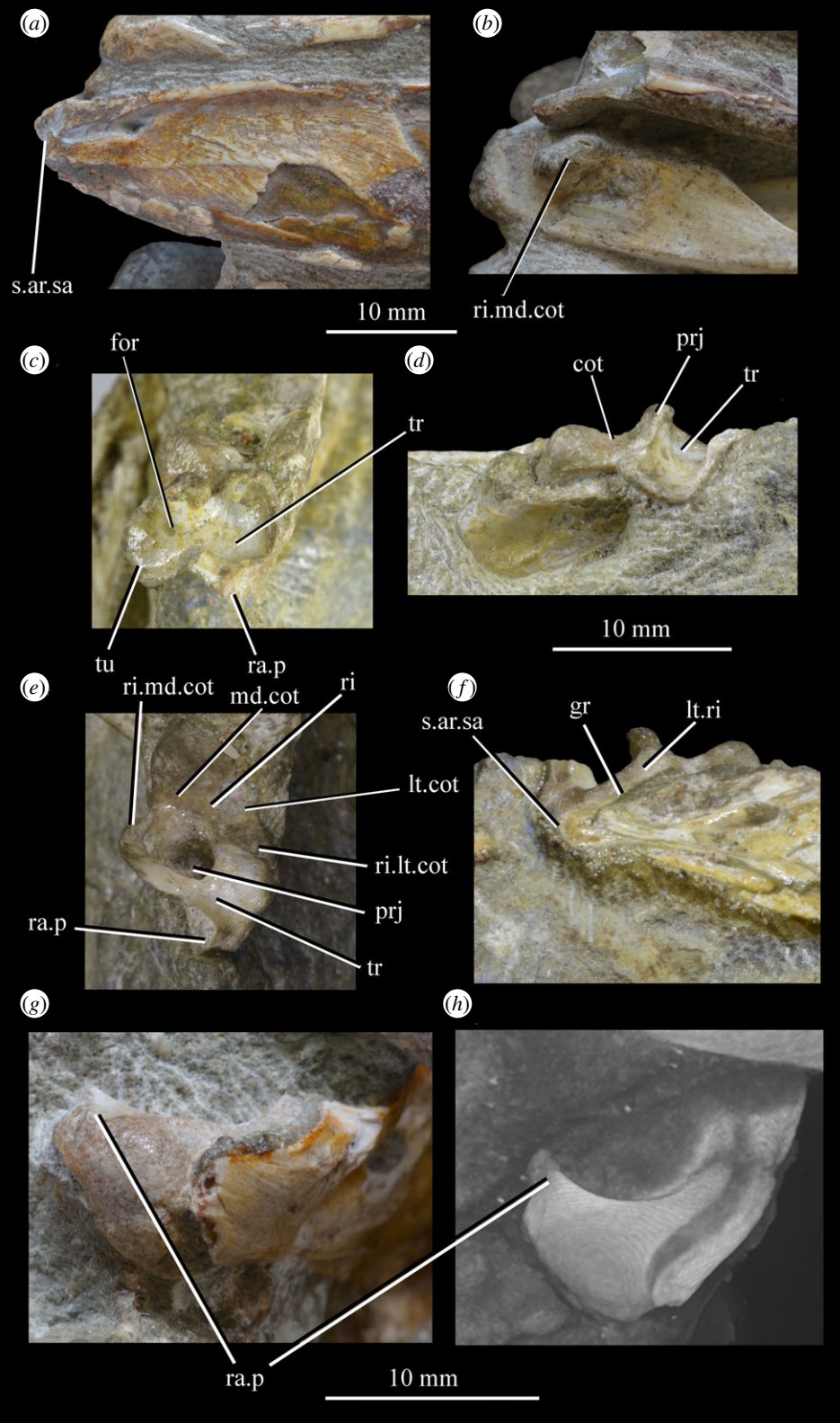

**Figure 48.** Examples and details of the articular of *Euparkeria capensis*. (*a*) Right articular of SAM-PK-5867 in lateral view; (*b*) left articular of SAM-PK-5867 in medial view; right articular of UMZC T.692 individual B in (*c*) posterior, (*d*) medial, (*e*) dorsal and (*f*) lateral view; right articular of SAM-PK-6047A in (*g*) right lateral and slightly dorsal view and (*h*) in right posterodorsal and slightly lateral view (CT rendering) to show the retroarticular process. cot, cotyle; for, foramen; gr, groove; lt.cot, lateral cotyle; lt.ri, lateral ridge; md.cot, medial cotyle; prj, projection; ra.p, retroarticular process; ri, ridge; ri.lt.cot, ridge bordering lateral cotyle; ri.md.cot, ridge bordering medial cotyle; s.ar.sa, suture between articular and surangular; tr, trough, tu, tuber. The upper scale applies to (*a,b*), the scale between (*d*) and (*f*) applies to (*c–f*), and lower scale under (*g*) and (*h*) applies to (*g*) and (*h*) only.

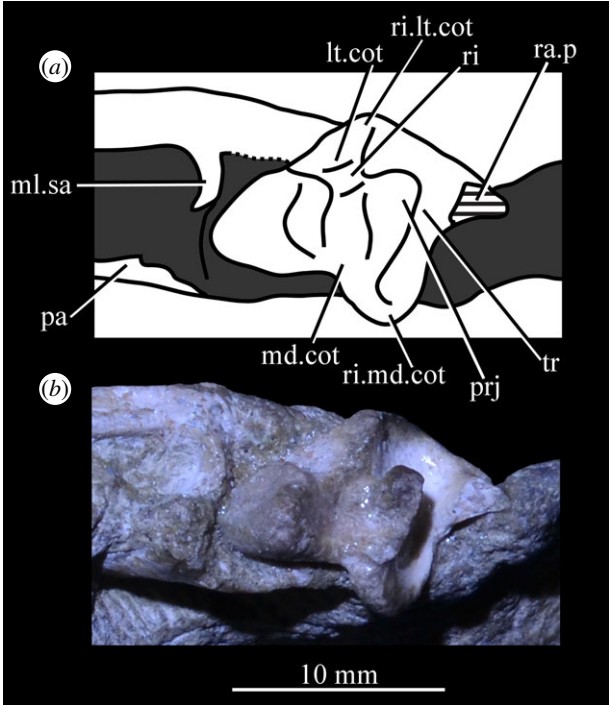

**Figure 49.** (*a*) line drawing and (*b*) flash lit photograph of right articular of *Euparkeria capensis* specimen UMZC T.692 individual B in dorsal view to show further details of morphology. lt.cot, lateral cotyle; ml.sa, medial lamina of surangular; md.cot, medial cotyle; pa, prearticular; prj, projection; ra.p, retroarticular process; ri, ridge; ri.lt.cot, ridge bordering lateral cotyle; ri.md.cot, ridge bordering medial cotyle; tr, trough. White (within external border of bone), exposed bone; ray (in line drawing), matrix (incl. matrix covering bones); cross-hatching, broken surface; dotted line, uncertain edge of exposed bone.

except ventrally and that of individual A visible in lateral view (figures 18, 19 and 48*c–e*). Other examples are preserved in SAM-PK-7699, SAM-PK-13665 and SAM-PK-13667 but were not used in the description. The articular forms most of the mandibular cotyles/glenoid, and thus the posterodorsal extremity of the mandible. Ventromedially, the articular contacts the prearticular, and laterally, anteriorly and dorsomedially, it contacts the surangular.

The dorsal surface of the articular just posterior to its anterior margin has two roughly circular, dorsally concave cotyles (i.e. fossae; figure 48, md.cot, lt.cot) for the condyles of the quadrate, one lying laterally to the other. In lateral view, the ventral margin of the cotyles is only just ventral to the dorsal margin of the dentary, unlike in ornithischians, sauropodomorphs and aetosaurs where it is well ventral of the dorsal margin [31,103]. The surface of the more medial cotyle is directed slightly medially, and that of the lateral cotyle is directed slightly laterally. These cotyles are separated by a low ridge (figure 48*e*, ri) almost half as broad as it is long. The more medial of the cotyles is larger in area. The anterior margin of the cotyles is bordered by a ridge formed by the surangular. The lateral margin of the lateral cotyle and the medial margin of the medial cotyle are low dorsally rounded ridges. The posterior margin of the medial cotyle is drawn up into a projection (figure 48*e*, prj) centred roughly posterior to the mediolateral centre of the cotyle. Either side of this projection, the posterior margin is composed of a low ridge, roughly the same height as that bounding the cotyle medially (figure 48*e*, ri.md.cot, ri.lt.ct).

Posterior to the ridge and projection forming the posterior border of the cotyles, the dorsal surface of the articular forms a smooth mediolaterally elongated trough (figure 48, tr) for attachment of the *depressor mandibulae* musculature. This trough is a mediolaterally elongate oval with a concave dorsal surface. It extends along a posterolateral to anteromedial diagonal of about 10° from mediolateral. The ridge posterior to the lateral cotyle (figure 48, ri.lt.cot) descends along the anterior and anterolateral margin of the trough and becomes gradually more laterally and less dorsally directed; by the time it reaches the bottom of the trough the ridge is entirely laterally directed, and the lateral margin of the trough is thus no higher than the surface at the mediolateral centre of the trough.

A similar ridge (figure 48, ri.md.cot) descends from the posteromedial extreme of the medial cotyle along the medial margin of the trough, but this ridge is lower even from its anterior onset. The medial side of the trough is more strongly expanded than the lateral side, and can thus be described as a

tubercle (see Gower [81] for *Erythrosuchus*; figure 48*c*, tu), corresponding to the 'ventromedially directed process' of Nesbitt [31]. Nesbitt, however, scores this process as absent in *Euparkeria* and other taxa outside the crown group; while this process may be more strongly developed in some crown taxa, we cannot see a clear difference between the state in *Postosuchus kirkpatricki* (fig. 26 in Nesbitt [31]) and that in *Euparkeria* (though the structure does appear to be relatively reduced in erythrosuchids and proterochampsids—[80,81]; see discussion in [33]).

The posterior margins of the medial and lateral projections of the trough are not raised. A small foramen (figure 48*c*, for) is present on the anteromedial surface of the trough (which is probably the entrance of the chorda tympani [74]), as in crown archosaurs and *Osmolskina* [74] but unlike in other non-archosaurian archosauromorphs [31]. In the CT scan of SAM-PK-6047A, a small channel through the articular on its medial surface below the ridge separating the cotyles and the depressor mandibulae may be the passage for the chorda tympani as identified in *Osmolskina* [74], but the channel is difficult to discern with certainty.

Posterior to the trough for attachment of the depressor mandibulae musculature, the articular narrows mediolaterally to around two-thirds of its anterior width. The medial margin moves further laterally than the lateral margin moves medially. Simultaneously, the articular increases in height to form a retroarticular process (*sensu* Gower [81]; figure 48, ra.p), which rises dorsally to at least the same height as the ridge posterior to the lateral cotyle. The anterior and posterior margins of the retroarticular process curve in an anterodorsally concave curve in lateral view. The process, preserved in its full extent on the right-hand side of SAM-PK-6047A (figure 48*g,h*), curves medially and slightly dorsally to end level with the medial margin of the cotyles. Nesbitt [31] terms this process the dorsomedial projection, and scores it as absent in non-archosaurian archosauromorphs. Although shorter than in some archosaurs, the structure in *Euparkeria* is undoubtedly homologous and is also present, though reduced, in erythrosuchids ([81]; PIN 951/46) and proterochampsids [80].

Ventral to the lateral cotyle, the lateral surface of the articular forms a smooth, bulging, anteroposteriorly elongated ridge arcing in a dorsally concave arc underneath the lateral cotyle (figure 48*f*, lt.ri). Below this, a groove (figure 48*f*, gr) of similar mediolateral width and dorsoventral depth to the ridge is present on the surangular, emphasizing the ridge above it. The convexity of the lateral surface of the articular becomes less pronounced posteriorly, as the lateral surface also becomes dorsoventrally deeper.

In medial view (figure 48*d*), the contact between the prearticular and the articular consists of a suture extending broadly posteroventrally to anterodorsally. At its posteroventral extreme, the suture is close to vertical then becomes much less steep level with the base of the medial tubercle of the articular and extends forward for a distance roughly equal to the depth of the tubercle. After this, the suture extends vertically once more in a posterodorsally concave arc. At the dorsal margin of the mandible, the prearticular forms a posterodorsally open cup into which the articular fits; the wall of the cup is concave (and the external margins of the articular convex) except for dorsolaterally where the situation is reversed.

Yet further dorsolaterally, the articular sits atop the surangular (which is dorsally concave); a deep groove (figure 48*f*, gr) mentioned above marks the boundary between the elements, and appears to lie entirely or largely on the surangular. Anteriorly, the groove has a rounded margin, but posteriorly it is open, with the ventral margin ending posteriorly anterior to the dorsal margin. At the posterior extreme of this groove, a suture between the articular and the surangular (figure 48*f*, s.ar.sa) extends down at a posterodorsal to anteroventral diagonal of about 10° to the vertical from the posterior extreme of the ventral margin of the groove separating the elements. Anterior to the articular, the medially extending process of the surangular is applied to the anteroventral surface of the articular (figure 49, ml.sa), cupping it in place.

### 3.2.8.2. Dentary (figure 50)

The left and right dentary of the same individual, visible in lateral and medial views, are preserved in SAM-PK-5867 (figures 4 and 50*a*) and in SAM-PK-6050 (figures 11 and 26*e,f*). Both dentaries are visible in lateral view in SAM-PK-6047A (figures 8 and 50*b*). In UMZC T.692 (individual A), the left dentary is visible in medial view (figure 50*f*), as is part of the right dentary of the same individual in lateral view (figure 19*a,b*) and that of individual B in lateral view (figure 18*d,f*). Other examples of the dentary, not used in the description, are present in SAM-PK-7699, SAM-PK-13665, SAM-PK-13666, SAM-PK-13667 and AMNH 2239. The dentary comprises approximately the anterior half of the mandible in lateral view, and is around five times longer than it is dorsoventrally deep at its deepest point. The dentary articulates posteroventrally with the angular, posterodorsally with the surangular

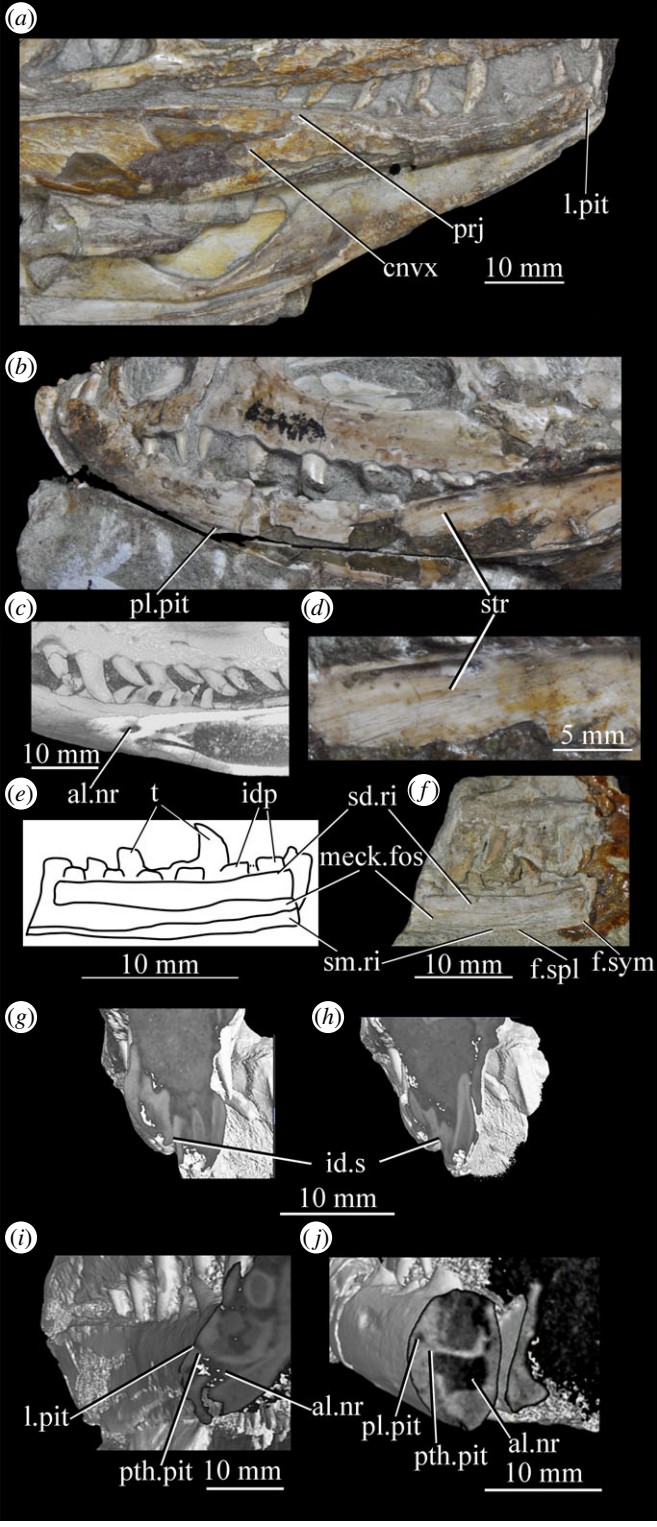

**Figure 50.** Examples and details of the dentary of *Euparkeria capensis*. (*a*) SAM-PK-5867 in right lateral view; (*b*) SAM-PK-6047A in left lateral view; (*c*) CT scan image of left dentary of SAM-PK-5867 in lateral view exposing alveolar nerve channel on the medial surface; (*d*) close-up image of SAM-PK-6047A in left lateral view to show striations; (*e*) line drawing of UMZC T.692 left dentary of individual A in medial view; (*f*) UMZC T.692 left dentary of individual A in medial view; CT rendering and coronal section of anterior dentaries of SAM-PK-5867 in anterior view at (*g*) slightly more posterior and (*h*) slightly more anterior positions; CT images in anterior view showing an approximately coronal cross-section through (*i*) right dentary of SAM-PK-5867 and (*j*) left dentary of SAM-PK-6047A showing the path of lateral and posterolateral pits joining alveolar nerve canal. al.nr, alveolar nerve channel; cnvx, convexity; f.spl, facet for splenial; f.sym, symphyseal facet; id.s, interdentary suture; idp, interdental plate; l.pit, laterally opening pit; meck.fos, Meckelian fossa; pl.pit; posterolaterally opening pit; prj, projection; pth.pit, path of pit through bone; sd.ri, subdental ridge; sm.ri, sub-Meckelian ridge; t, tooth. The scale below (*a*) applies to (*a*) and (*b*).

and medially with the splenial. The dentary forms the anterior margin of the external mandibular fenestra.

The lateral surface of the dentary is convex, and is ventromedially to dorsolaterally slanted in anterior view. In dorsal/ventral view, the dentary arcs towards the symphysis from posterolateral to anteromedial, and thus the lateral surface is also convex anteroposteriorly. Fine striations extend anteroposteriorly along the left and right dentaries in SAM-PK-6047A (figure 50b,d, str). Similar striations appear to be present in SAM-PK-5867, although the detail of the lateral surface is less well preserved. Anteroposteriorly elongated, posterolaterally opening pits are observable on the lateral surface of the left dentary of SAM-PK-6047A (figure 50, pl.pit), the most prominent of which is near the ventral margin, around half way along the element. Further, directly laterally opening pits (figure 50, l.pit) are visible in the right dentary of SAM-PK-6047A, and less clearly in the right dentary of SAM-PK-5867, below the anteriormost tooth crowns. All these pits connect within the dentary with the main alveolar nerve canal confluent with the Meckelian fossa (figure 50i,j, al.nr).

The ramus of the dentary forms a gently ventrally convex arc in lateral view. The entire anterior end of the dentary is dorsal to the dorsal margin at the posterior end. The curve of the dorsal margin levels off anteriorly, becoming approximately straight below the crowns of the three anteriormost teeth. There is no strong dorsal expansion of the anterior of the dentary accompanied with upturning of the anterodorsal margin as in some pseudosuchians (e.g. *Postosuchus kirkpatricki*, aetosaurs), erythrosuchids [81] and *Silesaurus opolensis*, nor ventral deflection as in sauropodomorphs (see [31]). The anterior margin of the dentary is rounded, curving in an anteroventrally convex arc from posteroventral to anterodorsal and does not taper to a point (unlike in some dinosauromorphs and aetosaurs [31]).

No foramina or ornamentation are visible on the medial surface of the dentary exposed in UMZC T.692 (figure 50f), but large foramina on the medial surface of the dentary (identified in *Osmolskina* as for the passage of the alveolar nerve) can be observed in the CT scans of SAM-PK-5867 and SAM-PK-6047A (figure 50c, al.nr). The Meckelian fossa (=Meckelian groove; figure 50e,f, meck.fos) is around 30% of the depth of the ramus of the dentary, with around 20% of the depth of the dentary below it (the sub-Meckelian ridge—figure 50e,f, sm.ri) and 50% above it (the sub-dental ridge—figure 50e,f, sd.ri); the Meckelian groove can thus be described as being located near the dorsoventral centre of the dentary rather than its ventral margin, unlike in some non-dinosaurian dinosauriforms and in ornithischians [31].

The dorsoventral depth of the Meckelian fossa increases gradually posteriorly at the expense of the sub-Meckelian ridge. The medial surface of the Meckelian fossa is recessed laterally in comparison with the ridges above and below it by around half of the maximum mediolateral width of the dentary. The Meckelian fossa terminates well posterior to the symphysis; among Triassic archosauromorphs only in *Silesaurus* and *Sacisaurus agudoensis* does it extend to the symphysis [31]. The medial surface of the sub-dental and sub-Meckelian ridge is mostly flat, with patches of gentle concavity and convexity.

The splenial articulated with the sub-dental and sub-Meckelian ridges. In the left dentary of individual A of UMZC T.692, the lower splenial facet of the dentary is visible as a groove extending from posteroventral to anterodorsal across the sub-Meckelian ridge (figure 50f, f.spl). This facet's anterior end is between the fifth and fourth dentary teeth, and its posterior end is roughly level mesiodistally with the seventh dentary tooth. The facet becomes gradually dorsoventrally deeper anteriorly, and the Meckelian fossa correspondingly shallower (the fossa disappears altogether anteriorly, immediately posterior to where the splenial ends and the dentary becomes the sole element of the jaw). This facet received the convex ventrolateral margin of the splenial, which moves dorsally away from the ventral jaw margin anteriorly. Anteriorly, the dentary contacts the splenial in a suture formed into two anteroposteriorly elongated interdigitations, with the splenial projecting forward dorsally, then the dentary backward, then the splenial forward again (see line drawing in figure 4a).

In the left dentary of individual A of UMZC T.692, the medial surface of the anterior extreme (extending back to the posterior of what is probably the fourth dentary tooth, and is the anteriormost tooth preserved in the specimen) of the sub-dental ridge is slightly depressed laterally; this is the start of the symphyseal facet which contacted the opposite dentary (figure 50f, f.sym). The symphysis was relatively short, unlike in ornithosuchids, phytosaurs, crocodyliforms, *Effigia* and *Shuvosaurus* [97] where it extends for one-third of the jaw ramus. In coronal section in the CT scan of SAM-PK-5867, the contact between the dentaries can be seen to be a simple appression, but seemingly with a slight medial convexity on the left dentary posteriorly (and concavity on the right) and vice versa more anteriorly (figure 50g,h).

The posterior of the dentary does not divide into the three clear processes of erythrosuchids [81,122] or two processes of *Herrerasaurus* [77]. Instead, there is a single process with a blunt posterior convexity (figure 50a, cnvx), and another very slight posterior projection (figure 50a, prj) at the dorsalmost part of the posterior margin of the dentary. The ventral margin of the lower convexity overlies the dorsal part

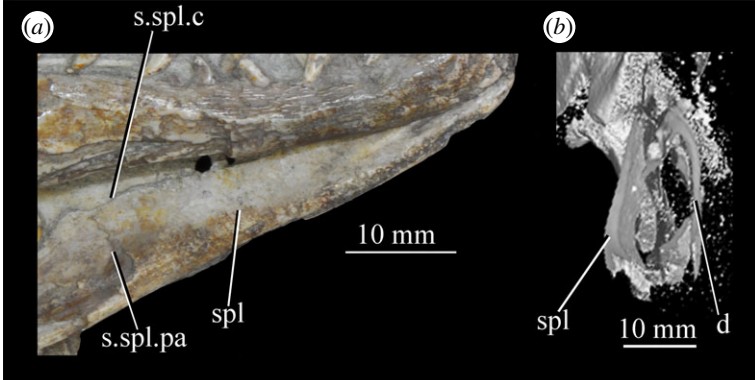

**Figure 51.** Example and details of the splenial of *Euparkeria capensis*. Left splenial of SAM-PK-5867 in (*a*) medial view and (*b*) in coronal cross-section (with CT rendering) in anterior view. c, coronoid; d, dentary; pa, prearticular; s., suture between; spl, splenial.

of the angular laterally, while the posterior part of the dorsal projection overlies the surangular laterally; both contacts are simple appresssions, with all elements slightly convex laterally and concave medially. The coronoid–dentary suture consists of a simple appression, with the coronoid slightly ventrolaterally concave and the dentary correspondingly dorsomedially convex in coronal section (see Coronoid).

There are 13 dentary teeth (nine visible externally in the right dentary and eight in the left of SAM-PK-5867). This is a similar number to basal archosauriforms with similarly short snouts (e.g. *Erythrosuchus* has 13 [81]; *Osmolskina* has around 13 [74]), and fewer than in long-snouted taxa (*Prolacerta* has around 16 [90]; *Proterosuchus fergusi* has around 20 [83]; *Chanaresuchus* has around 18 [80]). In SAM-PK-6047A, the left dentary shows 13 alveoli, with 10 teeth clearly visible externally; six teeth are visible in the less well-preserved right dentary externally, with at least 12 alveoli visible internally. The tooth row begins immediately posterior to the symphysis, and the posteriormost tooth in SAM-PK-5867 and SAM-PK-6047A is three-quarters of the length of the dorsal margin of the dentary from the anterior tip of the dentary, and thus, the dentary tooth row is around two-thirds the length of the maxillary tooth row.

The teeth are dorsoventrally elongated and recurved, anteriorly and posteriorly denticulated (= serrated; unlike those of *Prolacerta* [90]) and homodont, and thus differ in no consistent way from the maxillary teeth in form. As a whole, the dentary teeth are, however, smaller: the largest dentary tooth in the right dentary of SAM-PK-5867 is about two-thirds the size of the largest maxillary tooth. In the right dentary of SAM-PK-5867, the first and third dentary teeth are of roughly equal length and are longer than the other dentary teeth, with the third dentary tooth wider than the first. The second and fourth teeth of the right dentary of SAM-PK-5867 are around half of the size of the first, third and fifth, indicating alternating replacement. Interdental plates between the teeth are visible in medial view. These do not differ from those of the maxilla, and thus are subrectangular, mesiodistally elongated elements lingual to the teeth, overlapping the anterior of one tooth and the posterior of the next in lingual view. Denticle density does not appear to differ consistently along the tooth row, varying between 10 and 6 mm$^{-1}$ in SAM-PK-6047A and SAM-PK-5867.

### 3.2.8.3. Splenial (figures 51 and 27)

Both splenials are preserved in SAM-PK-5867, with the left splenial almost entirely exposed in medial view (figure 51*a*). The left splenial is also visible in medial view in UMZC T.692 (figure 27). A right and a left splenial are also well preserved in SAM-PK-6050 (figures 11*b* and 26*e*). Other examples of the splenial not used in the description are present in AMNH 2239 and possibly SAM-PK-13667. The splenial forms most of the anterior of the medial surface of the mandible. Following the overall shape of the mandible, the splenial thus becomes shallower anteriorly, with the ventral margin moving upwards. Posteriorly, the splenial articulates with the prearticular. Ventrally, on its lateral side, the splenial contacts the dentary, and at its posterior extreme the angular, and dorsally and posterodorsally, the splenial contacts the coronoid.

The medial surface of the splenial shows no ornamentation or foramina—there is thus certainly no foramen comparable to that found in eusaurischians [31,44]. The medial surface is mostly flat, curving laterally at its ventral margin in a ventromedially convex curve. The dorsoventrally central third of the medial surface of the splenial is slightly mediolaterally concave. The lateral surface of the splenial is smooth, concave in coronal section ventrally and slightly convex dorsally where it articulates with the dentary (figure 51*b*).

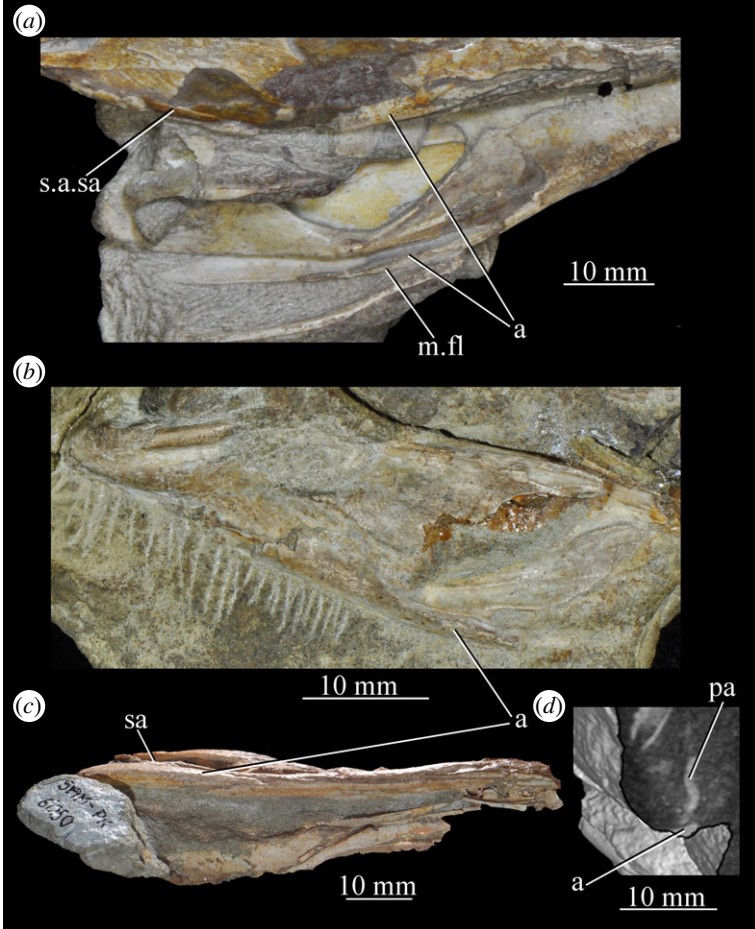

**Figure 52.** Examples and details of the angular of *Euparkeria capensis*. (*a*) SAM-PK-5867 in right lateral view; (*b*) UMZC T.692 individual A in right lateral view; (*c*) SAM-PK-6050 in ventral view; and (*d*) CT image of the right mandible of SAM-PK-5867 in right anterolateral view with coronal cross-section showing articulation of angular and prearticular. a, angular; m.fl, medial flange of angular; pa, prearticular; s.a.sa, surangular–angular suture; sa, surangular.

The articulation of the splenial with the prearticular consists of an anteriorly convex curved suture (figure 51, s.spl.pa), which reaches its most anterior point about one-quarter of the height of the splenial dorsal from the ventral margin of the splenial. The suture is non-interdigitating, but meanders slightly. The articulation of the splenial with the coronoid consists of a non-interdigitating suture extending posteroventrally from the dorsal margin of the mandible to the splenial–prearticular contact (figure 51, s.pl.c).

### 3.2.8.4. Angular (figure 52)

Both angulars are preserved in SAM-PK-5867 (the right in lateral view; the left is more complete, but partially obscured in lateral view and partially visible in medial view—figures 4 and 52*a*). The right angular of individual A and B in UMZC T.692 is visible in lateral view (figures 18, 19 and 52*b*), and the left angular of individual A is visible in medial view (figure 27). A right angular is preserved in SAM-PK-6050 exposed apart from its most dorsal medial portion (figure 26*f*). Other examples of the angular, not used in the description, are preserved in AMNH 2239, SAM-PK-7699, SAM-PK-13665, SAM-PK-13666 and SAM-PK-13667. The angular forms the ventral part of the deepest part of the jaw ramus (where the ventral margin of the mandible forms a point of inflection—henceforth the ventral point of inflection) on the lateral side, the ventral margin of the external mandibular fenestra and the posteroventral part of the ramus of the mandible. The angular extends posteriorly to close to the jaw joint, but does not reach the articular, with the surangular—which the angular contacts dorsally—separating angular and articular. Anteriorly, the angular contacts the dentary, and ventromedially the prearticular.

The lateral surface of the angular is ventrolaterally convex. Ventrally the lateral surface curves medially in a ventrolaterally convex arc (in anterior/posterior view) to reach the mediolateral midpoint of the jaw and contact the prearticular. Posterior to the ventral point of inflection of the jaw, the angular becomes progressively shallower dorsoventrally. Anterior to the ventral point of inflection of the jaw, the angular remains at the same depth as at the ventral point of inflection for two-thirds of the length of the element anterior to the ventral point of inflection, then decreases in depth and is overlapped dorsally by the dentary, accentuating this decrease. Foramina, striations and ornamentation are absent on the lateral surface of the angular. In dorsal or ventral view, the lateral surface of the angular, and indeed, the bone and mandible as a whole, is slightly laterally convex, with a point of inflection at the ventral point of inflection of the jaw (figure 52c).

The medial surface of the angular is slightly anteroposteriorly and dorsoventrally concave, mirroring the convexity of the lateral surface, and lacks ornamentation or foramina. The ventral margin of the angular is ventrally convex, with an arc of about 20° with a point of inflection at the ventral point of inflection of the jaw (i.e. at the anteroposterior midpoint of the angular and the external mandibular fenestra). This differs from the almost horizontal ventral margin of the angular in taxa with dorsoventrally shallow, long jaws such as *Chanaresuchus* [80] or *Gracilisuchus* [93], is relatively similar to, e.g. *Garjainia prima* (as reconstructed by Ochev [122]) and many carnivorous pseudosuchians (e.g. [75,76]), but is not as pronouncedly angular as in *Erythrosuchus* [81]. The dorsal margin of the angular fundamentally mirrors this shape, being dorsally concave, with a very gentle point of inflection at the anteroposterior midpoint of the external mandibular fenestra, thereby—together with the surangular—making the ventral margin of the fenestra into a dorsally concave arc centred at the midpoint of the fenestra.

The articulation of the surangular and the angular consists of a curved contact extending in an anterodorsally convex arc from anteroventral to posterodorsal in lateral view (figure 52a, s.a.sa). As preserved on the right-hand side in SAM-PK-5867 (figure 53c), the angular simply abuts the surangular, but based on the dorsal extent of the better preserved (but disarticulated) angular of the left-hand side of the same specimen, the angular may have laterally overlapped the ventral margin of the surangular; the exact dorsal extent of the angular is unclear. There is no pronounced facet for the angular as was tentatively identified in *Osmolskina* [74], and the dorsal margin of the 'facet' in the latter seems to correspond to the area above the central ridge on the lateral surface of the surangular in *Euparkeria* (figure 53, ri2); if this area has been misidentified in *Osmolskina*, this would indicate that the jaw is somewhat deeper posteriorly than reconstructed for the taxon. The angular does extend much further dorsally in some other taxa (e.g. *Batrachotomus* [75], *Prestosuchus chiniquensis* [135]), and there is a clear facet for the angular on the surangular, but the angular still does not immediately abut the upper lateral ridge on the surangular (figure 53, ri1) as is indicated to be the case in *Osmolskina*. The suture of the prearticular with the angular is via a medially depressed facet along the ventral margin of the prearticular which received the lateral surface of a dorsomedially directed medial flange (figure 52, m.fl) of the angular which overlapped the prearticular medially and cupped it ventrally (figure 52d), though the overlap becomes less pronounced posteriorly.

### 3.2.8.5. Surangular (figure 53)

Both surangulars are preserved in SAM-PK-5867, with the right element visible fully in lateral view and the left partly visible in medial view (figure 53a,b). Both are also preserved in SAM-PK-6047A, but are damaged, and that on the left only is exposed fully (in lateral view; figure 8). The right surangular of individual A (visible in lateral view) is preserved in UMZC T.692 (figure 19). A right surangular is preserved in SAM-PK-6050 in lateral view (figure 11a). Other examples of the surangular, not the primary basis of this description, are preserved in AMNH 2239, SAM-PK-7699, SAM-PK-13665, SAM-PK-13666 and SAM-PK-13667. The surangular forms most of the posterodorsal part of the lateral side of the mandible, the anterior portion of the cotyle fossa and the posterodorsal margin of the external mandibular fenestra. The surangular articulates ventrally with the angular, posterodorsally with the articular, anteriorly and anteroventrally with the dentary and anteromedially with the coronoid.

Most of the lateral surface of the surangular is a gently dorsoventrally convex sheet. A laterally convex, rounded ridge (figure 53, ri1) extends along the dorsal margin of the surangular, widening posteriorly and merging with the rounded ridge on the side of the articular. A similar ridge is present in many stem (e.g. *Erythrosuchus* [81]; *Chanaresuchus*—PULR 07) and crown (e.g. *Batrachotomus* [75]; *Alioramus altai* [130]) archosaurs. A groove (figure 53, gr) is formed below the posterior part of the ridge due to the overhang of the ridge and the convex curvature of the lateral surface of the surangular ventral to the ridge. A surangular foramen, as in *Erythrosuchus* [81], is present within the groove (figure 53, sa.for). The lateral surface of the right surangular in SAM-PK-5867 below the groove/dorsal ridge shows a ridge (shallower

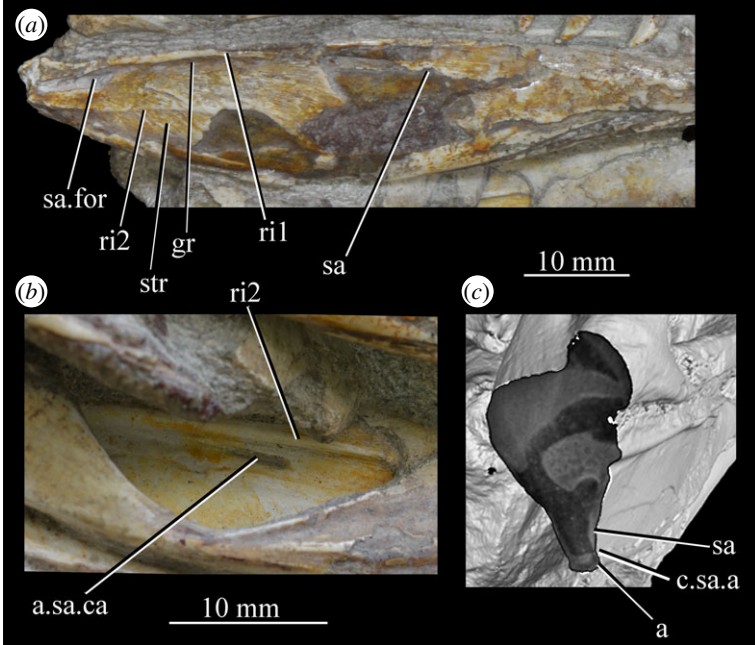

**Figure 53.** Examples and details of the surangular of *Euparkeria capensis*. (*a*) Right surangular of SAM-PK-5867 in lateral view; (*b*) left surangular of SAM-PK-5867 in medial view; and (*c*) CT image of the right mandible of SAM-PK-5867 in right posterolateral view with coronal cross-section showing angular–surangular contact. a, angular; a.sa.ca, anterior surangular canal; c., contact between; gr, groove; ri1–2, ridges 1–2; sa, surangular; sa.for, surangular foramen; str, striations.

and narrower than the dorsal ridge) extending from near the posteriormost point of the surangular down across the dorsoventral centre of the element to the margin of the external mandibular fenestra. The surface also shows striations (figure 53, str) radiating out anteriorly from the posterodorsal part of the element in anterodorsally convex arcs. *Euparkeria* lacks the vascular foramina on the lateral surface in *Osmolskina* [74].

The dorsal margin of the lateral surface forms a very gentle arc. This arc is less pronounced than in, for example, *Erythrosuchus* [81] (where it is also more sharply angular) or *Chanaresuchus* [80] and is similar to the shape in *Proterosuchus fergusi* [83], although the jaw ramus as a whole is differently shaped. The margin of the external mandibular fenestra formed by the surangular is chevron shaped (figure 53*a*), with the point of the chevron pointing posteriorly and level with the ridge extending across the dorsoventral centre of the lateral surface of the surangular. The sides of the chevron leading anteriorly from the point are gentle concave curves, making the posterior part of the external mandibular fenestra slightly rounded. The lateral surface of the surangular is slightly laterally convex in dorsal/ventral view, following the overall laterally convex arc of the mandible (figure 52*c*).

The surangular extends medially as a distally tapering process appressed to the anterior of the articular (the medial lamina—figure 49*a*, ml.sa), which curves posteromedially in an anteromedially convex curve. The anterodorsal part of the process is drawn up into a ridge to form the anterior border of the mandibular cotyles, with the surface posterior to this probably depressed to form the anterior of the cotyles themselves; the suture between the articular and surangular is difficult to discern in this region.

The centre of the medial surface of the surangular is visible through the internal mandibular fenestra on the left-hand side in SAM-PK-5867. The surface is smooth and dorsoventrally and anteroposteriorly concave. A ridge (figure 53, ri2) extends anteroposteriorly along the dorsoventral centre of the element, following the path of the central ridge visible laterally. Below this ridge, around one-third of the total length of the bone posterior to the anterior margin is a relatively small anteroposteriorly elongated foramen, presumably homologous with the foramen identified as the entrance for the anterior surangular canal in *Osmolskina* [74] (figure 53, a.sa.ca). This foramen opened on the lateral surface of the bone as in *Osmolskina* [74], but the canal also continued posterior to this point (though narrowing prior to re-expanding) to connect with the posterior surangular foramen.

### 3.2.8.6. Coronoid (figure 54)

Both coronoids are preserved in SAM-PK-5867 (figure 54) and a left coronoid is preserved in SAM-PK-6050 (figure 26*e*), and in SAM-PK-13666 (figure 14) and these form the basis of the description. A partial

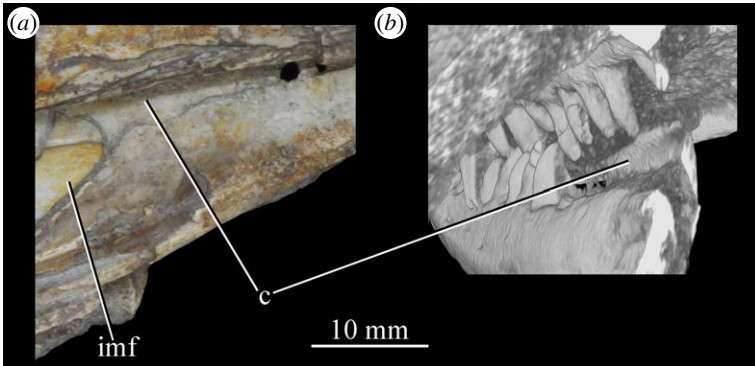

**Figure 54.** Left coronoids of *Euparkeria capensis* specimen SAM-PK-5867 in (*a*) medial view and (*b*) posterolateral view (CT scan rendering with coronal cross-section). c, coronoid; imf, internal mandibular fenestra. Same scale applies to (*a*) and (*b*).

left coronoid appears to be present in SAM-PK-6047A based on CT data. The coronoid forms the dorsalmost portion of the medial surface of the mandible anterior to the internal mandibular fenestra (of which it forms the anterodorsal margin). The coronoid was not extensively visible in lateral view, but may have projected a small way above the surangular and dentary (this is difficult to assess, as the element is slightly disarticulated in the specimens); in most archosauromorphs, it is not visible in lateral view, unlike in *Prolacerta* [90]. The coronoid is an elongated element which tapers dorsoventrally anteriorly, suturing with the splenial ventrally, with the prearticular posteroventrally and with the dentary dorsolaterally. It is wider mediolaterally dorsally than ventrally and tapers to a point dorsally in cross-section (figure 54*b*).

The prearticular and splenial overlap the ventral part of coronoid medially, while the surangular and dentary overlap the ventral part of the coronoid laterally (figure 54). The medial surface of the coronoid exposed above the prearticular and splenial bones is slightly convex in coronal section, and the lateral surface above the dentary and surangular is slightly concave (figure 54*b*). Below the point where the coronoid becomes obscured by the elements medial and lateral to it the medial and lateral surfaces become flattened and extend towards each other. The coronoid lacks ornamentation, foramina or dentition. The posterior margin of the coronoid is an anteriorly convex curve, forming the anterior extent of the internal mandibular fenestra (figure 54*a*). The contact of the coronoid with the bones surrounding it consists of simple appressions. The line of overlap between the prearticular and coronoid is convex dorsally, that between the coronoid and splenial is slightly concave dorsally more posteriorly then slightly convex more anteriorly, and that between the coronoid and dentary is roughly horizontal (figure 54).

### 3.2.8.7. Prearticular (figure 55)

The left and right prearticular is present in SAM-PK-5867 (figure 55) and SAM-PK-6050 (figure 26*e*) and is visible in medial view. SAM-PK-6047A preserves the prearticulars, but they are only visible using CT data as the mandibles are not exposed in medial view. Part of the left prearticular is visible in medial view in individual A of UMZC T.692 (figure 27) and in SAM-PK-13666 (figure 14). The prearticular forms the ventral point of inflection of the mandible on the medial side and the ventral margin of the internal mandibular fenestra. Anteriorly, the prearticular contacts the splenial, anterodorsally the coronoid, laterally and ventrally the angular, posterodorsally the articular and anterodorsal to this the surangular.

The prearticular is shallowest dorsoventrally at the ventral point of inflection of the jaw and fans out anterodorsally and posterodorsally in what can be described as a 'bow-tie' shape with the wings of the tie dorsally deflected. Thus, the dorsal and ventral margins form dorsally concave arcs with points of inflection at the ventral point of inflection of the jaw. This ventral point of inflection is more pronounced than in shallow, horizontal-jawed taxa (e.g. *Chanaresuchus* [80]), but less pronounced than in *Erythrosuchus* [81] (see Angular). The anterior and posterior ends of the element are rounded. The prearticular bulges medially where it is dorsoventrally narrowed at the ventral point of inflection (figure 55, blg); the anterior and posterior ends do not show this bulging and are flat and sheet-like.

A rounded ridge (figure 55, ri1) extends out from this bulge along the dorsal margin of the prearticular, and another similar ridge (figure 55, ri2) extends out around one-third of the way up the bone from the ventral margin; the area between the ridges forms a medially concave groove. The ridges end around

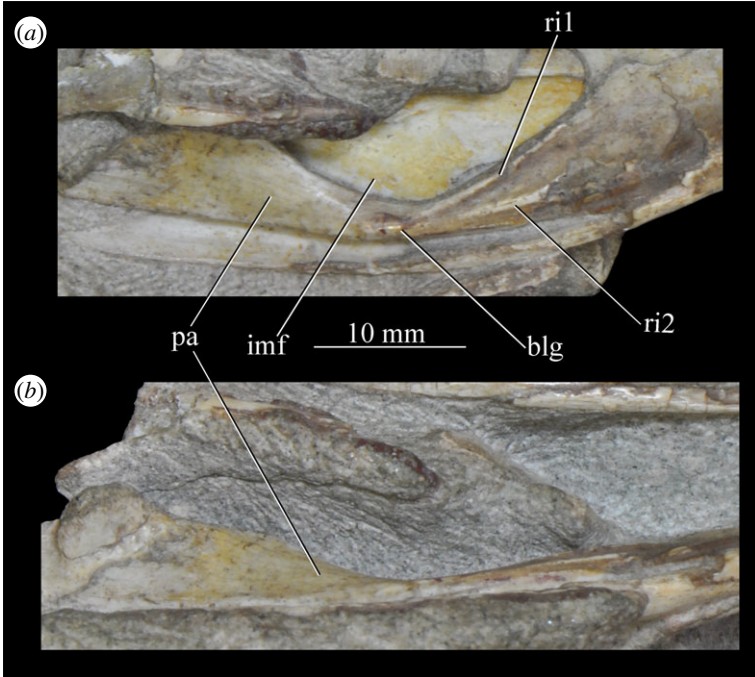

**Figure 55.** Left prearticular of *Euparkeria capensis* specimen SAM-PK-5867 in (*a*) medial and (*b*) ventral view showing details of morphology. blg, bulge; imf, internal mandibular fenestra; ri1–2, ridges 1–2; pa, prearticular.

two-thirds of the way from the ventral point of inflection to the anterior margin of the prearticular. Anterodorsal to the end of the ridges, the medial surface of the prearticular is flat, and shows no ornamentation or foramina. Posterior to the ventral point of inflection, the medial surface is smooth and dorsoventrally convex near the ventral point of inflection, becoming less so further posterodorsally. In dorsal/ventral view (figure 55*b*), the medial surface extends at a slight posterolateral to anteromedial diagonal until the ventral point of inflection, after which it moves at a similar angle posteromedial to anterolateral. Altogether, the element thus appears as if it were a vertically orientated rectangular sheet with the short sides of the rectangle directed anteriorly and posteriorly, and which had been pinched dorsoventrally together and pulled slightly medially at its centre.

## 3.3. Anatomical network analysis

### 3.3.1. Network parameters

Modularity parameters (figure 56) are here compared with six amniote species as described by Werneburg *et al.* [60] with archosaur representatives being *Tyrannosaurus rex*, *Gallus gallus* and *Alligator mississipiensis*. In addition, the marine turtle *Dermochelys coriacea*, the tuatara *Sphenodon punctatus* and the opossum *Didelphis virginiana* are compared (figure 56*a*). *Euparkeria capensis* has a relatively high number of skeletal elements in the skull (i.e. nodes; $N = 62$), which is between *Tyrannosaurus* ($N = 63$) and *Sphenodon* ($N = 58$) (figure 56*b*,*c*), both of which were considered to show some degree of cranial kinesis [60]. The same pattern observed for nodes is visible for the number of links (connections between bones: $K$) (figure 56*b*,*d*). The density of connections ($D$), often considered as one of the proxies of morphological complexity but also representing integration of the skull [60] and the 'burden' placed on each element (i.e. how connected each element on average is, and thus often how constrained it is by development) [136], is the lowest in *Euparkeria* ($D = 0.077$) among all compared species with only the tuatara ($D = 0.085$) being relatively close to that (figure 56*b*,*e*). The clustering coefficient ($C$), often considered a proxy of integration across the skull as a whole (i.e. between modules—its inverse represents integration within modules [136]), is relatively low in *Euparkeria* ($C = 0.354$), and is also a measure of complexity in so far as this relates to functional, evolutionary and developmental integration [136]. Only *Gallus* ($C = 0.304$) has a lower anatomical integration (figure 56*b*,*f*). The mean shortest path length ($L$) is also usually considered as a proxy of integration and again also as measure of complexity in so far as this relates to the integration of the system [136], and represents the mean shortest connection between all pairs of elements; for this measure *Euparkeria* ($L = 3.45$) falls into the range of all

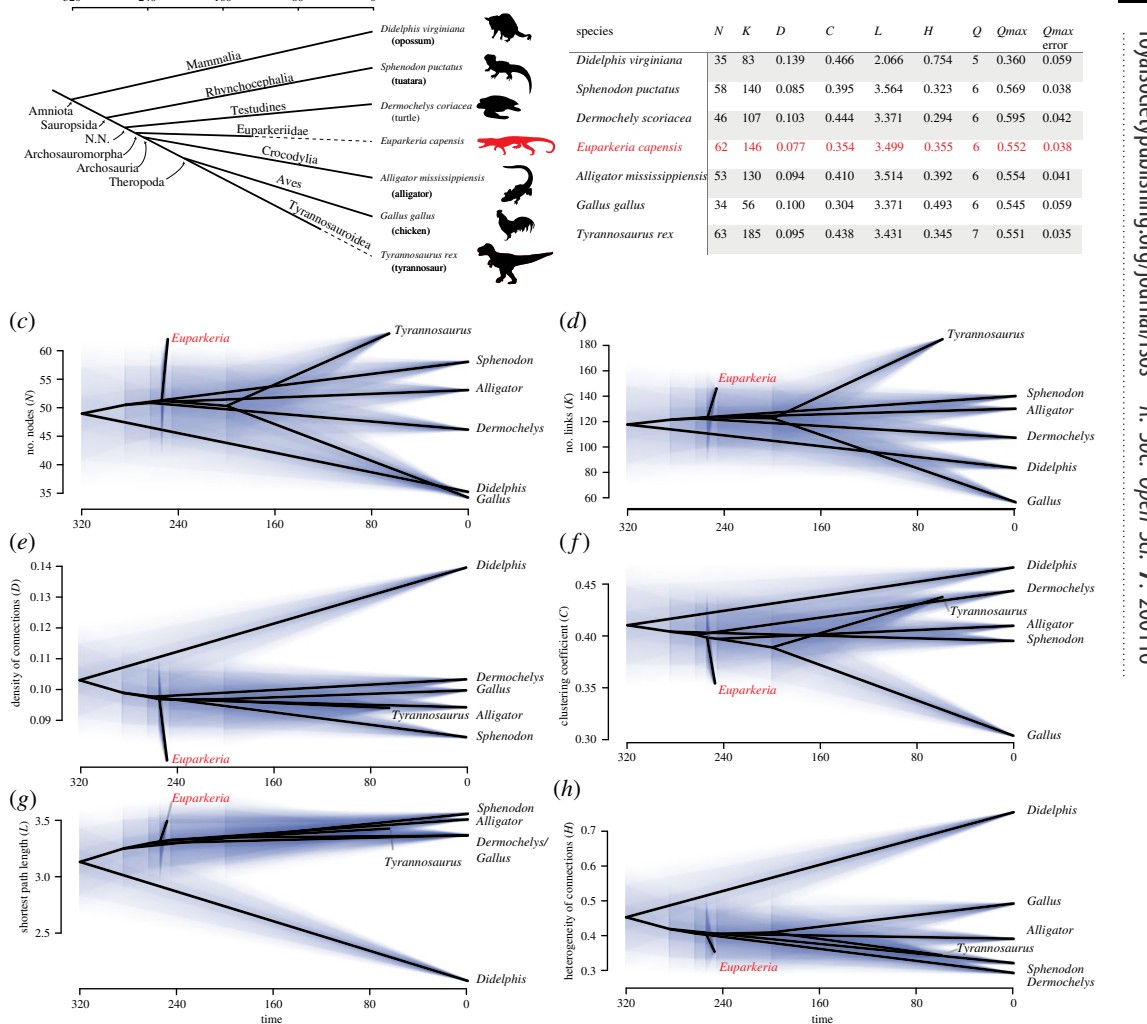

**Figure 56.** Phylogenetic arrangement and anatomical network parameters based on Werneburg *et al*. [60] and the present study. (*a*) Phylogenetic position of *Euparkeria capensis* as sister taxon to the archosaurs included in Werneburg *et al*. [60]. (*b*) Values of the network parameters and results of the modularity analysis. (*c–h*) Phylograms of network parameters. Shading represents 95% confidence intervals for ancestral state reconstructions using residual maximum likelihood. *N*, nodes, number of skeletal elements; *L*, links, number of contacts among skeletal elements (i.e. physical joints); *D*, density of connections, actual number of links divided by the maximum number possible; *C*, mean clustering coefficient, average of the ratio of a node's neighbours that connect among them; *L*, mean shortest path length, average number of links required to travel between two nodes; *H*, heterogeneity of connections, standard deviation divided mean of the number of connections of every node; *Q*, number of *Q*-modules, modules identified using optimization function *Q*; $Q_{max}$, maximum value of *Q* calculated for the best partition of the dendrogram; $Q_{max}$ error, expected error of $Q_{max}$. Time given in millions of years.

compared reptiles (figure 56*b,g*), with *Didelphis* showing an exceptionally short path length. The heterogeneity of the number of connections per element is considered as a proxy of anisomerism, which is effectively a measure of the average complexity of each element [136] (considered to increase despite the number of elements decreasing during tetrapod evolution in 'Williston's law' [65]). *Euparkeria* (*H* = 0.3553) shows a higher value only than *Tyrannosaurus*, *Dermochelys* and *Sphenodon* (figure 56*b,h*).

### 3.3.2. Modularity

The $Q_{max}$ value can be interpreted as representing the integration within modules, and its inverse as the integration across the network as a whole [136]; that of *Euparkeria* ($Q_{max}$ = 0.552) is similar to *Alligator* and *Tyrannosaurus*, but lower than *Dermochelys* and *Sphenodon* and higher than *Gallus* and *Didelphis* (the latter being exceptionally low); this indicates a relatively low integration across the cranium as a whole.

Regarding the number of *Q*-modules on each cranial side, *Euparkeria* has the same number of modules as all compared Reptilia (*Q*-modules = 6) with the exception of *Tyrannosaurus* (*Q*-modules = 7) (figure 56*b*). The left and right lower jaws form separate modules, which is a common pattern in all reptiles (blue in figure 57) [60]. The anterior region of the skull forms a 'snout module' (purple in figure 57) on each side, containing premaxilla, maxilla, prefrontal, lacrimal and nasal. A 'supraorbital module' on each side (red in figure 57) consists largely of dermal bones—the frontal, postfrontal, postorbital and parietal (but see below)—but also includes the laterosphenoid and epipterygoid. A 'cheek module' on each side (orange in figure 57) contains the jugal, ectopterygoid, quadratojugal, squamosal and quadrate.

The 'palate module' on each side (pink in figure 57) contains the vomer, palatine and pterygoid (but not the ectopterygoid, which is in the 'cheek' module). A separate left and a right braincase module (dark yellow in figure 57) are also found. Only the prootics, opisthotics and exoccipitals clearly correspond to the left or right braincase module. In different runs of the analysis, the interparietal, supraoccipital, parabasisphenoid and basioccipital either belong to the left or to the right braincase module (pale yellow 'labile bones' in figure 57). The reason for this pattern is that these unpaired bones equally attach to the left and right side skeletal elements and are, as such, randomly recovered to belong to only one braincase module in each run of the same analysis. A similar pattern of lability is found for the parietal, but here the right or left parietal either belongs to the supraorbital or the braincase module on the respective skull side module (pale red 'labile bones' in figure 57). This indicates that the parietal is less integrated into the skull than other bones that clearly belong to one particular module.

## 3.4. Phylogeny

The phylogenetic analysis recovered 248 most parsimonious trees of 1344 steps. The best score was hit 953 times out of the 1000 replications. In a strict consensus, *Euparkeria* was found to be in the same position as without this rescoring, placed outside of either the pseudosuchian or avemetatarsalian crown archosaur lineage as the sister taxon to a clade formed of *Osmolskina*, *Halazhaisuchus* and '*Turfanosuchus shageduensis*' (see electronic supplementary material). Absolute bootstrap and Bremer support for this euparkeriid clade was low (less than or equal to 30, 1), and were slightly or much higher for Pseudosuchia (40, 1), Ornithodira (80, 4) and Archosauriformes (38, 1). Excluding '*Turfanosuchus shageduensis*' had no effect on the analysis except slightly changing the bootstrap values (41, 39 and 85 for Euparkeriidae, Pseudosuchia and Ornithodira, respectively), and that the best score was found in 961 out of 1000 replications.

# 4. Discussion

This study provides a number of novel observations on the cranial anatomy of *Euparkeria capensis*, which, along with detailed redescriptions of all dermato- and splanchnocranial and mandibular elements, will facilitate the placement and use of the taxon in future phylogenetic analyses, which is likely to be particularly valuable as *Euparkeria* continues to be widely used as an outgroup in archosaur phylogenetic analyses [39–51]. Furthermore, the reassessment provided allows further elucidation of major patterns in the evolution of the cranium and ecomorphology of the Archosauria and their close relatives. The more complete and accurate understanding of the cranial and mandibular elements afforded by this study has also allowed an AnNA to be conducted, the results of which are discussed below, and which has bearing on the evolution of the diapsid skull and its relationship to changing diet.

## 4.1. Novel information on the cranium of *Euparkeria capensis* and its significance

Our reassessment of the cranial morphology of *Euparkeria* has revealed several major new pieces of information. These are highlighted below to facilitate their use by subsequent workers, especially in phylogenetic analysis, and their importance and context is discussed; many other more minor aspects of morphology, including the exact trajectories and forms of various sutures, were also clarified, and the relevant information can be found in the description and reconstruction (figure 3) as needed.

The morphology of the epipterygoid was revealed by CT scanning, but is not exceptional: it forms a curved, tapered element extending laterally (see above and figure 45). *Euparkeria* is clearly demonstrated to have four rather than three premaxillary teeth (*contra* Nesbitt [31] but agreeing with Ewer [54] and Ezcurra [36]); this is not unusual and the character is labile, but four premaxillary teeth are notably also found in 'rauisuchians' (e.g. *Postosuchus* [73], *Batrachotomus* [75]), which possess an overall similarity of

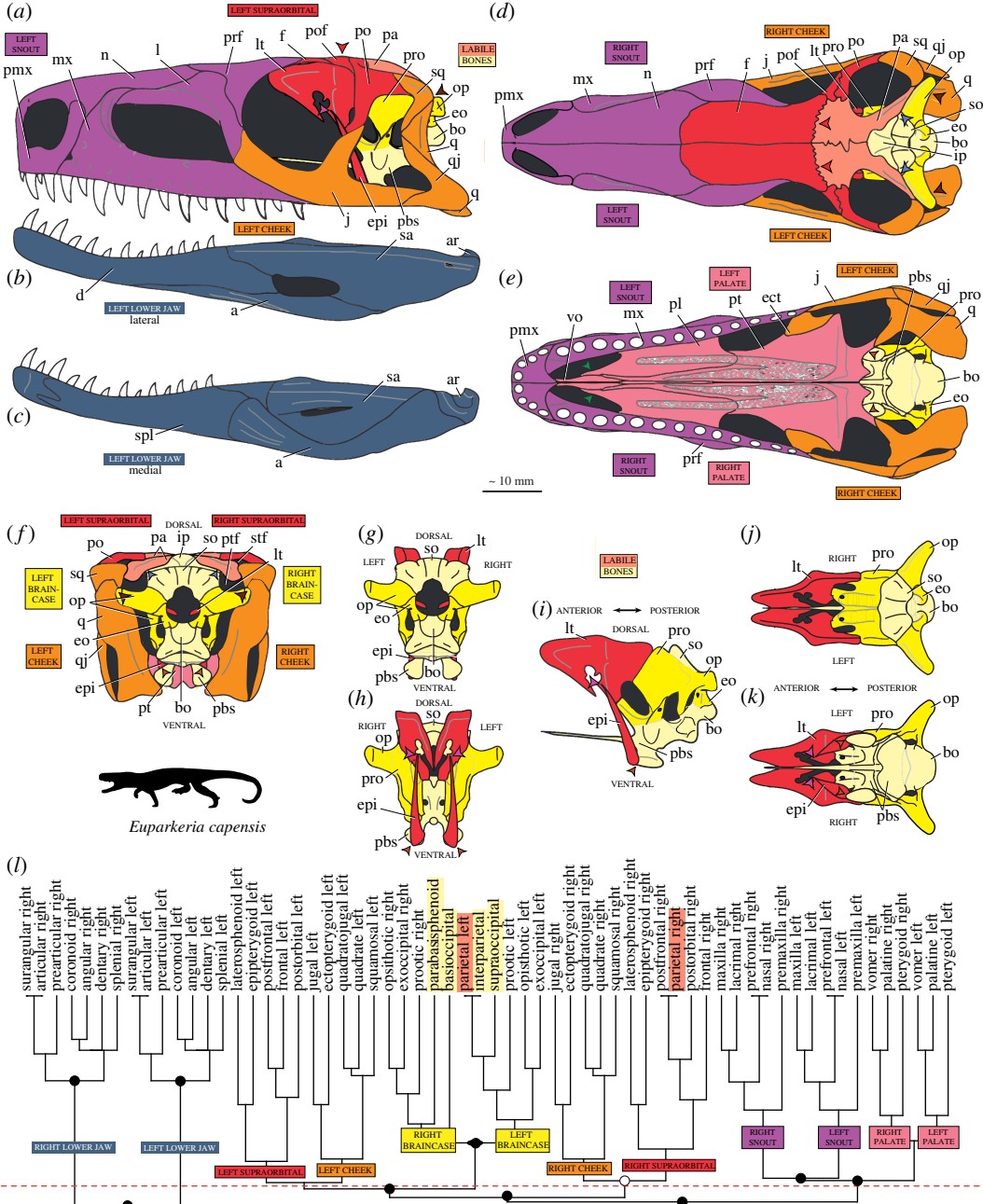

**Figure 57.** Modularity of the skull of *Euparkeria capensis*. Skull in lateral view (*a*). Left lower jaw in lateral (*b*) and medial view (*c*). Skull in dorsal (*d*), ventral (*e*) and posterior (*f*) view. Braincase in posterior (*g*), anterior (*h*), left lateral (*i*), dorsal (*j*) and ventral (*k*) view. (*l*) Dendrogram showing the hierarchical clustering of nodes for the anatomical network in *Euparkeria capensis*. Anatomical regions (modules) are coloured and labelled on the branches; labile bones, not strongly associated with a particular module and being placed on different sides in different analyses, are shaded in light red and light yellow (see text for further details). The horizontal red dashed line marks the best partition into *Q*-modules (above which identified modules best represent the network's modularity). Clusters with a circle are all S-modules; filled circles in clusters mark the statistical significance of this potential module (white, *p*-value < 0.05; black, *p*-value < 0.001). Arrow heads in the skull reconstruction indicate potential intracranial articulations (corresponding to Werneburg & Maier [137]); red: metakinesis (between frontal and parietal), blue: mesokinesis (between skull roof and braincase); orange: basipterygoid articulation (between parabasisphenoid and pterygoid), brown: articulation between quadrate and braincase; green: rhynchokinesis (between palate and ethmoid region), pink: articulation between epipterygoid and 'supraorbital' bones (note, also a ventral articulation of the epipterygoid with the pterygoid can be present). a, angular; ar, articular; bo, basioccipital; d, dentary; ect, ectopterygoid; eo, exoccipital; epi, epipterygoid; f, frontal; ip, interparietal; j, jugal; l, lacrimal; lt, laterosphenoid; mx, maxilla; n, nasal; p, parietal; pa, prearticular; pbs, parapasisphenoid; pl, palatine; pmx, premaxilla; po, postorbital; pof, postfrontal; pro, prootic; prf, prefrontal; pt, pterygoid; ptf, posttemporal fenestra; q, quadrate; qj, quadratojugal; sa, surangular; so, supraoccipital; spl, splenial; sq, squamosal; v, vomer.

skull shape to *Euparkeria* (see below). Furthermore, the ectopterygoid is shown to be 'double headed', again approximating the morphology of 'rauisuchians' and some other pseudosuchian-line archosaurs [31], but this is probably not a strong indicator of pseudosuchian affinities given that the morphology in ornithodirans is generally rather unclear due to preservation [77,78]. Even following the rescoring of these characters in the matrix of Nesbitt [31], the placement of the taxon remains unchanged, outside of crown Archosauria (figure 1). *Euparkeria* does, however, show similarity in overall skull shape, osteoderm shape and to some extent ankle morphology [21,31] with crown pseudosuchians, and although all of these characteristics are currently considered to be archosauriform plesiomorphies, the placement of *Euparkeria* itself must continue to be tested.

CT scanning has significantly improved our knowledge of the palatal dentition of *Euparkeria*. Palatal teeth revealed by CT scanning are not as small as described by Ewer [54] and have a greater recurvature; exposed palatal teeth, the source of previous observations, appear to be largely damaged, diminishing their true height and extent of curvature. The vomer, as described by Gow [124], is confirmed to have two palatal teeth. The overall arrangement of palatal teeth is broadly similar to that reconstructed by Ewer [54], with two fields of small, nonlinearly arranged teeth: T2 extending anterolaterally and T3 anteriorly—there is no T1 along the posterior margin of the pterygoid, and the overall arrangement is unique (see Diagnosis). Ezcurra [36] (character 196), scores the teeth in T2 of *Euparkeria* as a single row (state 2), but in fact they occupy a narrow field with no clearly defined rows, placed on a raised area (state 0); the teeth of field T3, which also form an irregular field on a raised platform, were, however, correctly scored as such by Ezcurra [36]. Both of these characters, although representing a much-needed attempt to incorporate further information from palatal dentition into phylogenetic analysis, were newly formulated by Ezcurra [36], and may require subsequent revision. In other taxa, different interpretations of the arrangements of teeth may be offered. For example, a pair of teeth next to one long row of teeth in T2 of *Howesia* [105] could be interpreted as two rows with one extremely short row or, equally, as a single row with a pair of additional teeth in which one or both could be part of the T3 field. There is also species-level variability in the arrangement of pterygoid teeth. Ezcurra [36] scored the tanystropheid *Macrocnemus bassanii* as having distinct T2 and T3 fields (character 195) and the T2 as two parallel rows (character 196). Recent work by Jaquier *et al.* [138] on another species, *Macrocnemus fuyuanensis*, revealed a pair of widely separated laterally oriented rows, the anterior one could be interpreted as a separate portion of T3 that would be scored as state 1 of character 195 [36]. Furthermore, *Tanystropheus* appears to show size-related differences in palatal dentition, with dentition being more extensive in smaller individuals [140]. Given the complexity and variety of different arrangements present, it may be preferable to score the presence of each field (T1–3), and the form of each field as present (i.e. row or nonlinear field) as separate characters, which could be downweighted as a block if considered appropriate. Irrespective of formulation, given the novelty of these characters they should be gradually rechecked by future workers.

## 4.2. *Euparkeria* in the context of archosauromorph cranial ossification changes

Through tetrapod evolution, there is a general tendency towards loss of ossified elements but the increasing complexity of the form of each element (increasing anisomerism), known as Williston's law [65,139,141,142], and this trend is also seen during ontogeny, with adult forms tending to have reduced numbers of elements and increased anisomerism [143]. The archosaur line also demonstrates this pattern, with separately ossified/retained elements tending to be the plesiomorphic condition [31]. As expected given its phylogenetic position, *Euparkeria* shows the highest number of nodes of all taxa (i.e. ossified elements) in the AnNA conducted (figure 56), and a relatively low value of heterogenity of connections, indicating low anisomerism. The skull of *Euparkeria* also shows a long shortest path length and low density of connections, indicating low integration of the skull (contrasting in the extreme with the highly integrated skull of *Didelphis*—figure 56) and also low complexity as measured in terms of functional, evolutionary and developmental integration and interdependence (see [136]). The integration of the skull as measured by the clustering coefficient is also very low, and modularity ($Q_{max}$) and number of modules are high. *Euparkeria* thus potentially represents a low-integration, low-complexity skull which would subsequently become more integrated—concurrent with decreased number of elements and increased anisomerism—in many later taxa, especially in crown birds, which show an exceptional, and arguably peramorphic pattern of increased cranial integration and fusion of elements [143]. However, the trend of integration is not clear cut because *Gallus* shows exceptionally low integration across the skull as a whole due to crown avian development of a high degree of kinesis *between* parts of the skull [144,145], although at least *Gallus* shows high integration *within* modules and a much reduced number of elements and

increased anisomerism of each element, and this is probably the case in other crown birds [143]. Crocodylians, on the other hand, lost kinesis to increase skull strength but did not develop such an integrated or complex skull [146], with—as quantitatively demonstrated here for *Alligator*—the number and heterogeneity of elements closer to the ancestral condition.

As a comparison, mammals developed a highly complex and anisomeric, but integrated, akinetic skull [147], as illustrated here by the divergence of *Didelphis* from reptilian taxa in many of the network parameters. *Didelphis* is closest to *Gallus* in terms of the number of nodes (elements) and heterogeneity of connections (anisomerism), demonstrating the similar development of a skull composed of fewer, more complex elements, but is at the opposite extreme in terms of integration (clustering coefficient) because the skull is completely akinetic and highly integrated as a whole. The low integration and high modularity of the skull indicate that some degree of ancestral kinesis may have been present in *Euparkeria*, but inferring kinesis from the fossil record is challenging [146], and there is certainly no indication of the kind of strong kinesis seen in birds and squamates [148,149] (see below).

In terms of specific elements, *Euparkeria* generally shows an ossification pattern which would be expected of a Triassic non-archosaur archosauriform or early crown archosaur (see phylogenetic position in figure 1). Unlike most non-archosauriform archosauromorphs (for example, *Prolacerta* [90], *Proterosuchus* [36]) but like other archosauriforms (e.g. [75,77,81]), *Euparkeria* lacks a septomaxilla; the loss of this element may be connected to sensory changes (see below). As in most archosauromorphs (e.g. [75,77,81,90]), but contrasting with stem saurians (e.g. *Youngina capensis*—GHG K 106), tabulars are lacking, but unlike in most archosauromorphs, the suture between an embryonic tabular and parietal may still be visible (figure 33). As in most stem archosaur taxa (with some exceptions, e.g. *Mesosuchus* [109]), *Euparkeria* retains the plesiomorphy of a separate ossified interparietal (= postparietal), distinguishing it from crown taxa [31,75,77]. Furthermore, the exoccipitals and opisthotics are not fused, contrasting with many crown taxa [86] and *Osmolskina* [74]. Equally, separate postfrontals are retained; this is not unusual in early stem and crown archosaurs, but does contrast with doswelliids [131,150] and proterochampsids [104,108].

Overall, compared with other taxa in a similar phylogenetic position, the ossification level of *Euparkeria* tends to be plesiomorphic, retaining separate elements wherever there is variation in archosauriforms. This is in concord with retention of relatively extensive palatal teeth (see below), and also indeed with the presence of ossified intercentra (SAM-PK-5867). Due to its plesiomorphic ossification pattern, it has previously been suggested for these reasons that the taxon could be juvenile [151], but this has seemingly been ruled out by histological analysis [152]. Furthermore, although there is certainly linkage of smaller size to increased palatal dentition in other taxa [140], this may represent intraspecific variation rather than ontogeny. Given that the taxon appears unlikely to be a juvenile, its broadly plesiomorphic morphology and completeness continue to make the taxon an excellent outgroup choice in phylogenetic analysis of early crown archosaurs.

## 4.3. Diet and evolution of the archosaur feeding apparatus

The overall shape of the skull of *Euparkeria capensis* and its dentition indicate that the animal was a terrestrial carnivore (but see below). The snout of *Euparkeria* is relatively deep dorsoventrally even at its anterior end, unlike the anteriorly tapered snout of *Prolacerta* [90] and proterosuchids [153], and strongly contrasting with the extreme elongation seen in probable aquatic or semiaquatic taxa such as proterochampsids [104] or phytosaurs (e.g. [154]). There is thus no indication of piscivory. A skull with similar anterior proportions is seen in hypercarnivorous taxa such as erythrosuchids (*Erythrosuchus* [81]; *Garjainia prima*—PIN 2394-5/1), early pseudosuchians (e.g. *Batrachotomus* [75]) and in theropod dinosaurs (e.g. *Herrerasaurus* [77]; tyrannosauroids [11]). Biomechanical and ecological studies would be needed to confirm this, but a tapering snout in some theropods (e.g. *Coelophysis* [155]) could conceivably be connected to overall importance of reach in feeding because animals such as *Coelophysis* were pursuing prey much smaller and at a different level (whether on the ground or above) than themselves; *Euparkeria* in contrast would presumably have taken prey at its own level and closer in size, as did hypercarnivorous taxa.

The dentition of *Euparkeria* is clearly indicative of carnivory, with mediolaterally compressed, recurved, serrated marginal teeth. Compared with some stem archosaurs (e.g. *Prolacerta* [90]; proterosuchids [126]), the palatal teeth are relatively small and do not form distinct rows. This may indicate reduced use in load-intensive activities such as capturing and shearing prey, and an increased focus on the marginal dentition for these activities. However, maintenance of the palatal teeth at all (contrasting with most crown archosaurs [31] and with erythrosuchids [127]) may reflect the continued importance of palatal teeth in retaining smaller prey once they enter the mouth, and have corresponded with retention of a relatively

flexible tongue in contrast with crown archosaurs [156]. The loss of palatal teeth during early archosauromorph evolution is, however, far from a clear, unequivocal trend. This is demonstrated by the optimization of the characters coding presence and type of palatal teeth from the dataset of Sookias [32,72] (see electronic supplementary material): although the phylogeny and characters need far greater scrutiny, and such character optimization is certainly not the focus of this study, this actually finds the absence of any palatal teeth to be the most parsimonious state for the node leading to the common ancestor of Archosauria and Euparkeriidae, indicating euparkeriids may have in fact *regained* palatal dentition. Further examples of the complexity of these changes are that insertion of palatal teeth in alveoli appears to have developed once in proterochampsids and a second time in crown taxa, the lack of vomerine dentition in *Youngina* and all palatal dentition in *Trilophosaurus*, and the lack of pterygoid teeth on the transverse process in *Mesosuchus* despite the presence in the more crownward *Prolacerta*; diet may well have played a role again here, with durophagy/herbivory possibly being associated with loss in rhynchosaurs and *Trilophosaurus*. There is, however, no doubt that many stem taxa possessed relatively extensive palatal dentition [90,109,131], and that palatal dentition was lost entirely in the majority of crown taxa [31]. During the Triassic archosauromorph radiation, palatal dentition was variously modified in different groups, usually—as is the wider tetrapod pattern [156]—involving loss or condensation, but several Triassic crown taxa do retain some form of palatal dentition [31,78,129]. Individual palatal teeth in *Euparkeria* are smaller than in, e.g. proterosuchids [126]; this may indicate the reduced importance of palatal dentition, but larger palatal teeth are also seen in crown taxa which retain them [78]. Further complicating the picture, the presence of tooth fields, rather than rows (as seen in proterosuchids [36]), is also probably plesiomorphic at a broad scale, with tooth fields not differentiated into rows being ancestral for tetrapods [156]. When compared with stem amniotes and with amphibians [156], the extent of these fields in *Euparkeria* is much reduced, however, and unlike *Prolacerta* [90] and proterosuchids [126,156], but like all crown taxa [31], *Euparkeria* lacks any dentition on the pterygoid flange. Overall, the deep skull of *Euparkeria*, potentially lightened by extension of the antorbital fenestra further anteriorly than in erythrosuchids [81] and possessing recurved, cutting marginal teeth and reduced palatal teeth, at the first sight appears to represent an intermediate morphology between that of some stem taxa (e.g. *Prolacerta* [90]; which probably approach more closely the ancestral archosauromorph condition [156]) and early diapsids (e.g. *Youngina* [157]), and crown archosaurs such as loricatans (e.g. *Batrachotomus* [75]). However, as clearly demonstrated by the phylogenetic placement and character optimizations for these taxa, there was certainly no clear 'trajectory' of change; rather, the development of a deepened skull was potentially an archosaur plesiomorphy carried over into loricatans, or possibly convergence, and—as demonstrated above—loss and change of palatal dentition was a phylogenetically widespread and non-uniform occurrence in early archosaurs and their close relatives.

An active lifestyle and mode of carnivory is supported by indications of elevated metabolism in *Euparkeria* compared with extant non-avian diapsids (which increased yet further in crown non-crocodylian pseudosuchians and theropods) [158,159], and is also supported by available information on the inner ear structures of *Euparkeria*. The elongation of the semicircular canals and the enlarged size of the floccular fossa indicate the presence of a refined mechanism for stabilization of the head, eye and neck movements. This mechanism is mediated by the vestibulo–ocular and vestibulochollic reflexes, which involve the semicircular canals and the floccular lobe, among others [160]. Although the exact ways in which these structures function remain obscure [161], a refined mechanism for stabilizing the movements of the head and the eyes is usually related to behaviours such as navigation in complex environments and active hunting [162–165], as also discussed by Sobral *et al.* [55].

The relatively low integration and high number of modules in the skull of *Euparkeria* (see above and figures 56 and 57) is broadly what would be expected of the taxon given its phylogenetic position, and is potentially indicative of a relatively flexible, at least partially kinetic skull, which would again fit its phylogenetic position [146]. However, inferring kinesis from fossils is difficult [146], with *in vivo* validation, of course, impossible, and there is no clear evidence of kinematically permissive linkages (one of the criteria as outlined by Holliday & Witmer [146]). Synovial basipterygopterygoid (basal) and quadratosquamosal (otic) joints (another criterion) may well have been present given the loose articulation at these points and their composition by rounded structures received by concavities, but preservation impedes the identification of the presence of hyaline cartilage or a fibrous capsule at the joints. Preservation similarly limits the identification of the correlates of protractor musculature (the remaining criterion), but a partially kinetic skull is by no means ruled out. The ancestral diapsid and amniote has long been considered to have had at least a partially kinetic skull [166,167], originally potentially developed to facilitate feeding on agile and small arthropod prey [60,167] (other adaptations included the reorganization and differentiation of the jaw musculature when compared with the ancestral anapsid

reptilian/amniote anatomy [168–170]); a partially kinetic skull is still considered likely both for these and the ancestral archosaur [146]. The modularity pattern visible in *Euparkeria* may thus correspond to an ancestral retention of a number of joints, or the remnants thereof, between skull modules which become more integrated in ornithischian and saurischian dinosaurs and fully fused crocodilians (and synapsids and turtles) [146]. An ancestral level of kinesis continued to be present in non-avian theropods and was expanded to full kinesis in birds, but especially in the latter, the skull has become more integrated within each module, with increased anisomerism and reduced number of elements.

The ancestral diapsid skull probably involved a movable epipterygoid that enabled some degree of prokinesis by articulating with the skull roof dorsally and the palate ventrally; the palate may have articulated anteriorly with the ethmoid region to enable rhynchokinesis and posteriorly with the parabasisphenoid to enable basipterygoid articulation (i.e. movement between parabasipterygoid and pterygoid) [137,167,171,172]. In the skull roof, mesokinesis was probably possible in the diapsid ancestor between frontal and parietal, and metakinesis was probably possible between parietal and braincase ([137], fig. 1). The modular pattern of *Euparkeria* with a separate palate module could suggest the presence of rhynchokinesis and basipterygoid articulation. Whether prokinesis was possible is questionable given the extensive suture of the ectopterygoid to the pterygoid, but they are resolved in separate modules. Compared with early diapsids such as *Youngina* [157,173], however, the palate is much more expanded anteriorly indicating a more robust palate. The parietal of *Euparkeria* was found to be a labile bone (see above and figure 57) in the AnNA due to its low integration with the surrounding modules, namely to the braincase and the supraorbital modules. The parietal forms an integrated part of the skull roof, together with the frontal in the alligator and tuatara, and together with the squamosal in the coelurosaurian theropod *Tyrannosaurus*. The extensive snout module of *Tyrannosaurus*, which included the frontal, might, following Werneburg *et al.* [60], have formed a secondary mesokinetic joint with the skull roof (including parietal) to enable powerful biting while allowing rostral flexibility, although recent finite-element analysis-based work has cast doubt on the presence of any functional cranial kinesis in *Tyrannosaurus* [174]. In *Euparkeria*, however, retention of an ancestral mesokinetic joint—or at least its remnant in terms of cranial integration—between parietal and the frontal, as a part of the supraorbital module, is likely. The labile integration of the parietal with the skull of *Euparkeria* might also indicate the maintenance of an ancestral diapsid metakinetic articulation to the braincase.

Despite the skull shape of *Euparkeria* showing some superficial similarity to hypercarnivorous taxa, including *Tyrannosaurus* examined by Werneburg *et al.* [60], major differences exist in regard to modularity as captured by the anatomical network analysis. *Euparkeria* has a uniform snout module, which is not separated into a dorsal and a ventral part as is seen in *Tyrannosaurus*. Also differing from *Tyrannosaurus*, a supraorbital module is separated from the snout, and the cheek module contains the jugal, squamosal and ectopterygoid. Also, a rather uniform palate module is formed in *Euparkeria*, whereas the palate bones are either integrated into the snout (vomer, palatine) or into the lower adductor chamber module (pterygoid) in *Tyrannosaurus*. Werneburg *et al.* [60] hypothesized that the unique skull modularity of *Tyrannosaurus* is—particularly the presence of an upper and lower snout module—related to the suggested behaviour of tearing flesh from its prey, although effective functional integration of the tyrannosaur cranium may preclude this [174]. The small body size of *Euparkeria* precludes a hypercarnivorous behaviour as in *Tyrannosaurus* and may explain some of these differences in modularity. Another archosaur carnivore, *Alligator mississippiensis*, and the lepidosaur *Sphenodon punctatus* differ in having a broadly more integrated skull than *Euparkeria*. In *Alligator* and *Sphenodon*, the snout forms a large, single entity, which involves at least the vomer and palatine in tuatara and also the pterygoid in the alligator. In *Alligator*, this is most likely explained by selection for forceful biting. In *Sphenodon*, it is probably due to the much shorter snout of the taxon, with the antorbital region only forming some 25% of the skull length compared with some 50% in *Euparkeria*; this short snout may, in turn, be connected to its diet, which necessitates a sturdy skull capable of crushing hard-shelled beetle and snail prey [175].

The modular pattern and morphology of the skull indicates a relatively flexible (i.e. with some elasticity and movement possible), but not fully kinetic cranial anatomy, probably inherited from the ancestral diapsid [167,171,176], and potentially facilitating predation on elusive but relatively small/soft bodied prey not requiring shearing or crushing (e.g. faster-moving insects; see [149]). The anatomy of teeth, with highly developed marginal rows and serrations, reduced palatal dentition, and an expanded palate, however, are indicative of a tendency towards a more carnivorous feeding behaviour (see above). Taken as whole, these data again indicate that *Euparkeria* represents a transitional taxon in regard to behaviour and ecomorphology from the typical form of an ancestral, insectivorous diapsid to a carnivorous crown archosaur. The cranial morphology of the taxon thus

provides at least a close analogue, if not a direct record, of the dawn of a key part of the archosaur 'body plan' which would underlie the Triassic archosaur radiation. Understanding of the exactitudes of kinetic movement and flexibility in the skull of *Euparkeria*, however, remains limited, with conclusions here based solely on interpretations of modularity; further work investigating the biomechanics of any potential kinesis in the skull of *Euparkeria* (as has recently been conducted for *Tyrannosaurus* [174]), would be potentially fruitful and would shed further light on this key transition.

## 4.4. *Euparkeria* in the context of the development of the archosaur sensory apparatus and 'body plan'

*Euparkeria* can also be viewed as exemplifying some changes seen in the development of the sensory system during archosauromorph evolution. Like crown archosaurs (e.g. *Batrachotomus* [75]; *Herrerasaurus* [77]) and other non-archosaurian archosauriforms, *Euparkeria* lacks a pineal foramen, but this contrasts with most non-archosauriform archosauromorphs (e.g. *Mesosuchus* [109]; *Prolacerta* [90]) and extant lepidosaurs (e.g. *Sphenodon punctatus* NHMUK 97.2.6.10). Loss of the pineal foramen (and concurrent loss of pineal photosensitivity) has been suggested to be connected to the development of nocturnality [177]. This would correspond to suggestions, based on the relative width of the sclerotic ring, that *Euparkeria* was nocturnal (scotopic [134]). There is thus the possibility that all archosaurs passed through a nocturnal phase in their evolution, leading to the loss of the pineal foramen.

*Euparkeria* also appears to have possessed a relatively refined hearing sense compared with other non-archosaurian archosauromorphs [85,89], as suggested by the elongate cochlea and the (albeit rudimentary) pressure-relief system. The basilar membrane is tonotopically arranged, so that an increased cochlear length indicates the extension of the hearing range. Elongation of the cochlea for increased sensitivity, which represents the plesiomorphic mechanism for increasing auditory sensitivity in amniotes, suggests that mechanical tuning was important for *Euparkeria*, in contrast with the electrical tuning mechanisms developed in crown archosaurs [178]. The overall degree of braincase ossification, together with the pressure-relief role played by the metotic foramen, also supports this hypothesis—see Sobral *et al.* [55] for further discussion.

Another change observable in archosauromorph evolution is the loss of the septomaxillae, with these elements lost in archosauriforms crownward of *Proterosuchus* [153]. While part of the broader trend of loss and fusion of cranial elements (see above), loss of the septomaxillae may also correspond to a shift in importance from olfactory to visual sensitivity, with the septomaxillae connected to the vomeronasal system [179,180], and the vomeronasal organ being lost in extant archosaurs [181]. However, near complete loss of the septomaxilla in extant mammals appears to be due to the reorganization of the vomeronasal system rather than reduction [179]. Phytosaurs show a similarly located ossification, but this appears to have developed separately and was not associated with the vomeronasal system, which appears to have been lacked by phytosaurs [180].

Although, as noted, there is not a complete correspondence between loss of the septomaxilla and reduction in the importance of the vomeronasal organ, reduction in the importance of olfaction, increased use of vision and corresponding increases in forebrain size can be seen as a recurring theme in archosaur evolution, seeing its extreme form in birds [16]. This trend is paralleled in primates, with similar increases in visual acuity, reduction in olfactory sense and increase in cerebral size [181,182]. This may, in turn, be connected with the capture of more mobile prey [182,183], and the corresponding shift to more cursorial locomotion, as also supported by evidence from the inner ear (see [55] and above). Given its phylogenetic position, *Euparkeria* serves well to demonstrate the beginnings of this shift in sensory, feeding and locomotor ecology in archosaurs, and a miniaturized skull with relatively large orbits and brain may well have been inherited by the (probably terrestrial, upright and predatory) common ancestor of Archosauria, with subsequent reversals in herbivorous taxa such as sauropods [16,184].

Overall, *Euparkeria* is of importance as it possesses many of the features of crown archosaurs but few features linking it to any group within the crown, and continues to be placed just outside the crown [32], making it a very useful outgroup. Furthermore, Sookias [32] used the phylogeny of early archosauriforms to estimate the morphology of the ancestor of Archosauria + Phytosauria using squared change parsimony, and this ancestor was suggested to be terrestrial, carnivorous and relatively small [32]; this approaches the general morphology of *Euparkeria*. This morphology, in turn, can be supposed to have underlain the cursorial, high metabolism [185], and relatively large-brained and large-eyed [181,182] body plan of the crown archosaurs which went on to radiate so spectacularly during the Mesozoic. *Euparkeria* probably already had an elevated metabolism compared with stem diapsids and had

developed some degree of endothermy, its relatively large eyes and brain and gracile morphology reflect the foundations of later developments, especially among theropod dinosaurs (including birds) and some pseudosuchians.

Data accessibility. Raw CT scan data for the cranial material of UMCZ T.692 have been made available at the Zenodo digital repository, and surface renderings of SAM-PK-5867 and SAM-PK-6047A, are also archived at Zenodo (doi:10.5281/zenodo.3887056). Other raw CT data are available from the authors on request, and will be archived following completion of further work by B.-A.S.B. Measurements, the matrix and analytical process used for phylogenetic analysis, and matrix used for the anatomical network analysis are included with the manuscript as electronic supplementary material. A photogrametric model of the skull of SAM-PK-5867, courtesy of B. Hendrick and R.B.J. Benson, and a bank of additional photographs, which were taken by the authors and colleagues (with permission) and used for the descriptive work have been uploaded to the Zenodo repository (doi:10.5281/zenodo.3887056).

Authors' contributions. R.B.S. led the study and conducted most of the descriptive and photographic work, studying all specimens, coded the network matrix, undertook the phylogenetic analysis, prepared the skull reconstruction, and wrote most of the manuscript. D.D. contributed significantly to the description and reconstruction and contributed to the manuscript in general. G.S. wrote much of the braincase description, contributed to the braincase reconstruction, and to points in the discussion. R.M.H.S. and F.P.W. conducted stratigraphic fieldwork on the section yielding *Euparkeria* and wrote much of the geological background section of the manuscript. A.B.A. contributed to the manuscript, provided photographs and discussions. B.-A.S.B. facilitated and assisted with CT data and contributed to writing the manuscript. I.W. conducted network analyses, wrote the related parts of the manuscript, and prepared related figures.

Competing interests. The authors declare no competing financial or non-financial interests.

Funding. During completion of this work, R.B.S. was supported by a European Research Council Starting Grant (TEMPO to Roger B.J. Benson), a postdoctoral research grant from the Alexander von Humboldt Foundation, a Marie Curie Career Integration Grant (PCIG14-GA-2013-630123 Archosaur rise to Richard J. Butler), the College of Life and Environmental Sciences of the University of Birmingham, UK, and an Emmy Noether Programme Award from the Deutschen Forschungsgemeinschaft (DFG; BU 2587/3-1 to Richard J. Butler). G.S. was supported by a research grant from the DFG. R.M.H.S. and F.P.W. were funded during their work by a grant to R.M.H.S. from the National Research Foundation of South Africa through the African Origins Platform (project UID117615). A.B.A. was supported by grant PROICO 2-1618 UNSL. I.W. was funded by DFG grant WE 5440/6-1. The publication of this article was funded by the Open Access Fund of the Leibniz Association.

Acknowledgements. We thank S. Kaal and Z. Erasmus (SAM) for access to specimens in their care and their hospitality to the authors when examining the material of *Euparkeria* in Cape Town. We also thank the following for access to specimens in their care: K. Mehling (AMNH); R. Hall, K. Wroe and A.L. Stewart (BSCUB); B.S. Rubidge (BP); L. Steel, P.D. Campbell and S.D. Chapman (NHMUK); M. Borsuk-Białynicka (ZPAL). We thank J. Müller (Museum für Naturkunde, Berlin—MfN) for support to R.B.S. when completing the work, and M. Kirchner and K. Mahlow (both MfN) for assistance with CT data. We thank B. Hendrick (Louisiana State University) and R.B.J. Benson (University of Oxford) for allowing use of light surface scans to create a 3D model of the holotype. We thank O. Yaryhin (Eberhard-Karls-Universität Tübingen) for assistance with anatomical network analysis. We thank A. Schmitt (University of Cambridge), P. Andreev (Qujing Normal University) for software and technical assistance. We thank H.E. Sookias for editorial assistance. We thank C. Foth (University of Fribourg) and M.D. Ezcurra (Museo Argentino de Ciencias Naturales 'Bernardino Rivadavia') for their reviews of the manuscript, which greatly improved its quality. We also greatly thank R.J. Butler (University of Birmingham) for providing extensive comments on earlier versions of the manuscript, discussion and for major efforts in facilitating the study's completion.

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
