## [Reviewer comments · Royal Society Open Science]

Review History

RSOS-200116.R0 (Original submission)

Review form: Reviewer 1 (Christian Foth)

Is the manuscript scientifically sound in its present form?

Yes

Are the interpretations and conclusions justified by the results?

Yes

Is the language acceptable?

Yes

Do you have any ethical concerns with this paper?

No

Have you any concerns about statistical analyses in this paper?

No

Recommendation?

Accept with minor revision (please list in comments)

Comments to the Author(s)

The manuscript by Sookias and colleagues describes the skull anatomy of the basal archosauriform Euparkeria. The description is very rich in detail, but easy to follow. In sum, I have only a few comments and the paper can be accepted after minor revision.

In general, the authors should decide if they want to use species or genera names. Furthermore, for the headings of the single bones, please decide if plural or singular.

P3L8: As the description also includes the mandible, change: "Here, the skull of Euparkeria ...".

P3L24: put "the ruling lizards" after Archosauria in line 23.

P4L21: If I got it right, the pedomorphosis in the bird evolution affects primarily the orbital and temporal region. The evolution of the rostrum seems to have a higher plasticity and was also affected by peramorphic events (see Bhullar et al. 2012).

P9L20: Please specify, which kind of cluster algorithm was used to calculate the dendrograms.

Furthermore, please list and describe all network parameters that were computed. Maybe the authors consider to estimate the network parameter Parcellation (P), too (Esteve-Altava et al. 2019).

P12L23f: move the examples *Herrerasaurus* and *Eoraptor* directly after *Avemetatarsalia* (I prefer *Ornithodira*).

Horizon and type locality section: could the authors provide the full names of the person that discovered the locality.

P24L21: To my knowledge the ear of reptiles consist only of a middle (external opening) and inner ear. The external ear is related to the fleshy bell mouth of mammals (at least in the German terminology).

P28L14: change: "other basal theropods ..."

I am not sure if the squamosal and the quadratojugal define the cheek. What is with the jugal?

P62L12. With *Buriolestes* we have at least one sauropodomorph (crown archosaur) with palatal teeth.

P82:L17f: *Silesaurus* and *Sacisaurus* are rather dinosauromorphs and thus crown archosaurs.

Please see that recent study of Cost et al. (2020) who say that the skull of *Tyrannosaurus* was akinetic. To be honest, for this matter I find their FEA approach more convincing than a network analysis. This is also relevant for the discussion.

P98L16: *Euparkeria* in italics.

P100fL23ff: the statement on the integration of bird skulls should be weaken, because it bases only on one species, *Gallus gallus*. How can the authors be sure that a parrot has the same kind of integration. Because of the small sample in the network analysis, all comparisons should refer only to the species sampled and not generalized to the clades they belong to.

P107: I am not sure if it is possible to compare the prey behaviour of *Euparkeria* and *Tyrannosaurus*. Looking at the coding, I realized that fused bones in *Tyrannosaurus* (like the nasals) were still treated as separate bones, implying that the modularity of *Tyrannosaurus* presented in Werneburg et al. (2019) might be not 100% correct.

P111: According to Legendre et al. (2013, 2016) the metabolic rate of *Euparkeria* was higher than that recent crocodylians and other recent non-avian reptiles.

In the modularity sections the authors describe correctly that the modularity in *Euparkeria* is slightly asymmetric, i.e., that the paired bones are not assigned to the same module on the left and right side. However, in the figure 55 everything looks symmetrically. Either the authors show the module distribution as estimated by the cluster analysis or they stated clearly that this is an interpretation of the dendrogram and that labile or conflicting bones are assigned a posteriori.

Christian Foth

References

Cost et al. 2020. Palatal Biomechanics and Its Significance for Cranial Kinesis in *Tyrannosaurus rex*. *The anatomical record* 303: 999-1017.

Esteve-Altava et al. 2019. Anatomical network analysis of the musculoskeletal system reveals integration loss and parcellation boost during the fins-to-limbs transition. *Evolution* 72: 601–618.

Legendre et al. 2013. Evidence for high bone growth rate in *Euparkeria* obtained using a new paleohistological inference model for the humerus. *Journal of Vertebrate Paleontology* 33:1343–1350.

Legendre et al. 2016. Palaeohistological Evidence for Ancestral High Metabolic Rate in Archosaurs. *Systematic Biology* 65: 989–996.

Review form: Reviewer 2 (Martín Ezcurra)

Is the manuscript scientifically sound in its present form?

Yes

Are the interpretations and conclusions justified by the results?

Yes

Is the language acceptable?

Yes

Do you have any ethical concerns with this paper?

No

Have you any concerns about statistical analyses in this paper?

No

Recommendation?

Accept with minor revision (please list in comments)

Comments to the Author(s)

This manuscript describes in deep detail the craniodental anatomy of the stem-archosaur *Euparkeria capensis*. Although species has been described and figured in some detail in the past, including the monographic work by Ewers (1965), the new information provided by the authors substantially enriches our knowledge of the anatomy of this key archosauriform species. This description and the detail of anatomical work provided in it will be very welcomed by all early archosaur workers. The text is well written and the manuscript is accompanied by multiple, very informative figures, including photographs of the actual specimen and captions from CT-data. The anatomical comparisons with other archosauriform species will be useful for future phylogenetic analyses.

The phylogenetic analysis conducted by the authors recovered very similar results to that of a paper published by the corresponding author some years ago. However, I think that the authors should provide more information on the results of the analysis, including changes in the branch supports with respect to the previous version of this analysis.

The Anatomical Network and Modularity analyses enrich the discussion of the manuscript and are very welcomed.

I haven't spotted major problems in the manuscript and I suggest its acceptance after minor modifications. However, I consider that the authors should address the following list of issues before the final acceptance of the manuscript:

- Horizon and locality section. Can you comment on the story behind the AMNH specimens of *Euparkeria*? Is the type locality currently exposed? Is the recently found long bone referable to *Euparkeria* or to other taxon?

- Diagnosis: in the Diagnosis it is described that *Euparkeria* lacks a downturned premaxilla, contrasting with *Osmolskina* –and I agree based on the holotype of the former taxon. However, the premaxilla is re-articulated as downturned in figure 9. I suggest you to comment in the

caption of this figure or somewhere in the text that this apparently downturned premaxilla is artificial.

- Section 3.4 Phylogeny: Please, provide more information here of how many times the optimum was hit during the replications and consistency and decay indices of the trees. Also provide measures of branch support (Bremer supports and resampling frequencies). Change the first sentence as follows: 'The phylogenetic analysis recovered 248 most parsimonious trees of 1344 steps'. Please, also explain here if the a priori exclusion of 'Turfanosuchus' shageduensis (a nomen dubium sensu Sookias et al., 2014) produces any change in the results. Also, 'Turfanosuchus' shageduensis should be changed to 'Turfanosuchus shageduensis', because the genus is valid, but the species is invalid (Sookias et al., 2014).

- Page 105: a general, simple differentiation between large theropods with high snouts and smaller theropods with tapering snouts is very problematic. Relatively large theropods, such as Zupaysaurus and Dilophosaurus, have tapering snout and they are bigger than Herrerasaurus. Similarly, Herrerasaurus has a moderately elongated neck that resembles the condition in Dilophosaurus, and both forms present high and tapering snouts, respectively. As a result, I suggest to delete those part of the discussion about neck elongation (see edits in the PDF).

- Page 106: there is a lot of discussion about the potential 'basal' or 'intermediate' condition of the morphology of the palatal teeth of Euparkeria. There is a lot of discussion about this and as it is currently worded it seems to be focused mainly on ideas of a directional trend of evolution. I strongly suggest the authors to discuss the evolution of the palatal dentition of Euparkeria and other archosauromorphs based on character optimizations of their phylogeny or of other authors (e.g. Pritchard and Sues 2019; Scheyer et al., 2020).

- Page 108: "Whether prokinesis was possible is questionable given the fusion of the ectopterygoid to the pterygoid". The description describes a suture with the pterygoid, but not fusion. The presence of fusion between pterygoid and ectopterygoid would be a very unusual condition among archosauromorphs. Please, check this statement that seems to contradict the description.

- Page 111: "Proterosuchus fergusi, where the presence of a pineal foramen varies between individuals [76], was found to be cathemeral (mesopic [130])." These authors recovered a mesopic behaviour for Proterosuchus based on the sclerotic ring reconstructed by Cruickshank (1972), but as far as I understand this reconstruction should have been based on other archosauriform specimens because there is not sclerotic ring preserved in Proterosuchus specimens. In addition, the pineal foramen, when present, of Proterosuchus is reduced to a small pit on the skull roof, thus I think that they couldn't have allowed the passage of a photosensitivity organ. I strongly suggest to delete this sentence.

- I strongly suggest moving figure 17 before all other anatomical figures because it represents a 'summary' of your observations and would be useful for the reader when going through the detail description of the bones. A similar organization has been chosen in other stem-archosaur osteologies, e.g. Erythrosuchus africanus (Gower, 2003) and Garjainia prima (Ezcurra et al., 2019).

- Figure 1 has several problems in my opinion. First, Dongusuchus and Dorosuchus should be interchanged between each other. Nesbitt et al. (2017: Nature) have recovered Yarasuchus and Dongusuchus as part of the clade Aphanosauria (same paper) at the base of Avemetatarsalia. This result should be shown here because it has been one of the key modifications in the archosaur tree of the last years. One of the phylogenies used by Nesbitt et al. (2017) is a modification of that of Ezcurra (2016), which is used by the authors for reference in this figure.

- Figure 2: add an arrowhead indicating that the Katberg Fm and the Lystrosaurus AZ extend to the Induan.

- Figure 15: I think that this figure is not very informative in its current state. Bones in (A) are basically indistinguishable and labels overlap most of the morphology in (B) and (C). I strongly suggest showing the block in different portions (maybe in six captions) and do not include the labels over the photographs.

- Figure 16: similar problem as figure 15. Try to show bigger and in more detail the cranial bones and the vertebrae.

- Figure 17: can you label the subnarial foramen? The position of the epipterygoid seems to be quite unusual among archosauromorphs, its base usually sits on the base of the quadrate wing of the pterygoid. Can you check that the position of this bone is correct? The posterior region of the

hemimandible seems to have an additional fenestra-size opening in the drawing, but as far as I remember from the holotype, this area has an elongated depression with the posterior surangular foramen on it. If this region of the hemimandible does not have an opening, please, modify the drawing to show a depression rather than a fenestra-like feature.

- I have made several, minor edits on the PDF of the manuscript.

- Finally, I tried to open the supplementary information uploaded in Zenodo but I couldn't. It is possible that the files are not still available for the public, but if it is the case, please check that these links are working properly.

Yours sincerely,
Martin Ezcurra

Decision letter (RSOS-200116.R0)

Dear Dr Sookias,

The editors assigned to your paper ("The craniomandibular anatomy of the early archosauriform *Euparkeria capensis* and the dawn of the archosaur skull") have now received comments from reviewers. We would like you to revise your paper in accordance with the referee and Associate Editor suggestions which can be found below (not including confidential reports to the Editor). As a marked up file, which is too large to send through the portal, has been appended to the report, I will send this via separate cover. Please note this decision does not guarantee eventual acceptance.

Please submit a copy of your revised paper before 16-May-2020. Please note that the revision deadline will expire at 00.00am on this date. If we do not hear from you within this time then it will be assumed that the paper has been withdrawn. In exceptional circumstances, extensions may be possible if agreed with the Editorial Office in advance. We do not allow multiple rounds of revision so we urge you to make every effort to fully address all of the comments at this stage. If deemed necessary by the Editors, your manuscript will be sent back to one or more of the original reviewers for assessment. If the original reviewers are not available, we may invite new reviewers.

- Data accessibility

If you wish to submit your supporting data or code to Dryad (<http://datadryad.org/>), or modify your current submission to dryad, please use the following link:
<http://datadryad.org/submit?journalID=RSOS&manu=RSOS-200116>

- Competing interests

- Authors' contributions

- Acknowledgements

- Funding statement

Kind regards,
 Andrew Dunn
 Royal Society Open Science Editorial Office
 Royal Society Open Science

on behalf of Dr Julia Brenda Desojo (Associate Editor) and Kevin Padian (Subject Editor)
openscience@royalsociety.org

Editor comments:

Thank you for submitting. The reviewers are clearly happy with the paper, which will make a nice contribution to the knowledge of an iconic taxon. However, the revisions they propose are fairly extensive and by issuing a "major revision" decision you have a few weeks to respond. Also, one of the reviews points out that the supplementary information uploaded in Zenodo is not available for the public. Could you check it and could we resolve it? Best wishes for your revision. I think it would not be necessary to send it out again for review; our AE can check it. Thanks

Comments to Author:

Reviewers' Comments to Author:

Reviewer: 1

Comments to the Author(s)

The manuscript by Sookias and colleagues describes the skull anatomy of the basal archosauriform *Euparkeria*. The description is very rich in detail, but easy to follow. In sum, I have only a few comments and the paper can be accepted after minor revision.

In general, the authors should decide if they want to use species or genera names. Furthermore, for the headings of the single bones, please decide if plural or singular.

P3L8: As the description also includes the mandible, change: "Here, the skull of *Euparkeria* ...".

P3L24: put "the ruling lizards" after Archosauria in line 23.

P4L21: If I got it right, the pedomorphosis in the bird evolution affects primarily the orbital and temporal region. The evolution of the rostrum seems to have a higher plasticity and was also affected by peramorphic events (see Bhullar et al. 2012).

P9L20: Please specify, which kind of cluster algorithm was used to calculate the dendograms. Furthermore, please list and describe all network parameters that were computed. Maybe the authors consider to estimate the network parameter Parcellation (P), too (Esteve-Altava et al. 2019).

P12L23f: move the examples *Herrerasaurus* and *Eoraptor* directly after *Avemetatarsalia* (I prefer *Ornithodira*).

Horizon and type locality section: could the authors provide the full names of the person that discovered the locality.

P24L21: To my knowledge the ear of reptiles consist only of a middle (external opening) and inner ear. The external ear is related to the fleshy bell mouth of mammals (at least in the German terminology).

P28L14: change: "other basal theropods ..."

I am not sure if the squamosal and the quadratojugal define the cheek. What is with the jugal?

P62L12. With *Buriolestes* we have at least one sauropodomorph (crown archosaur) with palatal teeth.

P82:L17f: *Silesaurus* and *Sacisaurus* are rather dinosauromorphs and thus crown archosaurs.

Please see that recent study of Cost et al. (2020) who say that the skull of *Tyrannosaurus* was akinetic. To be honest, for this matter I find their FEA approach more convincing than a network analysis. This is also relevant for the discussion.

P98L16: *Euparkeria* in italics.

P100fL23ff: the statement on the integration of bird skulls should be weakened, because it bases only on one species, *Gallus gallus*. How can the authors be sure that a parrot has the same kind of integration. Because of the small sample in the network analysis, all comparisons should refer only to the species sampled and not generalized to the clades they belong to.

P107: I am not sure if it is possible to compare the prey behaviour of Euparkeria and Tyrannosaurus. Looking at the coding, I realized that fused bones in Tyrannosaurus (like the nasals) were still treated as separate bones, implying that the modularity of Tyrannosaurus presented in Werneburg et al. (2019) might be not 100% correct.

P111: According to Legendre et al. (2013, 2016) the metabolic rate of Euparkeria was higher than that recent crocodylians and other recent non-avian reptiles.

In the modularity sections the authors describe correctly that the modularity in Euparkeria is slightly asymmetric, i.e., that the paired bones are not assigned to the same module on the left and right side. However, in the figure 55 everything looks symmetrically. Either the authors show the module distribution as estimated by the cluster analysis or they stated clearly that this is an interpretation of the dendrogram and that labile or conflicting bones are assigned a posteriori.

Christian Foth

References

Cost et al. 2020. Palatal Biomechanics and Its Significance for Cranial Kinesis in Tyrannosaurus rex. *The anatomical record* 303: 999-1017.

Esteve-Altava et al. 2019. Anatomical network analysis of the musculoskeletal system reveals integration loss and parcellation boost during the fins-to-limbs transition. *Evolution* 72: 601–618.

Legendre et al. 2013. Evidence for high bone growth rate in Euparkeria obtained using a new paleohistological inference model for the humerus. *Journal of Vertebrate Paleontology* 33:1343–1350.

Legendre et al. 2016. Palaeohistological Evidence for Ancestral High Metabolic Rate in Archosaurs. *Systematic Biology* 65: 989–996.

Reviewer: 2

Comments to the Author(s)

This manuscript describes in deep detail the craniodental anatomy of the stem-archosaur Euparkeria capensis. Although species has been described and figured in some detail in the past, including the monographic work by Ewers (1965), the new information provided by the authors substantially enriches our knowledge of the anatomy of this key archosauriform species. This description and the detail of anatomical work provided in it will be very welcomed by all early archosaur workers. The text is well written and the manuscript is accompanied by multiple, very informative figures, including photographs of the actual specimen and captions from CT-data. The anatomical comparisons with other archosauriform species will be useful for future phylogenetic analyses.

The phylogenetic analysis conducted by the authors recovered very similar results to that of a paper published by the corresponding author some years ago. However, I think that the authors should provide more information on the results of the analysis, including changes in the branch supports with respect to the previous version of this analysis.

The Anatomical Network and Modularity analyses enrich the discussion of the manuscript and are very welcomed.

I haven't spotted major problems in the manuscript and I suggest its acceptance after minor modifications. However, I consider that the authors should address the following list of issues before the final acceptance of the manuscript:

- Horizon and locality section. Can you comment on the story behind the AMNH specimens of Euparkeria? Is the type locality currently exposed? Is the recently found long bone referable to Euparkeria or to other taxon?

- Diagnosis: in the Diagnosis it is described that Euparkeria lacks a downturned premaxilla, contrasting with Osmolskina –and I agree based on the holotype of the former taxon. However, the premaxilla is re-articulated as downturned in figure 9. I suggest you to comment in the caption of this figure or somewhere in the text that this apparently downturned premaxilla is artificial.

- Section 3.4 Phylogeny: Please, provide more information here of how many times the optimum was hit during the replications and consistency and decay indices of the trees. Also provide

measures of branch support (Bremer supports and resampling frequencies). Change the first sentence as follows: 'The phylogenetic analysis recovered 248 most parsimonious trees of 1344 steps'. Please, also explain here if the a priori exclusion of 'Turfanosuchus' shageduensis (a nomen dubium sensu Sookias et al., 2014) produces any change in the results. Also, 'Turfanosuchus' shageduensis should be changed to 'Turfanosuchus shageduensis', because the genus is valid, but the species is invalid (Sookias et al., 2014).

- Page 105: a general, simple differentiation between large theropods with high snouts and smaller theropods with tapering snouts is very problematic. Relatively large theropods, such as Zupaysaurus and Dilophosaurus, have tapering snout and they are bigger than Herrerasaurus. Similarly, Herrerasaurus has a moderately elongated neck that resembles the condition in Dilophosaurus, and both forms present high and tapering snouts, respectively. As a result, I suggest to delete those part of the discussion about neck elongation (see edits in the PDF).

- Page 106: there is a lot of discussion about the potential 'basal' or 'intermediate' condition of the morphology of the palatal teeth of Euparkeria. There is a lot of discussion about this and as it is currently worded it seems to be focused mainly on ideas of a directional trend of evolution. I strongly suggest the authors to discuss the evolution of the palatal dentition of Euparkeria and other archosauromorphs based on character optimizations of their phylogeny or of other authors (e.g. Pritchard and Sues 2019; Scheyer et al., 2020).

- Page 108: "Whether prokinesis was possible is questionable given the fusion of the ectopterygoid to the pterygoid". The description describes a suture with the pterygoid, but not fusion. The presence of fusion between pterygoid and ectopterygoid would be a very unusual condition among archosauromorphs. Please, check this statement that seems to contradict the description.

- Page 111: "Proterosuchus fergusi, where the presence of a pineal foramen varies between individuals [76], was found to be cathemeral (mesopic [130])." These authors recovered a mesopic behaviour for Proterosuchus based on the sclerotic ring reconstructed by Cruickshank (1972), but as far as I understand this reconstruction should have been based on other archosauriform specimens because there is not sclerotic ring preserved in Proterosuchus specimens. In addition, the pineal foramen, when present, of Proterosuchus is reduced to a small pit on the skull roof, thus I think that they couldn't have allowed the passage of a photosensitivity organ. I strongly suggest to delete this sentence.

- I strongly suggest moving figure 17 before all other anatomical figures because it represents a 'summary' of your observations and would be useful for the reader when going through the detail description of the bones. A similar organization has been chosen in other stem-archosaur osteologies, e.g. Erythrosuchus africanus (Gower, 2003) and Garjainia prima (Ezcurra et al., 2019).

- Figure 1 has several problems in my opinion. First, Dongusuchus and Dorosuchus should be interchanged between each other. Nesbitt et al. (2017: Nature) have recovered Yarasuchus and Dongusuchus as part of the clade Aphanosauria (same paper) at the base of Avemetatarsalia. This result should be shown here because it has been one of the key modifications in the archosaur tree of the last years. One of the phylogenies used by Nesbitt et al. (2017) is a modification of that of Ezcurra (2016), which is used by the authors for reference in this figure.

- Figure 2: add an arrowhead indicating that the Katberg Fm and the Lystrosaurus AZ extend to the Induan.

- Figure 15: I think that this figure is not very informative in its current state. Bones in (A) are basically indistinguishable and labels overlap most of the morphology in (B) and (C). I strongly suggest showing the block in different portions (maybe in six captions) and do not include the labels over the photographs.

- Figure 16: similar problem as figure 15. Try to show bigger and in more detail the cranial bones and the vertebrae.

- Figure 17: can you label the subnarial foramen? The position of the epipterygoid seems to be quite unusual among archosauromorphs, its base usually sits on the base of the quadrate wing of the pterygoid. Can you check that the position of this bone is correct? The posterior region of the hemimandible seems to have an additional fenestra-size opening in the drawing, but as far as I remember from the holotype, this area has an elongated depression with the posterior surangular foramen on it. If this region of the hemimandible does not have an opening, please, modify the drawing to show a depression rather than a fenestra-like feature.

- I have made several, minor edits on the PDF of the manuscript.
- Finally, I tried to open the supplementary information uploaded in Zenodo but I couldn't. It is possible that the files are not still available for the public, but if it is the case, please check that these links are working properly.

Yours sincerely,
Martin Ezcurra

Author's Response to Decision Letter for (RSOS-200116.R0)

See Appendices A-C.

Decision letter (RSOS-200116.R1)

Dear Dr Sookias,

It is a pleasure to accept your manuscript entitled "The craniomandibular anatomy of the early archosauriform *Euparkeria capensis* and the dawn of the archosaur skull" in its current form for publication in Royal Society Open Science.

Please ensure that you send to the editorial office the correct numerical order for the figure files included with the submission? You do not need to resend the files, simply send a list identifying the numerical sequence the current files should appear in the typeset manuscript.

on behalf of Dr Julia Brenda Desojo (Associate Editor) and Kevin Padian (Subject Editor)
openscience@royalsociety.org

Appendix A

Dear editor,

On behalf of my coauthors, I would like to thank you for your time. We have tried to address fully all of the comments by the reviewers, and I present our responses to the comments below. I also attach a copy of the PDF with comments by one of the reviewers on, with the comments answered, and a version of the main document with tracked changes.

Very best wishes

Roland Sookias

Reviewers' Comments to Author:

Reviewer: 1

Comments to the Author(s)

The manuscript by Sookias and colleagues describes the skull anatomy of the basal archosauriform Euparkeria. The description is very rich in detail, but easy to follow. In sum, I have only a few comments and the paper can be accepted after minor revision.

In general, the authors should decide if they want to use species or genera names.

As stated on p. 7, l.13–14 “genus names are used alone after first mention in the case of monospecific genera”. We believe this provides this provides a good compromise between brevity and accuracy. We have added a section (2.2) in the methods which explicitly mentions this again.

Futhermore, for the headdings of the single bones, please decide if plural or singular.

We have changed all the headings to singular.

P3L8: As the description also includes the mandible, change: “Here, the skull of Euparkeria ...”.

We changed this to “cranium and mandible” because “skull” also can be interpreted as excluding the mandible.

P3L24: put “the ruling lizards” after Archosauria in line 23.

We have made this change. We also changed “lizards” to “reptiles” because although the original Ancient Greek usage seems to have been mostly for lizards, this is not completely clear, and is somewhat confusing in a phylogenetic context.

P4L21: If I got it right, the paedomorphosis in the bird evolution affects primarily the orbital and temporal region. The evolution of the rostrum seems to have a higher plasticity and was also affected by peramorphic events (see Bhullar et al. 2012).

Rereading Bhullar et al. 2012, it seems paedomorphosis is responsible for the relatively larger postrostral region and reduced face – a separate peramorphic event then extended the beak. We have reworded this to make it clearer:

“corresponding to the relatively larger postrostral region (and thus reduced face) connected with miniaturization and paedomorphosis in birds and their immediate relatives”

P9L20: Please specify, which kind of cluster algorithm was used to calculate the dendrograms. Furthermore, please list and describe all network parameters that were computed. Maybe the authors consider to estimate the network parameter Parcellation (P), too (Esteve-Altava et al. 2019).

We have now specified this (agglomerative hierarchical cluster), and listed all the parameters at this point in the text, although the parameters are listed in the table and elsewhere in the text already. Although Parcellation was used by Esteve-Altava et al. 2019 it was not considered to provide any usable biological information beyond the already calculated parameters. After discussing with the first author of that paper personally, we also found there is no urgent need for that addition to get a comprehensive picture of the evolution of the network.

P12L23f: move the examples *Herrerasaurus* and *Eoraptor* directly after *Avemetatarsalia* (I prefer *Ornithodira*).

We have made this change.

Horizon and type locality section: could the authors provide the full names of the person that discovered the locality.

We have now done this as best we can (it is a somewhat complicated story!) and attempted to clarify this section. There is ongoing work on this, however, which will bring further elucidation.

P24L21: To my knowledge the ear of reptiles consist only of a middle (external opening) and inner ear. The external ear is related to the fleshy bell mouth of mammals (at least in the German terminology).

We have changed this to “middle ear”.

P28L14: change: “other basal theropods ...”

We have made this change.

I am not sure if the squamosal and the quadratojugal define the cheek. What is with the jugal?

We understand the point made. The reason for this classification was that the jugal is already included in the circumorbital series section. Our arrangement follows Evans (1980 – *Gephyrosaurus*) – this taxon does however show an incomplete temporal bar, meaning that the jugal is more separated from the rest of the cheek region than in *Euparkeria*. We have thus renamed the “Circumorbital series” to “Circumorbital series and anterior cheek” and the “Cheek” to “Posterior cheek” for clarity.

P62L12. With *Buriolestes* we have at least one sauropodomorph (crown archosaur) with palatal teeth.

Indeed *Buriolestes* has palatal (pterygoid) teeth, as do other crown archosaurs (e.g. *Turfanosuchus*, *Eoraptor*), but not vomerine teeth, which are what is being discussed in the sentence at hand. We thus have left the text unchanged.

P82:L17f: *Silesaurus* and *Sacisaurus* are rather dinosauromorphs and thus crown archosaurs.

We thank the reviewer for flagging this up. The meaning here was intended to be early archosauromorphs including crown taxa, but this was not clear. We have changed “basal” to “Triassic” for clarity and accuracy.

Please see that recent study of Cost et al. (2020) who say that the skull of *Tyrannosaurus* was akinetic. To be honest, for this matter I find their FEA approach more convincing than a network analysis. This is also relevant for the discussion.

We thank the reviewer for bringing this recent work to our attention. We have now made reference to it, and adjusted our discussion accordingly:

On previous p. 106, l.22, we added (citing Cost et al.):

“although recent finite element analysis-based work has cast doubt on the presence of any functional cranial kinesis in *Tyrannosaurus* [166].

On previous p. 107, l. 14, we added:

“although effective functional integration of the tyrannosaur cranium may preclude this [166].”

On previous p. 108, approx. l. 15 (Discussion section), we added the sentences:

“Understanding of the exactitudes of kinetic movement and flexibility in the skull of *Euparkeria* however remains limited, with conclusions here based solely on interpretations of modularity; further work investigating the biomechanics of any potential kinesis in the skull of *Euparkeria* (as has recently been conducted for *Tyrannosaurus* [166]), would be potentially fruitful and would shed further light on this key transition.”

P98L16: *Euparkeria* in italics.

We have made this change.

P100fL23ff: the statement on the integration of bird skulls should be weakened, because it bases only on one species, *Gallus gallus*. How can the authors be sure that a parrot has the same kind of integration. Because of the small sample in the network analysis, all comparisons should refer only to the species sampled and not generalized to the clades they belong to.

We have toned down some of the statements to be clear they refer to indications of hypotheses which could be later more rigorously tested, and made them a bit more

specific. We have also more heavily referenced this part of the text. Of course, having a much larger sample would be ideal, and is a future avenue of research. Failing to hypothesize any trends or reach even tentative broader conclusions based on the results rather invalidates the point of the network analysis, however. Regarding the specific example mentioned in birds, although we don't provide quantitative support for it here, based on the first author's personal observation of the skulls, the pattern is broadly similar – the text has been modified to make it more specific to *Gallus* however. Regarding the general statement about crocodylians at the end of the paragraph, this is undoubtedly the case – the number and arrangement of elements is very conserved in most taxa; a reference and a qualifier highlighting that this is only quantified in *Alligator* has been added however.

P107: I am not sure if it is possible to compare the prey behaviour of Euparkeria and Tyrannosaurus. Looking at the coding, I realized that fused bones in Tyrannosaurus (like the nasals) were still treated as separate bones, implying that the modularity of Tyrannosaurus presented in Werneburg et al. (2019) might be not 100% correct.

Although the nasals are fused in tyrannosaurs, a clear midline suture is still visible (Hurum and Sabath 2003), indicating they are not fully fused as a single element. It is difficult to decide in such marginal cases, but given that they thus probably fuse during later ontogeny and are not fully integrated, this is a reasonable coding. Even if they fuse later in ontogeny, some flexibility is likely to remain.

P111: According to Legendre et al. (2013, 2016) the metabolic rate of Euparkeria was higher than that recent crocodylians and other recent non-avian reptiles.

We thank the reviewer for drawing our attention to these works, and have modified our discussion accordingly.

In the modularity sections the authors describe correctly that the modularity in Euparkeria is slightly asymmetric, i.e., that the paired bones are not assigned to the same module on the left and right side. However, in the figure 55 everything looks symmetrically. Either the authors show the module distribution as estimated by the cluster analysis or they stated clearly that this is an interpretation of the dendrogram and that labile or conflicting bones are assigned a posteriori.

We make clear in the figure caption and the text that the lighter colours (pale red, pale yellow) are used on the labile elements – these are assigned to different sides in different analyses. Illustrating them assigned to one side would misrepresent the results. We have made this yet more explicit in the figure caption.

Christian Foth

References

- Cost et al. 2020. Palatal Biomechanics and Its Significance for Cranial Kinesis in Tyrannosaurus rex. The anatomical record 303: 999-1017.
- Esteve-Altava et al. 2019. Anatomical network analysis of the musculoskeletal system reveals integration loss and parcellation boost during the fins-to-limbs transition. Evolution 72: 601–618.
- Legendre et al. 2013. Evidence for high bone growth rate in Euparkeria obtained using a

new paleohistological inference model for the humerus. *Journal of Vertebrate Paleontology* 33:1343–1350.

Legendre et al. 2016. Palaeohistological Evidence for Ancestral High Metabolic Rate in Archosaurs. *Systematic Biology* 65: 989–996.

Reviewer: 2

Comments to the Author(s)

This manuscript describes in deep detail the craniodental anatomy of the stem-archosaur *Euparkeria capensis*. Although species has been described and figured in some detail in the past, including the monographic work by Ewers (1965), the new information provided by the authors substantially enriches our knowledge of the anatomy of this key archosauriform species. This description and the detail of anatomical work provided in it will be very welcomed by all early archosaur workers. The text is well written and the manuscript is accompanied by multiple, very informative figures, including photographs of the actual specimen and captions from CT-data. The anatomical comparisons with other archosauriform species will be useful for future phylogenetic analyses.

The phylogenetic analysis conducted by the authors recovered very similar results to that of a paper published by the corresponding author some years ago. However, I think that the authors should provide more information on the results of the analysis, including changes in the branch supports with respect to the previous version of this analysis.

The Anatomical Network and Modularity analyses enrich the discussion of the manuscript and are very welcomed.

I haven't spotted major problems in the manuscript and I suggest its acceptance after minor modifications. However, I consider that the authors should address the following list of issues before the final acceptance of the manuscript:

- Horizon and locality section. Can you comment on the story behind the AMNH specimens of *Euparkeria*? Is the type locality currently exposed? Is the recently found long bone referable to *Euparkeria* or to other taxon?

We have added what details we could of the history of the AMNH specimens. The exact details are currently uncertain, but should be clarified in a forthcoming work by two of the authors. What is thought to be the type locality was exposed when observed in December 2019 – the date has been made clear now. The bone has not been referred beyond Archosauromorphes.

- Diagnosis: in the Diagnosis it is described that *Euparkeria* lacks a downturned premaxilla, contrasting with *Osmolskina* –and I agree based on the holotype of the former taxon. However, the premaxilla is re-articulated as downturned in figure 9. I suggest you to comment in the caption of this figure or somewhere in the text that this apparently downturned premaxilla is artificial.

We have now commented on this in the Premaxilla section.

- Section 3.4 Phylogeny: Please, provide more information here of how many times the optimum was hit during the replications and consistency and decay indices of the trees. Also provide measures of branch support (Bremer supports and resampling frequencies). Change the first sentence as follows: 'The phylogenetic analysis recovered 248 most parsimonious trees of 1344 steps'. Please, also explain here if the a priori exclusion of '*Turfanosuchus*' *shageduensis* (a nomen dubium sensu Sookias et al., 2014) produces any change in the

results. Also, 'Turfanosuchus' shageduensis should be changed to 'Turfanosuchus shageduensis', because the genus is valid, but the species is invalid (Sookias et al., 2014).

We have made these changes and carried out and present the additional analyses required. Excluding 'Turfanosuchus shageduensis' had no effect on topology and minimal effect on support.

- Page 105: a general, simple differentiation between large theropods with high snouts and smaller theropods with tapering snouts is very problematic. Relatively large theropods, such as Zupaysaurus and Dilophosaurus, have tapering snout and they are bigger than Herrerasaurus. Similarly, Herrerasaurus has a moderately elongated neck that resembles the condition in Dilophosaurus, and both forms present high and tapering snouts, respectively. As a result, I suggest to delete those part of the discussion about neck elongation (see edits in the PDF).

They have been deleted, as suggested.

- Page 106: there is a lot of discussion about the potential 'basal' or 'intermediate' condition of the morphology of the palatal teeth of Euparkeria. There is a lot of discussion about this and as it is currently worded it seems to be focused mainly on ideas of a directional trend of evolution. I strongly suggest the authors to discuss the evolution of the palatal dentition of Euparkeria and other archosauromorphs based on character optimizations of their phylogeny or of other authors (e.g. Pritchard and Sues 2019; Scheyer et al., 2020).

While this trend is not linear and continuous, there is no doubt that the ancestral condition is fundamentally with *more* palatal teeth, and that loss is very common in many later archosauromorphs. We also did not explicitly mention a directional trend and do discuss this morphology as being plesiomorphic – explicitly placing it in a phylogenetic framework. We now however added a sentence referring to the character optimisation on our and the suggested phylogenies, but, as mentioned elsewhere, the characters and character coding used are not optimal for the palatal dentition and do not capture all of the variation. Recoding all of this and examining patterns of change on the phylogeny would be outside of the scope of the work at hand, but would be of great interest. We have also made the discussion regarding palatal dentition more nuanced and cautioned overall, and attempt to assuage any notion of a clear direction trend.

- Page 108: "Whether prokinesis was possible is questionable given the fusion of the ectopterygoid to the pterygoid". The description describes a suture with the pterygoid, but not fusion. The presence of fusion between pterygoid and ectopterygoid would be a very unusual condition among archosauromorphs. Please, check this statement that seems to contradict the description.

We thank the reviewer for raising this point and have rewritten this part of the text accordingly.

- Page 111: "Proterosuchus fergusi, where the presence of a pineal foramen varies between individuals [76], was found to be cathemeral (mesopic [130])." These authors recovered a mesopic behaviour for Proterosuchus based on the sclerotic ring reconstructed by Cruickshank (1972), but as far as I understand this reconstruction should have been based

on other archosauriform specimens because there is not sclerotic ring preserved in Proterosuchus specimens. In addition, the pineal foramen, when present, of Proterosuchus is reduced to a small pit on the skull roof, thus I think that they couldn't have allowed the passage of a photosensitivity organ. I strongly suggest to delete this sentence.

We have deleted this sentence.

- I strongly suggest moving figure 17 before all other anatomical figures because it represents a 'summary' of your observations and would be useful for the reader when going through the detail description of the bones. A similar organization has been chosen in other stem-archosaur osteologies, e.g. *Erythrosuchus africanus* (Gower, 2003) and *Garjainia prima* (Ezcurra et al., 2019).

We have made this change, and refer first to the figure in the diagnosis section.

- Figure 1 has several problems in my opinion. First, *Dongusuchus* and *Dorosuchus* should be interchanged between each other. Nesbitt et al. (2017: Nature) have recovered *Yarasuchus* and *Dongusuchus* as part of the clade *Aphanosauria* (same paper) at the base of *Avemetatarsalia*. This result should be shown here because it has been one of the key modifications in the archosaur tree of the last years. One of the phylogenies used by Nesbitt et al. (2017) is a modification of that of Ezcurra (2016), which is used by the authors for reference in this figure.

We have revised the figure in accordance with these comments.

- Figure 2: add an arrowhead indicating that the Katberg Fm and the *Lystrosaurus* AZ extend to the Induan.

We have made this change.

- Figure 15: I think that this figure is not very informative in its current state. Bones in (A) are basically indistinguishable and labels overlap most of the morphology in (B) and (C). I strongly suggest showing the block in different portions (maybe in six captions) and do not include the labels over the photographs.

- Figure 16: similar problem as figure 15. Try to show bigger and in more detail the cranial bones and the vertebrae.

- Figure 17: can you label the subnarial foramen? The position of the epipterygoid seems to be quite unusual among archosauromorphs, its base usually sits on the base of the quadrate wing of the pterygoid. Can you check that the position of this bone is correct? The posterior region of the hemimandible seems to have an additional fenestra-size opening in the drawing, but as far as I remember from the holotype, this area has an elongated depression with the posterior surangular foramen on it. If this region of the hemimandible does not have an opening, please, modify the drawing to show a depression rather than a fenestra-like feature.

Figures 15 and 16: we have made the figure showing the blocks as a whole separate from those showing the close-up morphology, and provide a close-up image unlabelled next to the labelled image to prevent the morphology from being obscured by labelling. This is aimed to maximize the visible detail. We have not expanded the parts of the blocks showing non-cranial morphology as this is not relevant for the

paper at hand. Although not directly relevant to the description, and thus not included in the main text, larger images of the areas containing non-cranial material are made available in the supplementary photographs.

Figure 17: We have labelled the subnarial foramen, and changed the position of the epipterygoid, which the reviewer correctly pointed out was inaccurate. As far as we can see there is only one foramen marked on the posterior of the hemimandible, which is, as pointed out, the surangular foramen – this is now labelled. Its posterior extent was, however, exaggerated, and we have reduced this.

- I have made several, minor edits on the PDF of the manuscript.

We have made the suggested changes (unless there was a clear reason not to) and responses are documented in the PDF.

- Finally, I tried to open the supplementary information uploaded in Zenodo but I couldn't. It is possible that the files are not still available for the public, but if it is the case, please check that these links are working properly.

As mentioned in the letter to the editor, the supplement has not yet been uploaded to Zenodo but is available to the reviewers on Google Drive. We have informed the reviewer and passed on the link.

Yours sincerely,
Martin Ezcurra

Appendix B

[revised manuscript text omitted]

Formatted: Font: (Default) Arial, 12 pt
Formatted: Font: (Default) Arial, 12 pt
Formatted: Font: (Default) Arial, 12 pt
Formatted: Font: (Default) Arial, 12 pt
Formatted: Font: (Default) Arial, 12 pt
Formatted: Font: (Default) Arial, 12 pt
Formatted: Font: (Default) Arial, 12 pt
Formatted: Font: (Default) Arial, 12 pt
Formatted: Font: (Default) Arial, 12 pt
Formatted: English (United Kingdom)

2.5 Phylogenetic analysis

A phylogenetic analysis was carried out using the matrix of Sookias [32], including all
taxa and characters and (as in Sookias [32]) using *Youngina capensis* as an outgroup,
but rescaling character 90 (ectopterygoid single [0] or double [1] headed) to 1 (double
headed) for *Euparkeria* based on new information (see below). The modified matrix
and TNT code used to process it is given as supplementary information. A traditional
search was conducted with TNT 1.5 [64] using tree bisection-reconnection (TBR)
branch swapping and 1000 replicates. Standard and GC bootstrap values and decay
indices (Bremer support; using the Bremer script) were calculated for each node. The
same was also conducted a priori excluding ‘*Turfanosuchus shageduensis*’, as it has
been considered a *nomen dubium*.

**2.6 Institutional abbreviations**

AMNH, American Museum of Natural History, New York, USA; BPI, Evolutionary
Studies Institute, University of the Witwatersrand, Johannesburg, South Africa
[formerly Bernard Price Institute]; BSCUB, Biological Sciences Collection of the
University of Birmingham, Birmingham, UK; GPIT, Paläontologische Sammlung,
Eberhard-Karls-Universität Tübingen, Tübingen, Germany [formerly Geologisch-
Paläontologisches Institut Tübingen]; NHMUK, Natural History Museum, London, UK;
PULR, Paleontología, Universidad Nacional de La Rioja, La Rioja, Argentina; PVSJ,
División de Paleontología de Vertebrados del Museo de Ciencias Naturales y
Universidad Nacional de San Juan, San Juan, Argentina; SAM, Iziko South African
Museum, Cape Town, South Africa; UMZC, University Museum of Zoology Cambridge,
Cambridge, UK; ZPAL, Institute of Paleobiology, Polish Academy of Sciences,
Warsaw, Poland.

**3. Results**

**3.1 Systematic Paleontology**

Archosauriformes Gauthier, 1988 [65].

Euparkeriidae Huene, 1920 [66].

[revised manuscript text omitted]

SAM-PK-7699 (figure 124244), left mandible, left and possible right parietal and
possible left nasal.

SAM-PK-13664 (see figures 424240A, 434341A, 444442A,B), right pterygoid and
ectopterygoid, and left palatine exposed in ventral view.

SAM-PK-13665 (specimen 5 of Haughton [51]; figure 134342), crushed but largely
complete skull in block, exposed on left side with most elements roughly in articulation.

SAM-PK-13666 (figure 144443), slightly distorted, largely complete skull exposed on
left side.

SAM-PK-13667 (figure 154544), crushed partial skull including left and right mandibles
and right maxilla.

SAM-PK-K8050 (figures 164645, 17), four crushed skulls exposed in dorsal view
including nasals, frontals, parietals, postfrontals, prefrontals, quadrates, squamosals,

Formatted: Font: (Default) Arial

jugals and dentaries. The block also contains a skull and postcranium of the
rhynchosaur *Mesosuchus browni* (specimen SAM-PK-K8051).

UMZC T.692 (formerly R 527; figures 18, 19~~19~~16, and see figure 46~~46~~44B–D), cranial
remains of two individuals. Individual A includes frontals, parietals, right and left
prefrontal, right postfrontal, right and left lacrimal, right postorbital, largely complete
braincase, left pterygoid and ectopterygoid, partial right and left palatine and vomers,
left and right hyoid, left partial maxilla, left jugal, right quadrate, and anterior of left and
posterior of right mandible. Individual B includes left maxilla, anterior of left and
posterior of right mandible, part of anterior of left skull roof including nasal, posterior
of left jugal, parietals, and left quadrate.

Formatted: Font: (Default) Arial

[revised manuscript text omitted]

~~specimens were collected by Mr. A. Brown (or Mr. A.W. Higgins—see below), prior to~~
~~Watson's examination of them around the end of 1911 (see [50]), and the locality~~
~~was described as being "in the *Cynognathus* zone of the Karroo system" by Watson~~
~~[76]. Specimens from the collection of Brown were presented to the South African~~
~~Museum after his death in 1920, prior to Haughton's work in 1922 [51]. Following~~
~~Ewer [52], these specimens are labelled as "A. Brown collection, Krielfontein, Aliwal~~
~~North". Further material was also presented directly by Higgins to the SAM in 1924~~
~~and 1925 [52], with that presented in 1924 labelled "Higgins collection, Krielfontein,~~
~~Aliwal North" and that in 1925 labelled "A. W. Higgins collection, Quarry,~~
~~Commonage, Aliwal North". Haughton [51] reported the locality as "Krielfontein~~
~~Spruit on the Aliwal North Commonage" (with "Kriet" spelt with "t"), and indicated~~
~~that all specimens collected to date (1922) were from one locality. Based on a~~
~~communication from a certain Colonel de Wet, Ewer [52] suggested that some or~~
~~even all of the specimens labelled "A. Brown collection" may have been physically~~
~~collected by Higgins for Brown. Brown also directly presented a specimen (now~~
~~UMZC T.692) to Watson at University College London, from where it was transferred~~
~~to the University Museum of Zoology in Cambridge. Two SAM specimens (then SAM~~
~~7708 and 7698) were presented by the SAM to the Institut für Geologie und~~
~~Paläontologie at the University of Tübingen (now GPIT/RE/12913— which includes~~
~~cranial material; see above—and GPIT/RE/15029).~~

There has never been any doubt that all specimens were collected by either
Brown or Higgins from near Aliwal North (now in Walter Sisulu Municipality [77]),

Formatted: Don't hyphenate, Adjust space between Latin and Asian text, Adjust space between Asian text and numbers

Formatted: Font:

Eastern Cape, South Africa. An area Ewer [52] took to be the locality was located
after direction from a local resident, at an exposure of the appropriate sediments “on
one of the commonages over the brow of a hill on the left of the road to Lady Grey
just outside Aliwal North” (p. 381). ~~However, the recent discovery of detailed hand~~
~~written notes in the original journals of Brown has changed and clarified our~~
~~knowledge of the locality. These notes confirmed Haughton’s [51] name for the~~
~~locality (“Kriofontein Spruit” — “spruit” meaning “spring” in Afrikaans and referring to~~
~~small stream; this may have still — as suggested by Ewer [52] — been a private~~
~~collecting name, as it is not recorded in town records or known locally) and clearly~~
~~indicate that all of the specimens came from a quarry in a stream valley on the~~
~~western side of the English graveyard in Aliwal North. This is some 5 km away from~~
~~that determined by Ewer [52], who at the time of publishing was unaware of the~~
~~contents of Brown’s journals. The journals and the fossils have been in the SAM~~
~~collections since 1921, a year after the death of Brown, but have never been fully~~
~~documented or transcribed. [50, 76]~~

Following the directions of Brown to the abandoned quarry site next to the
English graveyard, R. M. H. S. and D.W. compared the sandstone outcropping in the
cliff face with the matrix of the *Euparkoria* blocks, and also the preservation style of a
newly discovered *in situ* archosaur long bone. Based on this we consider that the

[revised manuscript text omitted]

11 **Data accessibility**

Raw CT scan data for the cranial material of UMCZ T.692 have been made available
at the Zenodo digital repository (DOI: 10.5281/zenodo.3530854), and surface
renderings of SAM-PK-5867 and SAM-PK-6047A, are archived at Zenodo (DOI:
~~10.5281/zenodo.3530643~~10.5281/zenodo.3887056). Other raw CT data are available
from the authors on request, and will be archived following completion of further work
by B.A.B. Measurements, the matrix and analytical process used for phylogenetic
analysis, and matrix used for the anatomical network analysis are included with the
manuscript as supplementary information. A photogrammetric model of the skull of SAM-
PK-5867, courtesy of H. Mallison, is also archived at Zenodo (DOI:
~~10.5281/zenodo.3530643~~10.5281/zenodo.3887056). A bank of additional
photographs, which were taken by the authors and colleagues (with permission) and

used for the descriptive work have been uploaded to the Zenodo repository (DOI:
[10.5281/zenodo.3530643](https://doi.org/10.5281/zenodo.3530643) [10.5281/zenodo.3887056](https://doi.org/10.5281/zenodo.3887056)).

**Authors' contributions**

R.B.S. led the study and conducted most of the descriptive and photographic work,
studying all specimens, coded the network matrix, undertook the phylogenetic
analysis, and prepared the skull reconstruction, and wrote most of the manuscript.

D.D. contributed significantly to the description and reconstruction and contributed to
the manuscript in general. G.S. wrote much of the braincase description, contributed
to the braincase reconstruction, and to points in the discussion. R.M.H.S. and F.P.W.
conducted stratigraphic fieldwork on the section yielding *Euparkeria* and wrote most
of the geological background section of the manuscript. A.A. contributed to the
manuscript, provided photographs, and discussions. B.A.B. facilitated and assisted
with CT data and contributed to writing the manuscript. I.W. conducted network
analyses, wrote the related parts of the manuscript, and prepared related figures.

**Competing interests**

The authors declare no competing financial or non-financial interests.

**Funding**

During completion of this work, R.B.S. was supported by a European Research
Council Starting Grant (TEMPO to Roger B. J. Benson), a postdoctoral research grant
from the Alexander von Humboldt Foundation, a Marie Curie Career Integration Grant
(PCIG14-GA-2013-630123 Archosaur rise to Richard J. Butler), the College of Life and
Environmental Sciences of the University of Birmingham, UK, and an Emmy Noether
Programme Award from the Deutschen Forschungsgemeinschaft (DFG; BU 2587/3-1
to Richard J. Butler). G.S. was supported by a research grant from the DFG. R.M.H.S.
and F.P.W. were funded during their work by a grant to R.M.H.S. from the National
Research Foundation of South Africa through the African Origins Platform (project
UID117615). A. A. was supported by grant PROICO 2-1618 UNSL. I.W. was funded
by DFG grant WE 5440/6-1.

**Acknowledgements**

We thank S. Kaal and Z. Erasmus (SAM) for access to specimens in their care and
their hospitality to the authors when examining material of *Euparkeria* in Cape Town.

We also thank the following for access to specimens in their care: K. Mehling (AMNH);
R. Hall, K. Wroe and A. L. Stewart (BSCUB); B. S. Rubidge (BPI); L. Steel, P. D.
Campbell and S. D. Chapman (NHMUK); M. Borsuk-Białynicka (ZPAL). We thank J.
Müller (Museum für Naturkunde, Berlin - MfN) for support to R.B.S. when completing
the work, and M. Kirchner and K. Mahlow (both MfN) for assistance with CT data. We

thank B. Hendrick (Louisiana State University) and R. B. J. Benson (University of
Oxford) for allowing use of light surface scans to create a 3D model of the holotype.

~~We thank H. Mallison (Palaeo3D) for constructing a photogrammetric model of the~~

~~holotype~~. We thank O. Yaryhin (Eberhard-Karls-Universität Tübingen) for assistance
with anatomical network analysis. We thank A. Schmitt (University of Oxford) and P.
Andreev (Qujing Normal University) for software and technical assistance. We also
thank R. J. Butler (University of Birmingham) for providing extensive comments on
earlier versions of the manuscript, discussion, and for efforts in facilitating the study's
completion.

**References**

- 1 Jetz, W., Thomas, G., Joy, J., Hartmann, K., Mooers, A. 2012 The global diversity of birds
in space and time. *Nature*. **491**, 444.
- 2 Irmis, R. B., Nesbitt, S. J., Padian, K., Smith, N. D., Turner, A. H., Woody, D., Downs, A.
2007 A Late Triassic dinosauromorph assemblage from New Mexico and the rise of
dinosaurs. *Science*. **317**, 358-361.
- Brusatte, S. L., Benton, M. J., Lloyd, G. T., Ruta, M., Wang, S. C. 2010 Macroevolutionary
patterns in the evolutionary radiation of archosaurs (Tetrapoda: Diapsida). *Earth Environ. Sci.*
*Trans. R. Soc. Edinburgh*. **101**, 367-382.
- Brusatte, S. L., Nesbitt, S. J., Irmis, R. B., Butler, R. J., Benton, M. J., Norell, M. A. 2010
The origin and early radiation of dinosaurs. *Earth-Sci. Rev.* **101**, 68-100.
- Sookias, R. B., Butler, R. J., Benson, R. B. 2012 Rise of dinosaurs reveals major body-size
transitions are driven by passive processes of trait evolution. *Proc. R. Soc. B*. **279**, 2180-2187.
- Ezcurra, M. D., Butler, R. J. 2018 The rise of the ruling reptiles and ecosystem recovery
from the Permo-Triassic mass extinction. *Proceedings of the Royal Society B: Biological*
*Sciences*. **285**, (10.1098/rspb.2018.0361)
- Kaucza, M., Adameyko, I. Year Evolution and development of the cartilaginous skull: from
a lancelet towards a human face. *Seminars in Cell & Developmental Biology*; 2017:
Elsevier; 2017.
- Couly, G. F., Coltey, P. M., Le Douarin, N. M. J. D. 1993 The triple origin of skull in higher
vertebrates: a study in quail-chick chimeras. **117**, 409-429.
- Andjelković, M., Tomović, L., Ivanović, A. J. J. o. Z. 2017 Morphological integration of the
kinetic skull in Natrix snakes. **303**, 188-198.
- Rico-Guevara, A., Rubega, M., Hurme, K., Dudley, R. J. I. O. B. 2019 Shifting Paradigms
in the Mechanics of Nectar Extraction and Hummingbird Bill Morphology. **1**, oby006.
- Hurum, J. H., Sabath, K. 2003 Giant theropod dinosaurs from Asia and North America:
skulls of *Tarbosaurus bataar* and *Tyrannosaurus rex* compared. *Acta Palaeontologica*
*Polonica*. **48**,
- Nabavizadeh, A., Weishampel, D. B. J. T. A. R. 2016 The prementary bone and its
significance in the evolution of feeding mechanisms in ornithischian dinosaurs. **299**, 1358-
1388.

Hopson, J. A. J. A. R. o. E., Systematics. 1977 Relative brain size and behavior in
archosaurian reptiles. **8**, 429-448.

Bhullar, B.-A. S., Marugán-Lobón, J., Racimo, F., Bever, G. S., Rowe, T. B., Norell, M.
4 A., Abzhanov, A. J. N. 2012 Birds have paedomorphic dinosaur skulls. **487**, 223.

Lee, M. S. Y., Cau, A., Naish, D., Dyke, G. J. 2014 Sustained miniaturization and
anatomical innovation in the dinosaurian ancestors of birds. *Science*. **345**, 562-566.
(10.1126/science.1252243)

Hurlburt, G. R., Ridgely, R. C., Witmer, L. M. J. T. p. 2013 Relative size of brain and
cerebrum in tyrannosaurid dinosaurs: an analysis using brain-endocast quantitative
relationships in extant alligators. 1-21.

Bhullar, B.-A. S., Hanson, M., Fabbri, M., Pritchard, A., Bever, G. S., Hoffman, E. 2016
How to Make a Bird Skull: Major Transitions in the Evolution of the Avian Cranium,
Paedomorphosis, and the Beak as a Surrogate Hand. *Integrative and Comparative Biology*.
**56**, 389-403. (10.1093/icb/icw069)

Broom, R. 1913 On the South- African pseudosuchian *Euparkeria* and allied genera.
*Proceedings of the Zoological Society of London*. **83**, 619-633.

von Baczko, M. B., Ezcurra, M. D. 2016 Taxonomy of the archosaur *Ornithosuchus*:
reassessing *Ornithosuchus woodwardi* Newton, 1894 and *Dasygnathoides longidens* (Huxley
1877). *Earth Environ. Sci. Trans. R. Soc. Edinburgh*. **106**, 199-205.

Benton, M. J., Clark, J. M. 1988 Archosaur phylogeny and the relationships of the
Crocodylia. *The phylogeny and classification of the tetrapods*. **1**, 295-338.

Sereno, P. C., Arcucci, A. 1990 The monophyly of crurotarsal archosaurs and the origin of
bird and crocodile ankle joints. *N. Jb. Geol. Paläontol. Abh.* **180**, 21-52.

Sereno, P. C. 1991 Basal archosaurs: phylogenetic relationships and functional
implications. *J. Vert. Paleontol.* **11**, 1-53.

Parrish, J. M. 1993 Phylogeny of the Crocodylotarsi, with reference to archosaurian and
crurotarsan monophyly. *J. Vert. Paleontol.* **13**, 287-308.

Juul, L. 1994 The phylogeny of basal archosaurs. *Palaeontol. Afr.* **31**, 1-38.

Bennett, C. S. 1996 The phylogenetic position of the Pterosauria within the
Archosauromorpha. *Zool. J. Linn. Soc.* **118**, 261-308.

Benton, M. J. 1999 *Scleromochlus taylori* and the origin of dinosaurs and pterosaurs. *Phil.*
*Trans. R Soc. B.* **354**, 1423-1446.

Parker, W. G., Barton, B. J. 2008 New information on the Upper Triassic archosauriform
*Vancleavea campi* based on new material from the Chinle Formation of Arizona. *Palaeontol.*
*Electron.* **11**, 14A.

Nesbitt, S. J., Stocker, M. R., Small, B. J., Downs, A. 2009 The osteology and
relationships of *Vancleavea campi* (Reptilia: Archosauriformes). *Zool. J. Linn. Soc.* **157**, 814-
864.

Brusatte, S. L., Benton, M. J., Desojo, J. B., Langer, M. C. 2010 The higher-level
phylogeny of Archosauria (Tetrapoda: Diapsida). *J. Syst. Palaeontol.* **8**, 3-47.

Ezcurra, M. D., Lecuona, A., Martinelli, A. 2010 A new basal archosauriform diapsid from
the Lower Triassic of Argentina. *J. Vert. Paleontol.* **30**, 1433-1450.

Nesbitt, S. J. 2011 The early evolution of archosaurs: relationships and the origin of major
clades. *Bull. Am. Mus. Nat. Hist.*, 1-292.

Sookias, R. B. 2016 The relationships of the Euparkeriidae and the rise of Archosauria. *R.*
*Soc. Open Sci.* **3**, 150674.

- Sookias, R. B., Sennikov, A. G., Gower, D. J., Butler, R. J. 2014 The monophyly of
Euparkeriidae (Reptilia: Archosauriformes) and the origins of Archosauria: a revision of
*Dorosuchus neoetus* from the Mid- Triassic of Russia. *Palaeontology*. **57**, 1177-1202.
- Sookias, R. B., Sullivan, C., Liu, J., Butler, R. J. 2014 Systematics of putative euparkeriids
(Diapsida: Archosauriformes) from the Triassic of China. *PeerJ*. **2**, e658.
- Romer, A. S. 1972 The Chañares (Argentina) Triassic reptile fauna. XVI. Thecodont
classification. *Breviora*. **395**, 1-24.
- Gower, D. J., Weber, E. 1998 The braincase of Euparkeria, and the evolutionary
relationships of birds and crocodylians. *Biological reviews*. **73**, 367-411.
- Perry, S. F. 1992 Gas exchange strategies in reptiles and the origin of the avian lung.
*Physiological Adaptations in Vertebrates: respiration, circulation, and metabolism*. **56**, 149-
167.
- Carrier, D. R., Farmer, C. G. 2000 The evolution of pelvic aspiration in archosaurs.
*Paleobiology*. **26**, 271-293.
- Hutchinson, J. R. 2001 The evolution of pelvic osteology and soft tissues on the line to
extant birds (Neornithes). *Zool. J. Linn. Soc.* **131**, 123-168.
- Marugán-Lobón, J., Buscalioni, A. D. 2003 Disparity and geometry of the skull in
Archosauria (Reptilia: Diapsida). *Biological journal of the Linnean Society*. **80**, 67-88.
- Nesbitt, S. J. 2003 *Arizonasaurus* and its implications for archosaur divergence. *Proc. R.*
*Soc. B.* **270**, S234-S237.
- Rauhut, O. W. 2003 *Special Papers in Palaeontology, The Interrelationships and*
*Evolution of Basal Theropod Dinosaurs*. Blackwell Publishing.
- Seymour, R. S., Bennett-Stamper, C. L., Johnston, S. D., Carrier, D. R., Grigg, G. C. 2004
Evidence for endothermic ancestors of crocodiles at the stem of archosaur evolution.
*Physiological and Biochemical Zoology*. **77**, 1051-1067.
- de Ricqlès, A., Padian, K., Knoll, F., Horner, J. R. 2008 On the origin of high growth rates
in archosaurs and their ancient relatives: complementary histological studies on Triassic
archosauriforms and the problem of a “phylogenetic signal” in bone histology. *Annales de*
*Paléontologie*. **94**, 57-76.
- Sullivan, C. 2010 The role of the calcaneal ‘heel’ as a propulsive lever in basal archosaurs
and extant monitor lizards. *J. Vert. Paleontol.* **30**, 1422-1432.
- Maidment, S. C., Barrett, P. M. 2011 The locomotor musculature of basal ornithischian
dinosaurs. *J. Vert. Paleontol.* **31**, 1265-1291.
- Butler, R. J., Barrett, P. M., Gower, D. J. 2012 Reassessment of the evidence for
postcranial skeletal pneumaticity in Triassic archosaurs, and the early evolution of the avian
respiratory system. *PLoS ONE*. **7**, e34094.
- Foth, C., Rauhut, O. W. 2013 Macroevolutionary and morphofunctional patterns in
theropod skulls: a morphometric approach. *Acta Palaeontologica Polonica*. **58**, 1-16.
- Baron, M. G., Norman, D. B., Barrett, P. M. 2017 A new hypothesis of dinosaur
relationships and early dinosaur evolution. *Nature*. **543**, 501-506.
- Broom, R. 1913 Note on *Mesosuchus browni*, Watson, and on a new South African
Triassic pseudosuchian (*Euparkeria capensis*). *Records of the Albany Museum*. **2**, 394-396.
- Houghton, S. 1922 On the reptilian genera *Euparkeria* Broom, and *Mesosuchus* Watson.
*Transactions of the Royal Society of South Africa*. **10**, 81-88.
- Ewer, R. F. 1965 The anatomy of the thecodont reptile *Euparkeria capensis* Broom. *Phil.*
*Trans. R. Soc. Lond. B.* **248**, 379-435.

Sobral, G., Sookias, R. B., Bhullar, B.-A. S., Smith, R., Butler, R. J., Müller, J. 2016 New
information on the braincase and inner ear of *Euparkeria capensis* Broom: implications for
diapsid and archosaur evolution. *R. Soc. Open Sci.* **3**, 160072.
Hancox, P. J. 2000 The continental Triassic of South Africa. *Zentralblatt für Geologie und*
*Paläontologie Teil I.* **1998**, 1285-1324.
Hancox, P., Shishkin, M., Rubidge, B., Kitching, J. 1995 A threefold subdivision of the
*Cynognathus* Assemblage Zone (Beaufort Group, South-Africa) and its paleogeographic
implications. *South African Journal of Science.* **91**, 143-144.
Ottone, E. G., Monti, M., Marsicano, C. A., Marcelo, S., Naipauer, M., Armstrong, R.,
Mancuso, A. C. 2014 A new Late Triassic age for the Puesto Viejo Group (San Rafael
depocenter, Argentina): SHRIMP U–Pb zircon dating and biostratigraphic correlations across
southern Gondwana. *Journal of South American Earth Sciences.* **56**, 186-199.
Schneider, J. W., Lucas, S. G., Scholze, F., Voigt, S., Marchetti, L., Klein, H., Opluštil, S.,
Werneburg, R., Golubev, V. K., Barrick, J. E. 2019 Late Paleozoic–early Mesozoic
continental biostratigraphy—links to the Standard Global Chronostratigraphic Scale.
*Palaeoworld.*
Werneburg, I., Esteve-Altava, B., Bruno, J., Ladeira, M. T., Diogo, R. 2019 Unique skull
network complexity of *Tyrannosaurus rex* among land vertebrates. *Sci. Rep.* **9**, 1520.
R-Core-Team. R: A language and environment for statistical computing. Vienna, Austria:
R Foundation for Statistical Computing. 2014.
Csardi, G., Nepusz, T. 2006 The igraph software package for complex network research.
*InterJournal, Complex Systems.* **1695**, 1-9.
Esteve- Altava, B. J. J. o. M. 2017 Challenges in identifying and interpreting
organizational modules in morphology. **278**, 960-974.
Esteve- Altava, B., Marugán- Lobón, J., Botella, H., Bastir, M., Rasskin- Gutman, D. J.
26 J. o. E. Z. P. B. M., Evolution, D. 2013 Grist for Riedl's mill: a network model perspective on
the integration and modularity of the human skull. **320**, 489-500.
Esteve-Altava, B., Marugán-Lobón, J., Botella, H., Rasskin-Gutman, D. 2013 Structural
constraints in the evolution of the tetrapod skull complexity: Williston's law revisited using
network models. *Evolutionary Biology.* **40**, 209-219.
Goloboff, P. A., Catalano, S. A. 2016 TNT version 1.5, including a full implementation of
phylogenetic morphometrics. *Cladistics.* **32**, 221-238.
Gauthier, J., Kluge, A. G., Rowe, T. 1988 Amniote phylogeny and the importance of
fossils. *Cladistics.* **4**, 105-209.
Huene, F. v. von 1920. Osteologie von *Aetosaurus ferratus* O. Fraas. *Acta Zoologica.* **3**,
465-491.
Laurenti, J.-N. 1768 *Specimen medicum, exhibens synopin reptilium emendatam cum*
*experimentis circa venena et antidota reptilium Austriacorum.* Trattner.
von Linné, C. 1767 *Systema naturae per regna tria naturae: secundum classes, ordines,*
*genera, species, cum characteribus, differentiis, synonymis, locis.* Typis Ioannis Thomae.
Sookias, R. B., Butler, R. J. 2013 Euparkeriidae. *Geological Society, London, Special*
*Publications.* **379**, SP379. 376.
Weinbaum, J. C. 2011 The skull of *Postosuchus kirkpatricki* (Archosauria:
Paracrocodyliformes) from the Upper Triassic of the United States. *PaleoBios.* **30**,
Borsuk-Białynicka, M., Evans, S. 2009 Cranial and mandibular osteology of the early
triassic archosauriform *Osmolskina czatkowicensis* from Poland. *Palaeontologia Polonica.*
**65**, 235-281.

Formatted: German (Germany)

Formatted: German (Germany)

- Gower, D. J. 1999 The cranial and mandibular osteology of a new rauisuchian archosaur
from the Middle Triassic of southern Germany. *Stuttgarter Beiträge zur Naturkunde: Serie B*
*(Geologie und Paläontologie)*. **280**, 1-50.
- Walker, A. D. 1964 Triassic reptiles from the Elgin area: *Ornithosuchus* and the origin of
carnosaurs. *Phil. Trans. R. Soc. Lond. B.* **248**, 53-134.
- Sereno, P. C., Novas, F. E. 1994 The skull and neck of the basal theropod *Herrerasaurus*
*ischigualastensis*. *J. Vert. Paleontol.* **13**, 451-476.
- Sereno, P. C., Martínez, R. N., Alcober, O. A. 2012 Osteology of *Eoraptor lunensis*
(Dinosauria, Sauropodomorpha). *J. Vert. Paleontol.* **32**, 83-179.
- Cruickshank, A. 1972 The proterosuchian thecodonts. *Studies in vertebrate evolution*. **89**,
19.
- Romer, A. S. 1971 The Chañares (Argentina) Triassic reptile fauna. XI. Two new long-
snouted thecodonts, *Chanaresuchus* and *Gualosuchus*. *Breviora*. **379**, 1-22.
- Gower, D. J. 2003 Osteology of the early archosaurian reptile *Erythrosuchus africanus*
Broom. *Annals of the South African Museum*. **110**, 1-88.
- Walker, A. D. 1990 A revision of *Sphenosuchus acutus* Haughton, a crocodylomorph
reptile from the Elliot Formation (late Triassic or early Jurassic) of South Africa. *Phil. Trans.*
*R. Soc. Lond. B.* **330**, 1-120.
- Witmer, L. M. 1997 The evolution of the antorbital cavity of archosaurs: a study in soft-
tissue reconstruction in the fossil record with an analysis of the function of pneumaticity. *J.*
*Vert. Paleontol.* **17**, 1-76.
- Sobral, G., Müller, J. 2019 The braincase of *Mesosuchus browni* (Reptilia,
Archosauromorpha) with information on the inner ear and description of a pneumatic sinus.
*PeerJ*. **7**, e6798.
- Gower, D. J., Sennikov, A. 1996 Morphology and phylogenetic informativeness of early
archosaur braincases. *Palaeontology*. **39**, 883-906.
- Evans, S. 1986 The braincase of *Prolacerta broomi* (Reptilia: Triassic). *Neues Jahrbuch*
*für Geologie und Paläontologie: Abhandlungen*. **173**, 181-200.
- Heaton, M. J. 1979 Cranial anatomy of primitive captorhinid reptiles from the Late
Pennsylvanian and Early Permian, Oklahoma and Texas. *Oklahoma Geological Survey*
*Bulletin*. **127**, 1-81.
- Gardner, N. M., Holliday, C. M., O'Keefe, F. R. 2010 The braincase of *Youngina capensis*
(Reptilia, Diapsida): new insights from high-resolution CT scanning of the holotype.
- Modesto, S. P., Sues, H.-D. 2004 The skull of the Early Triassic archosauromorph reptile
*Prolacerta broomi* and its phylogenetic significance. *Zool. J. Linn. Soc.* **140**, 335-351.
- Lautenschlager, S., Rauhut, O. W. 2015 Osteology of *Rauisuchus tiradentes* from the Late
Triassic (Carnian) Santa Maria Formation of Brazil, and its implications for rauisuchid
anatomy and phylogeny. *Zool. J. Linn. Soc.* **173**, 55-91.
- Ochev, V. G. 1958 Novyye dannye po psevdozukkhiiyam SSSR. *Doklady Akademi Nauk.*
**123**, 749-751.
- Romer, A. S. 1972 The Chanares (Argentina) Triassic reptile fauna. An early
ornithosuchid pseudosuchian, *Gracilisuchus stipanicicorum*, gen. et sp. nov. *Breviora*. **389**, 1-
24.
- Wu, X. 1981 The discovery of a new thecodont from north-east Shensi. *Vertebr.*
*Palasiatica*. **19**, 122-132.
- Nesbitt, S. J. 2005 Osteology of the Middle Triassic pseudosuchian archosaur
*Arizonasaurus babbitti*. *Historical Biology*. **17**, 19-47.

- Walker, A. D. 1961 Triassic reptiles from the Elgin area: *Stagonolepis*, *Dasygnathus* and
their allies. *Phil. Trans. R. Soc. Lond. B.* **244**, 103-204.
- Nesbitt, S. 2007 The anatomy of *Effigia okeeffeae* (Archosauria, Suchia), theropod-like
convergence, and the distribution of related taxa. *Bull. Am. Mus. Nat. Hist.*, 1-84.
- Senter, P. 2003 New information on cranial and dental features of the Triassic
archosauriform reptile *Euparkeria capensis*. *Palaeontology*. **46**, 613-621.
- Brusatte, S. L., Butler, R. J., Sulej, T., Niedźwiedzki, G. 2009 The taxonomy and anatomy
of rauisuchian archosaurs from the Late Triassic of Germany and Poland. *Acta*
*Palaeontologica Polonica*. **54**, 221-230.
- Bonaparte, J. F. 1971 Los tetrápodos del sector superior de la formación Los Colorados,
La Rioja, Argentina (Triásico Superior): I Parte.
- von Baczko, M. B., Desojo, J. B. J. P. o. 2016 Cranial anatomy and palaeoneurology of the
archosaur *Riojasuchus tenuisiceps* from the Los Colorados Formation, La Rioja, Argentina.
**11**, e0148575.
- Yates, A. M. 2003 A new species of the primitive dinosaur *Thecodontosaurus* (Saurischia:
Sauropodomorpha) and its implications for the systematics of early dinosaurs. *J. Syst.*
*Palaeontol.* **1**, 1-42.
- Langer, M. C., Benton, M. J. 2006 Early dinosaurs: a phylogenetic study. *J. Syst.*
*Palaeontol.* **4**, 309-358.
- Arcucci, A. 2011 Sistemática y filogenia de los proterochampsidos (Amniota, Diápsida,
Archosauriformes) del Triásico de América del Sur, y sus implicancias en el origen de
Archosauria. *Unpublished PhD thesis, Universidad Nacional de San Luis*.
- Dilkes, D. W. 1995 The rhynchosaur *Howesia browni* from the Lower Triassic of South
Africa. *Palaeontology*. **38**, 665-686.
- Clark, J. M., Sues, H.-D. 2002 Two new basal crocodylomorph archosaurs from the
Lower Jurassic and the monophyly of the Sphenosuchia. *Zool. J. Linn. Soc.* **136**, 77-95.
- Clark, J. M., Xu, X., Forster, C. A., Wang, Y. 2004 A Middle Jurassic 'sphenosuchian'
from China and the origin of the crocodylian skull. *Nature*. **430**, 1021.
- Dilkes, D., Arcucci, A. 2012 *Proterochampsia barrionuevoi* (Archosauriformes:
Proterochampsia) from the Late Triassic (Carnian) of Argentina and a phylogenetic analysis
of Proterochampsia. *Palaeontology*. **55**, 853-885.
- Dilkes, D. W. 1998 The Early Triassic rhynchosaur *Mesosuchus browni* and the
interrelationships of basal archosauriform reptiles. *Phil. Trans. R Soc. B.* **353**, 501-541.
- Parrington, F. 1937 V.—A note on the supratemporal and tabular bones in reptiles.
*Journal of Natural History*. **20**, 69-76.
- Gower, D. J., Walker, A. D. 2002 New data on the braincase of the aetosaurian archosaur
(Reptilia: Diapsida) *Stagonolepis robertsoni* Agassiz. *Zool. J. Linn. Soc.* **136**, 7-23.
- Ochev, V. 1981 On *Erythrosuchus* (Garjainia) primus Ochev. *Voprosy Geologii Yuzhnogo*
*Urala I Povolzh'ya*. **22**, 3-22.
- Wang, R., Xu, S., Wu, X., Li, C., Wang, S. 2013 A new specimen of *Shansisuchus*
*shansisuchus* Young, 1964 (Diapsida: Archosauriformes) from the Triassic of Shanxi, China.
*Acta Geologica Sinica- English Edition*. **87**, 1185-1197.
- Young, C.-C. 1964 The pseudosuchians in China. *Palaeontologia Sinica*. 1-205.
- Rauhut, O. W. 1997 On the cranial anatomy of *Shuvosaurus inexpectatus* (Dinosauria;
Theropoda). *Terra Nostra*. **7**, 1-4.
- Gower, D. J., Hancox, P. J., Botha-Brink, J., Sennikov, A. G., Butler, R. J. 2014 A new
species of *Garjainia* Ochev, 1958 (Diapsida: Archosauriformes: Erythrosuchidae) from the
Early Triassic of South Africa. *PLoS ONE*. **9**, e111154.

- Müller, J. J. N. 2003 Early loss and multiple return of the lower temporal arcade in
diapsid reptiles. **90**, 473-476.
- Bever, G., Lyson, T. R., Field, D. J., Bhullar, B.-A. S. J. N. 2015 Evolutionary origin of
the turtle skull. **525**, 239.
- Heckert, A. B., Lucas, S. G. 1999 A new aetosaur (Reptilia: Archosauria) from the Upper
Triassic of Texas and the phylogeny of aetosaurs. *J. Vert. Paleontol.* **19**, 50-68.
- Parker, W. G. 2007 Reassessment of the aetosaur '*Desmatosuchus chamaensis* with a
reanalysis of the phylogeny of the Aetosauria (Archosauria: Pseudosuchia). *J. Syst.*
*Palaeontol.* **5**, 41-68.
- Sulej, T. 2005 A new rauisuchian reptile (Diapsida: Archosauria) from the Late Triassic
of Poland. *J. Vert. Paleontol.* **25**, 78-86.
- Ochev, V. 1958 New data on the Triassic vertebrate fauna from the Orenburg Cis-Urals.
*Doklady Akademii Nauk SSSR.* **22**, 749-752.
- Dzik, J., Sulej, T. 2007 A review of the early Late Triassic Krasiejów biota from Silesia,
Poland. *Phytopatologia Polonica.* 3-27.
- Gow, C. E. 1970 The anterior of the palate in *Euparkeria*. *Paleaeontologia Africana.* **13**,
61-62.
- Camp, C. L. 1945 Prolacerta and the protorosaurian reptiles; Part I. *American Journal of*
*Science.* **243**, 17-32.
- Welman, J. 1998 The taxonomy of the South African proterosuchids (Reptilia,
Archosauromorpha). *J. Vert. Paleontol.* **18**, 340-347.
- Ochev, V. 1975 On the proterosuchian palate. *Paleontologicheskii Zhurnal.* **1980**, 98-
105.
- Dollman, K. N., Clark, J. M., Norell, M. A., Xing, X., Choiniere, J. N. J. A. M. N. 2018
Convergent evolution of a eusuchian-type secondary palate within Shartegosuchidae. **2018**,
1-24.
- Butler, R. J., Sullivan, C., Ezcurra, M. D., Liu, J., Lecuona, A., Sookias, R. B. 2014 New
clade of enigmatic early archosaurs yields insights into early pseudosuchian phylogeny and
the biogeography of the archosaur radiation. *BMC Evol. Biol.* **14**, 128.
- Brusatte, S. L., Carr, T. D., Norell, M. A. 2012 The osteology of *Alioramus*, a gracile and
long-snouted tyrannosaurid (Dinosauria: Theropoda) from the Late Cretaceous of Mongolia.
*Bull. Am. Mus. Nat. Hist.*, 1-197.
- Dilkes, D., Sues, H.-D. 2009 Redescription and phylogenetic relationships of *Doswellia*
*kaltenbachi* (Diapsida: Archosauriformes) from the Upper Triassic of Virginia. *J. Vert.*
*Paleontol.* **29**, 58-79.
- Benton, M. J., Walker, A. D. 2002 *Erpetosuchus*, a crocodile-like basal archosaur from
the Late Triassic of Elgin, Scotland. *Zool. J. Linn. Soc.* **136**, 25-47.
- Nesbitt, S. J., Norell, M. A. 2006 Extreme convergence in the body plans of an early
suchian (Archosauria) and ornithomimid dinosaurs (Theropoda). *Proc. R. Soc. B.* **273**, 1045-
1048.
- Schmitz, L., Motani, R. 2011 Nocturnality in dinosaurs inferred from scleral ring and
orbit morphology. *Science.* **332**, 705-708.
- Mastrantonio, B., Von Baczko, M. B., Desojo, J. B., Schultz, C. L. 2019 The skull
anatomy and cranial endocast of the pseudosuchid archosaur *Prestosuchus chiniquensis* from
the Triassic of Brazil. **64**, 171-198.
- Rasskin-Gutman, D., Esteve-Altava, B. J. B. T. 2014 Connecting the dots: anatomical
network analysis in morphological EvoDevo. **9**, 178-193.

Ezcurra, M. D. 2016 The phylogenetic relationships of basal archosauromorphs, with an
emphasis on the systematics of proterosuchian archosauriforms. *PeerJ*. **4**, e1778.

Jaquier, V. P., Fraser, N. C., Furrer, H., Scheyer, T. M. 2017 Osteology of a new specimen
of *Macrocnemus* aff. *M. fuyuanensis* (Archosauromorpha, Protorosauria) from the Middle
Triassic of Europe: potential implications for species recognition and paleogeography of
tanystropheid protosaurs. *Frontiers in Earth Science*. **5**, 91.

Koyabu, D., Werneburg, I., Morimoto, N., Zollikofer, C. P., Forasiepi, A. M., Endo, H.,
Kimura, J., Ohdachi, S. D., Son, N. T., Sánchez-Villagra, M. R. 2014 Mammalian skull
heterochrony reveals modular evolution and a link between cranial development and brain
size. *Nature Comms*. **5**, 3625.

Spiekman, S., Scheyer, T. 2019 A taxonomic revision of the genus *Tanystropheus*
(Archosauromorpha, Tanystropheidae). *Palaeontol. Electron.*, Epub ahead of print.

Gregory, W. K. 1935 'Williston's law' relating to the evolution of skull bones in the
vertebrates. *American Journal of Physical Anthropology*. **20**, 123-152.

Sidor, C. A. 2001 Simplification as a trend in synapsid cranial evolution. *Evolution*. **55**,
1419-1442.

Plateau, O., Foth, C. 2020 Birds have peramorphic skulls, too: anatomical network
analyses reveal oppositional heterochronies in avian skull evolution. *Communications*
*Biology*. **3**, 1-12.

Bout, R. G., Zweers, G. A. J. C. B., Molecular, P. P. A., Physiology, I. 2001 The role of
cranial kinesis in birds. **131**, 197-205.

Gussekloo, S. W. S., Bout, R. G. 2005 Cranial kinesis in palaeognathous birds. **208**,
3409-3419. (10.1242/jeb.01768 %J Journal of Experimental Biology)

Holliday, C. M., Witmer, L. M. 2008 Cranial kinesis in dinosaurs: intracranial joints,
protractor muscles, and their significance for cranial evolution and function in diapsids. *J.*
*Vert. Paleontol.* **28**, 1073-1088.

Herring, S. W., Rafferty, K. L., Liu, Z. J., Marshall, C. D. J. C. B., Molecular, P. P. A.,
Physiology, I. 2001 Jaw muscles and the skull in mammals: the biomechanics of mastication.
**131**, 207-219.

Bock, W. J. 1964 Kinetics of the avian skull. *Journal of Morphology*. **114**, 1-41.

Herrel, A., Schaerlaeken, V., Meyers, J. J., Metzger, K. A., Ross, C. F. 2007 The
evolution of cranial design and performance in squamates: Consequences of skull-bone
reduction on feeding behavior. *Integrative and Comparative Biology*. **47**, 107-117.
(10.1093/icb/icm014)

Schoch, R., Sues, H. 2013 A new doswelliid archosauriform from the Middle Triassic of
Germany. *J Syst Palaeontol* [doi: 10.1080/14772019.2013.781066].

Gauthier, J. 1986 Saurischian monophyly and the origin of birds. *Memoirs of the*
*California Academy of sciences*. **8**, 1-55.

Botha-Brink, J., Smith, R. M. 2011 Osteohistology of the Triassic archosauromorphs
*Prolacerta*, *Proterosuchus*, *Euparkeria*, and *Erythrosuchus* from the Karoo Basin of South
Africa. *J. Vert. Paleontol.* **31**, 1238-1254.

Ezcurra, M. D., Butler, R. J. 2015 Taxonomy of the proterosuchid archosauriforms
(Diapsida: Archosauromorpha) from the earliest Triassic of South Africa, and implications for
the early archosauriform radiation. *Palaeontology*. **58**, 141-170.

Chatterjee, S. 1978 A primitive parasuchid (phytosaur) reptile from the Upper Triassic
Maleri Formation of India. *Palaeontology*. **21**, 83-127.

Colbert, E. H. 1989 The Triassic dinosaur *Coelophysis*. *Museum of Northern Arizona*
*Bulletin*. **53**, 1-160.

- Matsumoto, R., Evans, S. E. 2017 The palatal dentition of tetrapods and its functional
significance. *Journal of anatomy*. **230**, 47-65.
- Gow, C. E. 1975 The morphology and relationships of *Youngina capensis* Broom and
*Prolacerta broomi* Parrington. *Palaeontol. Afr.* **18**, 89-131.
- Legendre, L. J., Segalen, L., Cubo, J. J. J. o. V. P. 2013 Evidence for high bone growth
rate in Euparkeria obtained using a new paleohistological inference model for the humerus.
**33**, 1343-1350.
- Legendre, L. J., Guénard, G., Botha-Brink, J., Cubo, J. J. S. B. 2016 Palaeohistological
evidence for ancestral high metabolic rate in archosaurs. **65**, 989-996.
- Witmer, L., Ridgely, R., Dufeu, D., Semones, M. Anatomical imaging: towards a new
morphology. Tokyo: Springer-Verlag 2008.
- Walsh, S. A., Iwaniuk, A. N., Knoll, M. A., Bourdon, E., Barrett, P. M., Milner, A. C.,
Nudds, R. L., Abel, R. L., Sterpaio, P. D. 2013 Avian cerebellar floccular fossa size is not a
proxy for flying ability in birds. *PLoS ONE*. **8**, e67176.
- Bronzati, M., Benson, R. B., Rauhut, O. W. 2017 Rapid transformation in the braincase
of sauropod dinosaurs: integrated evolution of the braincase and neck in early sauropods?
*Palaeontology*. **61**, 289-302.
- Dudley, R., Yanoviak, S. P. 2011 Animal aloft: the origins of aerial behavior and flight.
*Integrative and comparative biology*. **51**, 926-936.
- Vasilopoulou- Kampitsi, M., Goyens, J., Baeckens, S., Van Damme, R., Aerts, P. 2019
Habitat use and vestibular system's dimensions in lacertid lizards. *Journal of anatomy*. **235**,
1-14.
- Vasilopoulou-Kampitsi, M., Goyens, J., Van Damme, R., Aerts, P. 2019 The ecological
signal on the shape of the lacertid vestibular system: simple versus complex microhabitats.
*Biological Journal of the Linnean Society*. **127**, 260-277.
- Versluys, J. 1912 Das Streptostylie Problem und die Bewegungen im Schädel bei
Sauropsiden. *Zool. Jb. Suppl.* **15**, 545-714.
- Evans, S. E. 2008 The skull of lizards and tuatara. *Biology of the Reptilia*. **20**, 1-347.
- Werneburg, I. 2019 Morphofunctional categories and ontogenetic origin of temporal
skull openings in amniotes. *Frontiers in Earth Science*. **7**, (10.3389/feart.2019.00013)
- Ferreira, G. S., Werneburg, I. 2019 Evolution, Diversity, and Development of the
Craniocervical System in Turtles with Special Reference to Jaw Musculature. In *Heads,*
*Jaws, and Muscles*. (ed.^eds. pp. 171-206: Springer.
- Diaz, R. E., Trainor, P. A. 2019 An Integrative View of Lepidosaur Cranial Anatomy,
Development, and Diversification. In *Heads, Jaws, and Muscles*. (ed.^eds. pp. 207-227:
Springer.
- Werneburg, I., Maier, W. 2019 Diverging development of akinetic skulls in cryptodire
and pleurodire turtles: an ontogenetic and phylogenetic study. *Vertebrate Zoology*. **69**, 113-
143. (<https://doi.org/10.26049/VZ69-2-2019-01>)
- Iordansky, N. N. 1989 Evolution of cranial kinesis in lower tetrapods. *Netherlands*
*Journal of Zoology*. **40**, 32-54.
- Iordansky, N. N. 1970 Structure and biomechanical analysis of functions of the jaw
muscles in the lizards. *Anatomischer Anzeiger*. **127**, 383-413.
- Olson, E. C., Broom, R. 1937 New genera and species of tetrapods from the Karroo beds
of South Africa. *Journal of Paleontology*. 613-619.
- Cost, I. N., Middleton, K. M., Sellers, K. C., Echols, M. S., Witmer, L. M., Davis, J. L.,
Holliday, C. M. A. R. 2020 Palatal biomechanics and its significance for cranial kinesis in
*Tyrannosaurus rex*. *Anatom. Rec.* **303**, 999-1017.

Formatted: German (Germany)

- Kitchin, J., Barratt, B. I., Jarvie, S., Adolph, S. C., Cree, A. J. N. Z. J. o. Z. 2017 Diet of
tuatara (*Sphenodon punctatus*) translocated to Ōrokonui Ecosanctuary in southern New
Zealand. **44**, 256-265.
- Iordansky, N. 2011 Cranial kinesis in lizards (Lacertilia): origin, biomechanics, and
evolution. *Biology Bulletin*. **38**, 868-877.
- Menaker, M., Moreira, L., Tosini, G. 1997 Evolution of circadian organization in
vertebrates. *Brazilian journal of medical and biological research*. **30**, 305-313.
- Mann, Z. F., Kelley, M. W. 2011 Development of tonotopy in the auditory periphery.
*Hearing research*. **276**, 2-15.
- Hillenius, W. J. 2000 Septomaxilla of nonmammalian synapsids: soft- tissue correlates
and a new functional interpretation. *Journal of Morphology*. **245**, 29-50.
- Senter, P. 2002 Lack of a pheromonal sense in phytosaurs and other archosaurs, and its
implications for reproductive communication. *Paleobiology*. **28**, 544-550.
- Keverne, E. B. 1999 The vomeronasal organ. *Science*. **286**, 716-720.
- Garamszegi, L. Z., Møller, A. P., Erritzøe, J. 2002 Coevolving avian eye size and brain
size in relation to prey capture and nocturnality. *Proc. R. Soc. B*. **269**, 961-967.
- Barton, R. A. 1998 Visual specialization and brain evolution in primates. *Proc. R. Soc. B*.
**265**, 1933-1937.
- Witmer, L. M., Ridgely, R. C., Dufeu, D. L., Semones, M. C. 2008 Using CT to peer
into the past: 3D visualization of the brain and ear regions of birds, crocodiles, and nonavian
dinosaurs. In *Anatomical Imaging*. (ed. eds. H. Endo, R. Frey), pp. 67-87: Springer.
- Summers, A. P. 2005 Evolution: warm-hearted crocs. *Nature*. **434**, 833.

Figures

1 FIGTREE_BASED_ON_EZCURRA

2 FIGMAPANDSTRAT

3347 FIGRECONSTRUCTION

Formatted: Font color: Accent 1

Formatted: Normal, Line spacing: single

Formatted: Font: (Default) Times New Roman

- ~~443~~FIGSAMPK5867LATERALDORSALVENTRAL
- ~~554~~FIGSAMPK5867ANTERIORPOSTERIOR
- ~~665~~FIGAMNH2239
- ~~776~~FIGGPIT16812
- ~~887~~FIGSAMPK6047ALATERALDORSALVENTRAL
- ~~998~~FIGSAMPK6047AANTERIORPOSTERIOR
- ~~10409~~ FIGSAMPK604
- ~~114110~~ FIGSAMPK6050
- ~~124244~~ FIGSAMPK7699
- ~~134342~~ FIGSAMPK13665
- ~~144443~~ FIGSAMPK13666
- ~~154544~~ FIGSAMPK13667
- ~~164645~~ FIGSAMPKK8050
- ~~1747~~FIGSAMPKK8050CLOSEUP
- ~~1848~~FIGUMZCT692BLOCK
- ~~194946~~ FIGUMZCT692POSTERIORSKULLINDIVIDUALA
- ~~47~~FIGRECONSTRUCTION
- ~~202018~~ FIGANTORBITALFENESTRA

Field Code Changed

Field Code Changed

Field Code Changed

Formatted: Font: (Default) Arial, 12 pt

Formatted: Line spacing: single

Formatted: Font: (Default) Arial, 12 pt

Formatted: Font: (Default) Arial, 12 pt

Formatted: Font: (Default) Arial, 12 pt

Formatted: Font: (Default) Times New Roman

Field Code Changed

Formatted: Font: 12 pt

Field Code Changed

- 212119 FIGPALATALFENESTRAE
- 222220 FIGBRAINCASE_RECONSTRUCTIONS
- 232324 FIGPREMAXILLA
- 242422 FIGPREMAXILLARYTEETH
- 252523 FIGVOMERCTSCAN
- 262624 FIGMAXILLA
- 272725 FIGUMZCT692MANDIBLEMAXILLAINDIVIDUALAMEDIA
- 282826 FIGSAMPK6047AMAXILLARYTOOTH
- 292927 FIGNASALS
- 303028 FIGMAXFACETOFNASAL
- 313129 FIGFRONTALS
- 323230 FIGPARIETALS
- 333334 FIGINTERPARIETAL
- 343432 FIGPREFRONTAL
- 353533 FIGPOSTFRONTAL
- 363634 FIGPOSTORBITAL
- 373735 FIGJUGAL
- 383836 FIGLACRIMAL

- ~~393937~~ FIGSQUAMOSAL
- ~~404038~~ FIGQUADRATOJUGAL
- ~~414139~~ FIGVOMER
- ~~424240~~ FIGPALATINE
- ~~434341~~ FIGECTOPTYERGOID
- ~~444442~~ FIGPTYERGOID
- ~~454543~~ FIGEPIPTYERGOID
- ~~464644~~ FIGQUADRATE
- ~~474745~~ FIGSCLERA
- ~~484846~~ FIGARTICULAR
- ~~494947~~ FIGARTICULARDRAWING
- ~~505048~~ FIGDENTARY
- ~~515149~~ FIGSPLENIAL
- ~~525250~~ FIGANGULAR
- ~~535351~~ FIGSURANGULAR
- ~~545452~~ FIGCORONOID
- ~~555553~~ FIGPREARTICULAR
- ~~565654~~ FIGIW1

1 ~~575755~~ FIGIW2

Appendix C**ROYAL SOCIETY
OPEN SCIENCE****The craniomandibular anatomy of the early archosauriform
Euparkeria capensis and the dawn of the archosaur skull**

Journal:	Royal Society Open Science
Manuscript ID	RSOS-200116
Article Type:	Research
Date Submitted by the Author:	20-Jan-2020
Complete List of Authors:	Sookias, Roland; Museum für Naturkunde - Leibniz-Institut für Evolutions- und Biodiversitätsforschung; University of Oxford, Department of Earth Sciences Dilkes, David; University of Wisconsin, Department of Biology Sobral, Gabriela; Staatliches Museum für Naturkunde Stuttgart Smith, Roger; University of the Witwatersrand, Evolutionary Studies Institute; Iziko South African Museum Wolvaardt, Fredrik P.; University of the Witwatersrand, Evolutionary Studies Institute Arcucci, Andrea; Universidad Nacional de San Luis Bhullar, Bhart-Anjan; Yale University, Geology & Geophysics Werneburg, Ingmar; Eberhard Karls Universität Tübingen, Senckenberg Center for Human Evolution and Palaeoenvironment; Eberhard Karls Universität Tübingen, Fachbereich Geowissenschaften
Subject:	palaeontology < BIOLOGY, evolution < BIOLOGY
Keywords:	Archosauria, Triassic, Euparkeria, Anatomical network analysis, Sensory and feeding evolution
Subject Category:	Earth and Environmental Science

Author-supplied statements

Relevant information will appear here if provided.

Ethics

Does your article include research that required ethical approval or permits?:

This article does not present research with ethical considerations

Statement (if applicable):

CUST_IF_YES_ETHICS :No data available.

Data

It is a condition of publication that data, code and materials supporting your paper are made publicly available. Does your paper present new data?:

Yes

Statement (if applicable):

Raw CT scan data for the cranial material of UMCZ T.692 have been made available at the Zenodo digital repository (DOI: 10.5281/zenodo.3530854), and surface renderings of SAM-PK-5867 and SAM-PK-6047A, are archived at Zenodo (DOI: 10.5281/zenodo.3530643). Other raw CT data are available from the authors on request, and will be archived following completion of further work by B.A.B. Measurements, the matrix and analytical process used for phylogenetic analysis, and matrix used for the anatomical network analysis are included with the manuscript as supplementary information. A photogrammetric model of the skull of SAM-PK-5867 is also archived at Zenodo (DOI: 10.5281/zenodo.3530643). A bank of additional photographs, which were taken by the authors and colleagues (with permission) and used for the descriptive work have been uploaded to the Zenodo repository (DOI: 10.5281/zenodo.3530643).

Conflict of interest

I/We declare we have no competing interests

Statement (if applicable):

CUST_STATE_CONFLICT :No data available.

Authors' contributions

This paper has multiple authors and our individual contributions were as below

Statement (if applicable):

[revised manuscript text omitted]
 (no-circle, $p\text{-value} \geq 0.5$, white, $p\text{-value} < 0.05$; grey, $p\text{-value} < 0.01$; black, $p\text{-value} < 0.001$).

2.4 Phylogenetic analysis

A phylogenetic analysis was carried out using the matrix of Sookias [32], including all taxa and characters and (as in Sookias [32]) using *Youngina capensis* as an outgroup, but rescaling character 90 (ectopterygoid single [0] or double [1] headed) to 1 (double headed) for *Euparkeria* based on new information (see below). The modified matrix and TNT code used to process it is given as supplementary information. A traditional search was conducted with TNT 1.5 [61] using tree bisection-reconnection (TBR) branch swapping and 1000 replicates.

2.5 Institutional abbreviations

AMNH, American Museum of Natural History, New York, USA; BPI, Evolutionary Studies Institute, University of the Witwatersrand, Johannesburg, South Africa [formerly Bernard Price Institute]; BSCUB, Biological Sciences Collection of the University of Birmingham, Birmingham, UK; GPIT, Paläontologische Sammlung, Eberhard-Karls-Universität Tübingen, Tübingen, Germany [formerly Geologisch-Paläontologisches Institut Tübingen]; NHMUK, Natural History Museum, London, UK; PULR, Paleontología, Universidad Nacional de La Rioja, La Rioja, Argentina; PVSJ, División de Paleontología de Vertebrados del Museo de Ciencias Naturales y Universidad Nacional de San Juan, San Juan, Argentina; SAM, Iziko South African Museum, Cape Town, South Africa; UMZC, University Museum of Zoology

Cambridge, Cambridge, UK; ZPAL, Institute of Paleobiology, Polish Academy of
Sciences, Warsaw, Poland.
4 **3. Results**

6 **3.1 Systematic Paleontology**

Archosauriformes Gauthier, 1988 [62].

Euparkeriidae Huene, 1920 [63].

[revised manuscript text omitted]

SAM-PK-13664 (see figures 40A, 41A, 42A,B), right pterygoid and ectopterygoid, and
left palatine exposed in ventral view.

SAM-PK-13665 (specimen 5 of Haughton [51]; figure 12), crushed but largely
complete skull in block, exposed on left side with most elements roughly in articulation.

SAM-PK-13666 (figure 13), slightly distorted, largely complete skull exposed on left
side.

SAM-PK-13667 (figure 14), crushed partial skull including left and right mandibles and
right maxilla.

SAM-PK-K8050 (figure 15), four crushed skulls exposed in dorsal view including
nasals, frontals, parietals, postfrontals, prefrontals, quadrates, squamosals, jugals and
dentaries. The block also contains a skull and postcranium of the rhynchosaur
*Mesosuchus browni* (specimen SAM-PK-K8051).

UMZC T.692 (formerly R 527; figure 16, and see figure 44B–D), cranial remains of two
individuals. Individual A includes frontals, parietals, right and left prefrontal, right
postfrontal, right and left lacrimal, right postorbital, largely complete braincase, left
pterygoid and ectopterygoid, partial right and left palatine and vomers, left and right
hyoid, left partial maxilla, left jugal, right quadrate, and anterior of left and posterior of
right mandible. Individual B includes left maxilla, anterior of left and posterior of right
mandible, part of anterior of left skull roof including nasal, posterior of left jugal,
parietals, and left quadrate.

**Specimens previously referred as cranial material.**

AMNH 19351 (supplementary figure 1), labelled as containing “fragment of jaw”, but
fragments of bone unidentifiable. Referral to either *Euparkeria* or *Mesosuchus* is

plausible based on its provenance, and the material is thus here considered

plausible based on its provenance, and the material is thus here considered
Archosauromorpha indet.

4 **Horizon and type locality (figure 2).** Most of the specimens were collected by Mr.
5 A. Brown (or Mr. A.W. Higgins – see below), prior to Watson’s examination of them
around the end of 1911 (see [50]), and the locality was described as being “in the
*Cynognathus* zone of the Karroo system” by Watson [72]. Specimens from the
collection of Brown were presented to the South African Museum after his death in
1920, prior to Haughton’s work in 1922 [51]. Following Ewer, these specimens are
labelled as “A. Brown collection, Krielfontein, Aliwal North”. Further material was also
presented directly by Higgins to the SAM in 1924 and 1925 [52], with that presented
in 1924 labelled “Higgins collection, Krielfontein, Aliwal North” and that in 1925
labelled “A. W. Higgins collection, Quarry, Commonage, Aliwal North”. Haughton [51]
reported the locality as “Krietfontein Spruit on the Aliwal North Commonage” (with
“Kriet-” spelt with “t”), and indicated that all specimens collected to date (1922) were
from one locality. Based on a communication from a certain Colonel de Wet, Ewer
[52] suggested that some or even all of the specimens labelled “A. Brown collection”
may have been physically collected by Higgins for Brown. Brown also directly
presented a specimen (now UMZC T.692) to Watson at University College London,
from where it was transferred to the University Museum of Zoology in Cambridge.
Two SAM specimens (then SAM 7708 and 7698) were presented by the SAM to the
Institut für Geologie und Paläontologie at the University of Tübingen (now
GPIT/RE/12913 - which includes cranial material; see above - and GPIT/RE/15029).

There has never been any doubt that all specimens were collected by either
Brown or Higgins from near Aliwal North (now in Walter Sisulu Municipality [73]),
Eastern Cape, South Africa. An area Ewer [52] took to be the locality was located
after direction from a local resident, at an exposure of the appropriate sediments “on
one of the commonages over the brow of a hill on the left of the road to Lady Grey
just outside Aliwal North” (p. 381). However, the recent discovery of detailed hand-
written notes in the original journals of Brown has changed and clarified our
knowledge of the locality. These notes confirmed Haughton’s [51] name for the
locality (“Krietfontein Spruit” – “spruit” meaning “spring” in Afrikaans and referring to
small stream; this may have still – as suggested by Ewer [52] - been a private
collecting name, as it is not recorded in town records or known locally) and clearly
indicate that all of the specimens came from a quarry in a stream valley on the
western side of the English graveyard in Aliwal North. This is some 5 km away from
that determined by Ewer [52], who at the time of publishing was unaware of the
contents of Brown’s journals. The journals and the fossils have been in the SAM
collections since 1921, a year after the death of Brown, but have never been fully
documented or transcribed.

Following the directions of Brown to the abandoned quarry site next to the
English graveyard, R. M. H. S. and D.W. compared the sandstone outcropping in the
cliff face with the matrix of the *Euparkeria* blocks, and also the preservation style of a

[revised manuscript text omitted]

tyrannosauroids [11]), and differs from the tapering snout of some smaller theropods
such as, for example, coelophysoids [148]. Biomechanical and ecological studies
would be needed to confirm this, but a tapering snout in some theropods (e.g.
*Coelophysis* [148]) could conceivably be connected to increased neck length and
overall importance of reach in feeding because animals such as *Coelophysis* were
pursuing prey much smaller and at a different level (whether on the ground or above)
than themselves; *Euparkeria* in contrast would presumably have taken prey at its own
level and closer in size, as did hypercarnivorous taxa.

The dentition of *Euparkeria* is clearly indicative of carnivory, with mediolaterally
compressed, recurved, serrated marginal teeth. Compared to some stem archosaurs
(e.g. *Prolacerta* [86]; proterosuchids [122]), the palatal teeth are relatively small and
do not form distinct rows. This may indicate reduced use in load-intensive activities
such as capturing and shearing prey, and an increased focus on the marginal dentition
for these activities. However, maintenance of the palatal teeth at all (contrasting with
most crown archosaurs [31] and with erythrosuchids [123]) may reflect continued

1 importance of palatal teeth in retaining smaller prey once they enter the mouth, and
2 have corresponded with retention of a relatively flexible tongue in contrast to crown
3 archosaurs [149]. Although each individual palatal tooth is smaller than in, e.g.
4 proterosuchids [122], the presence of tooth fields can be regarded as plesiomorphic,

[revised manuscript text omitted]

4 Brusatte, S. L., Nesbitt, S. J., Irmis, R. B., Butler, R. J., Benton, M. J., Norell, M. A.

2010 The origin and early radiation of dinosaurs. *Earth-Sci. Rev.* **101**, 68-100.

5 Sookias, R. B., Butler, R. J., Benson, R. B. 2012 Rise of dinosaurs reveals major

body-size transitions are driven by passive processes of trait evolution. *Proc. R. Soc.*
*B.* **279**, 2180-2187.

6 Ezcurra, M. D., Butler, R. J. 2018 The rise of the ruling reptiles and ecosystem

recovery from the Permo-Triassic mass extinction. *Proceedings of the Royal Society*
*B: Biological Sciences*. **285**, (10.1098/rspb.2018.0361)

Kaucka, M., Adameyko, I. Year Evolution and development of the cartilaginous
skull: from a lancelet towards a human face. *Seminars in Cell & Developmental*
*Biology*; 2017: Elsevier; 2017.

Couly, G. F., Coltey, P. M., Le Douarin, N. M. J. D. 1993 The triple origin of skull in
higher vertebrates: a study in quail-chick chimeras. **117**, 409-429.
Andjelković, M., Tomović, L., Ivanović, A. J. J. o. Z. 2017 Morphological integration
of the kinetic skull in *Natrix* snakes. **303**, 188-198.
Rico-Guevara, A., Rubega, M., Hurme, K., Dudley, R. J. I. O. B. 2019 Shifting
Paradigms in the Mechanics of Nectar Extraction and Hummingbird Bill Morphology.
**1**, oby006.
Hurum, J. H., Sabath, K. 2003 Giant theropod dinosaurs from Asia and North
America: skulls of *Tarbosaurus bataar* and *Tyrannosaurus rex* compared. *Acta*
*Palaeontologica Polonica*. **48**,
Nabavizadeh, A., Weishampel, D. B. J. T. A. R. 2016 The predentary bone and its
significance in the evolution of feeding mechanisms in ornithischian dinosaurs. **299**,
1358-1388.
Hopson, J. A. J. A. R. o. E., Systematics. 1977 Relative brain size and behavior in
archosaurian reptiles. **8**, 429-448.
14 Bhullar, B.-A. S., Marugán-Lobón, J., Racimo, F., Bever, G. S., Rowe, T. B.,
Norell, M. A., Abzhanov, A. J. N. 2012 Birds have paedomorphic dinosaur skulls.
**487**, 223.
15 Lee, M. S. Y., Cau, A., Naish, D., Dyke, G. J. 2014 Sustained miniaturization and
anatomical innovation in the dinosaurian ancestors of birds. *Science*. **345**, 562-566.
(10.1126/science.1252243)
16 Hurlburt, G. R., Ridgely, R. C., Witmer, L. M. J. T. p. 2013 Relative size of brain
and cerebrum in tyrannosaurid dinosaurs: an analysis using brain-endocast
quantitative relationships in extant alligators. 1-21.

17 Bhullar, B.-A. S., Hanson, M., Fabbri, M., Pritchard, A., Bever, G. S., Hoffman, E.
2016 How to Make a Bird Skull: Major Transitions in the Evolution of the Avian
Cranium, Paedomorphosis, and the Beak as a Surrogate Hand. *Integrative and*
*Comparative Biology*. **56**, 389-403. (10.1093/icb/icw069)
Broom, R. 1913 On the South-African pseudosuchian *Euparkeria* and allied
genera. *Proceedings of the Zoological Society of London*. **83**, 619-633.
von Baczko, M. B., Ezcurra, M. D. 2016 Taxonomy of the archosaur
*Ornithosuchus*: reassessing *Ornithosuchus woodwardi* Newton, 1894 and
*Dasygnathoides longidens* (Huxley 1877). *Earth Environ. Sci. Trans. R. Soc.*
*Edinburgh*. **106**, 199-205.
Benton, M. J., Clark, J. M. 1988 Archosaur phylogeny and the relationships of the
Crocodylia. *The phylogeny and classification of the tetrapods*. **1**, 295-338.
Sereno, P. C., Arcucci, A. 1990 The monophyly of crurotarsal archosaurs and the
origin of bird and crocodile ankle joints. *N. Jb. Geol. Paläontol. Abh.* **180**, 21-52.
Sereno, P. C. 1991 Basal archosaurs: phylogenetic relationships and functional
implications. *J. Vert. Paleontol.* **11**, 1-53.
Parrish, J. M. 1993 Phylogeny of the Crocodylotarsi, with reference to
archosaurian and crurotarsan monophyly. *J. Vert. Paleontol.* **13**, 287-308.
Juul, L. 1994 The phylogeny of basal archosaurs. *Palaeontol. Afr.* **31**, 1–38.
Bennett, C. S. 1996 The phylogenetic position of the Pterosauria within the
Archosauromorpha. *Zool. J. Linn. Soc.* **118**, 261-308.
Benton, M. J. 1999 *Scleromochlus taylori* and the origin of dinosaurs and
pterosaurs. *Phil. Trans. R Soc. B.* **354**, 1423-1446.

Parker, W. G., Barton, B. J. 2008 New information on the Upper Triassic
archosauriform *Vancleavea campi* based on new material from the Chinle Formation
of Arizona. *Palaeontol. Electron.* **11**, 14A.
Nesbitt, S. J., Stocker, M. R., Small, B. J., Downs, A. 2009 The osteology and
relationships of *Vancleavea campi* (Reptilia: Archosauriformes). *Zool. J. Linn. Soc.*
**157**, 814-864.
Brusatte, S. L., Benton, M. J., Desojo, J. B., Langer, M. C. 2010 The higher-level
phylogeny of Archosauria (Tetrapoda: Diapsida). *J. Syst. Palaeontol.* **8**, 3-47.
Ezcurra, M. D., Lecuona, A., Martinelli, A. 2010 A new basal archosauriform
diapsid from the Lower Triassic of Argentina. *J. Vert. Paleontol.* **30**, 1433-1450.
Nesbitt, S. J. 2011 The early evolution of archosaurs: relationships and the origin
of major clades. *Bull. Am. Mus. Nat. Hist.*, 1-292.
Sookias, R. B. 2016 The relationships of the Euparkeriidae and the rise of
Archosauria. *R. Soc. Open Sci.* **3**, 150674.
Sookias, R. B., Sennikov, A. G., Gower, D. J., Butler, R. J. 2014 The monophyly
of Euparkeriidae (Reptilia: Archosauriformes) and the origins of Archosauria: a
revision of *Dorosuchus neoetus* from the Mid-Triassic of Russia. *Palaeontology.* **57**,
1177-1202.
Sookias, R. B., Sullivan, C., Liu, J., Butler, R. J. 2014 Systematics of putative
euparkeriids (Diapsida: Archosauriformes) from the Triassic of China. *PeerJ.* **2**,
e658.
Romer, A. S. 1972 The Chañares (Argentina) Triassic reptile fauna. XVI.
Thecodont classification. *Breviora.* **395**, 1-24.

1 36 Gower, D. J., Weber, E. 1998 The braincase of Euparkeria, and the evolutionary
relationships of birds and crocodylians. *Biological reviews*. **73**, 367-411.
2
Perry, S. F. 1992 Gas exchange strategies in reptiles and the origin of the avian
lung. *Physiological Adaptations in Vertebrates: respiration, circulation, and*
lung. *Physiological Adaptations in Vertebrates: respiration, circulation, and*
*metabolism*. **56**, 149-167.
Carrier, D. R., Farmer, C. G. 2000 The evolution of pelvic aspiration in
archosaurs. *Paleobiology*. **26**, 271-293.
Hutchinson, J. R. 2001 The evolution of pelvic osteology and soft tissues on the
line to extant birds (Neornithes). *Zool. J. Linn. Soc.* **131**, 123-168.
Marugán-Lobón, J., Buscalioni, A. D. 2003 Disparity and geometry of the skull in
Archosauria (Reptilia: Diapsida). *Biological journal of the Linnean Society*. **80**, 67-88.
Nesbitt, S. J. 2003 Arizonasaurus and its implications for archosaur divergence.
*Proc. R. Soc. B*. **270**, S234-S237.
Rauhut, O. W. 2003 *Special Papers in Palaeontology, The Interrelationships and*
*Evolution of Basal Theropod Dinosaurs*. Blackwell Publishing.
Seymour, R. S., Bennett-Stamper, C. L., Johnston, S. D., Carrier, D. R., Grigg, G.
C. 2004 Evidence for endothermic ancestors of crocodiles at the stem of archosaur
evolution. *Physiological and Biochemical Zoology*. **77**, 1051-1067.
de Ricqlès, A., Padian, K., Knoll, F., Horner, J. R. 2008 On the origin of high
growth rates in archosaurs and their ancient relatives: complementary histological
studies on Triassic archosauriforms and the problem of a “phylogenetic signal” in
bone histology. *Annales de Paléontologie*. **94**, 57-76.
Sullivan, C. 2010 The role of the calcaneal ‘heel’ as a propulsive lever in basal
archosaurs and extant monitor lizards. *J. Vert. Paleontol.* **30**, 1422-1432.

Maidment, S. C., Barrett, P. M. 2011 The locomotor musculature of basal
ornithischian dinosaurs. *J. Vert. Paleontol.* **31**, 1265-1291.
Butler, R. J., Barrett, P. M., Gower, D. J. 2012 Reassessment of the evidence for
postcranial skeletal pneumaticity in Triassic archosaurs, and the early evolution of
the avian respiratory system. *PLoS ONE.* **7**, e34094.
Foth, C., Rauhut, O. W. 2013 Macroevolutionary and morphofunctional patterns
in theropod skulls: a morphometric approach. *Acta Palaeontologica Polonica.* **58**, 1-
16.
Baron, M. G., Norman, D. B., Barrett, P. M. 2017 A new hypothesis of dinosaur
relationships and early dinosaur evolution. *Nature.* **543**, 501-506.
Broom, R. 1913 Note on *Mesosuchus browni*, Watson, and on a new South
African Triassic pseudosuchian (*Euparkeria capensis*). *Records of the Albany*
*Museum.* **2**, 394-396.
Haughton, S. 1922 On the reptilian genera *Euparkeria* Broom, and *Mesosuchus*
Watson. *Transactions of the Royal Society of South Africa.* **10**, 81-88.
Ewer, R. F. 1965 The anatomy of the thecodont reptile *Euparkeria capensis*
Broom. *Phil. Trans. R. Soc. Lond. B.* **248**, 379-435.
Sobral, G., Sookias, R. B., Bhullar, B.-A. S., Smith, R., Butler, R. J., Müller, J.
2016 New information on the braincase and inner ear of *Euparkeria capensis* Broom:
implications for diapsid and archosaur evolution. *R. Soc. Open Sci.* **3**, 160072.
Hancox, P. J. 2000 The continental Triassic of South Africa. *Zentralblatt für*
*Geologie und Paläontologie Teil I.* **1998**, 1285-1324.

Hancox, P., Shishkin, M., Rubidge, B., Kitching, J. 1995 A threefold subdivision of
the *Cynognathus* Assemblage Zone (Beaufort Group, South-Africa) and its
paleogeographic implications. *South African Journal of Science*. **91**, 143-144.
Ottone, E. G., Monti, M., Marsicano, C. A., Marcelo, S., Naipauer, M., Armstrong,
R., Mancuso, A. C. 2014 A new Late Triassic age for the Puesto Viejo Group (San
Rafael depocenter, Argentina): SHRIMP U–Pb zircon dating and biostratigraphic
correlations across southern Gondwana. *Journal of South American Earth Sciences*.
**56**, 186-199.
Schneider, J. W., Lucas, S. G., Scholze, F., Voigt, S., Marchetti, L., Klein, H.,
Opluštil, S., Werneburg, R., Golubev, V. K., Barrick, J. E. 2019 Late Paleozoic–early
Mesozoic continental biostratigraphy—links to the Standard Global
Chronostratigraphic Scale. *Palaeoworld*.
Werneburg, I., Esteve-Altava, B., Bruno, J., Ladeira, M. T., Diogo, R. 2019
Unique skull network complexity of *Tyrannosaurus rex* among land vertebrates. *Sci.*
*Rep.* **9**, 1520.
R-Core-Team. R: A language and environment for statistical computing. Vienna,
Austria: R Foundation for Statistical Computing. 2014.
Csardi, G., Nepusz, T. 2006 The igraph software package for complex network
research. *InterJournal, Complex Systems*. **1695**, 1-9.
Goloboff, P. A., Catalano, S. A. 2016 TNT version 1.5, including a full
implementation of phylogenetic morphometrics. *Cladistics*. **32**, 221-238.
Gauthier, J., Kluge, A. G., Rowe, T. 1988 Amniote phylogeny and the importance
of fossils. *Cladistics*. **4**, 105-209.

Huene, F. v. von 1920. Osteologie von *Aetosaurus ferratus* O. Fraas. *Acta*
*Zoologica*. **3**, 465-491.
Laurenti, J.-N. 1768 *Specimen medicum, exhibens synopin reptilium emendatam*
*cum experimentis circa venena et antidota reptilium Austriacorum*. Trattner.
von Linné, C. 1767 *Systema naturae per regna tria naturae: secundum classes,*
*ordines, genera, species, cum characteribus, differentiis, synonymis, locis*. Typis
Ioannis Thomae.
Weinbaum, J. C. 2011 The skull of *Postosuchus kirkpatricki* (Archosauria:
Paracrocodyliformes) from the Upper Triassic of the United States. *PaleoBios*. **30**,
Borsuk-Białynicka, M., Evans, S. 2009 Cranial and mandibular osteology of the
10 early triassic archosauriform *Osmolskina czatkowicensis* from Poland.
*Palaeontologia Polonica*. **65**, 235-281.
Gower, D. J. 1999 The cranial and mandibular osteology of a new rauisuchian
archosaur from the Middle Triassic of southern Germany. *Stuttgarter Beiträge zur*
*Naturkunde: Serie B (Geologie und Paläontologie)*. **280**, 1-50.
Walker, A. D. 1964 Triassic reptiles from the Elgin area: *Ornithosuchus* and the
origin of carnosaurs. *Phil. Trans. R. Soc. Lond. B*. **248**, 53-134.
Sereno, P. C., Novas, F. E. 1994 The skull and neck of the basal theropod
*Herrerasaurus ischigualastensis*. *J. Vert. Paleontol.* **13**, 451-476.
Sereno, P. C., Martínez, R. N., Alcober, O. A. 2012 Osteology of *Eoraptor*
*lunensis* (Dinosauria, Sauropodomorpha). *J. Vert. Paleontol.* **32**, 83-179.
Watson, D. 1912 *Mesosuchus browni*, gen. et spec. nov. *Records of the Albany*
*Museum*. **2**, 298-299.

Walter Sisulu Local Municipality (EC145). 2019 [cited 14/12/2019]; Available
from: <https://municipalities.co.za/overview/1235/walter-sisulu-local-municipality>
Hancox, P. J. 1998 A stratigraphic, sedimentological and palaeoenvironmental
synthesis of the Beaufort-Molteno contact in the Karoo Basin. Johannesburg:
University of the Witwatersrand.
Neveling, J. 2004 Stratigraphic and sedimentological investigation of the contact
between the *Lystrosaurus* and the *Cynognathus* assemblage zones (Beaufort group:
Karoo supergroup). *Council for Geoscience Bulletin*. **137**, 1-165.
Cruickshank, A. 1972 The proterosuchian thecodonts. *Studies in vertebrate*
*evolution*. **89**, 19.
Romer, A. S. 1971 The Chañares (Argentina) Triassic reptile fauna. XI. Two new
long-snouted thecodonts, *Chanaresuchus* and *Gualosuchus*. *Breviora*. **379**, 1-22.
Gower, D. J. 2003 Osteology of the early archosaurian reptile *Erythrosuchus*
*africanus* Broom. *Annals of the South African Museum*. **110**, 1-88.
Walker, A. D. 1990 A revision of *Sphenosuchus acutus* Haughton, a
crocodylomorph reptile from the Elliot Formation (late Triassic or early Jurassic) of
South Africa. *Phil. Trans. R. Soc. Lond. B*. **330**, 1-120.
Witmer, L. M. 1997 The evolution of the antorbital cavity of archosaurs: a study in
soft-tissue reconstruction in the fossil record with an analysis of the function of
pneumaticity. *J. Vert. Paleontol.* **17**, 1-76.
Sobral, G., Müller, J. 2019 The braincase of *Mesosuchus browni* (Reptilia,
Archosauromorpha) with information on the inner ear and description of a pneumatic
sinus. *PeerJ*. **7**, e6798.

Gower, D. J., Sennikov, A. 1996 Morphology and phylogenetic informativeness of
early archosaur braincases. *Palaeontology*. **39**, 883-906.
Evans, S. 1986 The braincase of *Prolacerta broomi* (Reptilia: Triassic). *Neues*
*Jahrbuch für Geologie und Paläontologie: Abhandlungen*. **173**, 181-200.
Heaton, M. J. 1979 Cranial anatomy of primitive captorhinid reptiles from the Late
Pennsylvanian and Early Permian, Oklahoma and Texas. *Oklahoma Geological*
*Survey Bulletin*. **127**, 1-81.
Gardner, N. M., Holliday, C. M., O'Keefe, F. R. 2010 The braincase of *Youngina*
*capensis* (Reptilia, Diapsida): new insights from high-resolution CT scanning of the
holotype.
Modesto, S. P., Sues, H.-D. 2004 The skull of the Early Triassic
archosauromorph reptile *Prolacerta broomi* and its phylogenetic significance. *Zool. J.*
*Linn. Soc.* **140**, 335-351.
Lautenschlager, S., Rauhut, O. W. 2015 Osteology of *Rauisuchus tiradentes* from
the Late Triassic (Carnian) Santa Maria Formation of Brazil, and its implications for
rauisuchid anatomy and phylogeny. *Zool. J. Linn. Soc.* **173**, 55-91.
Ochev, V. G. 1958 Novyye dannye po psevozukkhiyam SSSR. *Doklady*
*Akademi Nauk.* **123**, 749-751.
Romer, A. S. 1972 The Chanares (Argentina) Triassic reptile fauna. An early
ornithosuchid pseudosuchian, *Gracilisuchus stipanicorum*, gen. et sp. nov.
*Breviora*. **389**, 1-24.
Wu, X. 1981 The discovery of a new thecodont from north-east Shensi. *Vertebr.*
*Palasiatica*. **19**, 122-132.

Nesbitt, S. J. 2005 Osteology of the Middle Triassic pseudosuchian archosaur
*Arizonasaurus babbitti*. *Historical Biology*. **17**, 19-47.
Walker, A. D. 1961 Triassic reptiles from the Elgin area: *Stagonolepis*,
*Dasygnathus* and their allies. *Phil. Trans. R. Soc. Lond. B*. **244**, 103-204.
Nesbitt, S. 2007 The anatomy of *Effigia okeeffeae* (Archosauria, Suchia),
theropod-like convergence, and the distribution of related taxa. *Bull. Am. Mus. Nat.*
*Hist.*, 1-84.
Senter, P. 2003 New information on cranial and dental features of the Triassic
archosauriform reptile *Euparkeria capensis*. *Palaeontology*. **46**, 613-621.
Brusatte, S. L., Butler, R. J., Sulej, T., Niedźwiedzki, G. 2009 The taxonomy and
anatomy of rauisuchian archosaurs from the Late Triassic of Germany and Poland.
*Acta Palaeontologica Polonica*. **54**, 221-230.
Bonaparte, J. F. 1971 Los tetrápodos del sector superior de la formación Los
Colorados, La Rioja, Argentina (Triásico Superior): I Parte.
von Baczko, M. B., Desojo, J. B. J. P. o. 2016 Cranial anatomy and
palaeoneurology of the archosaur *Riojasuchus tenuisiceps* from the Los Colorados
Formation, La Rioja, Argentina. **11**, e0148575.
Yates, A. M. 2003 A new species of the primitive dinosaur *Thecodontosaurus*
(Saurischia: Sauropodomorpha) and its implications for the systematics of early
dinosaurs. *J. Syst. Palaeontol.* **1**, 1-42.
Langer, M. C., Benton, M. J. 2006 Early dinosaurs: a phylogenetic study. *J. Syst.*
*Palaeontol.* **4**, 309-358.
Arcucci, A. 2011 Sistemática y filogenia de los proterochampsidos (Amniota,
Diápsida, Archosauriformes) del Triásico de América del Sur, y sus implicancias en

el origen de Archosauria. *Unpublished PhD thesis, Universidad Nacional de San*
*Luis.*
Dilkes, D. W. 1995 The rhynchosaur *Howesia browni* from the Lower Triassic of
South Africa. *Palaeontology*. **38**, 665-686.
Clark, J. M., Sues, H.-D. 2002 Two new basal crocodylomorph archosaurs from
the Lower Jurassic and the monophyly of the Sphenosuchia. *Zool. J. Linn. Soc.* **136**,
77-95.
Clark, J. M., Xu, X., Forster, C. A., Wang, Y. 2004 A Middle Jurassic
'sphenosuchian' from China and the origin of the crocodylian skull. *Nature*. **430**,
1021.
Dilkes, D., Arcucci, A. 2012 *Proterochampsia barrionuevoi* (Archosauriformes:
Proterochampsia) from the Late Triassic (Carnian) of Argentina and a phylogenetic
analysis of Proterochampsia. *Palaeontology*. **55**, 853-885.
Dilkes, D. W. 1998 The Early Triassic rhynchosaur *Mesosuchus browni* and the
interrelationships of basal archosauromorph reptiles. *Phil. Trans. R Soc. B.* **353**, 501-
541.
Parrington, F. 1937 V.—A note on the supratemporal and tabular bones in
reptiles. *Journal of Natural History*. **20**, 69-76.
Gower, D. J., Walker, A. D. 2002 New data on the braincase of the aetosaurian
archosaur (Reptilia: Diapsida) *Stagonolepis robertsoni* Agassiz. *Zool. J. Linn. Soc.*
**136**, 7-23.
Ochev, V. 1981 On Erythrosuchus (*Garjainia*) primus Ochev. *Voprosy Geologii*
*Yuzhnogo Urala I Povolzh'ya*. **22**, 3-22.

Wang, R., Xu, S., Wu, X., Li, C., Wang, S. 2013 A new specimen of
*Shansisuchus shansisuchus* Young, 1964 (Diapsida: Archosauriformes) from the
Triassic of Shanxi, China. *Acta Geologica Sinica-English Edition*. **87**, 1185-1197.
Young, C.-C. 1964 The pseudosuchians in China. *Palaeontologia Sinica*. 1-205.
Rauhut, O. W. 1997 On the cranial anatomy of *Shuvosaurus inexpectatus*
(Dinosauria; Theropoda). *Terra Nostra*. **7**, 1-4.
Gower, D. J., Hancox, P. J., Botha-Brink, J., Sennikov, A. G., Butler, R. J. 2014
A new species of *Garjainia* Ochev, 1958 (Diapsida: Archosauriformes:
Erythrosuchidae) from the Early Triassic of South Africa. *PLoS ONE*. **9**, e111154.
Müller, J. J. N. 2003 Early loss and multiple return of the lower temporal arcade
in diapsid reptiles. **90**, 473-476.
Bever, G., Lyson, T. R., Field, D. J., Bhullar, B.-A. S. J. N. 2015 Evolutionary
origin of the turtle skull. **525**, 239.
Heckert, A. B., Lucas, S. G. 1999 A new aetosaur (Reptilia: Archosauria) from
the Upper Triassic of Texas and the phylogeny of aetosaurs. *J. Vert. Paleontol.* **19**,
50-68.
Parker, W. G. 2007 Reassessment of the aetosaur '*Desmotosuchus*'
*chamaensis* with a reanalysis of the phylogeny of the Aetosauria (Archosauria:
Pseudosuchia). *J. Syst. Palaeontol.* **5**, 41-68.
Sulej, T. 2005 A new rauisuchian reptile (Diapsida: Archosauria) from the Late
Triassic of Poland. *J. Vert. Paleontol.* **25**, 78-86.
Ochev, V. 1958 New data on the Triassic vertebrate fauna from the Orenburg
Cis-Urals. *Doklady Akademii Nauk SSSR*. **22**, 749-752.

Dzik, J., Sulej, T. 2007 A review of the early Late Triassic Krasiejów biota from
Silesia, Poland. *Phytopatologia Polonica*. 3-27.
Gow, C. E. 1970 The anterior of the palate in *Euparkeria*. *Paleaeontologia*
*Africana*. **13**, 61-62.
Camp, C. L. 1945 Prolacerta and the protorosaurian reptiles; Part I. *American*
*Journal of Science*. **243**, 17-32.
Welman, J. 1998 The taxonomy of the South African proterosuchids (Reptilia,
Archosauromorpha). *J. Vert. Paleontol.* **18**, 340-347.
Ochev, V. 1975 On the proterosuchian palate. *Paleontologicheskii Zhurnal*.
**1980**, 98-105.
Dollman, K. N., Clark, J. M., Norell, M. A., Xing, X., Choiniere, J. N. J. A. M. N.
2018 Convergent evolution of a eusuchian-type secondary palate within
Shartegosuchidae. **2018**, 1-24.
Butler, R. J., Sullivan, C., Ezcurra, M. D., Liu, J., Lecuona, A., Sookias, R. B.
2014 New clade of enigmatic early archosaurs yields insights into early
pseudosuchian phylogeny and the biogeography of the archosaur radiation. *BMC*
*Evol. Biol.* **14**, 128.
Brusatte, S. L., Carr, T. D., Norell, M. A. 2012 The osteology of *Alioramus*, a
gracile and long-snouted tyrannosaurid (Dinosauria: Theropoda) from the Late
Cretaceous of Mongolia. *Bull. Am. Mus. Nat. Hist.*, 1-197.
Dilkes, D., Sues, H.-D. 2009 Redescription and phylogenetic relationships of
*Doswellia kaltenbachi* (Diapsida: Archosauriformes) from the Upper Triassic of
Virginia. *J. Vert. Paleontol.* **29**, 58-79.

Benton, M. J., Walker, A. D. 2002 *Erpetosuchus*, a crocodile-like basal
archosaur from the Late Triassic of Elgin, Scotland. *Zool. J. Linn. Soc.* **136**, 25-47.
Nesbitt, S. J., Norell, M. A. 2006 Extreme convergence in the body plans of an
early suchian (Archosauria) and ornithomimid dinosaurs (Theropoda). *Proc. R. Soc.*
*B.* **273**, 1045-1048.
Schmitz, L., Motani, R. 2011 Nocturnality in dinosaurs inferred from scleral ring
and orbit morphology. *Science.* **332**, 705-708.
Mastrantonio, B., Von Baczko, M. B., Desojo, J. B., Schultz, C. L. 2019 The skull
anatomy and cranial endocast of the pseudosuchid archosaur *Prestosuchus*
*chiquensis* from the Triassic of Brazil. **64**, 171-198.
Rasskin-Gutman, D., Esteve-Altava, B. J. B. T. 2014 Connecting the dots:
anatomical network analysis in morphological EvoDevo. **9**, 178-193.
Esteve-Altava, B., Marugán-Lobón, J., Botella, H., Rasskin-Gutman, D. 2013
Structural constraints in the evolution of the tetrapod skull complexity: Williston's law
revisited using network models. *Evolutionary Biology.* **40**, 209-219.
Ezcurra, M. D. 2016 The phylogenetic relationships of basal archosauromorphs,
with an emphasis on the systematics of proterosuchian archosauriforms. *PeerJ.* **4**,
e1778.
Jaquier, V. P., Fraser, N. C., Furrer, H., Scheyer, T. M. 2017 Osteology of a new
specimen of *Macrocnemus* aff. *M. fuyuanensis* (Archosauromorpha, Protorosauria)
from the Middle Triassic of Europe: potential implications for species recognition and
paleogeography of tanystropheid protorosaurs. *Frontiers in Earth Science.* **5**, 91.
Koyabu, D., Werneburg, I., Morimoto, N., Zollikofer, C. P., Forasiepi, A. M.,
Endo, H., Kimura, J., Ohdachi, S. D., Son, N. T., Sánchez-Villagra, M. R. 2014

Mammalian skull heterochrony reveals modular evolution and a link between cranial
development and brain size. *Nature Comms.* **5**, 3625.
Spiekman, S., Scheyer, T. 2019 A taxonomic revision of the genus
*Tanystropheus* (Archosauromorpha, Tanystropheidae). *Palaeontol. Electron.*, Epub
ahead of print.
Gregory, W. K. 1935 'Williston's law' relating to the evolution of skull bones in
the vertebrates. *American Journal of Physical Anthropology.* **20**, 123-152.
Sidor, C. A. 2001 Simplification as a trend in synapsid cranial evolution.
*Evolution.* **55**, 1419-1442.
Holliday, C. M., Witmer, L. M. 2008 Cranial kinesis in dinosaurs: intracranial
joints, protractor muscles, and their significance for cranial evolution and function in
diapsids. *J. Vert. Paleontol.* **28**, 1073-1088.
Bock, W. J. 1964 Kinetics of the avian skull. *Journal of Morphology.* **114**, 1-41.
Herrel, A., Schaerlaeken, V., Meyers, J. J., Metzger, K. A., Ross, C. F. 2007 The
evolution of cranial design and performance in squamates: Consequences of skull-
bone reduction on feeding behavior. *Integrative and Comparative Biology.* **47**, 107-
117. (10.1093/icb/icm014)
Schoch, R., Sues, H. 2013 A new doswelliid archosauriform from the Middle
Triassic of Germany. *J Syst Palaeontol* [doi: 10.1080/14772019.2013. 781066].
Gauthier, J. 1986 Saurischian monophyly and the origin of birds. *Memoirs of the*
*California Academy of sciences.* **8**, 1-55.
Botha-Brink, J., Smith, R. M. 2011 Osteohistology of the Triassic
archosauromorphs *Prolacerta*, *Proterosuchus*, *Euparkeria*, and *Erythrosuchus* from
the Karoo Basin of South Africa. *J. Vert. Paleontol.* **31**, 1238-1254.

Ezcurra, M. D., Butler, R. J. 2015 Taxonomy of the proterosuchid
archosauriforms (Diapsida: Archosauromorpha) from the earliest Triassic of South
Africa, and implications for the early archosauriform radiation. *Palaeontology*. **58**,
141-170.
Chatterjee, S. 1978 A primitive parasuchid (phytosaur) reptile from the Upper
Triassic Maleri Formation of India. *Palaeontology*. **21**, 83-127.
Colbert, E. H. 1989 The Triassic dinosaur *Coelophysis*. *Museum of Northern*
*Arizona Bulletin*. **53**, 1-160.
Matsumoto, R., Evans, S. E. 2017 The palatal dentition of tetrapods and its
functional significance. *Journal of anatomy*. **230**, 47-65.
Witmer, L., Ridgely, R., Dufeu, D., Semones, M. Anatomical imaging: towards
a new morphology. Tokyo: Springer-Verlag 2008.
Walsh, S. A., Iwaniuk, A. N., Knoll, M. A., Bourdon, E., Barrett, P. M., Milner, A.
C., Nudds, R. L., Abel, R. L., Sterpaio, P. D. 2013 Avian cerebellar floccular fossa
size is not a proxy for flying ability in birds. *PLoS ONE*. **8**, e67176.
Bronzati, M., Benson, R. B., Rauhut, O. W. 2017 Rapid transformation in the
braincase of sauropod dinosaurs: integrated evolution of the braincase and neck in
early sauropods? *Palaeontology*. **61**, 289-302.
Dudley, R., Yanoviak, S. P. 2011 Animal aloft: the origins of aerial behavior and
flight. *Integrative and comparative biology*. **51**, 926-936.
Vasilopoulou-Kampitsi, M., Goyens, J., Baeckens, S., Van Damme, R., Aerts, P.
2019 Habitat use and vestibular system's dimensions in lacertid lizards. *Journal of*
*anatomy*. **235**, 1-14.

Vasilopoulou-Kampitsi, M., Goyens, J., Van Damme, R., Aerts, P. 2019 The
ecological signal on the shape of the lacertid vestibular system: simple versus
complex microhabitats. *Biological Journal of the Linnean Society*. **127**, 260-277.
Versluys, J. 1912 Das Streptostylie Problem und die Bewegungen im Schädel
bei Sauropsiden. *Zool. Jb. Suppl.* **15**, 545-714.
Evans, S. E. 2008 The skull of lizards and tuatara. *Biology of the Reptilia*. **20**, 1-
347.
Werneburg, I. 2019 Morphofunctional categories and ontogenetic origin of
temporal skull openings in amniotes. *Frontiers in Earth Science*. **7**,
(10.3389/feart.2019.00013)
Ferreira, G. S., Werneburg, I. 2019 Evolution, Diversity, and Development of the
Craniocervical System in Turtles with Special Reference to Jaw Musculature. In
*Heads, Jaws, and Muscles*. (ed.^eds. pp. 171-206: Springer.
Diaz, R. E., Trainor, P. A. 2019 An Integrative View of Lepidosaur Cranial
Anatomy, Development, and Diversification. In *Heads, Jaws, and Muscles*. (ed.^eds.
pp. 207-227: Springer.
Werneburg, I., Maier, W. 2019 Diverging development of akinetic skulls in
cryptodire and pleurodire turtles: an ontogenetic and phylogenetic study. *Vertebrate*
*Zoology*. **69**, 113-143. (<https://doi.org/10.26049/VZ69-2-2019-01>)
lordansky, N. N. 1989 Evolution of cranial kinesis in lower tetrapods.
*Netherlands Journal of Zoology*. **40**, 32-54.
lordansky, N. N. 1970 Structure and biomechanical analysis of functions of the
jaw muscles in the lizards. *Anatomischer Anzeiger*. **127**, 383-413.

Olson, E. C., Broom, R. 1937 New genera and species of tetrapods from the
Karroo beds of South Africa. *Journal of Paleontology*. 613-619.
Gow, C. E. 1975 The morphology and relationships of *Youngina capensis*
Broom and *Prolacerta broomi* Parrington. *Palaeontol. Afr.* **18**, 89-131.
Kitchin, J., Barratt, B. I., Jarvie, S., Adolph, S. C., Cree, A. J. N. Z. J. o. Z. 2017
Diet of tuatara (*Sphenodon punctatus*) translocated to Ōrokonui Ecosanctuary in
southern New Zealand. **44**, 256-265.
Iordansky, N. 2011 Cranial kinesis in lizards (Lacertilia): origin, biomechanics,
and evolution. *Biology Bulletin*. **38**, 868-877.
Menaker, M., Moreira, L., Tosini, G. 1997 Evolution of circadian organization in
vertebrates. *Brazilian journal of medical and biological research*. **30**, 305-313.
Mann, Z. F., Kelley, M. W. 2011 Development of tonotopy in the auditory
periphery. *Hearing research*. **276**, 2-15.
Hillenius, W. J. 2000 Septomaxilla of nonmammalian synapsids: soft-tissue
correlates and a new functional interpretation. *Journal of Morphology*. **245**, 29-50.
Senter, P. 2002 Lack of a pheromonal sense in phytosaurs and other
archosaurs, and its implications for reproductive communication. *Paleobiology*. **28**,
544-550.
Keverne, E. B. 1999 The vomeronasal organ. *Science*. **286**, 716-720.
Garamszegi, L. Z., Møller, A. P., Erritzøe, J. 2002 Coevolving avian eye size
and brain size in relation to prey capture and nocturnality. *Proc. R. Soc. B*. **269**, 961-
967.
Barton, R. A. 1998 Visual specialization and brain evolution in primates. *Proc. R.*
*Soc. B*. **265**, 1933-1937.

Witmer, L. M., Ridgely, R. C., Dufeu, D. L., Semones, M. C. 2008 Using CT to
peer into the past: 3D visualization of the brain and ear regions of birds, crocodiles,
and nonavian dinosaurs. In *Anatomical Imaging*. (ed. ^eds. H. Endo, R. Frey), pp. 67-
87: Springer.
Summers, A. P. 2005 Evolution: warm-hearted crocs. *Nature*. **434**, 833.

**Figure captions (formatted correctly – thus differ from those entered under figures)**

Figure 1. Schematic phylogeny based on that of Ezcurra (2016) and Sookias (2016), showing position
of *Euparkeria capensis* (blue text, yellow background), other euparkeriids (blue text), taxa previously
considered euparkeriids (pink text), and other key early diapsid taxa. Although Proterochampsidae
was found to be the sister taxon to Archosauria by Ezcurra, many analyses have placed *Euparkeria*
as the sister taxon to crown Archosauria, including the the analysis of Nesbitt (2011) upon which the
analyses of Sookias et al. (2014a,b) and Sookias (2016) were based.

Figure 2. Maps showing (A) location of Aliwal North within South Africa, and (B) the approximate
position of the locality thought to have yielded all *Euparkeria* fossils (a stone quarry adjacent to a
small stream - "Krietfontein Spruit") – this locaton is based on the newly rediscovered notes of Mr.
Alfred Brown, and differs from the position identified by Ewer (1965) which was described as being on
the way out of Aliwal North in the direction of Lady Grey; (C) schematic stratigraphic column showing
current understanding of the correlation of *Cynognathus* Assemblage Zone Subzone B, within which
the locality is placed.

Figure 3. Holotype skull SAM-PK-5867 in (A) right lateral, (B) left lateral, (C) dorsal, and (D) ventral
view. ?, unidentified element or uncertainty in identification when preceding element; a, angular; ar,
articular; bo, basioccipital; bpt, basipterygoid process; c, coronoid; ch, choana; d, dentary; eaof,
external antorbital fenestra; ect, ectopterygoid; eo, exoccipital; f(r/l), frontal (right/left); fm, foramen
magnum; hy, hyoid; iaof, internal antorbital fenestra; ica, internal carotid artery foramen; ip,
interparietal; j, jugal; l, lacrimal; lt, laterosphenoid; ltf, lateral temporal fenestra; meck.for, Meckelian
foramen; mf, mandibular fenestra; mx, maxilla; n, nasal; nar, external naris; or, orbit; ost, osteoderms;
p, parietal; pa, prearticular; pof, postfrontal; ppo, postorbital; pp, paroccipital process; pl, palatine;
pmx, premaxilla; pt(r/l), pterygoid (right/left); ptf, posttemporal fenestra; q, quadrate; qf, quadrate
foramen; qj, quadratojugal; rib, rib; sa, surangular; scl, sclera; so, supraoccipital; spl, splenial; stf,
supratemporal fenestra; t, tooth; tu, tuber; v, vomer; vrop, ventral ramus of opsithotic. White=bone;
grey=matrix. Same scale applies to A-D.

Figure 4. Holotype skull SAM-PK-5867 in (A) posterior and (B) anterior view with interpretive
drawings. ?, unidentified element; a, angular; ar, articular; bo, basioccipital; bpt, basipterygoid
process; btbo, basal tuber of the basioccipital; d, dentary; eaof, external antorbital fenestra; eo,
exoccipital; f, frontal; fm, foramen magnum; hy, hyoid; ip, interparietal; j, jugal; l, lacrimal; mx, maxilla;
n, nasal; nar, naris; or, orbit; ost, osteoderms; p, parietal; pa, prearticular; po, postorbital; pof,
postfrontal; pmx, premaxilla; ppo, postorbital; pp, paroccipital process; preart, prearticular; pt,
pterygoid; ptf, posttemporal fenestra; q, quadrate; qf, quadrate foramen; qj, quadratojugal; rib, rib; so,
supraoccipital; sa, surangular; scl, sclera; spl, splenial; sq, squamosal; stf, supratemporal fenestra; t,
tooth; tu, tuber; vrop, ventral ramus of opisthotic. White=bone; grey=matrix. Same scale applies to A
and B.

Figure 5. Specimen AMNH 2239 showing partial skull roof and jaws in dorsal and left approximately
ventrolateral view respectively. ?pbs, possible parabasisphenoid; d, dentary; f, frontal; m, maxilla; p,
parietal; pof, postfrontal; prf, prefrontal.

Figure 6. Skull of GPIT/RE/12913 in dorsal view. f, frontal; l, lacrimal; m, maxilla; n, nasal; p, parietal;
pof, postfrontal; prf, prefrontal.

Figure 7. Skull of SAM-PK-6047A in (A) right lateral, (B) left lateral, (C) dorsal, and (D) ventral view. ?,
unidentified element or uncertainty in identification when preceding element; a, angular; ar, articular;
cor, coronoid; cor.for, coronoid foramen; d, dentary; f., facet for; f, frontal; hy, hyoid; j, jugal; l, lacrimal;
mx, maxilla; n, nasal; nar, external naris; ost, osteoderms; p, parietal; pof, postfrontal; po, postorbital;
pl, palatine; pmx, premaxilla; q, quadrate; qf, quadrate foramen; qj, quadratojugal; rib, rib; sa,
surangular; scl, sclera; so, supraoccipital; spl(l/r), splenial (left/right); t, tooth. Dark grey=matrix; light
grey=matrix with bone fragments, cross-hatching=broken surface. Same scale applies to A-D.

Figure 8. Skull of SAM-PK-6047A in (A) posterior and (B) anterior view with interpretive drawings. ?,
unidentified element; a, angular; ar, articular; d, dentary; j, jugal; l, lacrimal; mx, maxilla; n, nasal; ost,
osteoderm; t, tooth; p, parietal; po, postorbital; pof, postfrontal; prf, prefrontal; scl, sclera; sq,
squamosal. White=bone; dark grey=matrix; light grey=matrix with bone fragments; cross-
hatching=broken surface. Same scale applies to A and B.

Figure 9. Specimen SAM-PK-6048. (A) Block with nasal (dorsal view) and maxilla (right lateral view);
articulated partial premaxilla and maxilla in right lateral (B) and (C) medial view; (D) partial skull roof in
dorsal view. f, frontal; ip, interparietal; mx, maxilla; p, parietal; pmx, premaxilla; po, postorbital; pof,
postfrontal.

Figure 10. Right hand side of jaws and cheek region and both sides of palate of specimen SAM-PK-
6050 in (A) lateral and (B) medial view. d, dentary; j, jugal; mx, maxilla; sa, surangular; spl, splenial;
pl, palatine; po, postorbital; pt, pterygoid; prf, prefrontal. Same scale applies to A and B.

Figure 11. Cranial (ventral view) and mandibular (left lateral view) elements of specimen SAM-PK-
7699. a, angular; d, dentary; n, nasal; sa, surangular.

Figure 12. Cranium of SAM-PK-13665 in left lateral view. (A) interpretative line drawing; (B)
photograph. ?, uncertain identification; a, angular; ar, articular; bo, basioccipital; d, dentary; eo,
exoccipital; f, frontal; fem, femur; fib, fibula; gastr., gastralia; j(r), jugal (right); il, ilium; l, lacrimal; n,
nasal; op, opisthotic; ost, osteoderms; mtx, matrix; pa, prearticular; pof, postfrontal; pl, palatine; pmx,
premaxilla; prf, prefrontal; pt, pterygoid; q, quadrate; rib, rib; sa, surangular; so, supraoccipital; sq,
squamosal; tib, tibia.

Figure 13. Left hand side of cranium of SAM-PK-13666 in lateral view. ?, uncertain identification; c,
coronoid; d, dentary; f, frontal; j, jugal; l, lacrimal; mx, maxilla; pa, prearticular; po, postorbital; spl,
splenial.

Figure 14. Cranial material of specimen SAM-PK-13667 in right lateral (right jaw, upper) and medial
(left jaw, lower) view. ?, uncertain identification; a, angular; ar, articular; d, dentary; j, jugal; mx,
maxilla; sa, surangular; spl, splenial; t, tooth.

Figure 15. Specimen SAM-PK-K8050. (A) entire block; (B, C) close-up of cranial material. ?, uncertain
identification; f, frontal; l, lacrimal; mx, maxilla; n, nasal; p, parietal; p.mes, parietal of *Mesosuchus*
*browni*; po, postorbital; pof, postfrontal; q, quadrate. Upper scale applies to A and lower to B and C.
Skulls are in broadly dorsal view, but crushed dorsoventrally.

Figure 16. Overview of specimen UMZC T.692, which contains two individuals, showing (A) all blocks
fully assembled with skulls in left lateral view, (B) the same view but with blocks containing individual
A removed to expose the anterior skull and forelimb of individual B, and a close-up of the posterior of
the skull of individual A which preserves much of the braincase in (C) right and (D) left lateral views.
cran.A=cranium of individual A; cran.B=cranium of individual B; d, dentary; ec, ectopterygoid; eo,
exoccipital; f, frontal; f., facet for; h, head; op, opisthotic; qj, quadratojugal; l, lacrimal; n, nasal; nld,
nasolacrimal duct; pcran. A=postcranium of individual A; pcran.B, postcranium of individual B; pbs,

parabasisphenoid; pof, postfrontal; pr, prootic; prf, prefrontal; pt, pterygoid; sa, surangular. Scale
below B applies to A and B, and that between C and D applies to C and D.

Figure 17. Reconstruction of the skull and mandible of *Euparkeria capensis*. Cranium (A) and
mandible (B) in right lateral view; (B) left mandible in medial view; and cranium in (C) dorsal, (D)
ventral, and (E) posterior view. a, angular; ar, articular; bo, basioccipital; d, dentary; eo, exoccipital;
epi, epipterygoid; f, frontal; ip, interparietal; j, jugal; l, lacrimal; lt, laterosphenoid; op, opisthotic; mpr,
median pharyngeal recess; mx, maxilla; n, nasal; p, parietal; pa, prearticular; po, postorbital; pof,
postfrontal; pro, prootic; pbs, parabasisphenoid; pmx, premaxilla; q, quadrate; sa, surangular; so,
supraoccipital, sq, squamosal.

Figure 18. Right antorbital fenestra of SAM-PK-5867 in lateral view. eaof, external antorbital fenestra;
iaof, internal antorbital fenestra; j, jugal; l, lacrimal; mx, maxilla; n, nasal; prf, prefrontal. External
antorbital fenestra outlined in red, internal antorbital fenestra outlined in blue.

Figure 19. CT images showing anterior palate and palatal fenestrae. (A) palate of SAM-PK-5867 in
dorsal view; (B) surface rendering of anterior palate of SAM-PK-5867 viewed through naris in left
anterolateral and dorsal view. l., left; mx, maxilla; pal.p, palatal process of maxilla; pmx, premaxilla; r.,
right; sbof, suborbital fenestra; stf, subtemporal fenestra.

Figure 20. Reconstruction of the braincase of *Euparkeria capensis* in (A) ventral, (B) posterior, (C)
right lateral, (D) dorsal and (E) anterior view. aip, anterior inferior process; bpt, basipterygoid process;
bo, basioccipital; CN, cranial nerve (followed by number); c.p, cultriform process; eo, exoccipital; epi,
epipterygoid; fo, fenestra ovalis; fm, foramen magnum; ica, internal carotid artery foramen; ld, lateral
depression; lt, laterosphenoid; op, opisthotic; mf, metotic foramen; mpr, median pharyngeal recess;
pro, prootic; pbs, parabasisphenoid; so, supraoccipital; ug, unossified gap.

Figure 21. Examples and details of the premaxilla. Right (A) and left (B) premaxilla of SAM-PK-6047A
in lateral view; (C) right premaxilla of SAM-PK-5867 in lateral view; (D) coronal view of CT scan of the
premaxillae of SAM-PK-5867, showing their anterior palatal contact. ?for, potential foramen; ad.p,
anterodorsal process; mb, main body (of premaxilla); pd.p, posterodorsal process; pk, low peak;
32 s.l.r.pmx, palatal contact between left and right maxillae. Scale below B applies to A-C, and that below
33 D to D only.

Figure 22. CT slice (A) and reconstruction (B) showing premaxillary teeth of SAM-PK-6047A in cross
section in dorsal view. t1–4, premaxillary teeth 1–4.

Figure 23. CT image of premaxillae and vomers of SAM-PK-6047A in right lateral view. v.t, vomerine
teeth.

Figure 24. Examples and details of the maxilla. (A) left maxilla of SAM-PK-5867 in lateral view; (B)
right maxilla of SAM-PK-5867 in lateral view; (C) left maxilla of SAM-PK-6047A in lateral view; (D)
right maxilla of SAM-PK-6047A in lateral view; left maxilla of SAM-PK-6050 in (E) medial and (F)
lateral view; CT scan renderings of (G) right maxilla of SAM-PK-5867 in medial and slightly
posterodorsal view, with sagittal section to show entrance and path of superior alveolar canal, (H) left
maxilla of SAM-PK-6047A in medial and slightly dorsal view to show entrance of superior alveolar
canal, and (I,J) right maxilla of SAM-PK-5867 in right lateral view with more anterior (I) and more
posterior (J) cross section to show articulation with jugal. am.for, anterior maxillary foramen; a,
angular; ao.fos, antorbital fossa; c, coronoid; d, dentary; das, dorsally ascending sheet; d.p, dorsal
process; f.pl, facet for the palatine; f.pmx, facet for premaxilla; j, jugal; nv.for1, lower row of
neurovascular foramina; mx, maxilla; nv.for2, upper row of neurovascular foramina; pa, prearticular;
pal.p, palatal process; pnt, point; ri, ridge (see text for numbering); s.mxj, suture between maxilla and
jugal; salv.ca, entrance of superior alveolar canal; spl, splenial; tlp, thickened lower portion. Upper
scale applies to A-D, scale between E and F to both E and F, those below G and H to those images
respectively, and that below I and J to both I and J.

Figure 25. Left jaws of UMZC T.692 individual A in medial view. d, dentary; f.mb.ect, facet for main
body of ectopterygoid head; ?f.pd.p.ect, probable facet for posterodorsal process of ectopterygoid;

f.p.p.ect, facet for posterior process of ectopterygoid; idp, interdental plate; j, jugal; meck.fos, Meckelian fossa; mx, maxilla; pa, prearticular; spl, splenial; t, tooth.

Figure 26. Right maxillary tooth of SAM-PK-6047A in lateral view. dent, denticles; li, line.

Figure 27. Examples and details of the nasal. Nasals of SAM-PK-5867 in (A) right lateral, (B) left lateral, and (C) dorsal view, and of SAM-PK-6047A in (D) dorsal, (E) right lateral, (F) left lateral view. f.mx, maxillary facet; f.n, nasal facet; pit, pit. Same scale applies to all images.

Figure 28. Line drawing showing the maxillary and premaxillary facets of the nasal from SAM-PK-6047A in right lateral view. f.n, lacrimal facet for nasal; f.pmx, facet for premaxilla; lower, lower section of maxillary facet of nasal; upper, upper section of maxillary facet of nasal; mx, maxilla; n, nasal; pl, palatine; pmx, premaxilla; upper, upper section of maxillary facet of nasal. White=bone; dark grey=matrix; light grey=broken surface; medium grey=articulatory facet.

Figure 29. Examples and details of the frontal. Frontals of SAM-PK-5867 in (A) dorsal and (B) right lateral view and of SAM-PK-6047 in right lateral (C) and dorsal (D) view. f, frontal; gr, groove; p, parietal; pit, pit; prf, prefrontal; s., suture between. Scale below D applies to A-D.

Figure 30. Examples and details of articulation of the parietal. (A) parietals of SAM-PK-5867 in dorsal view; (B) right parietal of SAM-PK-6047A in dorsal view; (C) CT scan cross section of SAM-PK-5867 showing articulation of the parietal with squamosal. gr, groove; p, parietal; pl.p, posterolateral process; ri, ridge; sq, squamosal; str, striations.

Figure 31. Interparietal of SAM-PK-5867 in dorsal view (A) with and (B) without boundaries of elements indicated. ?t, possible embryonic tabulars; ip, interparietal; ri, ridge.

[revised manuscript text omitted]
 epiptyergoid with the pterygoid can be present). Abbreviations: a, angular; ar, articular; bo, basioccipital; bpt, basiptyergoid process; btbo, basal tuber of the basioccipital; ch, choana; d, dentary; ect, ectopterygoid; eo, exoccipital; f, frontal; ip, interparietal; j, jugal; l, lacrimal; lt, laterosphenoid; ltf, lateral temporal fenestra; mx, maxilla; n, nasal; p, parietal; pa, prearticular; pbs, parabasisphenoid; pl, palatine; pmx, premaxilla; pof, postfrontal; po, postorbital; prf, prefrontal; pt, pterygoid; ptf, posttemporal fenestra; q, quadrate; qj, quadratojugal; sa, surangular; so, supraoccipital; stf, supratemporal fenestra; v, vomer.

Figure 1. Schematic phylogeny based on that of Ezcurra (2016) and Sookias (2016), showing position of *Euparkeria capensis* (blue text, yellow background), other euparkeriids (blue text), taxa previously considered euparkeriids (pink text), and other key early diapsid taxa. Although Proterochampsidae was found to be the sister taxon to Archosauria by Ezcurra, many analyses have placed *Euparkeria* as the sister taxon to crown Archosauria, including the the analysis of Nesbitt (2011) upon which the analyses of Sookias et al. (2014a,b) and Sookias (2016) were based.

337x286mm (72 x 72 DPI)

Figure 2. Maps showing (A) location of Aliwal North within South Africa, and (B) the approximate position of the locality thought to have yielded all *Euparkeria* fossils (a stone quarry adjacent to a small stream - "Krietfontein Spruit") - this location is based on the newly rediscovered notes of Mr. Alfred Brown, and differs from the position identified by Ewer (1965) which was described as being on the way out of Aliwal North in the direction of Lady Grey; (C) schematic stratigraphic column showing current understanding of the correlation of *Cynognathus* Assemblage Zone Subzone B, within which the locality is placed.

1021x721mm (72 x 72 DPI)

Figure 3. Holotype skull SAM-PK-5867 in (A) right lateral, (B) left lateral, (C) dorsal, and (D) ventral view. ?, unidentified element or uncertainty in identification when preceding element; a, angular; ar, articular; bo, basioccipital; bpt, basipterygoid process; c, coronoid; ch, choana; d, dentary; eaof, external antorbital fenestra; ect, ectopterygoid; eo, exoccipital; f(r/l), frontal (right/left); fm, foramen magnum; hy, hyoid; iaof, internal antorbital fenestra; ica, internal carotid artery foramen; ip, interparietal; j, jugal; l, lacrimal; lt, laterosphenoid; ltf, lateral temporal fenestra; meck.for, Meckelian foramen; mf, mandibular fenestra; mx, maxilla; n, nasal; nar, external naris; or, orbit; ost, osteoderms; p, parietal; pa, prearticular; pof, postfrontal; ppo, postorbital; pp, paroccipital process; pl, palatine; pmx, premaxilla; pt(r/l), pterygoid (right/left); ptf, posttemporal fenestra; q, quadrate; qf, quadrate foramen; qj, quadratojugal; rib, rib; sa, surangular; scl, sclera; so, supraoccipital; spl, splenial; stf, supratemporal fenestra; t, tooth; tu, tuber; v, vomer; vrop, ventral ramus of opsithotic. White=bone; grey=matrix. Same scale applies to A-D.

723x972mm (72 x 72 DPI)

Figure 4. Holotype skull SAM-PK-5867 in (A) posterior and (B) anterior view with interpretive drawings. ?, unidentified element; a, angular; ar, articular; bo, basioccipital; bpt, basipterygoid process; btbo, basal tuber of the basioccipital; d, dentary; eaof, external antorbital fenestra; eo, exoccipital; f, frontal; fm, foramen magnum; hy, hyoid; ip, interparietal; j, jugal; l, lacrimal; mx, maxilla; n, nasal; nar, naris; or, orbit; ost, osteoderms; p, parietal; pa, prearticular; po, postorbital; pof, postfrontal; pmx, premaxilla; ppo, postorbital; pp, paroccipital process; preart, prearticular; pt, pterygoid; ptf, posttemporal fenestra; q, quadrate; qf, quadrate foramen; qj, quadratojugal; rib, rib; so, supraoccipital; sa, surangular; scl, sclera; spl, splenial; sq, squamosal; stf, supratemporal fenestra; t, tooth; tu, tuber; vrop, ventral ramus of opisthotic. White=bone; grey=matrix. Same scale applies to A and B.

723x973mm (72 x 72 DPI)

Figure 5. Specimen AMNH 2239 showing partial skull roof and jaws in dorsal and left approximately ventrolateral view respectively. ?pbs, possible parabasisphenoid; d, dentary; f, frontal; m, maxilla; p, parietal; pof, postfrontal; prf, prefrontal.

387x274mm (72 x 72 DPI)

Figure 6. Skull of GPIT/RE/12913 in dorsal view. f, frontal; l, lacrimal; m, maxilla; n, nasal; p, parietal; pof, postfrontal; prf, prefrontal.

722x457mm (72 x 72 DPI)

Figure 7. Skull of SAM-PK-6047A in (A) right lateral, (B) left lateral, (C) dorsal, and (D) ventral view. ?, unidentified element or uncertainty in identification when preceding element; a, angular; ar, articular; cor, coronoid; cor.for, coronoid foramen; d, dentary; f., facet for; f, frontal; hy, hyoid; j, jugal; l, lacrimal; mx, maxilla; n, nasal; nar, external naris; ost, osteoderms; p, parietal; pof, postfrontal; po, postorbital; pl, palatine; pmx, premaxilla; q, quadrate; qf, quadrate foramen; qj, quadratojugal; rib, rib; sa, surangular; scl, sclera; so, supraoccipital; spl(l/r), splenial (left/right); t, tooth. Dark grey=matrix; light grey=matrix with bone fragments, cross-hatching=broken surface. Same scale applies to A-D.

Figure 8. Skull of SAM-PK-6047A in (A) posterior and (B) anterior view with interpretive drawings. ?, unidentified element; a, angular; ar, articular; d, dentary; j, jugal; l, lacrimal; mx, maxilla; n, nasal; ost, osteoderm; t, tooth; p, parietal; po, postorbital; pof, postfrontal; prf, prefrontal; scl, sclera; sq, squamosal. White=bone; dark grey=matrix; light grey=matrix with bone fragments; cross-hatching=broken surface. Same scale applies to A and B.

723x827mm (72 x 72 DPI)

Figure 9. Specimen SAM-PK-6048. (A) Block with nasal (dorsal view) and maxilla (right lateral view); articulated partial premaxilla and maxilla in right lateral (B) and (C) medial view; (D) partial skull roof in dorsal view. f, frontal; ip, interparietal; mx, maxilla; p, parietal; pmx, premaxilla; po, postorbital; pof, postfrontal.

722x454mm (72 x 72 DPI)

Figure 10. Right hand side of jaws and cheek region and both sides of palate of specimen SAM-PK-6050 in
(A) lateral and (B) medial view. d, dentary; j, jugal; mx, maxilla; sa, surangular; spl, splenial; pl, palatine;
po, postorbital; pt, pterygoid; prf, prefrontal. Same scale applies to A and B.

722x656mm (72 x 72 DPI)

Figure 11. Cranial (ventral view) and mandibular (left lateral view) elements of specimen SAM-PK-7699. a, angular; d, dentary; n, nasal; sa, surangular.

722x826mm (72 x 72 DPI)

Figure 12. Cranium of SAM-PK-13665 in left lateral view. (A) interpretative line drawing; (B) photograph. ?, uncertain identification; a, angular; ar, articular; bo, basioccipital; d, dentary; eo, exoccipital; f, frontal; fem, femur; fib, fibula; gastr., gastralia; j(r), jugal (right); il, ilium; l, lacrimal; n, nasal; op, opisthotic; ost, osteoderms; mtx, matrix; pa, prearticular; pof, postfrontal; pl, palatine; pmx, premaxilla; prf, prefrontal; pt, pterygoid; q, quadrate; rib, rib; sa, surangular; so, supraoccipital; sq, squamosal; tib, tibia.

1397x1896mm (72 x 72 DPI)

Figure 13. Left hand side of cranium of SAM-PK-13666 in lateral view. ?, uncertain identification; c, coronoid; d, dentary; f, frontal; j, jugal; l, lacrimal; mx, maxilla; pa, prearticular; po, postorbital; spl, splenial.

722x564mm (72 x 72 DPI)

Figure 14. Cranial material of specimen SAM-PK-13667 in right lateral (right jaw, upper) and medial (left jaw, lower) view. ?, uncertain identification; a, angular; ar, articular; d, dentary; j, jugal; mx, maxilla; sa, surangular; spl, splenial; t, tooth.

722x657mm (72 x 72 DPI)

Figure 15. Specimen SAM-PK-K8050. (A) entire block; (B, C) close-up of cranial material. ?, uncertain identification; f, frontal; l, lacrimal; mx, maxilla; n, nasal; p, parietal; p.mes, parietal of *Mesosuchus browni*; po, postorbital; pof, postfrontal; q, quadrate. Upper scale applies to A and lower to B and C. Skulls are in broadly dorsal view, but crushed dorsoventrally.

722x761mm (72 x 72 DPI)

Figure 16. Overview of specimen UMZC T.692, which contains two individuals, showing (A) all blocks fully assembled with skulls in left lateral view, (B) the same view but with blocks containing individual A removed to expose the anterior skull and forelimb of individual B, and a close-up of the posterior of the skull of individual A which preserves much of the braincase in (C) right and (D) left lateral views. cran.A=cranium of individual A; cran.B=cranium of individual B; d, dentary; ec, ectopterygoid; eo, exoccipital; f, frontal; f., facet for; h, head; op, opisthotic; qj, quadratojugal; l, lacrimal; n, nasal; nld, nasolacrimal duct; pcran.A=postcranium of individual A; pcran.B, postcranium of individual B; pbs, parabasisphenoid; pof, postfrontal; pr, prootic; prf, prefrontal; pt, pterygoid; sa, surangular. Scale below B applies to A and B, and that between C and D applies to C and D.

846x772mm (72 x 72 DPI)

Figure 17. Reconstruction of the skull and mandible of *Euparkeria capensis*. Cranium (A) and mandible (B) in right lateral view; (B) left mandible in medial view; and cranium in (C) dorsal, (D) ventral, and (E) posterior view. a, angular; ar, articular; bo, basioccipital; d, dentary; eo, exoccipital; epi, epipterygoid; f, frontal; ip, interparietal; j, jugal; l, lacrimal; lt, laterosphenoid; op, opisthotic; mpr, median pharyngeal recess; mx, maxilla; n, nasal; p, parietal; pa, prearticular; po, postorbital; pof, postfrontal; pro, prootic; pbs, parabasisphenoid; pmx, premaxilla; q, quadrate; sa, surangular; so, supraoccipital, sq, squamosal.

Figure 18. Right antorbital fenestra of SAM-PK-5867 in lateral view. eaof, external antorbital fenestra; iaof, internal antorbital fenestra; j, jugal; l, lacrimal; mx, maxilla; n, nasal; prf, prefrontal. External antorbital fenestra outlined in red, internal antorbital fenestra outlined in blue.

781x670mm (72 x 72 DPI)

Figure 19. CT images showing anterior palate and palatal fenestrae. (A) palate of SAM-PK-5867 in dorsal view; (B) surface rendering of anterior palate of SAM-PK-5867 viewed through naris in left anterolateral and dorsal view. l., left; mx, maxilla; pal.p, palatal process of maxilla; pmx, premaxilla; r., right; sbof, suborbital fenestra; stf, subtemporal fenestra.

1261x510mm (72 x 72 DPI)

Figure 20. Reconstruction of the braincase of *Euparkeria capensis* in (A) ventral, (B) posterior, (C) right lateral, (D) dorsal and (E) anterior view. aip, anterior inferior process; bpt, basipterygoid process; bo, basioccipital; CN, cranial nerve (followed by number); c.p, cultriform process; eo, exoccipital; epi, epipterygoid; fo, fenestra ovalis; fm, foramen magnum; ica, internal carotid artery foramen; Id, lateral depression; lt, laterosphenoid; op, opisthotic; mf, metotic foramen; mpr, median pharyngeal recess; pro, prootic; pbs, parabasisphenoid; so, supraoccipital; ug, unossified gap.

Figure 21. Examples and details of the premaxilla. Right (A) and left (B) premaxilla of SAM-PK-6047A in lateral view; (C) right premaxilla of SAM-PK-5867 in lateral view; (D) coronal view of CT scan of the premaxillae of SAM-PK-5867, showing their anterior palatal contact. ?for, potential foramen; ad.p, anterodorsal process; mb, main body (of premaxilla); pd.p, posterodorsal process; pk, low peak; s.l.r.pmx, palatal contact between left and right maxillae. Scale below B applies to A-C, and that below D to D only.

1189x392mm (72 x 72 DPI)

Figure 22. CT slice (A) and reconstruction (B) showing premaxillary teeth of SAM-PK-6047A in cross section in dorsal view. t1–4, premaxillary teeth 1–4.

527x367mm (72 x 72 DPI)

Figure 23. CT image of premaxillae and vomers of SAM-PK-6047A in right lateral view. v.t, vomerine teeth.

822x631mm (72 x 72 DPI)

Figure 24. Examples and details of the maxilla. (A) left maxilla of SAM-PK-5867 in lateral view; (B) right maxilla of SAM-PK-5867 in lateral view; (C) left maxilla of SAM-PK-6047A in lateral view; (D) right maxilla of SAM-PK-6047A in lateral view; left maxilla of SAM-PK-6050 in (E) medial and (F) lateral view; CT scan renderings of (G) right maxilla of SAM-PK-5867 in medial and slightly posterodorsal view, with sagittal section to show entrance and path of superior alveolar canal, (H) left maxilla of SAM-PK-6047A in medial and slightly dorsal view to show entrance of superior alveolar canal, and (I,J) right maxilla of SAM-PK-5867 in right lateral view with more anterior (I) and more posterior (J) cross section to show articulation with jugal. am.for, anterior maxillary foramen; a, angular; ao.fos, antorbital fossa; c, coronoid; d, dentary; das, dorsally ascending sheet; d.p, dorsal process; f.pl, facet for the palatine; f.pmx, facet for premaxilla; j, jugal; nv.for1, lower row of neurovascular foramina; mx, maxilla; nv.for2, upper row of neurovascular foramina; pa, prearticular; pal.p, palatal process; pnt, point; ri, ridge (see text for numbering); s.mxj, suture between maxilla and jugal; salv.ca, entrance of superior alveolar canal; spl, splenial; tlp, thickened lower portion. Upper scale applies to A-D, scale between E and F to both E and F, those below G and H to those images respectively, and that below I and J to both I and J.

723x1481mm (72 x 72 DPI)

Figure 25. Left jaws of UMZC T.692 individual A in medial view. d, dentary; f.mb.ect, facet for main body of ectopterygoid head; ?f.p.d.p.ect, probable facet for posterodorsal process of ectopterygoid; f.p.p.ect, facet for posterior process of ectopterygoid; idp, interdental plate; j, jugal; meck.fos, Meckelian fossa; mx, maxilla; pa, prearticular; spl, splenial; t, tooth.

722x438mm (72 x 72 DPI)

Figure 26. Right maxillary tooth of SAM-PK-6047A in lateral view. dent, denticles; li, line.

236x335mm (72 x 72 DPI)

Figure 27. Examples and details of the nasal. Nasals of SAM-PK-5867 in (A) right lateral, (B) left lateral, and (C) dorsal view, and of SAM-PK-6047A in (D) dorsal, (E) right lateral, (F) left lateral view. f.mx, maxillary facet; f.n, nasal facet; pit, pit. Same scale applies to all images.

723x776mm (72 x 72 DPI)

Figure 28. Line drawing showing the maxillary and premaxillary facets of the nasal from SAM-PK-6047A in right lateral view. f.n, lacrimal facet for nasal; f.pmx, facet for premaxilla; lower, lower section of maxillary facet of nasal; upper; mx, maxilla; n, nasal; pl, palatine; pmx, premaxilla; upper section of maxillary facet of nasal. White=bone; dark grey=matrix; light grey=broken surface; medium grey=articular facet.

Figure 29. Examples and details of the frontal. Frontals of SAM-PK-5867 in (A) dorsal and (B) right lateral
view and of SAM-PK-6047 in right lateral (C) and dorsal (D) view. f, frontal; gr, groove; p, parietal; pit, pit;
prf, prefrontal; s., suture between. Scale below D applies to A-D.

613x972mm (72 x 72 DPI)

Figure 30. Examples and details of articulation of the parietal. (A) parietals of SAM-PK-5867 in dorsal view; (B) right parietal of SAM-PK-6047A in dorsal view; (C) CT scan cross section of SAM-PK-5867 showing articulation of the parietal with squamosal. gr, groove; p, parietal; pl.p, posterolateral process; ri, ridge; sq, squamosal; str, striations.

1799x1237mm (72 x 72 DPI)

Figure 31. Interparietal of SAM-PK-5867 in dorsal view (A) with and (B) without boundaries of elements
indicated. ?t, possible embryonic tabulars; ip, interparietal; ri, ridge.

743x371mm (72 x 72 DPI)

[revised manuscript text omitted]
